# Convergence Analysis of Two-Layer Neural Networks under Gaussian Input Masking

## Abstract

We investigate the convergence guarantee of two-layer neural network training with Gaussian randomly masked inputs. This scenario corresponds to Gaussian dropout at the input level, or noisy input training common in sensor networks, privacy-preserving training, and federated learning, where each user may have access to partial or corrupted features. Using a Neural Tangent Kernel (NTK) analysis, we demonstrate that training a two-layer ReLU network with Gaussian randomly masked inputs achieves linear convergence up to an error region proportional to the mask's variance. A key technical contribution is resolving the randomness within the non-linear activation, a problem of independent interest.

## 1 Introduction

Neural networks (NNs) have revolutionized AI applications, where their success largely stems from their ability to learn complex patterns when trained on well-curated datasets (Schuhmann et al., 2022; Li et al., 2023b; Gunasekar et al., 2023; Edwards, 2024). A component to the success of NNs is its ability to model a broad range of tasks and data distributions under various scenarios. Empirical evidence has suggested neural network's ability to learn even under noisy input (Kariotakis et al., 2024), gradient noise (Ruder, 2017), as well as modifications to the internal representations during training (Srivastava et al., 2014; Yuan et al., 2022). Leveraging such ability of the neural networks, many real-world deployment adopts a modification to the data representations during training to achieve particular goals such as robustness, privacy, or efficiency. Among the methods, perturbing the representations with an additive noise has been studied by a number of prior works (Gao et al., 2019b; Li et al., 2025; 2023a; Madry et al., 2018; Loo et al., 2022; Tsilivis & Kempe, 2022; Ilyas et al., 2019), showcasing both the benefit of such perturbation and the stable convergence of the training under this setting. Compared with additive noise, perturbing the representations by multiplying it with a mask has rarely been studied theoretically.

Perturbing the representations with multiplicative noise appears in many real-world settings, either by design or unintentionally. For instance, in federated learning (FL) settings (McMahan et al., 2017; Kairouz et al., 2021), particularly vertical FL (Cheng et al., 2020; Liu et al., 2021b; Romanini et al., 2021; He et al., 2020; Liu et al., 2022; 2024), different features of the input data may be available to different parties, effectively creating a form of sparsity-inducing multiplicative masking on the input space. Moreover, the drop-out family (Srivastava et al., 2014; Rey & Mnih, 2021) is a class of methods to prevent overfitting and improve generalization ability of neural networks during training. Lastly, training models under data-parallel protocol over a wireless channel incurs the channel effect that blurs the data passed to the workers through a multiplication Tse & Viswanath (2005).

Theoretically analyzing the training dynamics of neural networks under these settings are difficult, especially when the introduced randomness are intertwined with the nonlinearity of the activation function. While there has been previous work that studies the convergence of neural network training under drop-out (Liao & Kyrillidis, 2022; Mianjy & Arora, 2020), they often assume that the drop-out happens after the nonlinear activations are applied. From a technical perspective, statistics of the neural network outputs are easier to handle as the randomness are not affected by the nonlinearity.

In this paper, we take a step further into the understanding of multiplicative perturbations in neural network training by considering noise applied before the nonlinear activation. In particular, the setting we consider is the training of a two-layer MLP where the inputs bears a multiplicative Gaussian mask. This prototype provides a simplified scenario to study the noise-inside-activation difficulty, while generalizes various training scenarios ranging from input masking (Kariotakis et al., 2024) to Gaussian drop-out (Rey & Mnih, 2021), if one views the input in our setting as fixed embeddings from previous layers of a deep neural network. Under this setting, we aim to answer the following question:

*How do multiplicative perturbations at the input level propagate through the network and affect the training dynamics?*

**Our Contributions.** Analyzing the training dynamics under the Gaussian masks over the input means that we have to study the statistical properties of random variables inside a non-linear function. Our work takes a step towards resolving this technical difficulty. We utilize an NTK-based analysis (Du et al., 2018; Song & Yang, 2020; Oymak & Soltanolkotabi, 2019; Liao & Kyrillidis, 2022) to study the training convergence of the two-layer MLP under sufficient overparameterization. To our knowledge, this work provides the first convergence analysis for neural network training under Gaussian multiplicative input masking. Our main contributions are summarized as follows:

- *Theoretical Analysis of Input Masking.* We provide a rigorous characterization of the training dynamics for two-layer ReLU networks where noise is injected *before* the non-linear activation. We overcome the technical challenge of resolving the expectation of non-linear functions of random variables, proving that the expected loss decomposes into a smoothed objective plus an adaptive, data-dependent regularizer.
- *General Stochastic Training Framework.* We develop a general convergence theorem for overparameterized neural networks trained with biased stochastic gradient estimators. This result, which establishes linear convergence to a noise-dependent error ball, is of independent interest beyond the specific setting of Gaussian masking.
- *Explicit Convergence Guarantees.* We derive constructive bounds for the convergence rate and the final error radius. We show explicitly how these quantities depend on the mask variance $\kappa^2$, the network width $m$, and the initialization scale, demonstrating that the training converges linearly up to a floor determined by the noise level.
- *Empirical Validation.* We confirm our theoretical predictions about the expected gradient and loss landscape through simulations. We further illustrate the analysis in two practical regimes where the noise model is relevant: distributed training over wireless channels (where multiplicative noise arises naturally as channel fading) and Membership Inference Attack defense (where the noise is deliberately injected as a candidate regularizer).

## 2 Related Work

**NTK-based analysis.** The theoretical backbone of our work is the Neural Tangent Kernel. Jacot et al. (2018) showed that an infinitely wide network evolves as a Gaussian process governed by a stable kernel formed from the outer product of the network's tangent features. This observation was turned into non-asymptotic, finite-width convergence guarantees: Li & Liang (2019), Du et al. (2018; 2019) and Allen-Zhu et al. (2019) proved that (stochastic) gradient descent drives the training loss of two-layer and deep ReLU networks to zero at a linear rate under sufficient overparameterization, with the rate controlled by the least eigenvalue $\lambda_0$ of the limiting kernel $H^\infty$. The width requirements were progressively sharpened through matrix-concentration arguments (Song & Yang, 2020), fine-grained Jacobian control (Oymak & Soltanolkotabi, 2019), eigen-direction and functional-approximation analyses (Su & Yang, 2019; Arora et al., 2019), and improved over-parameterization bounds (Zou & Gu, 2019). The framework was extended to classification by treating the tangent features as a separating map (Ji & Telgarsky, 2020) and recast through the Polyak–Łojasiewicz condition implied by a well-conditioned NTK (Nguyen, 2021; Liu et al., 2021a).

Most relevant to us are NTK-based analyses of training under *perturbation*. For dropout-style masking, Mianjy & Arora (2020) prove $O(1/\epsilon)$ iteration complexity for two-layer ReLU networks under an NTK-margin assumption, and Liao & Kyrillidis (2022) establish linear convergence for dropout, multi-sample dropout, and

independent-subnetwork training by showing the masked surrogate kernel stays close to $H^\infty$. For adversarial (additive) perturbation, Gao et al. (2019a) gave the first NTK convergence guarantee, refined to polynomial width by Zhang et al. (2020); convergence of federated averaging has likewise been analyzed in this regime (Huang et al., 2021). Our analysis apart from the above work in two aspects. First, the masking schemes above are applied *after* the nonlinearity, so the randomness commutes with the activation and the hidden-layer statistics are available in closed form; our mask is multiplicative and applied to the *input*, hence *inside* the ReLU, so every expectation must be pushed through the activation. Second, the resulting masked gradient is a *biased* estimator of the clean gradient, whereas standard NTK proofs assume an exact or unbiased step, instead of a deterministic perturbation in the adversarial training scenario.

**Implicit bias of neural-network training.** A complementary literature asks not how fast training converges but the solution that the training algorithm converges to. For classification with exponentially tailed losses on separable data, gradient descent drives the parameter norm to infinity while its *direction* converges to the maximum-margin solution (Soudry et al., 2024; Ji & Telgarsky, 2019). Lyu & Li (2020) generalized this to homogeneous networks, showing the normalized margin increases and limit points satisfy the KKT conditions of a margin-maximization problem. In the wide two-layer limit, Chizat & Bach (2020) showed gradient flow on the logistic loss converges to a max-margin solution in a non-Hilbertian variation-norm space. We refer to Vardi (2022) for a more comprehensive survey of work in this direction. We believe our technical machinery is directly portable to this line of work when considering the implicit bias under Gaussian input masking, which require expectations of the ReLU or of its subgradient under a Gaussian distribution over inputs.

**Feature learning.** The kernel regime linearizes training, but this is also its central limitation: with the tangent features frozen at initialization, a lazy network provably cannot learn data-adaptive features and is matched by a fixed kernel (Chizat et al., 2020). A large recent literature studies the rich regime, where neurons travel far enough to outperform kernels. The mean-field view models the empirical distribution of neurons as a Wasserstein gradient flow and establishes global convergence for infinitely wide two-layer networks, including under noisy SGD (Chizat & Bach, 2018; Mei et al., 2018; 2019). A second line obtains separations in sample complexity on structured targets: a single gradient step already produces a rank-one feature update that beats kernels on single-index models (Ba et al., 2022), and gradient descent learns multi-index models, sparse parities, and staircase/leap functions with sample complexity unattainable by any rotationally invariant kernel (Damian et al., 2022; Abbe et al., 2024; 2023; Bietti et al., 2022; Daniely & Malach, 2020; Barak et al., 2023) .The lazy-to-rich transition is controlled by initialization scale (Woodworth et al., 2020) and, at infinite width, by the maximal-update ($\mu$P) parametrization (Yang & Hu, 2022). Our paper deliberately operates in the lazy regime and does not claim feature learning; the weight movement is bounded and the activation pattern is essentially frozen. However, our central technical contribution of evaluating the closed form expression of the Gaussian expectation of ReLU non-linearity has the potential to be transferrable to this line of work.

## 3 Problem Setup

Given a dataset $\{(\mathbf{x}_i, y_i)\}_{i=1}^n$, we are interested in training a neural network $f(\boldsymbol{\theta}, \cdot)$ that maps each input $\mathbf{x}_i \in \mathbb{R}^d$'s to an output $f(\boldsymbol{\theta}, \mathbf{x}_i)$ that fits the labels $y_i \in \mathbb{R}$. We consider $f(\boldsymbol{\theta}, \cdot)$ as a two-layer ReLU activated Multi-Layer Perceptron (MLP) under the NTK scaling:

$$f(\boldsymbol{\theta}, \mathbf{x}) = \frac{1}{\sqrt{m}} \sum_{r=1}^m a_r \sigma\left(\mathbf{w}_r^\top \mathbf{x}\right),$$

where $\boldsymbol{\theta} = (\{\mathbf{w}_r\}_{r=1}^m, \{a_r\}_{r=1}^m)$ denotes the neural network parameters, and $\sigma(\cdot) = \max\{0, \cdot\}$ denotes the ReLU activation function. We assume that the second-layer weights $a_r \in \{\pm 1\}$ are fixed, and only the first layer weights $\mathbf{w}_r$'s are trainable. Thus, we will be using $f(\mathbf{W}, \mathbf{x}) \equiv f(\boldsymbol{\theta}, \mathbf{x})$ where $\mathbf{W} \in \mathbb{R}^{m \times d}$, unless otherwise stated. This neural network set-up is studied widely in previous works (Du et al., 2018). We consider the training of the neural network by minimizing the MSE loss $\mathcal{L}(\mathbf{W})$ over the dataset $\{(\mathbf{x}_i, y_i)\}_{i=1}^n$:

$$\mathcal{L}(\mathbf{W}) = \frac{1}{2} \sum_{i=1}^n (f(\mathbf{W}, \mathbf{x}_i) - y_i)^2.$$

With the influence of the ReLU activation, the loss is both non-convex and non-smooth. However, a line of previous works (Du et al., 2018; Song & Yang, 2020; Oymak & Soltanolkotabi, 2019) proves a linear convergence rate of the loss function under the assumption that the number of hidden neurons is sufficiently large by adopting an NTK-based analysis (Jacot et al., 2018).

*While there have been theoretical approaches and assumptions that go beyond the NTK assumption, our focus is on a generalized scenario where the input data may be corrupted in each iteration under a multiplicative Gaussian noise:* Let $\mathbf{c} \sim \mathcal{N}\left(\mathbf{1}_d, \kappa^2 \mathbf{I}_d\right)$ be an isotropic Gaussian random vector centered at the all-one vector $\mathbf{1}_d$, the neural network output is given by $f\left(\mathbf{W}, \mathbf{x} \odot \mathbf{c}\right)$, where $\odot$ denotes the Hadamard (element-wise) product between two vectors. Under the multiplicative noise, the neural network is trained with gradient descent where each gradient is computed based on the surrogate loss $\mathcal{L}_{\mathbf{C}}\left(\mathbf{W}\right)$ defined over the neural network with the masked input:

$$\mathcal{L}_{\mathbf{C}}\left(\mathbf{W}\right) = \frac{1}{2}\sum_{i=1}^{n}\left(f\left(\mathbf{W}, \mathbf{x}_i \odot \mathbf{c}_i\right) - y_i\right)^2.$$

Here $\mathbf{C} = \{\mathbf{c}_i\}_{i=1}^{n}$ denotes the collection of the masks for all input $\mathbf{x}_i$. We assume that $\mathbf{c}_i$'s are independent. In real-world applications, this scheme can be considered as training on an imprecise hardware, where each input data point is read-in with noise. Alternatively, one could view each $\mathbf{x}_i$ as the output of a pre-trained large model, and our training scheme can be considered as fine-tuning the last two layers with the Gaussian drop-out (Wang & Manning, 2013; Kingma et al., 2015; Rey & Mnih, 2021) in the intermediate layer.

Let $\{\mathbf{W}_k\}_{k=1}^{K}$ be generated from the stochastic gradient descent given by:

$$\mathbf{W}_{k+1} = \mathbf{W}_k - \eta \nabla_{\mathbf{W}} \mathcal{L}_{\mathbf{C}_k}\left(\mathbf{W}_k\right), \tag{1}$$

where $\mathbf{C}_k$ is sampled independently in every iteration of the gradient descent. Our goal is to study the convergence of the loss sequence $\{\mathcal{L}\left(\mathbf{W}_k\right)\}_{k=1}^{\infty}$. Notice that the loss involved in the weight-update is the surrogate loss $\mathcal{L}_{\mathbf{C}_k}\left(\mathbf{W}_k\right)$, but the loss we aim to show convergence is the original loss $\mathcal{L}\left(\mathbf{W}\right)$.

Our set-up marks some differences from previous works. First, our set-up is distinct from unbiased estimators in current literature; our setup does not have such favorable property, since the randomness is applied at the input level of the neural network. Second, although there is a line of work that analyzes the convergence of vanilla drop-out tranining on two-layer neural networks (Liao & Kyrillidis, 2022; Mianjy & Arora, 2020), in their analysis the mask is applied to the hidden neurons after the activation function. On the contrary, our mask is applied directly to the input, which is contained in the non-linear function. Therefore, any analysis of the mask randomness must go through the ReLU function, which brings technical difficulty. We assume the following property for the training data.

**Assumption 3.1.** *The training dataset $\{(\mathbf{x}_i, y_i)\}_{i=1}^{n}$ satisfies $\|\mathbf{x}_i\|_2 \leq 1, |y_i| \leq O(1)$, and for any pair $i \neq j$, there exists no real number $q$ such that $\mathbf{x}_i = q \cdot \mathbf{x}_j$.*

This assumption guarantees the boundedness of the dataset, and that the input data are non-degenerate, which is a standard assumption in Du et al. (2018); Song & Yang (2020); Liao & Kyrillidis (2022).

## 4 Expectation of the Loss and Gradient under Gaussian Mask

A formal mathematical characterization of the expected loss and gradient is essential not only in prior literature of neural network training convergence (Liao & Kyrillidis, 2022; Mianjy & Arora, 2020) but also in the classical analysis of SGD even in the convex domain (Shamir & Zhang, 2013; Garrigos & Gower, 2023; Tang et al., 2013). In this section, we focus on the derivation of the explicit form of the expected surrogate loss $\mathbb{E}_{\mathbf{C}}\left[\mathcal{L}_{\mathbf{C}}\left(\mathbf{W}\right)\right]$ and the expected surrogate gradient $\mathbb{E}_{\mathbf{C}}\left[\nabla_{\mathbf{W}}\mathcal{L}_{\mathbf{C}}\left(\mathbf{W}\right)\right]$. Starting with gradient calculations, the surrogate gradient with respect to the $r$-th neuron can be written as:

$$\nabla_{\mathbf{w}_r}\mathcal{L}_{\mathbf{C}}\left(\mathbf{W}\right) = \frac{a_r}{\sqrt{m}}\sum_{i=1}^{n}\left(f\left(\mathbf{W}, \mathbf{x}_i \odot \mathbf{c}_i\right) - y_i\right)\left(\mathbf{x}_i \odot \mathbf{c}_i\right)\mathbb{I}\left\{\mathbf{w}_r^{\top}\left(\mathbf{x}_i \odot \mathbf{c}_i\right) \geq 0\right\} \tag{2}$$

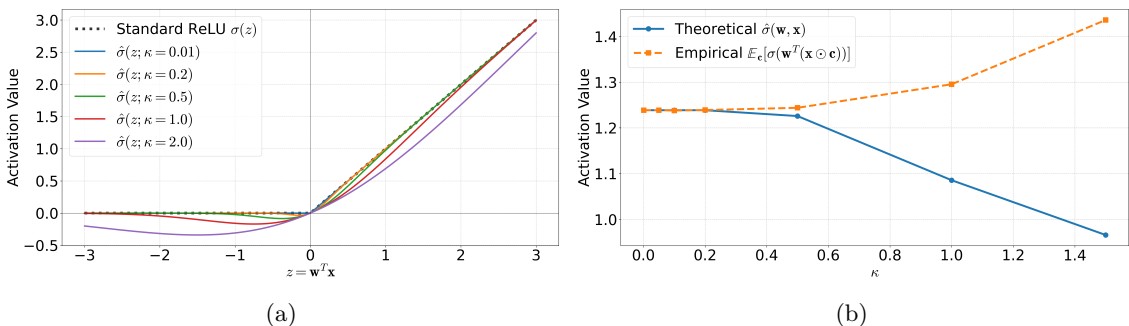

Figure 1: (a). Effect of the noise standard deviation $\kappa$ on the shape of the smoothed activation function $\hat{\sigma}(z;\kappa) = z \cdot \Phi_1(z/(\kappa\|\mathbf{w}\odot\mathbf{x}\|_2))$, where $z = \mathbf{w}^\top\mathbf{x}$. For this visualization, $\|\mathbf{w}\odot\mathbf{x}\|_2$ is held constant at 1.0. As $\kappa$ increases, the activation becomes progressively smoother compared to the standard ReLU (dotted black line). For small $\kappa$ (e.g., $\kappa = 0.01$), $\hat{\sigma}$ closely approximates the standard ReLU. (b). Theoretical smoothed activation $\hat{\sigma}(\mathbf{w},\mathbf{x})$ versus its empirical estimate $\mathbb{E}_\mathbf{c}[\sigma(\mathbf{w}^\top(\mathbf{x}\odot\mathbf{c}))]$ for a fixed pre-activation value $\mathbf{w}^\top\mathbf{x} \approx 0.77$ (actual value depends on fixed $\mathbf{w},\mathbf{x}$) as the noise standard deviation $\kappa$ varies. The close match across a range of—relatively small—$\kappa$ values validates the theoretical model for $\hat{\sigma}$. Note that this behavior consistently follows empirically for different $\mathbf{w},\mathbf{x}$ values.

Setting $\mathbf{c}_i = \mathbf{1}$ for all $i \in [n]$ gives the gradient of the original loss $\nabla_{\mathbf{w}_r}\mathcal{L}(\mathbf{W})$. Let $\Phi_1(\cdot)$ denote the CDF of the standard (one-dimensional) Gaussian random variable, and let $\phi, \psi : \mathbb{R} \to \mathbb{R}$ be defined as:

$$\phi(x) = \exp\left(-x^2\right); \quad \psi(x) = |x| \cdot \phi(x). \tag{3}$$

Observe that $\phi(x) \in (0,1]$ and $\psi(x) \in (0, 1/\sqrt{2e}]$. Before we state the results in this section, we need to define the following quantities.

**Definition 4.1.** Fix a first-layer weight $\mathbf{W} \in \mathbb{R}^{m\times d}$ and training data $\{(\mathbf{x}_i, y_i)\}_{i=1}^n$. We define the:

- Data-related quantity: $B_\mathbf{x} = \max_{i\in[n]}\|\mathbf{x}_i\|_\infty$, $B_y := \max_{i\in[n]}|y_i|$.

- Weight-related quantity (row-wise): $R_\mathbf{w} = \max_{r\in[m]}\|\mathbf{w}_r\|_2$.

- Mixed quantity: $R_\mathbf{u} := \max_{r\in[m],i\in[n]}\|\mathbf{w}_r\odot\mathbf{x}_i\|_2$ and

$$\psi_{\max} = \max_{r\in[m],i\in[n]} \psi\left(\frac{\mathbf{w}_r^\top\mathbf{x}_i}{2\kappa\|\mathbf{w}_r\odot\mathbf{x}_i\|_2}\right), \phi_{\max} = \max_{r\in[m],i\in[n]} \phi\left(\frac{\mathbf{w}_r^\top\mathbf{x}_i}{2\kappa\|\mathbf{w}_r\odot\mathbf{x}_i\|_2}\right).$$

**Expected Surrogate Loss.** To start, we focus on the expected loss under the Gaussian input mask. For a fixed neural network $f(\mathbf{W}, \cdot)$, we have the following result.

**Theorem 4.2.** Let $\mathbf{u}_{i,r} = \mathbf{w}_r \odot \mathbf{x}_i$. Define the smoothed activation and neural network as:

$$\hat{\sigma}_\kappa(\mathbf{w}, \mathbf{x}) = \mathbf{w}^\top\mathbf{x} \cdot \Phi_1\left(\frac{\mathbf{w}^\top\mathbf{x}}{\kappa\|\mathbf{w}\odot\mathbf{x}\|_2}\right), \quad \hat{f}(\mathbf{W}, \mathbf{x}) = \frac{1}{\sqrt{m}}\sum_{r=1}^m a_r\hat{\sigma}_\kappa(\mathbf{w}_r, \mathbf{x}).$$

Let $\phi_{\max}, \psi_{\max}, B_y, R_\mathbf{u}$, and $R_\mathbf{w}$ be defined in Definition 4.1. If $B_y \leq 3\sqrt{m}R_\mathbf{w}$, then we have that:

$$\mathbb{E}_\mathbf{C}[\mathcal{L}_\mathbf{C}(\mathbf{W})] = \mathcal{E} + \underbrace{\frac{1}{2}\sum_{i=1}^n\left(\hat{f}(\mathbf{W}, \mathbf{x}_i) - y_i\right)_2^2}_{\mathcal{T}_1} + \underbrace{\frac{\kappa^2}{2m}\sum_{i=1}^n\left\|\sum_{r=1}^m a_r\mathbf{u}_{i,r}\Phi_1\left(\frac{\mathbf{w}_r^\top\mathbf{x}_i}{\kappa\|\mathbf{u}_{i,r}\|_2}\right)\right\|_2^2}_{\mathcal{T}_2},$$

with the magnitude of $\mathcal{E}$ bounded by:

$$|\mathcal{E}| \leq mn\left(\kappa^2R_\mathbf{u}^2\psi_{\max}^2 + \left(\kappa^2R_\mathbf{u}^2 + \kappa R_\mathbf{w}\right)\phi_{\max}^2\right). \tag{4}$$

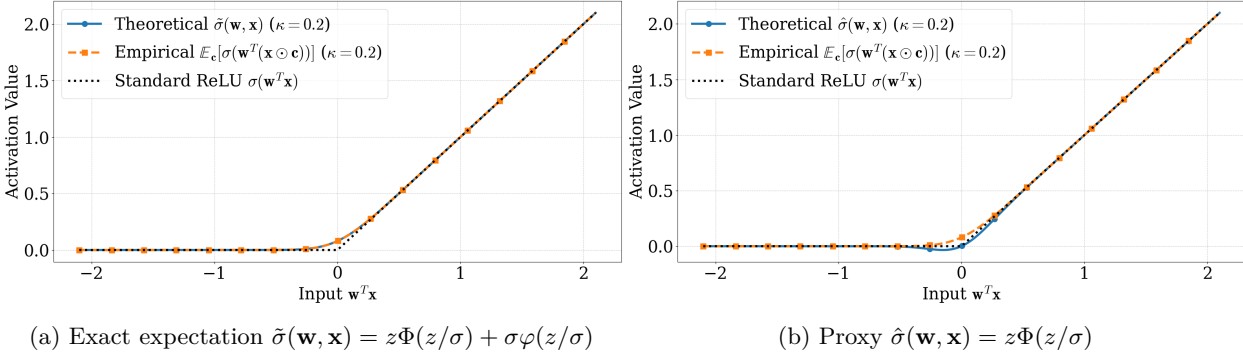

(a) Exact expectation $\tilde{\sigma}(\mathbf{w}, \mathbf{x}) = z\Phi(z/\sigma) + \sigma\varphi(z/\sigma)$    (b) Proxy $\hat{\sigma}(\mathbf{w}, \mathbf{x}) = z\Phi(z/\sigma)$

Figure 2: Smoothed ReLU under multiplicative Gaussian input masking for fixed $\kappa = 0.2$, where $z = \mathbf{w}^\top \mathbf{x}$ and $\sigma = \kappa \|\mathbf{w} \odot \mathbf{x}\|_2$. **(a)** Exact closed-form expectation $\tilde{\sigma}(\mathbf{w}, \mathbf{x}) = \mathbb{E}_{\mathbf{c}}[\sigma(\mathbf{w}^\top(\mathbf{x} \odot \mathbf{c}))] = z\Phi(z/\sigma) + \sigma\varphi(z/\sigma)$ (as shown in 19)) matches the Monte Carlo estimate. **(b)** Proxy smoothed activation $\hat{\sigma}(\mathbf{w}, \mathbf{x}) = z\Phi(z/\sigma)$ (used in Theorem 4.2) differs mainly near $z \approx 0$ due to the missing $\sigma\varphi(z/\sigma)$ term.

**Remark 4.3.** The core of our analysis of the expected loss involves understanding how the ReLU activation behaves under the multiplicative Gaussian input mask. Lemma D.14 in the appendix provides the analytical form for the expectation of a truncated Gaussian random variable. This leads to the definition of a smoothed activation function, as presented in Theorem 4.2. Figure 1b demonstrates the correspondence between the theoretical and empirical values of this smoothed activation across a range of noise levels $\kappa$ for a fixed input $\mathbf{w}^\top \mathbf{x}$: being an approximation of ReLU, as $\kappa$ increases, it is expected the two curves to deviate, yet for small enough $\kappa$ values (here, $\kappa \lesssim 0.2$) the two curves coincide. Figure 2b visually compares this theoretical smoothed activation $\hat{\sigma}$ with its empirical estimate $\mathbb{E}_{\mathbf{c}}[\sigma(\mathbf{w}^\top(\mathbf{x} \odot \mathbf{c}))]$ for a fixed $\kappa = 0.2$ as the input $\mathbf{w}^\top \mathbf{x}$ varies. The close agreement validates our analytical derivation of $\hat{\sigma}$ and illustrates its smoothing effect compared to the standard ReLU.

Thus, the term $\mathbf{\Phi}_1\left(\frac{\mathbf{w}^\top \mathbf{x}}{\kappa \|\mathbf{w} \odot \mathbf{x}\|_2}\right)$ can be interpreted as a smoothed version of the indicator function $\mathbb{I}\{\mathbf{w}^\top \mathbf{x} \geq 0\}$. To visualize the impact of the noise variance $\kappa^2$ on the shape of this smoothed activation, see Figure 1a for various values $\mathbf{w}, \mathbf{x}$ (For this illustration, we assume a fixed value for $\|\mathbf{w} \odot \mathbf{x}\|_2 = 1.0$ to isolate the effect of $z = \mathbf{w}^\top \mathbf{x}$ and $\kappa$). As $\kappa$ increases, the transition of $\hat{\sigma}$ around the origin becomes progressively gentler compared to the sharp kink of the standard ReLU activation.

**Remark 4.4.** Theorem 4.2 shows that the expected loss can be approximated by the combination of terms $\mathcal{T}_1$ and $\mathcal{T}_2$, with an additive error term defined by $\mathcal{E}$. Notice that the smoothed activation $\hat{\sigma}_\kappa(\mathbf{w}, \mathbf{x})$ satisfies (see (19)):

$$\hat{\sigma}(\mathbf{w}, \mathbf{x}) = \mathbf{w}^\top \mathbf{x} \cdot \Phi_1\left(\frac{\mathbf{w}^\top \mathbf{x}}{\kappa \|\mathbf{w} \odot \mathbf{x}\| 2}\right) = \mathbb{E}_{\mathbf{c} \sim \mathcal{N}(\mathbf{1}, \kappa^2 \mathbf{I})}[\sigma(\mathbf{w}^\top(\mathbf{x} \odot \mathbf{c}))] \pm O\left(\kappa \|\mathbf{w} \odot \mathbf{x}\|_2 \phi\left(\frac{\mathbf{w}^\top \mathbf{x}}{\kappa \|\mathbf{w} \odot \mathbf{x}\|_2}\right)\right)$$

Therefore, here $\mathcal{T}_1$ can be seen as loss defined on the smoothed neural network $\hat{f}(\mathbf{W}, \cdot)$ with the same weights and dataset.

**Remark 4.5.** One may notice that the form of $\mathcal{T}_2$ is similar to the $\ell_2$ regularization in the ridge regression. To understand $\mathcal{T}_2$, we first notice that:

$$\nabla_{\mathbf{w}_r}\hat{f}(\mathbf{W}, \mathbf{x}_i) \approx \frac{1}{\sqrt{m}}\sum_{r=1}^m a_r \mathbf{x}_i \mathbf{\Phi}_1\left(\frac{\mathbf{w}_r^\top \mathbf{x}_i}{\kappa \|\mathbf{u}_{i,r}\|_2}\right).$$

Therefore, $\mathcal{T}_2$ can approximately be written as:

$$\mathcal{T}_2 \approx \frac{\kappa^2}{2}\sum_{i=1}^n \left\|\sum_{r=1}^m \nabla_{\mathbf{w}_r}\hat{f}(\mathbf{W}, \mathbf{x}_i) \odot \mathbf{w}_r\right\|_2^2 = \frac{\kappa^2}{2}\sum_{i=1}^n\sum_{j=1}^d\left(\nabla_{\hat{\mathbf{w}}_j}f(\mathbf{W}, \mathbf{x}_i)^\top \hat{\mathbf{w}}_j\right)^2 = \mathsf{vec}(\mathbf{W})^\top \hat{\mathbf{H}}\mathsf{vec}(\mathbf{W}).$$

Here $\hat{\mathbf{w}}_j$ is the $j$th row of the matrix $\mathbf{W} = [\mathbf{w}_1, \ldots, \mathbf{w}_m] \in \mathbb{R}^{d \times m}$, $\text{vec}(\mathbf{W}) = \text{concat}(\hat{\mathbf{w}}_1, \ldots, \hat{\mathbf{w}}_d)$ is the concatenation of the $\hat{\mathbf{w}}_j$'s, and $\hat{\mathbf{H}} \in \mathbb{R}^{md \times md}$ is the block-diagonal matrix whose $j$th diagonal block is $\hat{\mathbf{H}}_j := \sum_{i=1}^{n} \nabla_{\hat{\mathbf{w}}_j} f(\mathbf{W}, \mathbf{x}_i) \nabla_{\hat{\mathbf{w}}_j} f(\mathbf{W}, \mathbf{x}_i)^\top \in \mathbb{R}^{m \times m}$ for $j \in [d]$. Intuitively, $\hat{\mathbf{H}}$ can be seen as a matrix consisting of the tangent features' (Baratin et al., 2021; LeJeune & Alemohammad, 2024) outer products. As a result, $\mathcal{T}_2$ can be seen as the regularization term of $\mathbf{W}$ in terms of a norm defined by the tangent feature outer product matrix $\hat{\mathbf{H}}$.

**Remark 4.6.** The magnitude of $\mathcal{E}$ is given in (4). At a first glance, one could see that the term decreases monotonically as $\kappa$ decrease, implying a smaller error when $\kappa$ is small. As discussed in the beginning of this section, $\phi_{\max}$ and $\psi_{\max}$ are upper-bounded by some constant. Therefore, in the worst case, we have $|\mathcal{E}| \leq O\left(mn\left(\kappa^2 R_{\mathbf{u}} + \kappa R_{\mathbf{w}}\right)\right)$, which scales linearly with $\kappa$.

Below, we provide the intuition and sketch the proof of Theorem 4.2. The full proof of Theorem 4.2 is deferred to Appendix B.2.

*Proof sketch.* The analysis of both the expected loss (Theorem 4.2) and the expected gradient (Theorem 4.9) follows the same idea: writing $\mathbf{w}_r^\top(\mathbf{x} \odot \mathbf{c}) = \mathbf{c}^\top(\mathbf{w}_r \odot \mathbf{x})$ and using $\mathbf{c} \sim \mathcal{N}(\mathbf{1}, \kappa^2 \mathbf{I})$ turns the masked pre-activation into a univariate Gaussian with mean $\mathbf{w}_r^\top \mathbf{x}$ and variance $\kappa^2 \|\mathbf{w}_r \odot \mathbf{x}\|_2^2$. Each expectation then becomes an integral of a (possibly rectified) Gaussian, which we evaluate explicitly using integration by parts. This produces two pieces: a dominating term governed by the Gaussian CDF $\mathbf{\Phi}_1$—which gives the smoothed activation $\hat{\sigma}$ and the regularizer $\mathcal{T}_2$—and an exponentially decaying remainder, which we bound to obtain the stated error term.

Our proof starts with the decomposition of the expected loss as:

$$\mathbb{E}_{\mathbf{C}}\left[\mathcal{L}_{\mathbf{C}}(\mathbf{W})\right] = \frac{1}{2} \sum_{i=1}^{n} \mathbb{E}_{\mathbf{C}}\left[(f(\mathbf{W}, \mathbf{x}_i \odot \mathbf{c}_i))^2\right] + \frac{1}{2} \sum_{i=1}^{n} y_i^2 - \sum_{i=1}^{n} y_i \mathbb{E}_{\mathbf{C}}\left[(f(\mathbf{W}, \mathbf{x}_i \odot \mathbf{c}_i))\right].$$

It boils down to analyzing the terms $\mathbb{E}_{\mathbf{C}}\left[(f(\mathbf{W}, \mathbf{x}_i \odot \mathbf{c}_i))^2\right]$ and $\mathbb{E}_{\mathbf{C}}\left[(f(\mathbf{W}, \mathbf{x}_i \odot \mathbf{c}_i))\right]$. Plugging in $f(\mathbf{W}, \mathbf{x}_i \odot \mathbf{c}_i)$, it suffices to analyze the following expectations:

$$E_1 = \mathbb{E}_{\mathbf{c}}\left[\sigma\left(\mathbf{w}_r^\top(\mathbf{x} \odot \mathbf{c})\right) \sigma\left(\mathbf{w}_{r'}^\top(\mathbf{x} \odot \mathbf{c})\right)\right]; \quad E_2 = \mathbb{E}_{\mathbf{c}}\left[\sigma\left(\mathbf{w}_r^\top(\mathbf{x} \odot \mathbf{c})\right)\right].$$

The trick of evaluating $E_1$ and $E_2$ is to notice that $\mathbf{w}_r^\top(\mathbf{x} \odot \mathbf{c}) = \mathbf{c}^\top(\mathbf{w}_r \odot \mathbf{x}_i)$. Since $\mathbf{c} \sim \mathcal{N}(\mathbf{1}, \kappa^2 \mathbf{I})$, we must have that $\mathbf{c}^\top(\mathbf{w}_r \odot \mathbf{x}) \sim \mathcal{N}\left(\mathbf{w}_r^\top \mathbf{x}, \kappa^2 \|\mathbf{w}_r \odot \mathbf{x}\|_2^2\right)$. Therefore, we can define $z_1 = \mathbf{c}^\top(\mathbf{w}_r \odot \mathbf{x})$ and $z_2 = \mathbf{c}^\top(\mathbf{w}_{r'} \odot \mathbf{x})$. Then the problem of evaluating $E_1$ and $E_2$ becomes computing:

$$E_1 = \mathbb{E}_{z_1, z_2}\left[z_1 z_2 \mathbb{I}\{z_1 \geq 0; z_2 \geq 0\}\right]; \quad E_2 = \mathbb{E}_{z_1}\left[z_1 \mathbb{I}\{z_1 \geq 0\}\right].$$

Here $\text{Cov}(z_1, z_2) = (\mathbf{w}_r \odot \mathbf{x}_i)^\top(\mathbf{w}_{r'} \odot \mathbf{x}_i)$. To complete the proof, we prove the following two lemmas.

**Lemma 4.7.** *Let $z_1 \sim \mathcal{N}(\mu_1, \kappa_1^2)$. Then, we have that:*

$$\mathbb{E}\left[z_1 \mathbb{I}\{z_1 \geq 0\}\right] = \frac{\kappa}{\sqrt{2\pi}} \exp\left(-\frac{\mu^2}{2\kappa^2}\right) + \mu \mathbf{\Phi}_1\left(\frac{\mu}{\kappa}\right).$$

**Lemma 4.8.** *Let $z_1 \sim \mathcal{N}(\mu_1, \kappa_1^2)$ and $z_2 \sim \mathcal{N}(\mu_2, \kappa_2^2)$, with $\text{Cov}(z_1, z_2) = \kappa_1 \kappa_2 \rho$. Let $\mathbf{\Phi}_2(a, b, \rho)$ denote the joint CDF of standard Gaussian random variables $\hat{z}_1, \hat{z}_2$ with covariance $\rho$ at $z_1 = a, z_2 = b$. Then, we have:*

$$\mathbb{E}\left[z_1 z_2 \mathbb{I}\{z_1 \geq 0; z_2 \geq 0\}\right] = (\mu_1 \mu_2 + \kappa_1 \kappa_2 \rho) \mathbf{\Phi}_2\left(\frac{\mu_1}{\kappa_1}, \frac{\mu_2}{\kappa_2}, \rho\right) + \frac{1}{\sqrt{2\pi}}\left(\kappa_1 \mu_2 T_1 + \kappa_2 \mu_1 T_2\right)$$
$$+ \frac{\kappa_1 \kappa_2}{2\pi} \exp\left(-\frac{1}{2(1 - \rho^2)}\left(\frac{\mu_1^2}{\kappa_1^2} - \frac{2\rho\mu_1\mu_2}{\kappa_1 \kappa_2} + \frac{\mu_2^2}{\kappa_2^2}\right)\right)$$

*Here, $T_1, T_2$ are defined as:*

$$T_1 = \exp\left(-\frac{\mu_1^2}{2\kappa_1^2}\right) \mathbf{\Phi}_1\left(\frac{1}{\sqrt{1 - \rho^2}}\left(\frac{\mu_2}{\kappa_2} - \frac{\rho\mu_1}{\kappa_1}\right)\right); \quad T_2 = \exp\left(-\frac{\mu_2^2}{2\kappa_2^2}\right) \mathbf{\Phi}_1\left(\frac{1}{\sqrt{1 - \rho^2}}\left(\frac{\mu_1}{\kappa_1} - \frac{\rho\mu_2}{\kappa_2}\right)\right)$$

Plugging $z_1 = \mathbf{c}^\top (\mathbf{w}_r \odot \mathbf{x})$ and $z_2 = \mathbf{c}^\top (\mathbf{w}_{r'}^\top \mathbf{x})$ back into $\mathbb{E}_{\mathbf{C}} \left[ (f (\mathbf{W}, \mathbf{x}_i \odot \mathbf{c}_i))^2 \right]$ and $\mathbb{E}_{\mathbf{C}} [(f (\mathbf{W}, \mathbf{x}_i \odot \mathbf{c}_i))]$ and bounding the emerging error terms would give the desired result. Details are deferred into the appendix.

**Expected Surrogate Gradient.** In the following part, we study the expectation of the surrogate gradient.

**Theorem 4.9.** *Assume that $\mathbf{c}_i \sim \mathcal{N} \left( \mathbf{1}, \kappa^2 \mathbf{I} \right)$ for some $\kappa \leq 1$. Let $\phi_{\max}, \psi_{\max}, B_y, R_{\mathbf{u}}$, and $R_{\mathbf{w}}$ be defined in Definition 4.1. Then, we have that:*

$$\mathbb{E}_{\mathbf{C}} \left[ \nabla_{\mathbf{w}_r} \mathcal{L}_{\mathbf{C}} (\mathbf{W}) \right] = \nabla_{\mathbf{w}_r} \mathcal{L} (\mathbf{W}) + \mathbf{g}_r + \frac{3\kappa^2}{m} \sum_{r'=1}^m a_{r'} \mathrm{Diag} (\mathbf{x}_i)^2 \mathbf{w}_{r'} \cdot \mathbb{I} \left\{ \mathbf{w}_r^\top \mathbf{x}_i \geq 0; \mathbf{w}_{r'}^\top \mathbf{x}_i \geq 0 \right\} \tag{5}$$

*where the magnitude of $\mathbf{g}_r$ can be bounded as:*

$$\|\mathbf{g}_r\|_2 \leq \left( 6n\kappa^2 B_{\mathbf{x}}^2 R_{\mathbf{w}} + 5n\kappa R_{\mathbf{u}} \sqrt{d} \right) \phi_{\max} + \frac{\sigma_{\max} (\mathbf{X})}{\sqrt{m}} \phi_{\max} \mathcal{L} (\mathbf{W})^{\frac{1}{2}} + 6n\kappa R_{\mathbf{u}} \psi_{\max}. \tag{6}$$

**Remark 4.10.** Theorem 4.9 shows that the expected gradient can be written as the summation of the vanilla loss gradient $\nabla_{\mathbf{w}_r} \mathcal{L} (\mathbf{W})$, a term $\mathcal{T}_3$ above, and a gradient error $\mathbf{g}_r$. The magnitude of $\mathbf{g}_r$ is controlled in (6). As discussed previously, when $\left| \mathbf{w}_r^\top \mathbf{x}_i \right| > 0$, both $\phi_{\max}$ and $\psi_{\max}$ decreases exponentially as $\kappa$ decreases. Note that, although the second term scales with $\mathcal{L} (\mathbf{W})$, when $\mathcal{L} (\mathbf{W})$ decreases during training, that term will also contribute less to the overall gradient error.

**Remark 4.11.** One could observe that the third term on the right-hand side of (5) is the gradient of the function $\mathcal{R} (\mathbf{W})$ with respect to $\mathbf{w}_r$, where $\mathcal{R} (\mathbf{W})$ is given by:

$$\mathcal{R} (\mathbf{W}) = \frac{3\kappa^2}{m} \left\| \sum_{r'=1}^m a_{r'} \mathbf{w}_{r'} \odot \mathbf{x}_i \mathbb{I} \left\{ \mathbf{w}_{r'}^\top \mathbf{x}_i \geq 0 \right\} \right\|_2.$$

Therefore, $\mathcal{R} (\mathbf{W})$ can be seen as a scaled version of the $\mathcal{T}_2$-term in Theorem 4.2. This again verifies the regularization effect of the Gaussian random mask.

The proof of Theorem 4.9 is provided in Appendix B.3, and we provide the proof sketch below.

*Proof sketch.* By the form of the surrogate gradient in (2), we focus on the following two terms:

$$\mathcal{T}_1 = f (\mathbf{W}, \mathbf{x}_i \odot \mathbf{c}_i) (\mathbf{x}_i \odot \mathbf{c}_i) \mathbb{I} \left\{ \mathbf{w}_r^\top (\mathbf{x}_i \odot \mathbf{c}_i) \geq 0 \right\}; \mathcal{T}_2 = y_i (\mathbf{x}_i \odot \mathbf{c}_i) \mathbb{I} \left\{ \mathbf{w}_r^\top (\mathbf{x}_i \odot \mathbf{c}_i) \geq 0 \right\}.$$

Plug in the form of $f(\mathbf{W}, \mathbf{x}_i \odot \mathbf{c}_i)$, we can write $\mathcal{T}_1$ as:

$$\mathcal{T}_1 = \frac{1}{\sqrt{m}} \sum_{r'=1}^m a_{r'} \sigma(\mathbf{w}_{r'}^\top (\mathbf{x}_i \odot \mathbf{c}_i)) \cdot (\mathbf{x}_i \odot \mathbf{c}_i) \mathbb{I} \left\{ \mathbf{w}_r^\top (\mathbf{x}_i \odot \mathbf{c}_i) \geq 0 \right\}$$

$$= \frac{1}{\sqrt{m}} \sum_{r'=1}^m a_{r'} \mathrm{Diag} (\mathbf{x}_i) \mathbf{c}_i \mathbf{c}_i^\top \mathbb{I} \left\{ \mathbf{c}_i^\top (\mathbf{w}_r \odot \mathbf{x}_i) \geq 0 \right\} \cdot \mathbb{I} \left\{ \mathbf{c}_i^\top (\mathbf{w}_{r'} \odot \mathbf{x}_i \geq 0) \right\} (\mathbf{w}_{r'} \odot \mathbf{x}_i),$$

while $\mathcal{T}_2$ can be written as:

$$\mathcal{T}_2 = y_i \mathbf{x}_i \odot \left( \mathbf{c}_i \mathbb{I} \left\{ \mathbf{c}_i^\top (\mathbf{w}_r \odot \mathbf{x}_i) \geq 0 \right\} \right).$$

This allows us to focus on the following quantities instead

$$\mathbb{E} \left[ \mathbf{c}\mathbf{c}^\top \mathbb{I} \left\{ \mathbf{c}^\top \mathbf{u} \geq 0; \mathbf{c}^\top \mathbf{v} \geq 0 \right\} \right]; \mathbb{E} \left[ \mathbf{c}\mathbb{I} \left\{ \mathbf{c}^\top \mathbf{u} \geq 0 \right\} \right]$$

with multi-variate Gaussian random variable analysis. The rest of the proof then proceeds similarly as in the proof of Theorem 4.2.

## 5 Convergence Guarantee of Training with Gaussian Mask

Here, we study the convergence property of a general framework of stochastic neural network training, which gives us a theoretical result that can be of independent interest. Recall also the setting in Section 3: Consider

$f(\mathbf{W}, \cdot)$ as a two-layer ReLU activated MLP, as described above. Let $\boldsymbol{\xi}$ denote the randomness in one step of (stochastic) gradient descent. Let $\hat{\mathcal{L}}(\mathbf{W}, \boldsymbol{\xi})$ and $\nabla_{\mathbf{w}_r}\hat{\mathcal{L}}(\mathbf{W}, \boldsymbol{\xi})$ denote the stochastic loss and the stochastic gradient induced by $\xi$, respectively. We consider the sequence $\{\mathbf{W}_k\}_{k=1}^K$ generated by the following updates:

$$\mathbf{W}_{k+1} = \mathbf{W}_k - \eta \nabla_{\mathbf{W}}\hat{\mathcal{L}}(\mathbf{W}_k, \boldsymbol{\xi}_k). \tag{7}$$

Instead, the connection between $\nabla_{\mathbf{w}_r}\hat{\mathcal{L}}(\mathbf{W}, \boldsymbol{\xi})$ and $\hat{\mathcal{L}}(\mathbf{W}, \boldsymbol{\xi})$ with respect to $\mathbf{w}_r$, along with other requirements, are stated in the assumption below.

**Assumption 5.1.** *For all $\boldsymbol{\xi}, \mathbf{W}$, we assume that the following properties hold:*

$$\mathbb{E}_{\boldsymbol{\xi}}\left[\hat{\mathcal{L}}(\mathbf{W}, \boldsymbol{\xi})\right] \leq 2\mathcal{L}(\mathbf{W}) + \varepsilon_1, \tag{8}$$

$$\left\|\mathbb{E}_{\boldsymbol{\xi}}\left[\nabla_{\mathbf{w}_r}\hat{\mathcal{L}}(\mathbf{W}, \boldsymbol{\xi})\right] - \nabla_{\mathbf{w}_r}\mathcal{L}(\mathbf{W})\right\|_2 \leq \varepsilon_3 \mathcal{L}(\mathbf{W})^{\frac{1}{2}} + \varepsilon_2, \tag{9}$$

$$\left\|\nabla_{\mathbf{w}_r}\hat{\mathcal{L}}(\mathbf{W}, \boldsymbol{\xi})\right\|_2^2 \leq \gamma \hat{\mathcal{L}}(\mathbf{W}, \boldsymbol{\xi}). \tag{10}$$

Here, (8) and (9) provide an upper bound on the expected loss and the error of the expected gradient. (10) can be seen as a relaxed form of the smoothness. Our analysis is based on the standard NTK-type argument as in (Du et al., 2018; Song & Yang, 2020; Liao & Kyrillidis, 2022), which considers the infinite-width NTK $\mathbf{H}^\infty$ given by:

$$\mathbf{H}_{ij}^\infty = \mathbf{x}_i^\top \mathbf{x}_j \mathbb{E}_{\mathbf{w} \sim \mathcal{N}(\mathbf{0}, \mathbf{I})}\left[\mathbb{I}\left\{\mathbf{w}^\top \mathbf{x}_i \geq 0; \mathbf{w}^\top \mathbf{x}_j \geq 0\right\}\right]. \tag{11}$$

It is shown in Du et al. (2018) that $\mathbf{H}^\infty$ is positive definite. We define $\lambda_0 := \lambda_{\min}(\mathbf{H}^\infty) > 0$.

**Theorem 5.2.** *Assume that the first-layer weights of a neural network are initialized according to $\mathbf{w}_{0,r} \sim \mathcal{N}(\mathbf{0}, \tau^2 \mathbf{I})$ for some $\tau > 0$, and the second-layer weights are initialized according to $\mathbf{a}_r \sim \mathrm{Unif}\{\pm 1\}$. Let the number of hidden neurons satisfy $m = \Omega\left(\frac{n^4 K^2}{\lambda_0^4 \delta^2 \tau^2}\right)$ and the step size satisfy $\eta = O\left(\frac{\lambda_0}{n^2}\right)$. Assume that Assumptions 3.1, 5.1 hold for some $\gamma = C_1 \cdot \frac{n}{m}$ with some small enough $\varepsilon_1, \varepsilon_2, \varepsilon_3$ satisfying:*

$$\varepsilon_1 \leq O\left(\frac{\delta m}{nK^4}\right), \quad \varepsilon_2 \leq O\left(\frac{\delta \lambda_0}{nK^2}\right), \quad \varepsilon_3 \leq O\left(\frac{\lambda_0}{\sqrt{mn}}\right). \tag{12}$$

*Then, with probability at least $1 - 2\delta - n^2 \exp\left(-\frac{n^3}{\delta^2 \tau^2 \lambda_0^3}\right)$, for all $k \in [K]$, the sequence $\{\mathbf{W}_k\}_{k=1}^K$ generated by (7) satisfies:*

$$\mathbb{E}_{\boldsymbol{\xi}_0, \ldots, \boldsymbol{\xi}_{k-1}}\left[\mathcal{L}(\mathbf{W}_k)\right] \leq \left(1 - \frac{\eta \lambda_0}{2}\right)^k \mathcal{L}(\mathbf{W}_0) + O\left(\frac{mn}{\lambda_0^2} \cdot \varepsilon_2^2 + \varepsilon_1\right). \tag{13}$$

*Furthermore, we can guarantee that $\|\mathbf{w}_{k,r} - \mathbf{w}_{0,r}\|_2 \leq O\left(\frac{\tau \lambda_0}{n}\right)$ for all $r \in [m]$ and $k \in [K]$.*

In short, Theorem 5.2 shows that under a small enough $\varepsilon_1, \varepsilon_2, \varepsilon_3$ and $\gamma$, if the neural network is sufficiently overparameterized, then, with a small enough step size $\eta$, we can guarantee the convergence under the training given by (7) and that the change in each $\mathbf{w}_r$ is bounded by $O\left(\frac{\tau \lambda_0}{n}\right)$. As shown in (13), the expected loss converges linearly up to a ball around the global minimum with radius given by $O\left(\frac{mn}{\lambda_0^2} \cdot \varepsilon_2^2 + \varepsilon_1\right)$. This error region monotonically decreases as the error in the expected loss and gradient, namely $\varepsilon_1$ and $\varepsilon_2$, decreases.

*Proof sketch.* The argument follows the standard NTK trajectory analysis (Du et al., 2018; Song & Yang, 2020). The weights stay in a ball around initialization, so the neurons split into a large "unchanging" set, whose activation pattern is fixed, and a small "changing" set; this keeps the finite-width kernel close to $\mathbf{H}^\infty$ with $\lambda_{\min}(\mathbf{H}_k) \geq \lambda_0/2$, which produces the geometric factor $(1 - \eta\lambda_0/2)^k$. The novelty relative to deterministic or unbiased SGD analyses is that the stochastic gradient here is *biased*; the three conditions in Assumption 5.1—bounding the expected loss, the gradient bias, and a relaxed smoothness—are injected at the appropriate steps to absorb this stochasticity, and the additive error region in (13) is introduced by this bias.

**Training Convergence with Gaussian Input Mask.** We apply the general result in Theorem 5.2 to the scenarios of Gaussian input masking, as given by (1). To apply Theorem 5.2, one need to make sure that the requirements in Assumption 5.1 are guaranteed. Here, we present two corollaries as extensions of Theorem 4.2 and Theorem 4.9, with the goal of showing (8) and (9).

**Corollary 5.3.** *Let $B_y, \phi_{\max}, \psi_{\max}, R_{\mathbf{u}}$, and $R_{\mathbf{w}}$ be defined in Definition 4.1. If $B_y \leq 3\sqrt{m} R_{\mathbf{w}}$, then we have*

$$\mathbb{E}_{\mathbf{C}}\left[\mathcal{L}_{\mathbf{C}}\left(\mathbf{W}\right)\right] \leq 2\mathcal{L}\left(\mathbf{W}\right) + 2mn\kappa^2 R_{\mathbf{u}}^2 + mn\left(\kappa^2 R_{\mathbf{u}}^2 + \kappa R_{\mathbf{w}}\right)\phi_{\max}^2 + mn\kappa^2\left(R_{\mathbf{u}}^2 + 1\right)\psi_{\max}^2$$

Corollary 5.3 follows simply from Theorem 4.2 by upper-bounding the difference between the smoothed neural network function $\hat{f}(\mathbf{W}, \cdot)$ and the vanilla neural network function $f(\mathbf{W}, \cdot)$, and by upper-bounding the regularization term. In particular, Corollary 5.3 implies the bound of the error $\varepsilon_1$ as $2mn\kappa^2 R_{\mathbf{u}} + mn\left(\kappa^2 R_{\mathbf{u}}^2 + \kappa R_{\mathbf{w}}\right)\phi_{\max}^2 + mn\kappa^2\left(R_{\mathbf{u}}^2 + 1\right)\psi_{\max}^2$.

**Corollary 5.4.** *Let $B_y, \phi_{\max}, \psi_{\max}, R_{\mathbf{u}}$, and $R_{\mathbf{w}}$ be defined in Definition 4.1. If $B_y \leq 3\sqrt{m} R_{\mathbf{w}}$, then we have*

$$\left\|\mathbb{E}_{\mathbf{C}}\left[\nabla_{\mathbf{w}_r}\mathcal{L}_C\left(\mathbf{W}\right)\right] - \nabla_{\mathbf{w}_r}\mathcal{L}\left(\mathbf{W}\right)\right\|_2 \leq O\left(\left(n\kappa^2 B_{\mathbf{x}}^2 R_{\mathbf{w}} + n\kappa R_{\mathbf{u}}\sqrt{d}\right)\phi_{\max}\right) + O\left(\frac{\sigma_{\max}\left(\mathbf{X}\right)\phi_{\max}}{\sqrt{m}}\right)\mathcal{L}\left(\mathbf{W}\right)^{\frac{1}{2}}$$
$$+ O\left(n\kappa R_{\mathbf{u}}\psi_{\max} + \kappa^2\sqrt{m}B_{\mathbf{x}}^2 R_{\mathbf{w}}\right)$$

Similar to Corollary 5.3, Corollary 5.4 follows from upper bounding the regularization term in Theorem 4.9. By Corollary 5.4, we can write $\varepsilon_2$ and $\varepsilon_3$ in Assumption 5.1 as $\varepsilon_2 = O\left(\left(n\kappa^2 B_{\mathbf{x}}^2 R_{\mathbf{w}} + n\kappa R_{\mathbf{u}}\sqrt{d}\right)\phi_{\max}\right) + O\left(n\kappa R_{\mathbf{u}}\psi_{\max} + \kappa^2\sqrt{m}B_{\mathbf{x}}^2 R_{\mathbf{w}}\right)$ and likely $\varepsilon_3 = O\left(\frac{\sigma_{\max}(\mathbf{X})\phi_{\max}}{\sqrt{m}}\right)$. The proof of Corollary 5.3 and Corollary 5.4 are deferred to Appendix C.2. To complete the requirements in Assumption 5.1, we can show the following lemma for (10).

**Lemma 5.5.** *Assume that Assumption 3.1 holds. Then, we have:*

$$\left\|\nabla_{\mathbf{w}_r}\mathcal{L}_{\mathbf{C}}\left(\mathbf{W}\right)\right\|_2^2 \leq C_1\frac{n}{m}\mathcal{L}_{\mathbf{C}}\left(\mathbf{W}\right).$$

The proof can be found in the appendix D.21

With the help of Corollary 5.3,5.4, and Lemma D.21, we can derive the convergence guarantee of training the two-layer ReLU neural network under Gaussian input mask.

**Theorem 5.6.** *Assume that the first-layer weights are initialized according to $\mathbf{w}_{0,r} \sim \mathcal{N}\left(\mathbf{0}, \tau^2\mathbf{I}\right)$ for some $\tau > 0$, and the second-layer weights are initialized according to $\mathbf{a}_r \sim \mathrm{Unif}\{\pm 1\}$. Let the number of hidden neurons satisfy $m = \Omega\left(\frac{n^4 K^2}{\lambda_0^4 \delta^2 \tau^2}\right)$ and the step size satisfy $\eta = O\left(\frac{\lambda_0}{n^2}\right)$. Assume that for all $\mathbf{W} \in \{\mathbf{W}_k\}_{k=1}^K$, the following hold:*

$$\kappa = O\left(\frac{\sqrt{\delta\lambda_0}}{\tau^2 K^2\left(m^{\frac{1}{4}}\sqrt{d} + nd\right)\left(\hat{\phi}_{\max} + \hat{\psi}_{\max}\right)}\right) \tag{14}$$

$$\sigma_{\max}\left(\mathbf{X}\right)\hat{\phi}_{\max} \leq O\left(\frac{\lambda_0}{\sqrt{n}}\right). \tag{15}$$

*Then, we have that, with probability at least $1 - 2\delta - n^2\exp\left(-\frac{n^3}{\delta^2\tau^2\lambda_0^3}\right)$, for all $k \in [K]$, the sequence $\{\mathbf{W}_k\}_{k=1}^K$ generated by (7) satisfies:*

$$\mathbb{E}_{\mathbf{C}_0,\ldots,\mathbf{C}_{k-1}}\left[\mathcal{L}\left(\mathbf{W}_k\right)\right] \leq \left(1 - \frac{\eta\lambda_0}{2}\right)^k\mathcal{L}\left(\mathbf{W}_0\right) + O\left(\kappa\tau^2 mn^2 d^2\left(\hat{\phi}_{\max}^2 + \hat{\psi}_{\max}^2\right)\right) \tag{16}$$

$$+ O\left(\kappa^2\tau^2 m^2 nd\right) + O\left(\kappa\tau mn\sqrt{d}\hat{\phi}_{\max}^2\right) \tag{17}$$

*where $\hat{\phi}_{\max} = \max_{k\in[K]}\phi_{\max}\left(\mathbf{W}_k\right)$ and $\hat{\psi}_{\max} = \max_{k\in[K]}\psi_{\max}\left(\mathbf{W}_k\right)$.*

Theorem 5.6 is an instantiation of Theorem 5.2, in which the three conditions of Assumption 5.1 are verified for the Gaussian mask: Corollary 5.3 supplies $\varepsilon_1$, Corollary 5.4 supplies $\varepsilon_2$ and $\varepsilon_3$ (both from the Section 4 analysis), and Lemma D.21 supplies the relaxed-smoothness constant $\gamma$. The $\kappa$-dependence of the final error floor in (16) is then inherited directly from the $\kappa$-dependence of these three error terms.

In short, Theorem 5.6 guarantees the convergence of training a two-layer ReLU neural network under Gaussian input mask in the form of (16) under the condition of sufficient overparameterization, proper step size, and the requirement in (14) and (15). In particular, (14) requires a sufficiently small Gaussian variance $\kappa$. The condition in (15) requires either a small maximum singular value of the input data matrix $\mathbf{X}$, or a small $\phi_{\max}$. Lastly, (16) shows a linear convergence of the expected loss up to some error region. Notice that the first part of the error region depends both on $\kappa, \tau$ and on $\phi_{\max}$ and $\psi_{\max}$, and the second part of the error region depends solely on $\kappa$ and $\tau$. This means that one can guarantee an arbitrarily small error region when the Gaussian noise $\kappa$ and the initialization scale $\tau$ is sufficiently small.

**Remark 5.7.** Both the requirement and the error region in Theorem 5.6 depend on the quantity $\hat{\phi}_{\max}$ and $\hat{\psi}_{\max}$. Recall that:

$$\hat{\phi}_{\max} = \max_{k,r,i} \left\{ \exp\left( -\frac{\left(\mathbf{w}_r^\top \mathbf{x}_i\right)}{4\kappa^2 \left\| \mathbf{w}_r \odot \mathbf{x}_i \right\|_2^2} \right) \right\}; \quad \hat{\psi}_{\max} = \max_{k,r,i} \left\{ \left| \mathbf{w}_r^\top \mathbf{x}_i \right| \cdot \exp\left( -\frac{\left(\mathbf{w}_r^\top \mathbf{x}_i\right)}{4\kappa^2 \left\| \mathbf{w}_r \odot \mathbf{x}_i \right\|_2^2} \right) \right\}.$$

Both quantities decay exponentially fast as $\kappa$ decays, when $\mathbf{w}_r^\top \mathbf{x}_i \neq 0$ for all $k, r, i$. As the sequence $\{\mathbf{W}_k\}_{k=1}^K$ is generated under the randomness of $\mathbf{C}_k$'s, intuitively it is almost never the case where $\mathbf{w}_{k,r}^\top \mathbf{x}_i = 0$. Therefore, in most cases Theorem 5.6 should require only a log-dependency of $\kappa$ on other parameters in order for (14) and (15) to be satisfied. However, it should be noticed that $\kappa$ still need to decay in powerlaw if one want to sufficiently decrease the second part of the error region.

## 6 Experiments

### 6.1 Empirical Validation of Training Convergence with Gaussian Mask.

Theorem 5.6 asserts that training a sufficiently overparameterized two-layer ReLU network with Gaussian multiplicative input noise results in linear convergence of the expected loss to an error ball. The radius of this error ball is proportional to the noise variance (controlled by $\kappa$) and other network and data-dependent terms. We empirically verify this convergence behavior.

*Simulation Setup.* We train a two-layer ReLU MLP: As a toy example, the network has $d = 20$ input features and $m = 100$ hidden units. The training dataset comprised $n = 500$ synthetic samples, with input features $\mathbf{x}_i$ normalized such that $\|\mathbf{x}_i\|_2 \leq 1$, and target values $y_i$ generated from a non-linear function of $\mathbf{x}_i$ with small added noise. First-layer weights $\mathbf{W}$ were initialized using Kaiming uniform initialization, and second-layer weights $a_r \in \{\pm 1\}$ were fixed. The network was trained for 2000 iterations using full-batch gradient descent with a learning rate of 0.005. We performed separate training runs for different noise levels: $\kappa \in \{0.0, 0.05, 0.2, 0.4, 0.6, 1.0, 2.0\}$. For each run, we tracked the evolution of the clean training loss $\mathcal{L}(\mathbf{W}_k)$.

*Results and Discussion.* The training trajectories, plotted in Figure 3a, illustrate the theoretical predictions. For clean training ($\kappa = 0.0$), the loss exhibits an initial linear convergence phase. When multiplicative Gaussian noise is introduced, the initial linear convergence trend is preserved. However, as training progresses, the loss converges not to the same minimal value but to a distinct error ball, plateauing at a value higher than the clean case. As expected, the size of this error ball, indicated by the final converged loss value, systematically increases with the noise level $\kappa$. This direct relationship between $\kappa$ and the size of the error ball provides strong empirical support for the convergence guarantees outlined in Theorem 5.6.

### 6.2 Impact of Multiplicative Gaussian Noise on Model Accuracy

We trained a 1-hidden-layer MLP (4096 hidden units, GELU activation, dropout=0.2) on CIFAR-10 for 80 epochs using AdamW optimization with cosine annealing, label smoothing, and standard data augmentation. During training, we injected multiplicative Gaussian noise $x \leftarrow x \cdot (1 + \kappa Z)$ where $Z \sim \mathcal{N}(0, 1)$. and

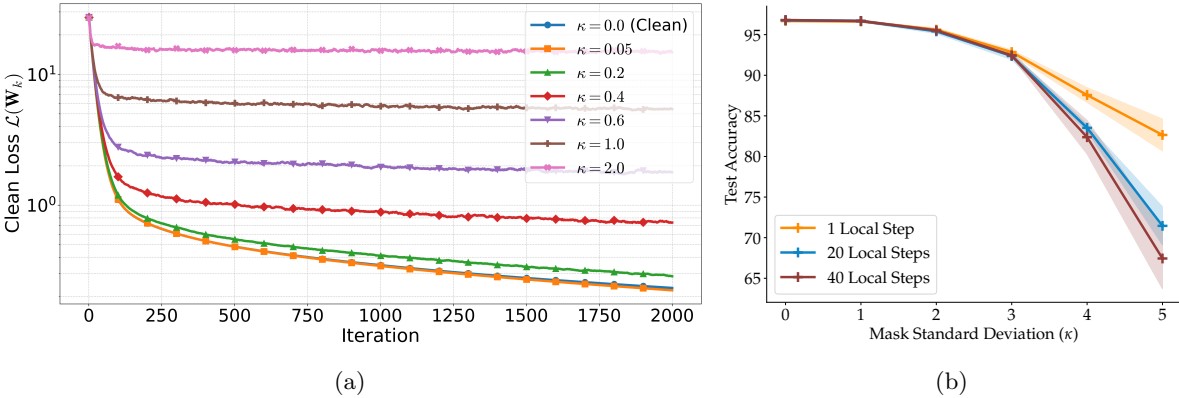

Figure 3: (a). Training loss $\mathcal{L}(\mathbf{W}_k)$ (log-scale) versus training iteration for a two-layer ReLU network ($n = 500, d = 20, m = 100$) trained with full-batch gradient descent under different levels of input multiplicative Gaussian noise standard deviation $\kappa$. (b). Distributed training with Gaussian mask for different $\kappa$ and number of local steps.

evaluated the impact of noise strength $\kappa$ on clean test accuracy. The results reveal that small amounts of MG noise $\kappa \approx 0.2$ improve generalization, acting as an effective regularizer beyond the existing random cropping and flipping. This suggests that modest input perturbations help the model learn more robust features that transfer better to the test set. However, accuracy degrades monotonically beyond this point, dropping to 49.88% at $\kappa = 1.8$. Next, to assess the impact of multiplicative Gaussian noise on a well-regularized

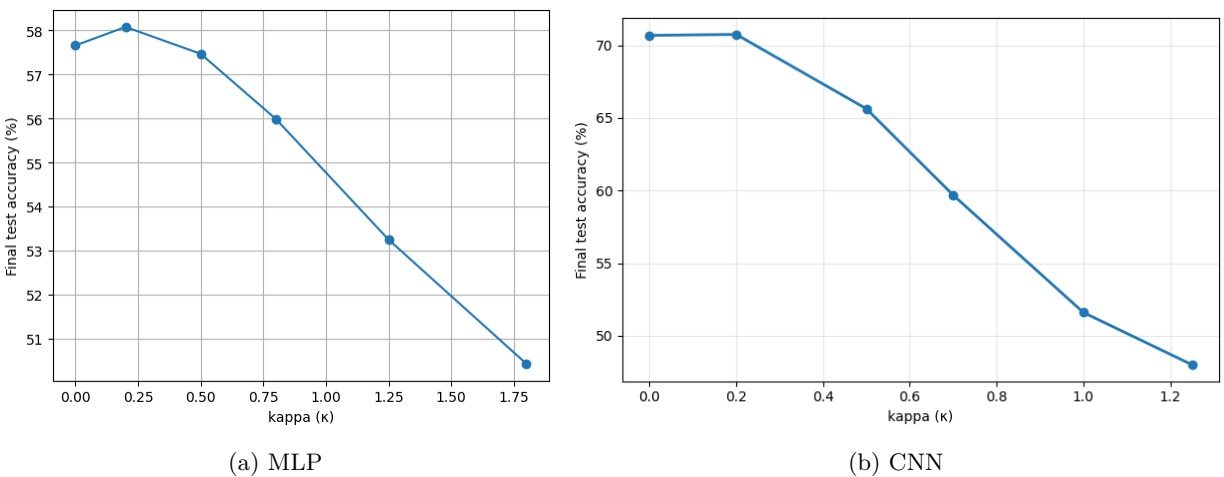

Figure 4: Test accuracy versus multiplicative Gaussian noise strength $\kappa$ for (a) a 1-hidden-layer MLP and (b) a CNN, trained on CIFAR-10. Small noise levels ($\kappa \approx 0.2$) can improve generalization for the MLP, likely due to regularization effects. In contrast, the CNN exhibits robustness by maintaining baseline accuracy. Beyond this point, accuracy degrades monotonically for both architectures as noise corrupts the training signal.

convolutional architecture, we trained a CNN on CIFAR-10 with varying noise strengths $\kappa$. The architecture consists of 4 convolutional layers ($32 \to 32 \to 64 \to 64$ filters with $3 \times 3$ kernels), batch normalization after each convolutional layer, three dropout layers with rates $0.2, 0.3$ and $0.5$ respectively, and a fully connected layer with ReLU activation. We trained the CNN for 80 epochs using the Adam optimizer with $lr = 10^{-3}$, weight decay$= 10^{-5}$ and batch size$= 128$. During training, we applied Multiplicative Gaussian (MG) noise $x \leftarrow x \cdot (1 + \kappa Z)$ where $Z \sim \mathcal{N}(0, 1)$, while all accuracy measurements were performed on the clean test set (noiseless). As shown in Figure 4b, we observe that the model exhibits robustness to low-magnitude MG

noise, maintaining its baseline accuracy ($\approx 71\%$) at $\kappa = 0.2$. Beyond that point though, accuracy degrades monotonically and excessive noise corrupts the training signal.

## 6.3 Application: Distributed Training over Wireless Channels

Communicating signals over wireless channels incurs fading phenomena to the signals (Tse & Viswanath, 2005). Specifically, a time-varying signal $x(t)$ transmitted over channel given by $h(t)$ and additive noise $z(t)$ result in $y(t) = x(t)h(t) + z(t)$. For data parallel distributed training over wireless channels, each input data is passed through the channel to the workers to perform local training. We consider using the Gaussian masked input training scheme studied in this paper as a simplified setup to model the channel fading in wireless communication. In particular, we let $\mathbf{x}$ be the signals transmitted $x(t)$, and $\mathbf{c} \sim \mathcal{N}\left(\mathbf{1}, \kappa^2 \mathbf{I}_d\right)$ be the channel effect $h(t)$. For simplicity, we set the additive noise to 0. The time-depending behavior of $x(t)$ and $h(t)$ are transformed into the masking scheme that in each iteration, a new mask is applied to the sample.

Under this setup, we train a two-layer MLP with 128 hidden neurons for the MNIST dataset using FedAvg. That is, we assume that the total training process is partitioned into multiple global iterations, where in each global iteration, the central server passes the updated model parameter together with the current copy of training data through a wireless channel. Each worker receives the training data with channel fading (in our case, modeled with Gaussian multiplicative noise), and updated its local copy of the model using gradient descent starting from the parameter shared by the server for some number of local steps. After the local update, the workers sends the updated parameter to the central server to perform an aggregation by averaging the worker's weights. Under this setup, we train an aggregated model with 5 workers and the choice of {1, 20, 40} local steps with a batch size of 128. We also vary the variance of the Gaussian mask to study the relationship between the number of local steps and the noise scales. For each combination of local steps and Gaussian variance, we run 5 trials and record the mean and standard deviation of the resulting accuracy.

We plot the result in Figure 3b. In general, for all choices of the number of local steps, we observe a decay in the test accuracy as the input masking variance grows larger, indicating the negative influence of the noise to the overall training performance. In particular, we can also observe that, in the low noise regime ($\kappa \approx 0$), the resulting final accuracy is not influenced much by the number of local iterations. However, as the noise scale grows larger (larger $\kappa$), the final accuracy decays drastically as we increase the number of local iterations. We hypothesis that this behavior is due to the fact more local iterations allows the workers to fit more to the noise instead of the original signals in the data.

## 6.4 Membership Inference Attacks: Multiplicative Gaussian Noise as Deliberate Injection

In this section, we empirically evaluate the effectiveness of training with input multiplicative Gaussian (MG) noise as a defense against Membership Inference Attacks (MIAs). In simple words, the primary goal of a MIA is to determine if a specific data point $\mathbf{x}$ was part of the training set of a target model $f$.

**Threat Model and Attack Methodology.** We adopt the black-box shadow model attack methodology (Adversary 1) from the ML-Leaks framework (Salem et al., 2019). In this setup, the adversary aims to determine whether a specific data point was part of a target model's training set using only the model's output posteriors. Because the adversary lacks the target's training labels, they employ a shadow model trained on a proxy dataset to mimic the target's behavior. By observing how the shadow model treats its own members versus non-members, the adversary generates labeled data to train an attack model. This binary classifier learns to identify the statistical "signatures" of membership—such as increased confidence or reduced entropy—enabling it to perform membership inference on the original target model. A detailed breakdown of the data partitioning and the five-stage attack pipeline is provided in Appendix A.2.

**Experimental Setup.** We evaluate the effectiveness of Multiplicative Gaussian (MG) noise as a privacy defense using the CIFAR-10 dataset. The data is partitioned into four disjoint sets to train and audit both target and shadow models independently. Our evaluation covers two architectures—a fully-connected MLP and a multi-block CNN—to ensure the defense generalizes across different model complexities. We measure privacy leakage by training a Logistic Regression attack model against target models subjected to varying noise intensities ($\kappa \in \{0.0, 0.5, 1.2, 1.8\}$) and training durations (20–120 epochs). The defense is quantified

via Precision, Recall, and AUC, where an AUC of 0.5 indicates perfect privacy (random guessing). Detailed hyperparameters, partitioning sizes, and architectural specifications are provided in Appendix A.2.

### 6.4.1 Results and Discussion

Our experiments confirm that, in this experimental setup, training with multiplicative Gaussian noise systematically reduces attack success. The results for the MLP and CNN model are presented in Figure 6 and 7 respectively, with the AUC values of our experiments illustrated in Figure 5.

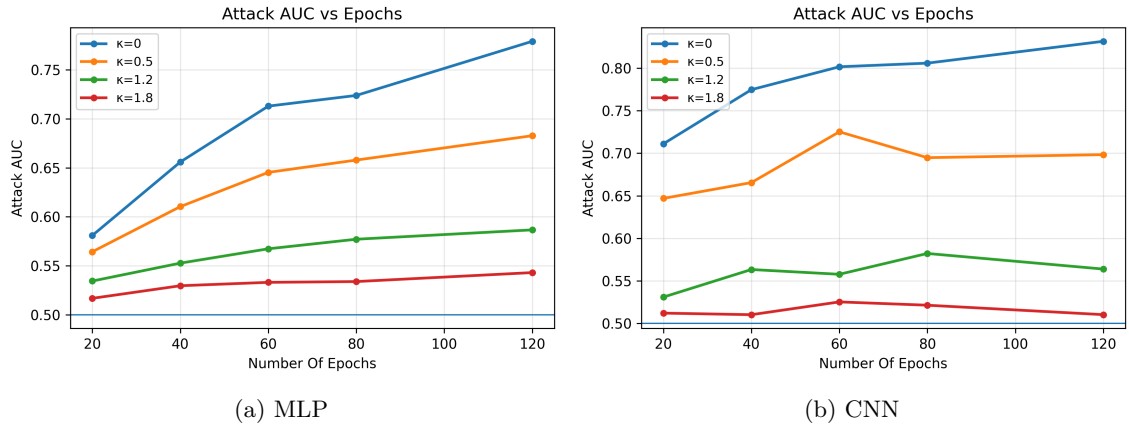

(a) MLP  (b) CNN

Figure 5: Attack AUC on the target model when it is (a) an MLP and (b) a CNN. Higher values indicate greater privacy leakage. Training with MG noise (larger $\kappa$) consistently reduces attack success.

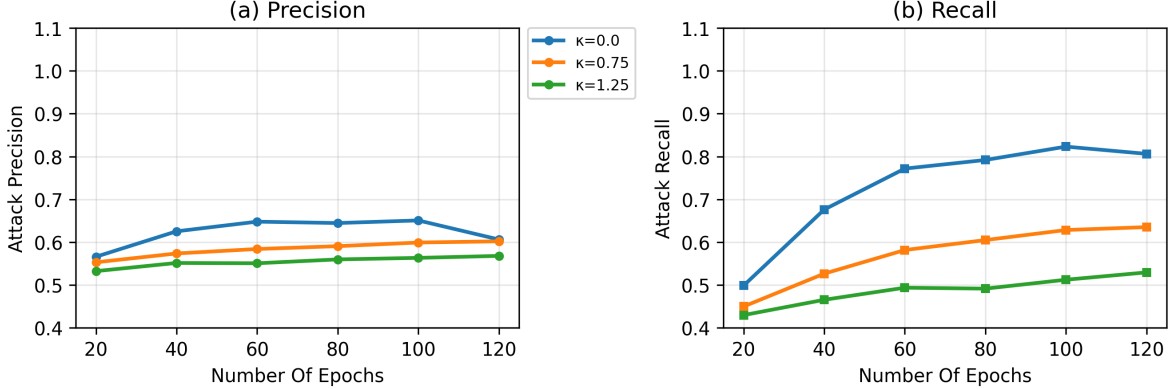

Figure 6: MLP target. Multiplicative Gaussian noise provides resilience against the attacks as $\kappa$ increases.

Figure 7 demonstrates the attack success for $\kappa = 0.0$, in which case as the number of training epochs increases from 20 to 120, AUC rises from 0.578 to a significant 0.782. This validates that our attack implementation correctly captures privacy leakage.

The central observation is that applying MG noise (increasing $\kappa$) reduces attack success on the tested architectures. For example, after 120 epochs of training, the standard model is highly vulnerable (Attack AUC = 0.782). In contrast, the model trained with $\kappa = 0.5$ reduces this leakage (AUC = 0.692), and models with stronger noise achieve even better privacy (AUC = 0.585 for $\kappa = 1.2$ and AUC = 0.543 for $\kappa = 1.8$). In our experiments, increasing the noise variance $\kappa$ leads to lower attack success against this MIA. Similar trends were observed for the CNN architecture (see Figure 7). Figure 4 shows the corresponding utility cost: by sacrificing some test accuracy, training with multiplicative Gaussian noise reduces the statistical signature that the attack exploits in this setup. We do not claim a formal privacy guarantee, nor that MG noise is a

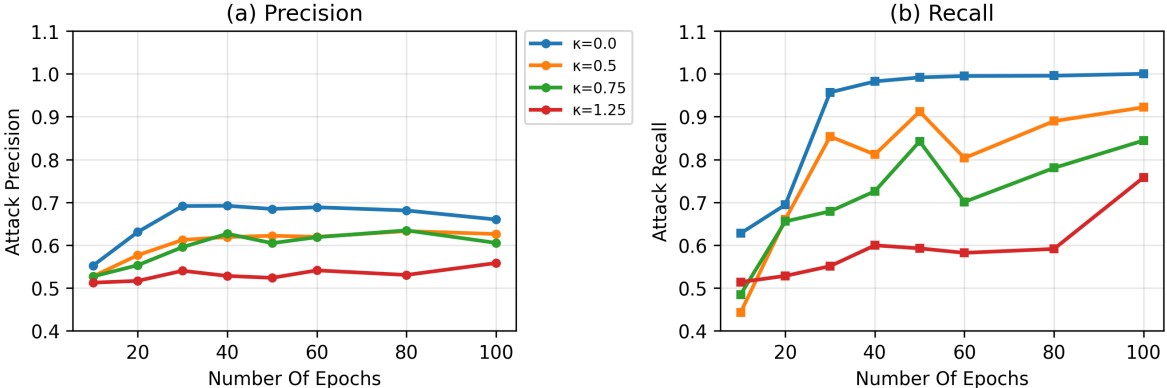

Figure 7: CNN target. We observe a similar trend as the MLP

substitute for differential privacy; we report these results only as evidence that the analyzed noise model has a measurable effect on membership-inference vulnerability.

## 7 Conclusion

This work investigates the fundamental question of how independent multiplicative Gaussian masking affects training dynamics. Focusing on a two-layer ReLU network in the NTK regime, we demonstrate that the masked objective admits a closed-form decomposition into a smoothed square loss plus an explicit, data-dependent regularizer. This structure allows training with the gradient from the masked objective to achieve linear convergence toward a noise-controlled error ball given small step size and large enough over-parameterization. Beyond our theory, we provided experimental result to validate the convergence to small ball, and presented applications in distributed training under channel fading and how the masking can defend against attacks.

Though our work is theoretical by nature, the results could inform practitioners in the following ways:

- **A closed-form for how multiplicative input noise smooths ReLU.** Our paper derives expected masked activation (up to a small correction term), which can be seen as a smoothed ReLU or a more general form of the popular GELU activation, whose smoothness is controlled by the noise scale $\kappa$. This result offers a way to conceptualize what input-level multiplicative noise does to a ReLU unit, and provides an intuitive connection between Gaussian drop-out and GELU activation.
- **Multiplicative masking induces a data-dependent regularizer.** The $\mathcal{T}_2$ term in Theorem 4.2 has the form of a regularization that depends on the tangent-feature outer-product matrix (as discussed in Remark 4.5). Qualitatively, this is a penalty shaped by the data and the current weights, unlike weight decay's isotropic Frobenius norm penalty.
- **Convergence reaches a noise-dependent floor.** Theorem 5.6 gives a linear convergence up to an error term where the error is a sum of $\kappa$-dependent terms. The exact constants are tied to the NTK regime, but the qualitative shape—a geometric decrease followed by a plateau whose height increases with $\kappa$—is observable beyond that regime. Figure 3(a) shows it on the synthetic regression task the theory describes.

**Limitations and Future Work.** Our theoretical analysis is currently constrained to the small-noise regime and the extreme over-parameterization typical of NTK models. Furthermore, our proofs rely on the independence of masks across iterations and do not yet incorporate a formal privacy accounting pipeline, such as subsampling or composition. Despite these constraints, multiplicative Gaussian masking represents a provable and practically viable method for injecting input-level uncertainty. These results provide a principled foundation for future exploration of deep networks and more complex noise settings in feature-wise training.

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

## A    Additional Experimental Results and Related Details.

### A.1    Empirical Validation of Expected Gradient Properties (Theorem 4.9).

Theorem 4.9 characterizes the expected gradient under Gaussian input masking as $\mathbb{E}_{\mathbf{C}}\left[\nabla_{\mathbf{w}_r}\mathcal{L}_{\mathbf{C}}\left(\mathbf{W}\right)\right] = \nabla_{\mathbf{w}_r}\mathcal{L}\left(\mathbf{W}\right) + \mathcal{T}_{3,r} + \mathbf{g}_r$. Here, $\nabla_{\mathbf{w}_r}\mathcal{L}\left(\mathbf{W}\right)$ is the clean input gradient, $\mathcal{T}_{3,r}$ is a systematic deviation term proportional to $\kappa^2$, and $\mathbf{g}_r$ is a residual error bounded by Eq. (6).

*Simulation Setup.* We used a two-layer ReLU MLP with $d = 20$ input features, $m = 100$ hidden units, on $n = 500$ synthetic samples ($\|\mathbf{x}_i\|_2 \leq 1$, $y_i \sim \mathcal{N}(0, 0.5^2)$). First-layer weights $\mathbf{W}$ are from $\mathcal{N}(0, 0.1^2)$; second-layer $a_r \in \{\pm 1\}$ are fixed. We analyze $\nabla_{\mathbf{w}_r}\mathcal{L}_{\mathbf{C}}(\mathbf{W})$ by averaging $N = 2000$ Monte Carlo samples for $\kappa \in [0.001, 1.0]$, for a representative neuron $r$. For this setup, the clean loss $\mathcal{L}(\mathbf{W}) \approx 71.52$ and $\|\nabla_{\mathbf{w}_r}\mathcal{L}(\mathbf{W})\|_2 \approx 0.981$.

*Results and Discussion.* Our simulations validate the decomposition in Theorem 4.9. Figure 8a displays the $\ell_2$-norms of the gradient components versus $\kappa$. The clean gradient norm is constant. The $\mathcal{T}_{3,r}$'s norm, $\|\mathcal{T}_{3,r}\|_2$, scales with $\kappa$ (e.g., from $\approx 8.1 \times 10^{-7}$ at $\kappa = 0.001$ to $\approx 0.81$ at $\kappa = 1.0$), confirming its theoretical dependence. The norm of the empirically estimated expected masked gradient, $\|\mathbb{E}_{\mathbf{C}}[\nabla_{\mathbf{w}_r}\mathcal{L}_{\mathbf{C}}(\mathbf{W})]\|_2$, follows the clean gradient for small $\kappa$ and reflects the vector sum with the growing teal term for larger $\kappa$ in Eq. 5.

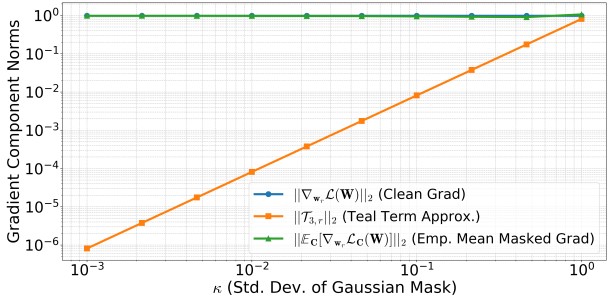
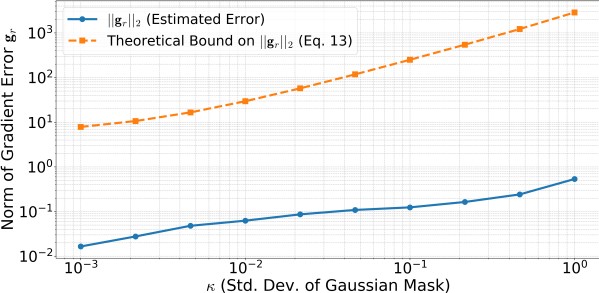

(a) $\ell_2$-norms of key components of the expected gradient $\mathbb{E}_{\mathbf{C}}[\nabla_{\mathbf{w}_r}\mathcal{L}_{\mathbf{C}}(\mathbf{W})]$:the clean gradient norm ($\|\nabla_{\mathbf{w}_r}\mathcal{L}(\mathbf{W})\|_2$), the $\mathcal{T}_{3,r}$'s norm ($\|\mathcal{T}_{3,r}\|_2$), and the total expected masked gradient norm. $\mathcal{T}_{3,r}$ scales with $\kappa^2$.

(b) Comparison of the $\ell_2$-norm of the empirically estimated gradient error term, $\|\mathbf{g}_r\|_2$, against its theoretical upper bound from Eq. (6). The empirical error (solid line) remains below the derived bound (dashed line) across all tested $\kappa$. Log-log scale.

Figure 8: $\ell_2$-norms of the gradient components (left) and residual error bound check (right).

Figure 8b examines the residual error term $\mathbf{g}_r$. It compares the $\ell_2$-norm of the empirically estimated $\mathbf{g}_0$ with its theoretical upper bound from Eq. (6). The estimated error norm, $\|\mathbf{g}_r\|_{\text{est}}$, increases with $\kappa$ (from $\approx 1.65 \times 10^{-2}$ at $\kappa = 0.001$ to $\approx 0.53$ at $\kappa = 1.0$). Importantly, the theoretical bound on $\|\mathbf{g}_r\|_2$ consistently upper-bounds the empirical error across the entire range of $\kappa$. For instance, at $\kappa = 0.001$, $\|\mathbf{g}_r\|_{\text{est}} \approx 0.0165$ while its bound is $\approx 7.85$.

In summary, the simulations confirm that the expected gradient under Gaussian input masking, with sufficiently small $\kappa$ values, is well-approximated by the sum of the clean gradient and the $\kappa^2$-dependent term, with a residual error that is effectively bounded by our theoretical derivation.

### A.2    Experimental Details for Section 6.4

We adopt the black-box threat model and the shadow model attack methodology (Adversary 1) proposed in the ML-Leaks framework (Salem et al., 2019).

*Threat Model.* The adversary has black-box access to a trained target model $f$. This means the adversary can query the model with any input $\mathbf{x}$ and observe its output posterior probability vector $\mathbf{p} = f(\mathbf{x})$ (i.e., the softmax output over the classes), but has no access to the model's parameters, gradients or original training

data. The adversary's goal is to train an *attack model $A$* that, given the posterior $\mathbf{p}_f(\mathbf{x})$ from the target model for a point $\mathbf{x}$, predicts whether $\mathbf{x}$ was a member of the target's training set.

*Shadow Model Attack Pipeline.* Since the attacker does not have access to the target model's training set, they cannot directly generate labeled data (member vs. non-member posteriors) to train their attack model. The shadow model technique circumvents this by creating a proxy environment where the attacker controls data membership. The pipeline is as follows:

1. *Data Partitioning:* The attacker possesses a dataset $D_{\text{shadow}}$, disjoint from the target's training set but drawn from the same data distribution. This set is split into $D_{\text{Shadow}}^{\text{Train}}$ and $D_{\text{Shadow}}^{\text{Out}}$.

2. *Shadow Model Training:* A shadow model $S$, which mimics the target model's architecture and training process, is trained on $D_{\text{Shadow}}^{\text{Train}}$.

3. *Attack Dataset Generation:* The trained shadow model $S$ is queried on its own training data (members, $D_{\text{Shadow}}^{\text{Train}}$) and its hold-out data (non-members, $D_{\text{Shadow}}^{\text{Out}}$). The resulting posterior vectors $\mathbf{p}_S(\mathbf{x})$ are collected. Following (Salem et al., 2019), the top-3 sorted probabilities of each posterior are used as features: $\phi(\mathbf{p}_S(\mathbf{x})) = (\mathrm{p}_{(1)}, \mathrm{p}_{(2)}, \mathrm{p}_{(3)})$. These feature vectors are labeled "1" if $\mathbf{x} \in D_{\text{Shadow}}^{\text{Train}}$ and "0" otherwise.

4. *Attack Model Training:* A binary classifier, the attack model $A$, is trained on this generated dataset of "(feature, label)" pairs. It learns to distinguish the statistical "signature" of a member's posterior from a non-member's. This signature often manifests as higher confidence (larger $\mathrm{p}_{(1)}$) and lower entropy for members, a result of the target/shadow model overfitting to its training data.

5. *Inference on Target Model:* To attack the original target model $f$, the adversary queries it with a point of interest $\mathbf{x}$, extracts the features $\phi(\mathbf{p}_f(\mathbf{x}))$, and feeds them to the trained attack model $A$ to get a membership prediction.

**Dataset and Partitioning.** We use the CIFAR-10 dataset, consisting of 60,000 images. The full pool is shuffled and divided into four disjoint sets of 10,520 images each: `target_train` (training MG-protected models), `target_test` (non-member audit data), `shadow_train` (training shadow models), and `shadow_test` (shadow non-member data). All data is normalized using the mean and standard deviation of their respective training sets. For protected models, inputs $\mathbf{x}$ are modified via elementwise multiplication with a random mask $\mathbf{m}$, where $m_i \sim \mathcal{N}(1, \kappa^2)$.

**Model Architectures.**

- **MLP ("nn"):** A fully-connected network with one hidden layer of 100 neurons (Tanh activation). Input layer: 3,072 features.

- **CNN ("cnn"):** Two `Conv-ReLU-MaxPool` blocks, followed by a Tanh-activated fully-connected layer with 100 hidden units.

**Training and Hyperparameters.** Both models are trained using the Adam optimizer (Learning Rate: $10^{-3}$, $\ell_2$ Regularization: $10^{-7}$) for intervals between 20 and 120 epochs. The attack model $A$ is a `LogisticRegression` classifier (scikit-learn), trained on a balanced dataset of member and non-member posteriors.

### A.2.1 Multiplicative Gaussian noise vs Differential privacy

We evaluate the privacy-utility tradeoff of Multiplicative Gaussian (MG) noise against Differential Privacy (DP-SGD) using the CIFAR-10 image classification dataset. Following the standard ML-Leaks evaluation protocol [Shokri et al., 2017], we utilize the dataset partitioning described in the previous section and employ a `Standard CNN` architecture, i.e. a shallow baseline consisting of two convolutional layers ($5 \times 5$ kernels, 32 filters) followed by max-pooling and a fully connected layer. For the multiplicative gaussian defense, we sweep the noise parameter $\kappa \in \{0.0, 0.2, 0.4, 0.6, 0.8, 1.0, 1.2\}$ where $\kappa = 0.0$ represents the undefended baseline. For

each training batch, we apply element-wise multiplicative noise to input features $\tilde{x} = x \odot (1 + \kappa Z)$ ›where $Z \sim \mathcal{N}(0, I)$ is the standard Gaussian noise. For the the Differential Privacy, (DP-SGD) part, we sweep the noise multiplier $\sigma \in \{0.3, 0.5, 0.8, 1.0, 1.5, 2.0, 3.0\}$ using the opacus library (Yousefpour et al. (2021)). We set the per-sample gradient clipping norm $C = 1.0$ and target $\delta = 10^{-5}$. The empirical findings for this experimet are illustrated in Figure 9. The plots map the privacy leakage (Attack Precision and Recall) against the model's utility (Target Accuracy) across the swept noise parameters.

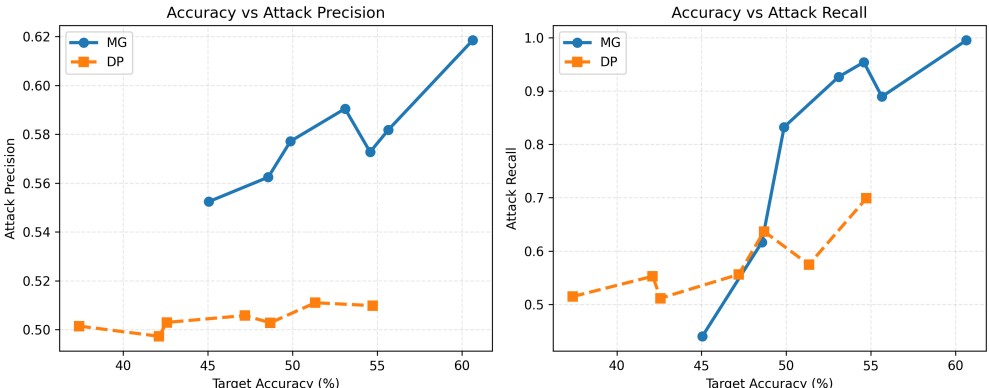

Figure 9: Privacy-utility tradeoff for the Standard CNN. The left panel shows Attack Precision vs. Accuracy, and the right panel shows Attack Recall vs. Accuracy.

**Evaluation on High-Capacity Architecture**: We repeated the evaluation using an `Improved CNN` architecture. This model features a deeper convolutional structure (convolutional blocks with increasing filter sizes $(32 \rightarrow 64 \rightarrow 128)$ using $3 \times 3$ kernels) and also incorporates Batch Normalization and Dropout to achieve higher baseline utility. Figure 10 below illustrates the results for the improved CNN architecture. We observe that the performance gap between the two methods narrows and the MG noise curve remains much closer to the near-random guess of the attacker for a longer stretch of the accuracy spectrum.

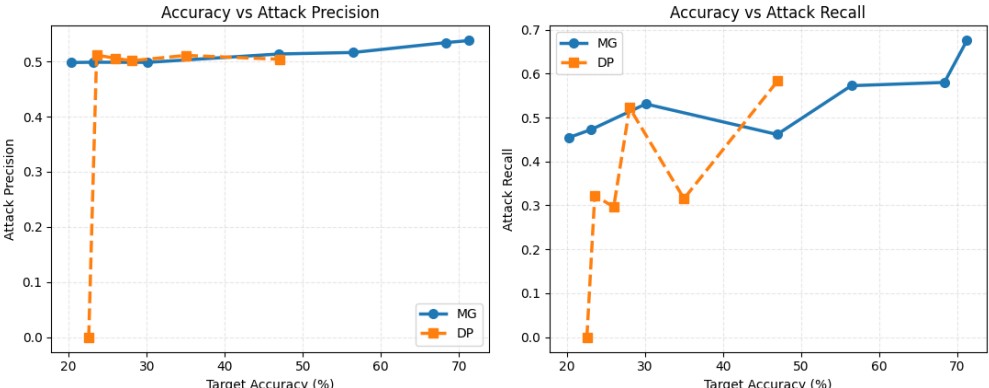

Figure 10: Privacy-utility tradeoff for the Improved CNN. Unlike the Standard CNN, the MG noise curve stays much closer to the DP-SGD curve across the accuracy range

### A.3    Additional Datasets and Broader Noise Range

To probe whether the qualitative behavior predicted by our analysis persists beyond the setting of Section 6, we repeat the accuracy-versus-noise study on two further datasets, SVHN and Fashion-MNIST, and over a broader range $\kappa \in \{0, 0.2, \dots, 3.0\}$, repeating the experiment we did for CIFAR-10, the results of which are illustrated in Figure 4b. These runs use a Tanh-based architecture family for a one-hidden-layer MLP and a

two-block convolutional network. During training we inject multiplicative Gaussian noise $\mathbf{x} \leftarrow \mathbf{x} \odot (1 + \kappa Z)$ with $Z \sim \mathcal{N}(\mathbf{0}, \mathbf{I})$, while all accuracy is measured on the clean test set; we average over three seeds and report mean and standard deviation.

The results, shown in Figures 11 and 12, are consistent with our analysis. For large $\kappa$, accuracy degrades monotonically on every dataset and architecture. The two architectures differ in the small-noise regime: The MLPs exhibit a mild beneficial regularization trend as accuracy improves slightly over the $\kappa = 0$ baseline before declining (most visibly on SVHN, where the peak near $\kappa \approx 1.0$ exceeds the clean baseline by a few points), whereas the CNNs retain their baseline accuracy at small $\kappa$ and then degrade—gently on SVHN and more sharply on Fashion-MNIST—with the run-to-run variance widening as the noise grows and the training signal is increasingly corrupted.

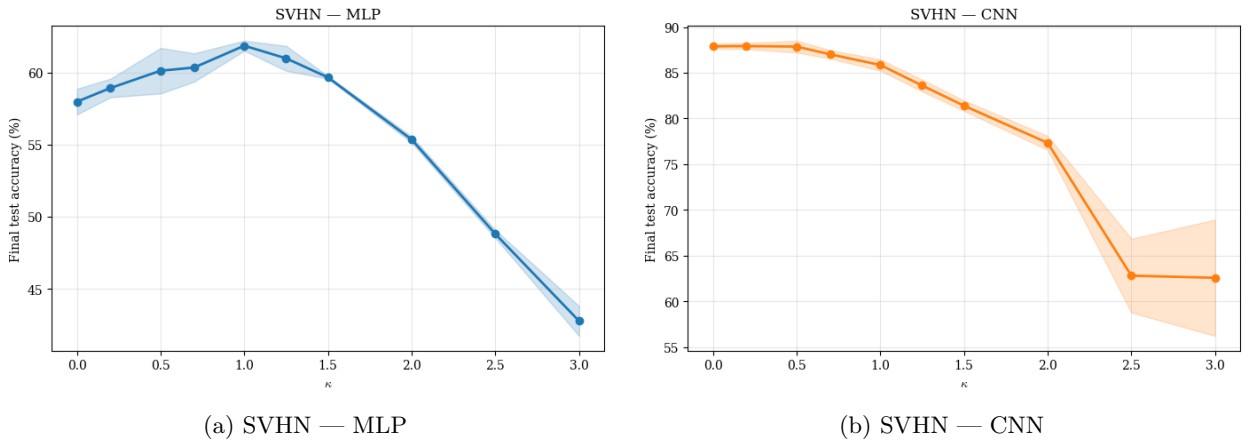

(a) SVHN — MLP                   (b) SVHN — CNN

Figure 11: Final clean test accuracy versus mask standard deviation $\kappa$ o SVHN, for (a) the MLP and (b) the CNN, under train-time multiplicative Gaussian input noise. Bands show $\pm 1$ standard deviation over three seeds. The MLP shows a mild beneficial-regularization regime near $\kappa \approx 1.0$; both architectures degrade monotonically as $\kappa$ grows.

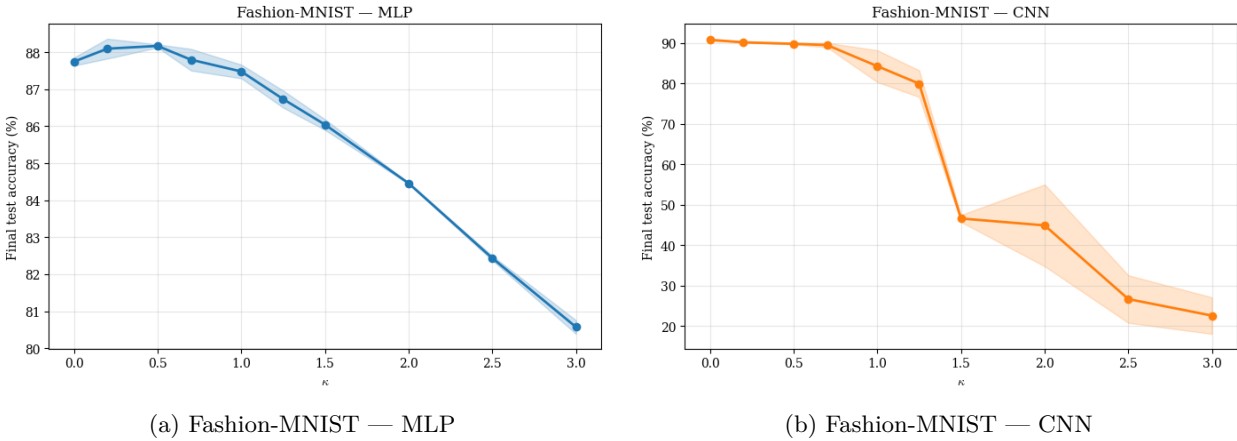

(a) Fashion-MNIST — MLP            (b) Fashion-MNIST — CNN

Figure 12: Final clean test accuracy versus mask standard deviation $\kappa$ on Fashion-MNIST, for (a) the MLP and (b) the CNN. Bands show $\pm 1$ standard deviation over three seeds. The MLP benefits from a small beneficial-regularization at low $\kappa$; the CNN retains its baseline before degrading sharply beyond $\kappa \approx 1.25$.

**A deeper architecture on a harder dataset.** To check that the same qualitative behavior appears on a more challenging benchmark, we additionally evaluate CIFAR-100 with a deeper convolutional network (the "improved CNN": four convolutional layers with batch normalization and dropout, and a ReLU-activated

fully-connected layer). We chose this architecture because CIFAR-100 is substantially harder than the other datasets. Training again injects multiplicative Gaussian noise $\mathbf{x} \leftarrow \mathbf{x} \odot (1 + \kappa Z)$, $Z \sim \mathcal{N}(\mathbf{0}, \mathbf{I})$, with accuracy measured on the clean test set and averaged over three seeds. As shown in Figure 13, accuracy decreases monotonically with $\kappa$, from roughly 52% at $\kappa = 0$ to about 14% at $\kappa = 1.8$. Unlike the MLPs, this network shows no beneficial-regularization regime at small $\kappa$; the heavier built-in regularization (batch normalization and dropout) likely leaves little room for additional input noise to help but the monotone, $\kappa$-controlled degradation is nonetheless consistent with the noise-dependent error region.

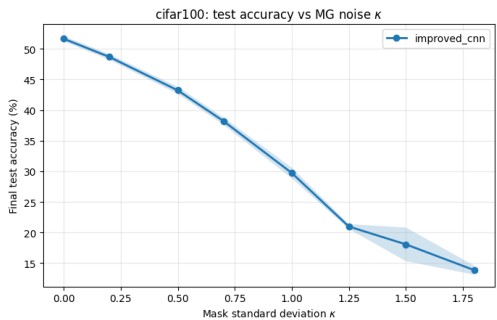

Figure 13: Final clean test accuracy versus mask standard deviation $\kappa$ on CIFAR-100, for a deeper CNN with batch normalization and dropout, under train-time multiplicative Gaussian input noise. The band shows $\pm 1$ standard deviation over three seeds. Accuracy degrades monotonically with $\kappa$, with no beneficial-regularization regime.

**A practical analogue of the error ball on real data.** Figure 3a illustrates the noise-controlled error ball on the synthetic regression task that our theory describes. To see whether the same qualitative behavior appears on real data, we reproduce that experiment on Fashion-MNIST. For a classifier trained with multiplicative Gaussian input noise $\mathbf{x} \leftarrow \mathbf{x} \odot (1 + \kappa Z)$, $Z \sim \mathcal{N}(\mathbf{0}, \mathbf{I})$, we track the *clean* training loss (cross-entropy on the unmasked training set) as a function of the iteration count, for several values of $\kappa$. As shown in Figure 14, every trajectory decreases rapidly and then plateaus, and—crucially—the plateaus are ordered by $\kappa$: the clean run ($\kappa = 0$) and the smallest noise level settle lowest, while each larger $\kappa$ settles at a progressively higher floor, with the $\kappa = 2.0$ curve both highest and noisiest.

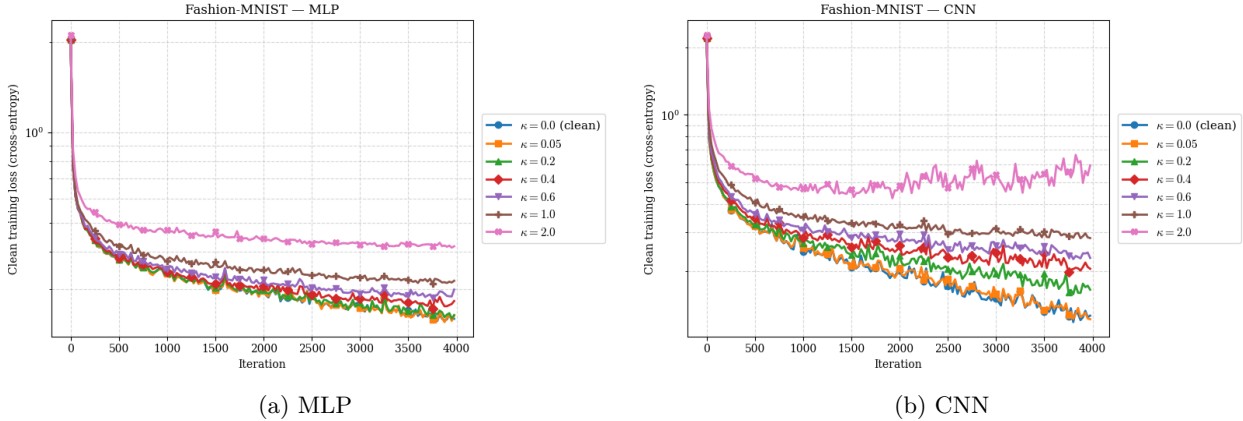

(a) MLP                                      (b) CNN

Figure 14: Clean training loss (cross-entropy on the unmasked training set, log scale) versus iteration on Fashion-MNIST, under train-time multiplicative Gaussian input noise, for (a) an MLP and (b) a CNN. Each curve corresponds to a mask standard deviation $\kappa$. The loss decreases and then plateaus at a floor whose height increases with $\kappa$—the practical counterpart of the noise-controlled error ball in Theorem 5.6.

# B    Proofs in Section 4

In this section, we first prove an exact form of the expected surrogate loss function.

## B.1    General Form of Expected Loss

**Lemma B.1.** *Let* $\mathbf{u}_{i,r} = \mathbf{w}_r \odot \mathbf{x}_i$, *let* $\hat{\sigma}_\kappa(\mathbf{w}, \mathbf{x}) = \mathbf{w}^\top \mathbf{x} \cdot \Phi_1\left(\frac{\mathbf{w}^\top \mathbf{x}}{\kappa \|\mathbf{w} \odot \mathbf{x}\|_2}\right)$, *and let* $\hat{f}(\boldsymbol{\theta}, \mathbf{x}) = \frac{1}{\sqrt{m}} \sum_{r=1}^m a_r \hat{\sigma}_\kappa(\mathbf{w}_r, \mathbf{x})$. *Then we have*

$$
\mathbb{E}_{\mathbf{C}}[\mathcal{L}_{\mathbf{C}}(\boldsymbol{\theta})] = \frac{1}{2} \sum_{i=1}^n \left(\hat{f}(\boldsymbol{\theta}, \mathbf{x}_i) - y_i\right)_2^2 + \frac{\kappa^2}{2m} \sum_{i=1}^n \left\| \sum_{r=1}^m a_r \mathbf{u}_{i,r} \Phi\left(\frac{\mathbf{w}_r^\top \mathbf{x}_i}{\kappa \|\mathbf{u}_{i,r}\|_2}\right) \right\|_2^2
$$
$$
+ \frac{1}{m} \sum_{i=1}^n \sum_{r,r'=1}^m a_r a_{r'} \left(C_{i,r,r'} + \frac{\kappa^2}{2\pi} E_{i,r,r'}\right)
$$
$$
+ \frac{2\kappa}{\sqrt{2\pi m}} \sum_{i=1}^n \sum_{r=1}^m a_r G_{i,r} \left(\frac{1}{\sqrt{m}} \sum_{r'=1}^m a'_r T_{i,r,r'} - y\right)
$$

*where* $C_{i,r,r'}, E_{i,r,r'}, T_{i,r,r'}$ *and* $G_{i,r}$ *are defined as*

$$
C_{i,r,r'} = \left((\mathbf{w}_r^\top \mathbf{x}_i)(\mathbf{w}_{r'}^\top \mathbf{x}_i) + \kappa^2 \mathbf{u}_{i,r}^\top \mathbf{u}_{i,r'}\right) \mathcal{C}\left(\frac{\mathbf{w}_r^\top \mathbf{x}_i}{\kappa \|\mathbf{u}_{i,r}\|_2}, \frac{\mathbf{w}_{r'}^\top \mathbf{x}_i}{\kappa \|\mathbf{u}_{i,r'}\|_2}, \frac{\mathbf{u}_{i,r}^\top \mathbf{u}_{i,r'}}{\|\mathbf{u}_{i,r}\|_2 \|\mathbf{u}_{i,r'}\|_2}\right)
$$

$$
E_{i,r,r'} = \|\mathbf{u}_{i,r}\|_2 \|\mathbf{u}_{i,r'}\|_2 \exp\left(-\frac{\|\mathbf{u}_{i,r'}\|_2^2 (\mathbf{w}_r^\top \mathbf{x}_i)^2 - 2(\mathbf{u}_{i,r}^\top \mathbf{u}_{i,r'})(\mathbf{w}_r^\top \mathbf{x}_i)(\mathbf{w}_{r'}^\top \mathbf{x}_i) + \|\mathbf{u}_{i,r}\|_2^2 (\mathbf{w}_{r'}^\top \mathbf{x}_i)^2}{2\kappa^2 \left(\|\mathbf{u}_{i,r}\|_2^2 \|\mathbf{u}_{i,r'}\|_2^2 - (\mathbf{u}_{i,r}^\top \mathbf{u}_{i,r'})^2\right)}\right)
$$

$$
T_{i,r,r'} = \mathbf{w}_{r'}^\top \mathbf{x}_i \cdot \Phi_1\left(\frac{\|\mathbf{u}_{i,r}\|_2 \|\mathbf{u}_{i,r'}\|_2 \cdot \mathbf{w}_{r'}^\top \mathbf{x} - \mathbf{u}_{i,r}^\top \mathbf{u}_{i,r'} \cdot \mathbf{w}_r^\top \mathbf{x}_i}{\kappa^2 \|\mathbf{u}_{i,r}\|_2 \|\mathbf{u}_{i,r'}\|_2 \left(\|\mathbf{u}_{i,r}\|_2^2 \|\mathbf{u}_{i,r'}\|_2^2 - (\mathbf{u}_{i,r}^\top \mathbf{u}_{i,r'})^2\right)}\right)
$$

$$
G_{i,r} = \|\mathbf{u}_{i,r}\|_2 \exp\left(-\frac{(\mathbf{w}_r^\top \mathbf{x}_i)^2}{2\kappa^2 \|\mathbf{u}_r\|_2^2}\right)
$$

*Proof.* By definition, we have

$$
\mathbb{E}_{\mathbf{C}}[\mathcal{L}_{\mathbf{C}}(\boldsymbol{\theta})] = \frac{1}{2} \sum_{i=1}^n \mathbb{E}_{\mathbf{c}_i}\left[(f(\boldsymbol{\theta}, \mathbf{x}_i \odot \mathbf{c}_i) - y_i)^2\right]
$$

For simplicity, we fix $i \in [n]$, and study $\mathbb{E}_{\mathbf{c}}\left[(f(\boldsymbol{\theta}, \mathbf{x} \odot \mathbf{c}) - y)^2\right]$. In the analysis below, we let $\mathbf{u}_r = \mathbf{w}_r \odot \mathbf{x}$. In particular, we have

$$
\mathbb{E}_{\mathbf{c}}\left[(f(\boldsymbol{\theta}, \mathbf{x} \odot \mathbf{c}) - y)^2\right] = \mathbb{E}_{\mathbf{c}}\left[f(\boldsymbol{\theta}, \mathbf{x} \odot \mathbf{c})^2\right] - 2y\mathbb{E}_{\mathbf{c}}[f(\boldsymbol{\theta}, \mathbf{x} \odot \mathbf{c})] + y^2
$$

Here we shall evaluate the two expectations separately. To start, for the first-order term, we have

$$
\mathbb{E}_{\mathbf{c}}[f(\boldsymbol{\theta}, \mathbf{x} \odot \mathbf{c})] = \frac{1}{\sqrt{m}} \sum_{r=1}^m a_r \mathbb{E}\left[\sigma\left(\mathbf{w}_r^\top (\mathbf{c} \odot \mathbf{x})\right)\right] = \frac{1}{\sqrt{m}} \sum_{r=1}^m a_r \mathbb{E}\left[\sigma\left(\mathbf{c}^\top (\mathbf{w}_r \odot \mathbf{x})\right)\right] \tag{18}
$$

Notice that since $\mathbf{c} \sim \mathcal{N}(\mathbf{1}, \kappa^2 \mathbf{I})$. By Lemma D.13 we have that $\mathbf{c}^\top (\mathbf{w}_r \odot \mathbf{x}) \sim \mathcal{N}\left(\mathbf{w}_r^\top \mathbf{x}, \kappa^2 \|\mathbf{w}_r \odot \mathbf{x}\|_2^2\right)$, since $\mathbf{1}^\top (\mathbf{w}_r \odot \mathbf{x}) = \mathbf{w}_r^\top \mathbf{x}$. Applying Lemma D.18 with $z = \mathbf{c}^\top (\mathbf{w}_r \odot \mathbf{x})$, mean $\mathbf{w}_r^\top \mathbf{x}$, and standard deviation $\kappa \|\mathbf{w}_r \odot \mathbf{x}\|$, we have that

$$
\mathbb{E}\left[\sigma\left(\mathbf{c}^\top (\mathbf{w}_r \odot \mathbf{x})\right)\right] = \frac{\kappa \|\mathbf{w}_r \odot \mathbf{x}\|_2}{\sqrt{2\pi}} \exp\left(-\frac{(\mathbf{w}_r^\top \mathbf{x})^2}{2\kappa^2 \|\mathbf{w}_r \odot \mathbf{x}\|_2^2}\right) + \mathbf{w}_r^\top \mathbf{x} \Phi_1\left(\frac{\mathbf{w}_r^\top \mathbf{x}}{\kappa \|\mathbf{w}_r \odot \mathbf{x}\|_2}\right) \tag{19}
$$

Plugging in to the form of (18) gives

$$\mathbb{E}_{\mathbf{c}}\left[f\left(\boldsymbol{\theta}, \mathbf{x} \odot \mathbf{c}\right)\right] = \frac{1}{\sqrt{m}} \sum_{r=1}^{m} a_r \hat{\sigma}_\kappa\left(\mathbf{w}_r, \mathbf{x}\right) + \frac{\kappa}{\sqrt{2\pi m}} \sum_{r=1}^{m} a_r \left\|\mathbf{u}_r\right\|_2 \exp\left(-\frac{\left(\mathbf{w}_r^\top \mathbf{x}\right)^2}{2\kappa^2 \left\|\mathbf{u}_r\right\|_2^2}\right) \tag{20}$$

Next, we focus on the second-order term. Since $\mathbf{w}_r^\top\left(\mathbf{c} \odot \mathbf{x}\right) = \mathbf{c}^\top\left(\mathbf{w}_r \odot \mathbf{x}\right)$, we have

$$\mathbb{E}_{\mathbf{c}}\left[f\left(\boldsymbol{\theta}, \mathbf{x} \odot \mathbf{c}\right)^2\right] = \frac{1}{m} \sum_{r,r'=1}^{m} a_r a_{r'} \mathbb{E}\left[\sigma\left(\mathbf{c}^\top \mathbf{u}_r\right) \sigma\left(\mathbf{c}^\top \mathbf{u}_{r'}\right)\right]$$

Let $z_1 = \mathbf{c}^\top \mathbf{u}_r$ and $z_2 = \mathbf{c}^\top \mathbf{u}_{r'}$, we have that $z_1 \sim \mathcal{N}\left(\mathbf{w}_r^\top\mathbf{x}, \kappa^2 \left\|\mathbf{u}_r\right\|_2^2\right), z_2 \sim \mathcal{N}\left(\mathbf{w}_{r'}^\top\mathbf{x}, \kappa^2 \left\|\mathbf{u}_{r'}\right\|_2^2\right)$, and $\mathrm{Cov}\left(z_1, z_2\right) = \kappa^2 \mathbf{u}_r^\top \mathbf{u}_{r'}$. Applying Lemma D.9 with $a = b = 0$ gives

$$\mathbb{E}\left[\sigma\left(\mathbf{c}^\top \mathbf{u}_r\right) \sigma\left(\mathbf{c}^\top \mathbf{u}_{r'}\right)\right]$$

$$= \left(\left(\mathbf{w}_r^\top \mathbf{x}\right)\left(\mathbf{w}_{r'}^\top\mathbf{x}\right) + \kappa^2 \mathbf{u}_r^\top \mathbf{u}_{r'}\right) \boldsymbol{\Phi}_2\left(\frac{\mathbf{w}_r^\top \mathbf{x}}{\kappa \left\|\mathbf{u}_r\right\|_2}, \frac{\mathbf{w}_{r'}^\top\mathbf{x}}{\kappa \left\|\mathbf{u}_{r'}\right\|_2}, \frac{\mathbf{u}_r^\top \mathbf{u}_{r'}}{\left\|\mathbf{u}_r\right\|_2 \left\|\mathbf{u}_{r'}\right\|_2}\right)$$

$$+ \frac{\kappa^2}{2\pi} \left\|\mathbf{u}_r\right\|_2 \left\|\mathbf{u}_{r'}\right\|_2 \exp\left(-\frac{\left\|\mathbf{u}_{r'}\right\|_2^2 \left(\mathbf{w}_r^\top \mathbf{x}\right)^2 - 2\left(\mathbf{u}_r^\top \mathbf{u}_{r'}\right)\left(\mathbf{w}_r^\top \mathbf{x}\right)\left(\mathbf{w}_{r'}^\top\mathbf{x}\right) + \left\|\mathbf{u}_r\right\|_2^2 \left(\mathbf{w}_{r'}^\top\mathbf{x}\right)^2}{2\kappa^2 \left(\left\|\mathbf{u}_r\right\|_2^2 \left\|\mathbf{u}_{r'}\right\|_2^2 - \left(\mathbf{u}_r^\top \mathbf{u}_{r'}\right)^2\right)}\right)$$

$$+ \frac{\kappa}{\sqrt{2\pi}}\left(\left\|\mathbf{u}_r\right\|_2 \cdot \mathbf{w}_{r'}^\top\mathbf{x} \cdot \hat{T}_{1,r,r'} + \left\|\mathbf{u}_{r'}\right\|_2 \cdot \mathbf{w}_r^\top\mathbf{x} \cdot \hat{T}_{2,r,r'}\right)$$

where $\hat{T}_{1,r,r'}, \hat{T}_{2,r,r'}$ are defined as

$$\hat{T}_{1,r,r'} = \exp\left(-\frac{\left(\mathbf{w}_r^\top \mathbf{x}\right)^2}{2\kappa^2 \left\|\mathbf{u}_r\right\|_2^2}\right) \boldsymbol{\Phi}_1\left(\frac{\left\|\mathbf{u}_r\right\|_2^2 \cdot \mathbf{w}_{r'}^\top\mathbf{x} - \mathbf{u}_r^\top \mathbf{u}_{r'} \cdot \mathbf{w}_r^\top\mathbf{x}}{\kappa \left\|\mathbf{u}_r\right\|_2 \left(\left\|\mathbf{u}_r\right\|_2^2 \left\|\mathbf{u}_{r'}\right\|_2^2 - \left(\mathbf{u}_r^\top \mathbf{u}_{r'}\right)^2\right)^{\frac{1}{2}}}\right)$$

$$\hat{T}_{2,r,r'} = \exp\left(-\frac{\left(\mathbf{w}_{r'}^\top\mathbf{x}\right)^2}{2\kappa^2 \left\|\mathbf{u}_{r'}\right\|_2^2}\right) \boldsymbol{\Phi}_1\left(\frac{\left\|\mathbf{u}_{r'}\right\|_2^2 \cdot \mathbf{w}_r^\top \mathbf{x} - \mathbf{u}_r^\top \mathbf{u}_{r'} \cdot \mathbf{w}_{r'}^\top\mathbf{x}}{\kappa \left\|\mathbf{u}_{r'}\right\|_2 \left(\left\|\mathbf{u}_r\right\|_2^2 \left\|\mathbf{u}_{r'}\right\|_2^2 - \left(\mathbf{u}_r^\top \mathbf{u}_{r'}\right)^2\right)^{\frac{1}{2}}}\right)$$

For the simplicity of notations, we define $T_{1,r,r'} = \left\|\mathbf{u}_r\right\|_2 \cdot \mathbf{w}_{r'}^\top\mathbf{x} \cdot \hat{T}_{1,r,r'}$ and $T_{2,r,r'} = \left\|\mathbf{u}_{r'}\right\|_2 \cdot \mathbf{w}_r^\top\mathbf{x} \cdot \hat{T}_{2,r,r'}$. Moreover, we define

$$E_{r,r'} = \left\|\mathbf{u}_r\right\|_2 \left\|\mathbf{u}_{r'}\right\|_2 \exp\left(-\frac{\left\|\mathbf{u}_{r'}\right\|_2^2 \left(\mathbf{w}_r^\top \mathbf{x}\right)^2 - 2\left(\mathbf{u}_r^\top \mathbf{u}_{r'}\right)\left(\mathbf{w}_r^\top \mathbf{x}\right)\left(\mathbf{w}_{r'}^\top\mathbf{x}\right) + \left\|\mathbf{u}_r\right\|_2^2 \left(\mathbf{w}_{r'}^\top\mathbf{x}\right)^2}{2\kappa^2 \left(\left\|\mathbf{u}_r\right\|_2^2 \left\|\mathbf{u}_{r'}\right\|_2^2 - \left(\mathbf{u}_r^\top \mathbf{u}_{r'}\right)^2\right)}\right)$$

Lastly, we use the definition of the Gaussian Copula function $\mathcal{C}\left(a, b, \rho\right) = \boldsymbol{\Phi}_2\left(a, b, \rho\right) - \boldsymbol{\Phi}_1\left(a\right) \boldsymbol{\Phi}_1\left(b\right)$ and define

$$C_{r,r'} = \left(\left(\mathbf{w}_r^\top \mathbf{x}\right)\left(\mathbf{w}_{r'}^\top\mathbf{x}\right) + \kappa^2 \mathbf{u}_r^\top \mathbf{u}_{r'}\right) \mathcal{C}\left(\frac{\mathbf{w}_r^\top \mathbf{x}}{\kappa \left\|\mathbf{u}_r\right\|_2}, \frac{\mathbf{w}_{r'}^\top\mathbf{x}}{\kappa \left\|\mathbf{u}_{r'}\right\|_2}, \frac{\mathbf{u}_r^\top \mathbf{u}_{r'}}{\left\|\mathbf{u}_r\right\|_2 \left\|\mathbf{u}_{r'}\right\|_2}\right)$$

Under these definitions, we have that

$$\mathbb{E}\left[\sigma\left(\mathbf{c}^\top \mathbf{u}_r\right) \sigma\left(\mathbf{c}^\top \mathbf{u}_{r'}\right)\right] = \left(\left(\mathbf{w}_r^\top \mathbf{x}\right)\left(\mathbf{w}_{r'}^\top\mathbf{x}\right) + \kappa^2 \mathbf{u}_r^\top \mathbf{u}_{r'}\right) \boldsymbol{\Phi}_1\left(\frac{\mathbf{w}_r^\top \mathbf{x}}{\kappa \left\|\mathbf{u}_r\right\|_2}\right) \boldsymbol{\Phi}_1\left(\frac{\mathbf{w}_{r'}^\top\mathbf{x}}{\kappa \left\|\mathbf{u}_{r'}\right\|_2}\right)$$

$$+ C_{r,r'} + \frac{\kappa^2}{2\pi} E_{r,r'} + \frac{\kappa}{\sqrt{2\pi}}\left(T_{1,r,r'} + T_{2,r,r'}\right)$$

$$= \hat{\sigma}_\kappa\left(\mathbf{w}_r, \mathbf{x}\right) \hat{\sigma}_\kappa\left(\mathbf{w}_{r'}, \mathbf{x}\right) + \kappa^2 \mathbf{u}_r^\top \mathbf{u}_{r'} \boldsymbol{\Phi}_1\left(\frac{\mathbf{w}_r^\top \mathbf{x}}{\kappa \left\|\mathbf{u}_r\right\|_2}\right) \boldsymbol{\Phi}_1\left(\frac{\mathbf{w}_{r'}^\top\mathbf{x}}{\kappa \left\|\mathbf{u}_{r'}\right\|_2}\right)$$

$$+ C_{r,r'} + \frac{\kappa^2}{2\pi} E_{r,r'} + \frac{\kappa}{\sqrt{2\pi}}\left(T_{1,r,r'} + T_{2,r,r'}\right)$$

Plugging back into the expression of $\mathbb{E}_{\mathbf{c}}\left[f\left(\boldsymbol{\theta}, \mathbf{x} \odot \mathbf{c}\right)^2\right]$ gives

$$
\begin{aligned}
\mathbb{E}_{\mathbf{c}}&\left[f\left(\boldsymbol{\theta}, \mathbf{x} \odot \mathbf{c}\right)^2\right] \\
&= \frac{1}{m}\sum_{r,r'=1}^{m} a_r a_{r'} \hat{\sigma}_\kappa\left(\mathbf{w}_r, \mathbf{x}\right)\hat{\sigma}_\kappa\left(\mathbf{w}_{r'}, \mathbf{x}\right) + \frac{1}{m}\sum_{r,r'=1}^{m} a_r a_{r'}\kappa^2 \mathbf{u}_r^\top \mathbf{u}_{r'}\boldsymbol{\Phi}_1\left(\frac{\mathbf{w}_r^\top\mathbf{x}}{\kappa\|\mathbf{u}_r\|_2}\right)\boldsymbol{\Phi}_1\left(\frac{\mathbf{w}_{r'}^\top\mathbf{x}}{\kappa\|\mathbf{u}_{r'}\|_2}\right) \\
&\quad + \frac{1}{m}\sum_{r,r'=1}^{m} a_r a_{r'}\left(C_{r,r'} + \frac{\kappa^2}{2\pi}E_{r,r'} + \frac{\kappa}{\sqrt{2\pi}}\left(T_{1,r,r'} + T_{2,r,r'}\right)\right) \\
&= \left(\frac{1}{\sqrt{m}}\sum_{r=1}^{m} a_r\hat{\sigma}_\kappa\left(\mathbf{w}_r, \mathbf{x}\right)\right)^2 + \kappa^2\left\|\frac{1}{\sqrt{m}}\sum_{r=1}^{m} a_r\mathbf{u}_r\boldsymbol{\Phi}\left(\frac{\mathbf{w}_r^\top\mathbf{x}}{\kappa\|\mathbf{u}_r\|_2}\right)\right\|_2^2 \\
&\quad + \frac{1}{m}\sum_{r,r'=1}^{m} a_r a_{r'}\left(C_{r,r'} + \frac{\kappa^2}{2\pi}E_{r,r'} + \frac{\kappa}{\sqrt{2\pi}}\left(T_{1,r,r'} + T_{2,r,r'}\right)\right)
\end{aligned}
$$

Combining the expression of $\mathbb{E}_{\mathbf{c}}\left[f\left(\boldsymbol{\theta}, \mathbf{x} \odot \mathbf{c}\right)^2\right]$ and $\mathbb{E}_{\mathbf{c}}\left[f\left(\boldsymbol{\theta}, \mathbf{x} \odot \mathbf{c}\right)\right]$, and noticing $T_{1,r,r'} = T_{2,r',r}$, we have

$$
\begin{aligned}
\mathbb{E}_{\mathbf{c}}&\left[\left(f\left(\boldsymbol{\theta}, \mathbf{x} \odot \mathbf{c}\right) - y\right)^2\right] = \left(\frac{1}{\sqrt{m}}\sum_{r=1}^{m} a_r\hat{\sigma}_\kappa\left(\mathbf{w}_r, \mathbf{x}\right)\right)^2 + \kappa^2\left\|\frac{1}{\sqrt{m}}\sum_{r=1}^{m} a_r\mathbf{u}_r\boldsymbol{\Phi}\left(\frac{\mathbf{w}_r^\top\mathbf{x}}{\kappa\|\mathbf{u}_r\|_2}\right)\right\|_2^2 \\
&\quad + \frac{1}{m}\sum_{r,r'=1}^{m} a_r a_{r'}\left(C_{r,r'} + \frac{\kappa^2}{2\pi}E_{r,r'} + \frac{\kappa}{\sqrt{2\pi}}\left(T_{1,r,r'} + T_{2,r,r'}\right)\right) \\
&\quad - \frac{2y}{\sqrt{m}}\sum_{r=1}^{m} a_r\hat{\sigma}_\kappa\left(\mathbf{w}_r, \mathbf{x}\right) - \frac{2\kappa y}{\sqrt{2\pi m}}\sum_{r=1}^{m} a_r\|\mathbf{u}_r\|_2\exp\left(-\frac{\left(\mathbf{w}_r^\top\mathbf{x}\right)^2}{\|\mathbf{u}_r\|_2^2}\right) + y^2 \\
&= \left(\frac{1}{\sqrt{m}}\sum_{r=1}^{m} a_r\hat{\sigma}_\kappa\left(\mathbf{w}_r, \mathbf{x}\right) - y\right)^2 + \frac{\kappa^2}{m}\left\|\sum_{r=1}^{m} a_r\mathbf{u}_r\boldsymbol{\Phi}\left(\frac{\mathbf{w}_r^\top\mathbf{x}}{\kappa\|\mathbf{u}_r\|_2}\right)\right\|_2^2 \\
&\quad + \frac{1}{m}\sum_{r,r'=1}^{m} a_r a_{r'}\left(C_{r,r'} + \frac{\kappa^2}{2\pi}E_{r,r'} + \frac{2\kappa}{\sqrt{2\pi}}T_{1,r,r'}\right) \\
&\quad - \frac{2\kappa y}{\sqrt{2\pi m}}\sum_{r=1}^{m} a_r\|\mathbf{u}_r\|_2\exp\left(-\frac{\left(\mathbf{w}_r^\top\mathbf{x}\right)^2}{2\kappa^2\|\mathbf{u}_r\|_2^2}\right)
\end{aligned}
$$

To extend to the case of $\mathbf{x}_i, y_i$, we need to re-define

$$
C_{i,r,r'} = \left(\left(\mathbf{w}_r^\top\mathbf{x}_i\right)\left(\mathbf{w}_{r'}^\top\mathbf{x}_i\right) + \kappa^2\mathbf{u}_{i,r}^\top\mathbf{u}_{i,r'}\right)\mathcal{C}\left(\frac{\mathbf{w}_r^\top\mathbf{x}_i}{\kappa\|\mathbf{u}_{i,r}\|_2}, \frac{\mathbf{w}_{r'}^\top\mathbf{x}_i}{\kappa\|\mathbf{u}_{i,r'}\|_2}, \frac{\mathbf{u}_{i,r}^\top\mathbf{u}_{i,r'}}{\|\mathbf{u}_{i,r}\|_2\|\mathbf{u}_{i,r'}\|_2}\right)
$$

$$
E_{i,r,r'} = \|\mathbf{u}_{i,r}\|_2\|\mathbf{u}_{i,r'}\|_2\exp\left(-\frac{\|\mathbf{u}_{i,r'}\|_2^2\left(\mathbf{w}_r^\top\mathbf{x}_i\right)^2 - 2\left(\mathbf{u}_{i,r}^\top\mathbf{u}_{i,r'}\right)\left(\mathbf{w}_r^\top\mathbf{x}_i\right)\left(\mathbf{w}_{r'}^\top\mathbf{x}_i\right) + \|\mathbf{u}_{i,r}\|_2^2\left(\mathbf{w}_{r'}^\top\mathbf{x}_i\right)^2}{2\kappa^2\left(\|\mathbf{u}_{i,r}\|_2^2\|\mathbf{u}_{i,r'}\|_2^2 - \left(\mathbf{u}_{i,r}^\top\mathbf{u}_{i,r'}\right)^2\right)}\right)
$$

$$
T_{i,r,r'} = \mathbf{w}_{r'}^\top\mathbf{x}_i\cdot\boldsymbol{\Phi}_1\left(\frac{\|\mathbf{u}_{i,r}\|_2^2\cdot\mathbf{w}_{r'}^\top\mathbf{x}_i - \mathbf{u}_{i,r}^\top\mathbf{u}_{i,r'}\cdot\mathbf{w}_r^\top\mathbf{x}_i}{\kappa\|\mathbf{u}_{i,r}\|_2\left(\|\mathbf{u}_{i,r}\|_2^2\|\mathbf{u}_{i,r'}\|_2^2 - \left(\mathbf{u}_{i,r}^\top\mathbf{u}_{i,r'}\right)^2\right)^{\frac{1}{2}}}\right)
$$

$$
G_{i,r} = \|\mathbf{u}_{i,r}\|_2\exp\left(-\frac{\left(\mathbf{w}_r^\top\mathbf{x}_i\right)^2}{2\kappa^2\|\mathbf{u}_r\|_2^2}\right)
$$

Moreover, let $\hat{f}(\boldsymbol{\theta}, \mathbf{x}) = \frac{1}{\sqrt{m}} \sum_{r=1}^{m} a_r \hat{\sigma}_\kappa(\mathbf{w}_r, \mathbf{x})$. Then we have that

$$
\begin{aligned}
\mathbb{E}_{\mathbf{C}}[\mathcal{L}_{\mathbf{C}}(\boldsymbol{\theta})] &= \frac{1}{2} \sum_{i=1}^{n} \left( \hat{f}(\boldsymbol{\theta}, \mathbf{x}_i) - y_i \right)_2^2 + \frac{\kappa^2}{2m} \sum_{i=1}^{n} \left\| \sum_{r=1}^{m} a_r \mathbf{u}_{i,r} \boldsymbol{\Phi} \left( \frac{\mathbf{w}_r^\top \mathbf{x}_i}{\kappa \|\mathbf{u}_{i,r}\|_2} \right) \right\|_2^2 \\
&\quad + \frac{1}{m} \sum_{i=1}^{n} \sum_{r,r'=1}^{m} a_r a_{r'} \left( C_{i,r,r'} + \frac{\kappa^2}{2\pi} E_{i,r,r'} + \kappa \sqrt{\frac{2}{\pi}} T_{i,r,r'} G_{i,r} \right) \\
&\quad - \kappa y \sqrt{\frac{2}{\pi m}} \sum_{i=1}^{n} \sum_{r=1}^{m} a_r G_{i,r} \\
&= \frac{1}{2} \sum_{i=1}^{n} \left( \hat{f}(\boldsymbol{\theta}, \mathbf{x}_i) - y_i \right)_2^2 + \frac{\kappa^2}{2m} \sum_{i=1}^{n} \left\| \sum_{r=1}^{m} a_r \mathbf{u}_{i,r} \boldsymbol{\Phi} \left( \frac{\mathbf{w}_r^\top \mathbf{x}_i}{\kappa \|\mathbf{u}_{i,r}\|_2} \right) \right\|_2^2 \\
&\quad + \frac{1}{m} \sum_{i=1}^{n} \sum_{r,r'=1}^{m} a_r a_{r'} \left( C_{i,r,r'} + \frac{\kappa^2}{2\pi} E_{i,r,r'} \right) \\
&\quad + \frac{2\kappa}{\sqrt{2\pi m}} \sum_{i=1}^{n} \sum_{r=1}^{m} a_r G_{i,r} \left( \frac{1}{\sqrt{m}} \sum_{r'=1}^{m} a_r' T_{i,r,r'} - y \right)
\end{aligned}
$$

$\square$

## B.2 Proof of Theorem 4.2

*Proof.* Let $\mathbf{u}_{i,r} = \mathbf{w}_r \odot \mathbf{x}_i$. By Lemma B.1, we have that

$$
\begin{aligned}
\mathbb{E}_{\mathbf{C}}[\mathcal{L}_{\mathbf{C}}(\boldsymbol{\theta})] &= \frac{1}{2} \sum_{i=1}^{n} \left( \hat{f}(\boldsymbol{\theta}, \mathbf{x}_i) - y_i \right)_2^2 + \frac{\kappa^2}{2m} \sum_{i=1}^{n} \left\| \sum_{r=1}^{m} a_r \mathbf{u}_{i,r} \boldsymbol{\Phi} \left( \frac{\mathbf{w}_r^\top \mathbf{x}_i}{\kappa \|\mathbf{u}_{i,r}\|_2} \right) \right\|_2^2 \\
&\quad + \frac{1}{m} \sum_{i=1}^{n} \sum_{r,r'=1}^{m} a_r a_{r'} \left( C_{i,r,r'} + \frac{\kappa^2}{2\pi} E_{i,r,r'} \right) \\
&\quad + \frac{2\kappa}{\sqrt{2\pi m}} \sum_{i=1}^{n} \sum_{r=1}^{m} a_r G_{i,r} \left( \frac{1}{\sqrt{m}} \sum_{r'=1}^{m} a_r' T_{i,r,r'} - y \right)
\end{aligned}
$$

where $C_{i,r,r'}, E_{i,r,r'}, T_{i,r,r'}$ and $G_{i,r}$ are defined as

$$
C_{i,r,r'} = \left( (\mathbf{w}_r^\top \mathbf{x}_i)(\mathbf{w}_{r'}^\top \mathbf{x}_i) + \kappa^2 \mathbf{u}_{i,r}^\top \mathbf{u}_{i,r'} \right) \mathcal{C} \left( \frac{\mathbf{w}_r^\top \mathbf{x}_i}{\kappa \|\mathbf{u}_{i,r}\|_2}, \frac{\mathbf{w}_{r'}^\top \mathbf{x}_i}{\kappa \|\mathbf{u}_{i,r'}\|_2}, \frac{\mathbf{u}_{i,r}^\top \mathbf{u}_{i,r'}}{\|\mathbf{u}_{i,r}\|_2 \|\mathbf{u}_{i,r'}\|_2} \right)
$$

$$
E_{i,r,r'} = \|\mathbf{u}_{i,r}\|_2 \|\mathbf{u}_{i,r'}\|_2 \exp \left( -\frac{\|\mathbf{u}_{i,r'}\|_2^2 (\mathbf{w}_r^\top \mathbf{x}_i)^2 - 2(\mathbf{u}_{i,r}^\top \mathbf{u}_{i,r'})(\mathbf{w}_r^\top \mathbf{x}_i)(\mathbf{w}_{r'}^\top \mathbf{x}_i) + \|\mathbf{u}_{i,r}\|_2^2 (\mathbf{w}_{r'}^\top \mathbf{x}_i)^2}{2\kappa^2 \left( \|\mathbf{u}_{i,r}\|_2^2 \|\mathbf{u}_{i,r'}\|_2^2 - (\mathbf{u}_{i,r}^\top \mathbf{u}_{i,r'})^2 \right)} \right)
$$

$$
T_{i,r,r'} = \mathbf{w}_{r'}^\top \mathbf{x}_i \cdot \boldsymbol{\Phi}_1 \left( \frac{\|\mathbf{u}_{i,r}\|_2^2 \cdot \mathbf{w}_{r'}^\top \mathbf{x}_i - \mathbf{u}_{i,r}^\top \mathbf{u}_{i,r'} \cdot \mathbf{w}_r^\top \mathbf{x}_i}{\kappa \|\mathbf{u}_{i,r}\|_2 \left( \|\mathbf{u}_{i,r}\|_2^2 \|\mathbf{u}_{i,r'}\|_2^2 - (\mathbf{u}_{i,r}^\top \mathbf{u}_{i,r'})^2 \right)^{\frac{1}{2}}} \right)
$$

$$
G_{i,r} = \|\mathbf{u}_{i,r}\|_2 \exp \left( -\frac{(\mathbf{w}_r^\top \mathbf{x}_i)^2}{2\kappa^2 \|\mathbf{u}_r\|_2^2} \right)
$$

Therefore, the proof of the theorem relies on the upper bound of $C_{i,r,r'}, E_{i,r,r'}, T_{i,r,r'}$ and $G_{i,r}$. To upper-bound $C_{i,r,r'}$, we utilize the result in  that

$$
|C(a,b,\rho)| \le \frac{|\arcsin \rho|}{2\pi} \exp \left( -\frac{a^2 - 2\rho ab + b^2}{2(1-\rho^2)} \right) \le \frac{|\rho|}{4} \exp \left( -\frac{a^2 + b^2}{4} \right)
$$

where we used Lemma D.19 that $|\arcsin x| \leq \frac{\pi}{2} \cdot |x|$. Plugging in $a = \frac{\mathbf{w}_r^\top \mathbf{x}_i}{\kappa \|\mathbf{u}_{i,r}\|_2}, b = \frac{\mathbf{w}_{r'}^\top \mathbf{x}_i}{\kappa \|\mathbf{u}_{i,r'}\|_2}$ and $\rho = \frac{\mathbf{u}_{i,r}^\top \mathbf{u}_{i,r'}}{\|\mathbf{u}_{i,r}\|_2 \|\mathbf{u}_{i,r'}\|_2}$ gives

$$
\begin{aligned}
|C_{i,r,r'}| &\leq \left|\left(\mathbf{w}_r^\top \mathbf{x}_i\right)\left(\mathbf{w}_{r'}^\top \mathbf{x}_i\right) + \kappa^2 \mathbf{u}_{i,r}^\top \mathbf{u}_{i,r'}\right| \cdot \frac{\left|\mathbf{u}_{i,r}^\top \mathbf{u}_{i,r'}\right|}{4 \|\mathbf{u}_{i,r}\|_2 \|\mathbf{u}_{i,r'}\|_2} \exp\left(-\frac{1}{4\kappa^2}\left(\frac{\left(\mathbf{w}_r^\top \mathbf{x}_i\right)^2}{\|\mathbf{u}_{i,r}\|_2^2} + \frac{\left(\mathbf{w}_{r'}^\top \mathbf{x}_i\right)^2}{\|\mathbf{u}_{i,r'}\|_2^2}\right)\right) \\
&\leq \frac{1}{4}\left(\left|\left(\mathbf{w}_r^\top \mathbf{x}_i\right)\left(\mathbf{w}_{r'}^\top \mathbf{x}_i\right)\right| + \kappa^2 \|\mathbf{u}_{i,r}\|_2 \|\mathbf{u}_{i,r'}\|_2\right) \phi\left(\frac{\mathbf{w}_r^\top \mathbf{x}_i}{2\kappa \|\mathbf{u}_{i,r}\|_2}\right) \phi\left(\frac{\mathbf{w}_{r'}^\top \mathbf{x}_i}{2\kappa \|\mathbf{u}_{i,r'}\|_2}\right) \\
&= \frac{\kappa^2}{4} \|\mathbf{u}_{i,r}\|_2 \|\mathbf{u}_{i,r'}\|_2 \left(\frac{\left|\mathbf{w}_r^\top \mathbf{x}_i\right|}{\kappa \|\mathbf{u}_{i,r}\|_2} \cdot \frac{\left|\mathbf{w}_{r'}^\top \mathbf{x}_i\right|}{\kappa \|\mathbf{u}_{i,r}\|_2} + 1\right) \phi\left(\frac{\mathbf{w}_r^\top \mathbf{x}_i}{2\kappa \|\mathbf{u}_{i,r}\|_2}\right) \phi\left(\frac{\mathbf{w}_{r'}^\top \mathbf{x}_i}{2\kappa \|\mathbf{u}_{i,r'}\|_2}\right) \\
&= \frac{\kappa^2}{4} \|\mathbf{u}_{i,r}\|_2 \|\mathbf{u}_{i,r'}\|_2 \left(\psi\left(\frac{\mathbf{w}_r^\top \mathbf{x}_i}{2\kappa \|\mathbf{u}_{i,r}\|_2}\right) \psi\left(\frac{\mathbf{w}_{r'}^\top \mathbf{x}_i}{2\kappa \|\mathbf{u}_{i,r'}\|_2}\right) + \phi\left(\frac{\mathbf{w}_r^\top \mathbf{x}_i}{2\kappa \|\mathbf{u}_{i,r}\|_2}\right) \phi\left(\frac{\mathbf{w}_{r'}^\top \mathbf{x}_i}{2\kappa \|\mathbf{u}_{i,r'}\|_2}\right)\right)
\end{aligned}
$$

where we use the definition $P_{i,r} = \left|\mathbf{w}_r^\top \mathbf{x}_i\right| \cdot \exp\left(-\frac{\left(\mathbf{w}_r^\top \mathbf{x}_i\right)^2}{4\kappa^2 \|\mathbf{u}_{i,r}\|_2^2}\right)$. For the term $E_{i,r,r'}$, we notice that by letting $a = \frac{\mathbf{w}_r^\top \mathbf{x}_i}{\kappa \|\mathbf{u}_{i,r}\|_2}, b = \frac{\mathbf{w}_{r'}^\top \mathbf{x}_i}{\kappa \|\mathbf{u}_{i,r'}\|_2}$ and $\rho = \frac{\mathbf{u}_{i,r}^\top \mathbf{u}_{i,r'}}{\|\mathbf{u}_{i,r}\|_2 \|\mathbf{u}_{i,r'}\|}$, we have

$$
\exp\left(-\frac{\|\mathbf{u}_{i,r'}\|_2^2 \left(\mathbf{w}_r^\top \mathbf{x}_i\right)^2 - 2\left(\mathbf{u}_{i,r}^\top \mathbf{u}_{i,r'}\right)\left(\mathbf{w}_r^\top \mathbf{x}_i\right)\left(\mathbf{w}_{r'}^\top \mathbf{x}_i\right) + \|\mathbf{u}_{i,r}\|_2^2 \left(\mathbf{w}_{r'}^\top \mathbf{x}_i\right)^2}{2\kappa^2 \left(\|\mathbf{u}_{i,r}\|_2^2 \|\mathbf{u}_{i,r'}\|_2^2 - \left(\mathbf{u}_{i,r}^\top \mathbf{u}_{i,r'}\right)^2\right)}\right) = \exp\left(-\frac{a^2 - 2\rho ab + b^2}{2\left(1 - \rho^2\right)}\right)
$$

Using $\exp\left(-\frac{a^2 - 2\rho ab + b^2}{2(1-\rho^2)}\right) \leq \exp\left(-\frac{a^2 + b^2}{4}\right)$, we have that

$$
\begin{aligned}
|E_{i,r,r'}| &\leq \|\mathbf{u}_{i,r}\|_2 \|\mathbf{u}_{i,r'}\|_2 \exp\left(-\frac{1}{4\kappa^2}\left(\frac{\left(\mathbf{w}_r^\top \mathbf{x}_i\right)^2}{\|\mathbf{u}_{i,r}\|_2^2} + \frac{\left(\mathbf{w}_{r'}^\top \mathbf{x}_i\right)^2}{\|\mathbf{u}_{i,r'}\|_2^2}\right)\right) \\
&= \|\mathbf{u}_{i,r}\|_2 \|\mathbf{u}_{i,r'}\|_2 \phi\left(\frac{\mathbf{w}_r^\top \mathbf{x}_i}{2\kappa \|\mathbf{u}_{i,r}\|_2}\right) \phi\left(\frac{\mathbf{w}_{r'}^\top \mathbf{x}_i}{2\kappa \|\mathbf{u}_{i,r'}\|_2}\right)
\end{aligned}
$$

Therefore, we have

$$
\begin{aligned}
&\left|C_{i,r,r'} + \frac{\kappa^2}{2\pi} E_{i,r,r'}\right| \\
&\leq \frac{\kappa^2}{4} \|\mathbf{u}_{i,r}\|_2 \|\mathbf{u}_{i,r'}\|_2 \left(\psi\left(\frac{\mathbf{w}_r^\top \mathbf{x}_i}{2\kappa \|\mathbf{u}_{i,r}\|_2}\right) \psi\left(\frac{\mathbf{w}_{r'}^\top \mathbf{x}_i}{2\kappa \|\mathbf{u}_{i,r'}\|_2}\right) + \phi\left(\frac{\mathbf{w}_r^\top \mathbf{x}_i}{2\kappa \|\mathbf{u}_{i,r}\|_2}\right) \phi\left(\frac{\mathbf{w}_{r'}^\top \mathbf{x}_i}{2\kappa \|\mathbf{u}_{i,r'}\|_2}\right)\right)
\end{aligned}
$$

This gives that

$$
\begin{aligned}
&\left|\sum_{r,r'=1}^m a_r a_{r'}\left(C_{i,r,r'} + \frac{\kappa^2}{2\pi} E_{i,r,r'}\right)\right| \\
&\leq \frac{\kappa^2}{4}\left(\left(\sum_{r=1}^m \|\mathbf{u}_{i,r}\|_2 \psi\left(\frac{\mathbf{w}_r^\top \mathbf{x}_i}{2\kappa \|\mathbf{u}_{i,r}\|_2}\right)\right)^2 + \left(\sum_{r=1}^m \|\mathbf{u}_{i,r}\|_2 \phi\left(\frac{\mathbf{w}_r^\top \mathbf{x}_i}{2\kappa \|\mathbf{u}_{i,r}\|_2}\right)\right)^2\right)
\end{aligned}
\tag{21}
$$

By definition, we have $\|\mathbf{u}_{i,r}\|_2 \leq R_{\mathbf{u}}$. Therefore

$$
\left|\sum_{r,r'=1}^m a_r a_{r'}\left(C_{i,r,r'} + \frac{\kappa^2}{2\pi} E_{i,r,r'}\right)\right| \leq \frac{1}{4}\kappa^2 R_{\mathbf{u}}^2 \left(\left(\sum_{r=1}^m \psi\left(\frac{\mathbf{w}_r^\top \mathbf{x}_i}{2\kappa \|\mathbf{u}_{i,r}\|_2}\right)\right)^2 + \left(\sum_{r=1}^m \phi\left(\frac{\mathbf{w}_r^\top \mathbf{x}_i}{2\kappa \|\mathbf{u}_{i,r}\|_2}\right)\right)^2\right)
$$

Next, we focus on the term $T_{i,r,r'}$ and $G_{i,r}$. By the property of CDF, we have that $|T_{i,r,r'}| \leq \left|\mathbf{w}_{r'}^\top \mathbf{x}_i\right| \leq \|\mathbf{w}_r\|_2$. Therefore

$$\left|\frac{1}{\sqrt{m}}\sum_{r'=1}^m a_{r'}T_{i,r,r'} - y_i\right| \leq \frac{1}{\sqrt{m}}\sum_{r'=1}^m \|\mathbf{w}_{r'}\|_2 + |y_i| \leq \sqrt{m}R_\mathbf{w} + B_y \leq 2\sqrt{m}R_\mathbf{w}$$

where we applied $\|\mathbf{w}_r\|_2 \leq R_\mathbf{w}$ and $B_y \leq 3\sqrt{m}R_\mathbf{w}$. Thus

$$\left|\sum_{r=1}^m a_r G_{i,r}\left(\frac{1}{\sqrt{m}}\sum_{r'=1}^m a_{r'}T_{i,r,r'} - y_i\right)\right| \leq \sum_{r=1}^m G_{i,r}\cdot 2\sqrt{m}R_\mathbf{w} \leq \sqrt{m}R_\mathbf{w}\sum_{r=1}^m \phi\left(\frac{\mathbf{w}_r^\top \mathbf{x}_i}{2\kappa\|\mathbf{u}_{i,r}\|_2}\right)$$

where we used $\|\mathbf{u}_{i,r}\|_2 \leq R_\mathbf{u}$. Combining the inequality above and (21), we have

$$|\mathcal{E}| \leq \frac{n\kappa^2 R_\mathbf{u}^2}{4m}\left(\left(\sum_{r=1}^m \psi\left(\frac{\mathbf{w}_r^\top \mathbf{x}_i}{2\kappa\|\mathbf{u}_{i,r}\|_2}\right)\right)^2 + \left(\sum_{r=1}^m \phi\left(\frac{\mathbf{w}_r^\top \mathbf{x}_i}{2\kappa\|\mathbf{u}_{i,r}\|_2}\right)\right)^2\right) + \frac{n\kappa R_\mathbf{w}}{2}\sum_{r=1}^m \phi\left(\frac{\mathbf{w}_r^\top \mathbf{x}_i}{2\kappa\|\mathbf{u}_{i,r}\|_2}\right)$$

Applying the definition of $\psi_{\max}$ and $\phi_{\max}$ gives the desired results. $\qquad\square$

### B.3 Proof of Theorem 4.9

*Proof.* By the form of the gradient, we have

$$\mathbb{E}_{\mathbf{C}_k}\left[\nabla_{\mathbf{w}_r}\mathcal{L}_\mathbf{C}(\boldsymbol{\theta})\right] = \frac{a_r}{\sqrt{m}}\sum_{i=1}^n \mathbb{E}_\mathbf{C}\left[(f(\boldsymbol{\theta}, \mathbf{x}_i \odot \mathbf{c}_i) - y_i)\mathbf{x}_i \odot \mathbf{c}_i \mathbb{I}\{\langle \mathbf{w}_r, \mathbf{x}_i \odot \mathbf{c}_i\rangle \geq 0\}\right]$$

$$= \frac{a_r}{\sqrt{m}}\sum_{i=1}^n \underbrace{\mathbb{E}_{\mathbf{c}_i}\left[f(\boldsymbol{\theta}, \mathbf{x}_i \odot \mathbf{c}_i)\mathbf{x}_i \odot \mathbf{c}_i \mathbb{I}\{\langle \mathbf{w}_r, \mathbf{x}_i \odot \mathbf{c}_i\rangle \geq 0\}\right]}_{\mathcal{T}_{1,i}} \qquad (22)$$

$$- \frac{a_r}{\sqrt{m}}\sum_{i=1}^n y_i \underbrace{\mathbb{E}_{\mathbf{c}_i}\left[\mathbf{x}_i \odot \mathbf{c}_i \mathbb{I}\{\langle \mathbf{w}_r, \mathbf{x}_i \odot \mathbf{c}_i\rangle \geq 0\}\right]}_{\mathcal{T}_{2,i}}$$

Let $\mathbf{u}_{r,i} = \mathbf{w}_r \odot \mathbf{x}_i$. For $\mathcal{T}_{1,i}$, we further have

$$\mathcal{T}_{1,i} = \frac{1}{\sqrt{m}}\sum_{r'=1}^m a_{r'}\mathbb{E}_{\mathbf{c}_i}\left[\sigma\left(\mathbf{w}_{r'}^\top(\mathbf{x}_i \odot \mathbf{c}_i)\right)\mathbf{x}_i \odot \mathbf{c}_i \mathbb{I}\{\mathbf{w}_r^\top(\mathbf{x}_i \odot \mathbf{c}_i) \geq 0\}\right]$$

$$= \frac{1}{\sqrt{m}}\sum_{r'=1}^m a_{r'}\mathbb{E}_{\mathbf{c}_i}\left[(\mathbf{x}_i \odot \mathbf{c}_i)(\mathbf{x}_i \odot \mathbf{c}_i)^\top \mathbf{w}_{r'}\mathbb{I}\{\mathbf{w}_r^\top(\mathbf{x}_i \odot \mathbf{c}_i) \geq 0; \mathbf{w}_{r'}^\top(\mathbf{x}_i \odot \mathbf{c}_i) \geq 0\}\right]$$

$$= \frac{1}{\sqrt{m}}\sum_{r'=1}^m a_{r'}\left(\mathbb{E}_{\mathbf{c}_i}\left[\mathbf{c}_i\mathbf{c}_i^\top \mathbb{I}\{\mathbf{w}_r^\top(\mathbf{x}_i \odot \mathbf{c}_i) \geq 0; \mathbf{w}_{r'}^\top(\mathbf{x}_i \odot \mathbf{c}_i) \geq 0\}\right] \odot (\mathbf{x}_i\mathbf{x}_i^\top)\right)\mathbf{w}_{r'}$$

$$= \frac{1}{\sqrt{m}}\sum_{r'=1}^m a_{r'}\left(\mathbb{E}_{\mathbf{c}_i}\left[\mathbf{c}_i\mathbf{c}_i^\top \mathbb{I}\{\mathbf{u}_{r,i}^\top\mathbf{c}_i \geq 0; \mathbf{u}_{r',i}^\top\mathbf{c}_i \geq 0\}\right] \odot (\mathbf{x}_i\mathbf{x}_i^\top)\right)\mathbf{w}_{r'}$$

$$= \frac{1}{\sqrt{m}}\sum_{r'=1}^m a_{r'}\mathrm{Diag}(\mathbf{x})_i\,\mathbb{E}_{\mathbf{c}_i}\left[\mathbf{c}_i\mathbf{c}_i^\top \mathbb{I}\{\mathbf{u}_{r,i}^\top\mathbf{c}_i \geq 0; \mathbf{u}_{r',i}^\top\mathbf{c}_i \geq 0\}\right]\mathbf{u}_{r',i}$$

For $\mathcal{T}_{2,i}$, we can easily obtain

$$\mathcal{T}_{2,i} = \mathbb{E}_{\mathbf{c}_i}\left[\mathbf{c}_i\mathbb{I}\{\mathbf{u}_{r,i}^\top\mathbf{c}_i \geq 0\}\right] \odot \mathbf{x}_i$$

Abstractly, we are thus interested in the following quantity:

$$\mathbb{E}_\mathbf{c}\left[\mathbf{c}\mathbf{c}^\top\mathbb{I}\{\mathbf{c}^\top\mathbf{u} \geq 0; \mathbf{c}^\top\mathbf{v} \geq 0\}\right]; \quad \mathbb{E}_\mathbf{c}\left[\mathbf{c}\mathbb{I}\{\mathbf{c}^\top\mathbf{u} \geq 0\}\right]$$

where $\mathbf{c} \sim \mathcal{N}(\boldsymbol{\mu}, \kappa^2\mathbf{I})$, and $\mathbf{u}, \mathbf{v}$ are fixed vectors. Let $z_1 = \mathbf{c}^\top\mathbf{u}$ and $z_2 = \mathbf{c}^\top\mathbf{v}$. Then we have

$$z_1 \sim \mathcal{N}\left(\mathbf{c}^\top\mathbf{u}, \kappa^2\|\mathbf{u}\|_2^2\right); \quad z_2 \sim \mathcal{N}\left(\mathbf{c}^\top\mathbf{v}, \kappa^2\|\mathbf{v}\|_2^2\right)$$

According to Lemma D.11 and Lemma D.12, and by defining $\boldsymbol{\mu} = \mathbf{1}$, we have that $\boldsymbol{\Delta}_{k,r,i}^{(1)} \in \mathbb{R}^d$ and $\boldsymbol{\Delta}_{k,r,r',i}^{(2)} \in \mathbb{R}^{d \times d}$ defined below

$$\boldsymbol{\Delta}_{k,r,i}^{(1)} := \mathbb{E}_{\mathbf{c}_i} \left[ \mathbf{c}_i \mathbb{I} \left\{ \mathbf{c}_i^\top \mathbf{u}_{r,i} \geq 0 \right\} \right] - \mathbf{1} \cdot \boldsymbol{\Phi}_1 \left( \frac{\mathbf{w}_r^\top \mathbf{x}_i}{\kappa \left\| \mathbf{w}_r \odot \mathbf{x}_i \right\|_2} \right)$$

$$\boldsymbol{\Delta}_{k,r,r',i} := \mathbb{E}_{\mathbf{c}} \left[ \mathbf{c}\mathbf{c}^\top \mathbb{I} \left\{ \mathbf{u}_{r,i}^\top \mathbf{c}_i \geq 0; \mathbf{u}_{r',i}^\top \mathbf{c}_i \geq 0 \right\} \right] \mathbf{u}_{r',i}$$
$$- \left( \mathbf{1}\mathbf{1}^\top \mathbf{u}_{r',i} + 3\kappa^2 \mathbf{u}_{r',i} \right) \boldsymbol{\Phi}_1 \left( \frac{\mathbf{w}_r^\top \mathbf{x}_i}{\kappa \left\| \mathbf{w}_r \odot \mathbf{x}_i \right\|_2} \right) \boldsymbol{\Phi}_1 \left( \frac{\mathbf{w}_{r'}^\top \mathbf{x}_i}{\kappa \left\| \mathbf{w}_{r'} \odot \mathbf{x}_i \right\|_2} \right)$$

satisfies

$$\left\| \boldsymbol{\Delta}_{r,i}^{(1)} \right\|_\infty \leq \kappa R_{\mathbf{u}} \phi_{\max}$$
$$\left\| \boldsymbol{\Delta}_{r,r',i}^{(2)} \right\|_\infty \leq 4\kappa \left\| \mathbf{v} \right\|_2 \left( \sqrt{d} \phi_{\max} + \psi_{\max} \right)$$

Here we used $\left\| \boldsymbol{\mu} \right\|_\infty = 1$ and $\boldsymbol{\mu}^\top \left( \mathbf{w}_r \odot \mathbf{x}_i \right) = \mathbf{w}_{k,r}^\top \mathbf{x}_i$ when $\boldsymbol{\mu} = \mathbf{1}$. Therefore, for $\mathcal{T}_{1,i}$, we have

$$\mathcal{T}_{1,i} = \frac{1}{\sqrt{m}} \sum_{r'=1}^{m} a_{r'} \mathrm{Diag}\left( \mathbf{x}_i \right) \left( \left( \mathbf{w}_{r'}^\top \mathbf{x}_i \cdot \mathbf{1} \right) \boldsymbol{\Phi}_1 \left( \frac{\mathbf{w}_{r'}^\top \mathbf{x}_i}{\kappa \left\| \mathbf{w}_{r'} s \odot \mathbf{x}_i \right\|_2} \right) \boldsymbol{\Phi}_1 \left( \frac{\mathbf{w}_{r'}^\top \mathbf{x}_i}{\kappa \left\| \mathbf{w}_{r'} \odot \mathbf{x}_i \right\|_2} \right) + \boldsymbol{\Delta}_{r,r',i}^{(2)} \right)$$
$$+ \frac{3\kappa^2}{\sqrt{m}} \sum_{r'=1}^{m} a_{r'} \mathrm{Diag}\left( \mathbf{x}_i \right) \left( \mathbf{w}_r \odot \mathbf{x}_i \right) \boldsymbol{\Phi}_1 \left( \frac{\mathbf{w}_r^\top \mathbf{x}_i}{\kappa \left\| \mathbf{w}_r \odot \mathbf{x}_i \right\|_2} \right) \boldsymbol{\Phi}_1 \left( \frac{\mathbf{w}_{r'}^\top \mathbf{x}_i}{\kappa \left\| \mathbf{w}_{r'} \odot \mathbf{x}_i \right\|_2} \right)$$
$$= \frac{1}{\sqrt{m}} \sum_{r'=1}^{m} a_{r'} \mathbf{w}_{r'}^\top \mathbf{x}_i \cdot \mathbf{x}_i \boldsymbol{\Phi}_1 \left( \frac{\mathbf{w}_{r'}^\top \mathbf{x}_i}{\kappa \left\| \mathbf{w}_{r'} s \odot \mathbf{x}_i \right\|_2} \right) \boldsymbol{\Phi}_1 \left( \frac{\mathbf{w}_{r'}^\top \mathbf{x}_i}{\kappa \left\| \mathbf{w}_{r'} \odot \mathbf{x}_i \right\|_2} \right) + \frac{1}{\sqrt{m}} \sum_{r'=1}^{m} a_{r'} \left( \mathbf{x}_i \odot \boldsymbol{\Delta}_{r,r',i}^{(2)} \right)$$
$$+ \frac{3\kappa^2}{\sqrt{m}} \sum_{r'=1}^{m} a_{r'} \mathrm{Diag}\left( \mathbf{x}_i \right)^2 \mathbf{w}_{r'} \boldsymbol{\Phi}_1 \left( \frac{\mathbf{w}_{r'}^\top \mathbf{x}_i}{\kappa \left\| \mathbf{w}_{r'} s \odot \mathbf{x}_i \right\|_2} \right) \boldsymbol{\Phi}_1 \left( \frac{\mathbf{w}_{r'}^\top \mathbf{x}_i}{\kappa \left\| \mathbf{w}_{r'} \odot \mathbf{x}_i \right\|_2} \right)$$
$$= f\left( \boldsymbol{\theta}, \mathbf{x}_i \right) \mathbf{x}_i \mathbb{I} \left\{ \mathbf{w}_r^\top \mathbf{x}_i \geq 0 \right\} + \frac{3\kappa^2}{\sqrt{m}} \sum_{r'=1}^{m} a_{r'} \mathrm{Diag}\left( \mathbf{x}_i \right)^2 \mathbf{w}_{r'} \mathbb{I} \left\{ \mathbf{w}_r^\top \mathbf{x}_i \geq 0; \mathbf{w}_{r'}^\top \mathbf{x}_i \geq 0 \right\}$$
$$+ \frac{1}{\sqrt{m}} \sum_{r'=1}^{m} a_{r'} \left( \mathbf{x}_i \odot \boldsymbol{\Delta}_{r,r',i}^{(2)} \right) + \mathbf{g}_{1,i} + \mathbf{g}_{2,i}$$

where

$$\mathbf{g}_{1,i} = \frac{1}{\sqrt{m}} \sum_{r'=1}^{m} a_{r'} \mathbf{w}_{r'}^\top \mathbf{x}_i \cdot \mathbf{x}_i \left( \boldsymbol{\Phi}_1 \left( \frac{\mathbf{w}_{r'}^\top \mathbf{x}_i}{\kappa \left\| \mathbf{w}_{r'} s \odot \mathbf{x}_i \right\|_2} \right) \boldsymbol{\Phi}_1 \left( \frac{\mathbf{w}_{r'}^\top \mathbf{x}_i}{\kappa \left\| \mathbf{w}_{r'} \odot \mathbf{x}_i \right\|_2} \right) - \mathbb{I} \left\{ \mathbf{w}_r^\top \mathbf{x}_i \geq 0; \mathbf{w}_{r'}^\top \mathbf{x}_i \geq 0 \right\} \right)$$
$$\mathbf{g}_{2,i} = \frac{3\kappa^2}{\sqrt{m}} \sum_{r'=1}^{m} a_{r'} \mathrm{Diag}\left( \mathbf{x}_i \right)^2 \mathbf{w}_{r'} \left( \boldsymbol{\Phi}_1 \left( \frac{\mathbf{w}_{r'}^\top \mathbf{x}_i}{\kappa \left\| \mathbf{w}_{r'} s \odot \mathbf{x}_i \right\|_2} \right) \boldsymbol{\Phi}_1 \left( \frac{\mathbf{w}_{r'}^\top \mathbf{x}_i}{\kappa \left\| \mathbf{w}_{r'} \odot \mathbf{x}_i \right\|_2} \right) - \mathbb{I} \left\{ \mathbf{w}_r^\top \mathbf{x}_i \geq 0; \mathbf{w}_{r'}^\top \mathbf{x}_i \geq 0 \right\} \right)$$

Likely, for $\mathcal{T}_{2,i}$ we have

$$\mathcal{T}_{2,i} = \left( \mathbf{1} \boldsymbol{\Phi}_1 \left( \frac{\mathbf{w}_r^\top \mathbf{x}_i}{\kappa \left\| \mathbf{w}_r \odot \mathbf{x}_i \right\|_2} \right) + \boldsymbol{\Delta}_{r,i}^{(1)} \right) \odot \mathbf{x}_i$$
$$= \mathbf{x}_i \boldsymbol{\Phi}_1 \left( \frac{\mathbf{w}_r^\top \mathbf{x}_i}{\kappa \left\| \mathbf{w}_r \odot \mathbf{x}_i \right\|_2} \right) + \boldsymbol{\Delta}_{r,i}^{(1)} \odot \mathbf{x}_i$$
$$= \mathbf{x}_i \mathbb{I} \left\{ \mathbf{w}_r^\top \mathbf{x}_i \geq 0 \right\} + \boldsymbol{\Delta}_{r,i}^{(1)} \odot \mathbf{x}_i + \mathbf{g}_{3,i}$$

where $\mathbf{g}_{3,i} = \mathbf{x}_i \cdot \left( \mathbf{\Phi}_1 \left( \frac{\mathbf{w}_r^\top \mathbf{x}_i}{\kappa \| \mathbf{w}_r \odot \mathbf{x}_i \|_2} \right) - \mathbb{I} \left\{ \mathbf{w}_r^\top \mathbf{x}_i \geq 0 \right\} \right)$. Therefore, the final gradient is given by

$$
\begin{aligned}
\mathbb{E}_{\mathbf{C}} \left[ \nabla_{\mathbf{w}_r} \mathcal{L}_{\mathbf{C}} \left( \boldsymbol{\theta} \right) \right] = {} & \frac{a_r}{\sqrt{m}} \sum_{i=1}^{n} \left( f \left( \boldsymbol{\theta}, \mathbf{x}_i \right) - y_i \right) \mathbf{x}_i \mathbb{I} \left\{ \mathbf{w}_r^\top \mathbf{x}_i \right\} \\
& + \frac{a_r}{\sqrt{m}} \sum_{i=1}^{n} \left( \frac{1}{\sqrt{m}} \sum_{r'=1}^{m} a_{r'} \left( \mathbf{x}_i \odot \boldsymbol{\Delta}_{r,r',i}^{(2)} \right) + y_i \boldsymbol{\Delta}_{r,i}^{(1)} \odot \mathbf{x}_i \right) \\
& + \frac{3\kappa^2}{\sqrt{m}} \sum_{r'=1}^{m} a_{r'} \mathrm{Diag} \left( \mathbf{x}_i \right)^2 \mathbf{w}_{r'} \mathbb{I} \left\{ \mathbf{w}_r^\top \mathbf{x}_i \geq 0; \mathbf{w}_{r'}^\top \mathbf{x}_i \geq 0 \right\} \\
& + \frac{a_r}{\sqrt{m}} \sum_{i=1}^{n} \left( \mathbf{g}_{1,i} + \mathbf{g}_{2,i} - y_i \cdot \mathbf{g}_{3,i} \right) \\
= {} & \nabla_{\mathbf{w}_r} \mathcal{L} \left( \boldsymbol{\theta} \right) + \frac{3\kappa^2}{\sqrt{m}} \sum_{r'=1}^{m} a_{r'} \mathrm{Diag} \left( \mathbf{x}_i \right)^2 \mathbf{w}_{r'} \mathbb{I} \left\{ \mathbf{w}_r^\top \mathbf{x}_i \geq 0; \mathbf{w}_{r'}^\top \mathbf{x}_i \geq 0 \right\} \\
& + \underbrace{\frac{a_r}{\sqrt{m}} \sum_{i=1}^{n} \left( \frac{1}{\sqrt{m}} \sum_{r'=1}^{m} a_{r'} \left( \mathbf{x}_i \odot \boldsymbol{\Delta}_{r,r',i}^{(2)} \right) + y_i \boldsymbol{\Delta}_{r,i}^{(1)} \odot \mathbf{x}_i \right)}_{\mathbf{g}_4} \\
& + \frac{a_r}{\sqrt{m}} \sum_{i=1}^{n} \left( \mathbf{g}_{1,i} + \mathbf{g}_{2,i} - y_i \cdot \mathbf{g}_{3,i} \right)
\end{aligned}
$$

Notice that we can re-write $\mathbf{g}_{1,i}$ as

$$
\begin{aligned}
\mathbf{g}_{1,i} = {} & \mathbf{x}_i \left( \mathbf{\Phi}_1 \left( \frac{\mathbf{w}_r^\top \mathbf{x}_i}{\kappa \| \mathbf{w}_r \odot \mathbf{x}_i \|_2} \right) - \mathbb{I} \left\{ \mathbf{w}_r^\top \mathbf{x}_i \geq 0 \right\} \right) \cdot \frac{1}{\sqrt{m}} \sum_{r'=1}^{m} a_{r'} \mathbf{w}_{r'}^\top \mathbf{x}_i \mathbb{I} \left\{ \left( \mathbf{w}_{r'}^\top \mathbf{x}_i \geq 0 \right) \right\} \\
& + \mathbf{x}_i \mathbf{\Phi}_1 \left( \frac{\mathbf{w}_r^\top \mathbf{x}_i}{\kappa \| \mathbf{w}_r \odot \mathbf{x}_i \|_2} \right) \cdot \frac{1}{\sqrt{m}} \sum_{r'=1}^{m} a_{r'} \mathbf{w}_{r'}^\top \mathbf{x}_i \left( \mathbf{\Phi}_1 \left( \frac{\mathbf{w}_{r'}^\top \mathbf{x}_i}{\kappa \| \mathbf{w}_{r'} \odot \mathbf{x}_i \|_2} \right) - \mathbb{I} \left\{ \mathbf{w}_{r'}^\top \mathbf{x}_i \geq 0 \right\} \right) \\
= {} & \mathbf{x}_i \mathbf{\Phi}_1 \left( \frac{\mathbf{w}_r^\top \mathbf{x}_i}{\kappa \| \mathbf{w}_r \odot \mathbf{x}_i \|_2} \right) \cdot \frac{1}{\sqrt{m}} \sum_{r'=1}^{m} a_{r'} \mathbf{w}_{r'}^\top \mathbf{x}_i \left( \mathbf{\Phi}_1 \left( \frac{\mathbf{w}_{r'}^\top \mathbf{x}_i}{\kappa \| \mathbf{w}_{r'} \odot \mathbf{x}_i \|_2} \right) - \mathbb{I} \left\{ \mathbf{w}_{r'}^\top \mathbf{x}_i \geq 0 \right\} \right) \\
& + \mathbf{x}_i \left( \mathbf{\Phi}_1 \left( \frac{\mathbf{w}_r^\top \mathbf{x}_i}{\kappa \| \mathbf{w}_r \odot \mathbf{x}_i \|_2} \right) - \mathbb{I} \left\{ \mathbf{w}_r^\top \mathbf{x}_i \geq 0 \right\} \right) \cdot f \left( \boldsymbol{\theta}, \mathbf{x}_i \right)
\end{aligned}
$$

Then, by the definition of $\mathbf{g}_{3,i}$, we have that

$$
\begin{aligned}
\mathbf{g}_{1,i} - y_i \cdot \mathbf{g}_{3,i} = {} & \mathbf{x}_i \mathbf{\Phi}_1 \left( \frac{\mathbf{w}_r^\top \mathbf{x}_i}{\kappa \| \mathbf{w}_r \odot \mathbf{x}_i \|_2} \right) \cdot \frac{1}{\sqrt{m}} \sum_{r'=1}^{m} a_{r'} \mathbf{w}_{r'}^\top \mathbf{x}_i \left( \mathbf{\Phi}_1 \left( \frac{\mathbf{w}_{r'}^\top \mathbf{x}_i}{\kappa \| \mathbf{w}_{r'} \odot \mathbf{x}_i \|_2} \right) - \mathbb{I} \left\{ \mathbf{w}_{r'}^\top \mathbf{x}_i \geq 0 \right\} \right) \\
& + \left( f \left( \boldsymbol{\theta}, \mathbf{x}_i \right) - y_i \right) \mathbf{x}_i \left( \mathbf{\Phi}_1 \left( \frac{\mathbf{w}_r^\top \mathbf{x}_i}{\kappa \| \mathbf{w}_r \odot \mathbf{x}_i \|_2} \right) - \mathbb{I} \left\{ \mathbf{w}_r^\top \mathbf{x}_i \geq 0 \right\} \right)
\end{aligned}
$$

Using Lemma D.4, we have that

$$
\left| \mathbf{\Phi}_1 \left( a \right) - \mathbb{I} \left\{ a \geq 0 \right\} \right| \leq \exp \left( -\frac{a^2}{2} \right) \leq \phi \left( \frac{a}{2} \right)
$$

Therefore, we have that

$$\left\| \sum_{i=1}^{n} \left( \mathbf{g}_{1,i} - y_i \cdot \mathbf{g}_{3,i} \right) \right\|_2 \leq \frac{n}{\sqrt{m}} \left\| \sum_{r'=1}^{m} a_{r'} \mathbf{w}_{r'}^{\top} \mathbf{x}_i \left( \mathbf{\Phi}_1 \left( \frac{\mathbf{w}_{r'}^{\top} \mathbf{x}_i}{\kappa \left\| \mathbf{w}_{r'} \odot \mathbf{x}_i \right\|_2} \right) - \mathbb{I} \left\{ \mathbf{w}_{r'}^{\top} \mathbf{x}_i \geq 0 \right\} \right) \right\|_2$$

$$+ \left\| \sum_{i=1}^{n} \left( f \left( \boldsymbol{\theta}, \mathbf{x}_i \right) - y_i \right) \cdot \mathbf{x}_i \left( \mathbf{\Phi}_1 \left( \frac{\mathbf{w}_r^{\top} \mathbf{x}_i}{\kappa \left\| \mathbf{w}_r \odot \mathbf{x}_i \right\|_2} \right) - \mathbb{I} \left\{ \mathbf{w}_r^{\top} \mathbf{x}_i \geq 0 \right\} \right) \right\|_2$$

$$\leq \frac{n}{\sqrt{m}} \sum_{r'=1}^{m} \left| \mathbf{w}_{r'}^{\top} \mathbf{x}_i \right| \phi \left( \frac{\mathbf{w}_{r'}^{\top} \mathbf{x}_i}{2\kappa \left\| \mathbf{w}_{r'} \odot \mathbf{x}_i \right\|_2} \right) + \left\| \mathrm{Diag} \left( \mathbf{\Delta} \right) \mathbf{X} \left( f \left( \boldsymbol{\theta} - \mathbf{y} \right) \right) \right\|_2$$

$$= \kappa \sqrt{m} R_{\mathbf{u}} \psi_{\max} + \sigma_{\max} \left( \mathbf{X} \right) \phi_{\max} \mathcal{L} \left( \boldsymbol{\theta} \right)^{\frac{1}{2}}$$

Moreover, we can bound $\mathbf{g}_{2,i}$ as

$$\left\| \mathbf{g}_{2,i} \right\| \leq \frac{3\kappa^2}{\sqrt{m}} \sum_{r'=1}^{m} \left\| \mathbf{x} \right\|_{\infty}^2 \left\| \mathbf{w}_r \right\| \cdot 2\phi_{\max} \leq 6\kappa^2 \sqrt{m} B_{\mathbf{x}}^2 R_{\mathbf{w}} \phi_{\max}$$

Lastly, we can bound $\mathbf{g}_3$ as

$$\left\| \mathbf{g}_3 \right\|_2 \leq \frac{1}{m} \sum_{i=1}^{n} \sum_{r'=1}^{m} \left\| \mathbf{x}_i \odot \mathbf{\Delta}_{r,r',i}^{(2)} \right\| + \frac{1}{\sqrt{m}} \sum_{i=1}^{n} \left| y_i \right| \left\| \mathbf{\Delta}_{r,i}^{(1)} \odot \mathbf{x}_i \right\|_2$$

$$\leq \frac{1}{m} \sum_{i=1}^{n} \sum_{r'=1}^{m} \left\| \mathbf{\Delta}_{r,r',i}^{(2)} \right\|_{\infty} \left\| \mathbf{x}_i \right\|_2 + \frac{1}{\sqrt{m}} \sum_{i=1}^{n} \left| y_i \right| \left\| \mathbf{\Delta}_{r,r',i}^{(1)} \right\|_{\infty} \left\| \mathbf{x}_i \right\|_2$$

$$\leq \frac{1}{m} \sum_{i=1}^{n} \sum_{r'=1}^{m} \left\| \mathbf{\Delta}_{r,r',i}^{(2)} \right\|_{\infty} + \frac{B_y}{\sqrt{m}} \sum_{i=1}^{n} \left\| \mathbf{\Delta}_{r,r',i}^{(1)} \right\|_{\infty}$$

$$\leq 4n\kappa R_{\mathbf{u}} \left( \sqrt{d} \phi_{\max} + \psi_{\max} \right) + \frac{B_y}{\sqrt{m}} \cdot n\kappa R_{\mathbf{u}} \phi_{\max}$$

$$\leq 5n\kappa R_{\mathbf{u}} \left( \sqrt{d} \phi_{\max} + \psi_{\max} \right)$$

when $B_y \leq \sqrt{md}$. Therefore, we have that

$$\left\| \mathbb{E}_{\mathbf{C}} \left[ \nabla_{\mathbf{w}_r} \mathcal{L}_{\mathbf{C}} \left( \boldsymbol{\theta} \right) \right] - \left( \nabla_{\mathbf{w}_r} \mathcal{L} \left( \boldsymbol{\theta} \right) + \frac{3\kappa^2}{\sqrt{m}} \sum_{r'=1}^{m} a_{r'} \mathrm{Diag} \left( \mathbf{x}_i \right)^2 \mathbf{w}_{r'} \mathbb{I} \left\{ \mathbf{w}_r^{\top} \mathbf{x}_i \geq 0; \mathbf{w}_{r'}^{\top} \mathbf{x}_i \geq 0 \right\} \right) \right\|_2$$

$$\leq n\kappa R_{\mathbf{u}} \psi_{\max} + \frac{\sigma_{\max} \left( \mathbf{X} \right) \phi_{\max}}{\sqrt{m}} \mathcal{L} \left( \boldsymbol{\theta} \right)^{\frac{1}{2}} + 6n\kappa^2 B_{\mathbf{x}}^2 R_{\mathbf{w}} \phi_{\max} + 5n\kappa R_{\mathbf{u}} \left( \sqrt{d} \phi_{\max} + \psi_{\max} \right)$$

$$\leq \left( \frac{\sigma_{\max} \left( \mathbf{X} \right)}{\sqrt{m}} \mathcal{L} \left( \boldsymbol{\theta} \right)^{\frac{1}{2}} + 6n\kappa^2 B_{\mathbf{x}}^2 R_{\mathbf{w}} + 5n\kappa R_{\mathbf{u}} \sqrt{d} \right) \phi_{\max} + 6n\kappa R_{\mathbf{u}} \psi_{\max}$$

$\square$

## C  Proofs in Section 5

### C.1  Proof of Theorem 5.2

*Proof.* To start the proof, we define the following quantity in the standard NTK-based analysis of two-layer ReLU neural network. Let $R = C_1 \cdot \frac{\tau \lambda_0}{n}$ for some $C_1 > 0$, we define event $A_{i,r}$ and set $S_i, S_i^{\perp}$ as

$$A_{i,r} = \left\{ \exists \mathbf{w} \in \mathcal{B} \left( \mathbf{w}_{0,r}, R \right) : \mathbb{I} \left\{ \mathbf{w}_{0,r}^{\top} \mathbf{x}_i \geq 0 \right\} \neq \mathbb{I} \left\{ \mathbf{w}^{\top} \mathbf{x}_i \geq 0 \right\} \right\} \tag{23}$$

$$S_i = \left\{ r \in [m] : \neg A_{i,r} \right\}; \quad S_i^{\perp} = [m] \setminus S_i \tag{24}$$

Lemma 16 from    shows that with probability at least $1 - n \exp\left(-\frac{mR}{\tau}\right)$, we have that $\left|S_i^\perp\right| \leq \frac{4mR}{\tau}$. In the following of the proof, we assume that such event holds. Define $K' = \min\left\{k \in \mathbb{N} : \exists r \in [m] \text{ s.t. } \|\mathbf{w}_{k,r} - \mathbf{w}_{0,r}\|_2 > R\right\}$. Then for all $k < K'$, we have that $\mathbf{w}_{k,r} \in \mathcal{B}\left(\mathbf{w}_{0,r}, R\right)$. Fix any $k < K' - 1$. Consider the expansion of $\mathcal{L}\left(\boldsymbol{\theta}_{k+1}\right)$ as the following

$$
\begin{aligned}
\mathcal{L}\left(\boldsymbol{\theta}_{k+1}\right) &= \frac{1}{2} \sum_{i=1}^n \left(f\left(\boldsymbol{\theta}_{k+1}, \mathbf{x}_i\right) - y_i\right)^2 \\
&= \frac{1}{2} \sum_{i=1}^n \left(\left(f\left(\boldsymbol{\theta}_{k+1}, \mathbf{x}_i\right) - f\left(\boldsymbol{\theta}_k, \mathbf{x}_i\right)\right) + \left(f\left(\boldsymbol{\theta}_k, \mathbf{x}_i\right) - y_i\right)\right)^2 \\
&= \frac{1}{2} \sum_{i=1}^n \left(f\left(\boldsymbol{\theta}_{k+1}, \mathbf{x}_i\right) - f\left(\boldsymbol{\theta}_k, \mathbf{x}_i\right)\right)^2 + \sum_{i=1}^n \left(f\left(\boldsymbol{\theta}_{k+1}, \mathbf{x}_i\right) - f\left(\boldsymbol{\theta}_k, \mathbf{x}_i\right)\right)\left(f\left(\boldsymbol{\theta}_k, \mathbf{x}_i\right) - y_i\right) \\
&\quad + \frac{1}{2} \sum_{i=1}^n \left(f\left(\boldsymbol{\theta}_k, \mathbf{x}_i\right) - y_i\right)^2
\end{aligned}
\tag{25}
$$

We will analyze the three terms separately. To start, notice that

$$
\frac{1}{2} \sum_{i=1}^n \left(f\left(\boldsymbol{\theta}_k, \mathbf{x}_i\right) - y_i\right)^2 = \mathcal{L}\left(\boldsymbol{\theta}_k\right)
\tag{26}
$$

For the first term, by the definition of $f\left(\boldsymbol{\theta}, \mathbf{x}\right)$, we have

$$
\begin{aligned}
\left|f\left(\boldsymbol{\theta}_{k+1}, \mathbf{x}_i\right) - f\left(\boldsymbol{\theta}_k, \mathbf{x}_i\right)\right| &= \left|\frac{1}{\sqrt{m}} \sum_{r=1}^m a_r \left(\sigma\left(\mathbf{w}_{k+1,r}^\top \mathbf{x}_i\right) - \sigma\left(\mathbf{w}_{k,r}^\top \mathbf{x}_i\right)\right)\right| \\
&\leq \frac{1}{\sqrt{m}} \sum_{r=1}^m \left|\sigma\left(\mathbf{w}_{k+1,r}^\top \mathbf{x}_i\right) - \sigma\left(\mathbf{w}_{k,r}^\top \mathbf{x}_i\right)\right| \\
&\leq \frac{1}{\sqrt{m}} \sum_{r=1}^m \left|\left(\mathbf{w}_{k+1} - \mathbf{w}_k\right)^\top \mathbf{x}_i\right| \\
&\leq \frac{1}{\sqrt{m}} \sum_{r=1}^m \left\|\mathbf{w}_{k+1} - \mathbf{w}_k\right\| \\
&= \frac{\eta}{\sqrt{m}} \sum_{r=1}^m \left\|\nabla_{\mathbf{w}_r} \hat{\mathcal{L}}\left(\boldsymbol{\theta}_k, \boldsymbol{\xi}_k\right)\right\|_2
\end{aligned}
$$

where in the first inequality we use the fact that $a = \pm 1$, and in the second inequality we use the 1-Lipschitzness of ReLU. Applying Assumption 5.1, we have that

$$
\begin{aligned}
\sum_{i=1}^n \left(f\left(\boldsymbol{\theta}_{k+1}, \mathbf{x}_i\right) - f\left(\boldsymbol{\theta}_k, \mathbf{x}_i\right)\right)^2 &\leq \frac{\eta^2}{m} \sum_{i=1}^n \left(\sum_{r=1}^m \left\|\nabla_{\mathbf{w}_r} \hat{\mathcal{L}}\left(\boldsymbol{\theta}_k, \boldsymbol{\xi}_k\right)\right\|_2\right) \\
&\leq \frac{\eta^2 n}{m} \cdot \left(m \cdot \sqrt{\gamma \hat{\mathcal{L}}\left(\boldsymbol{\theta}_k, \boldsymbol{\xi}_k\right)}\right)^2 \\
&= \eta^2 m n \gamma \hat{\mathcal{L}}\left(\boldsymbol{\theta}_k, \boldsymbol{\xi}_k\right)
\end{aligned}
\tag{27}
$$

Lastly, to analyze the second term, we use the following definition of $I_{i,k}$ and $I_{i,k}^\perp$

$$
I_{i,k} = \frac{1}{\sqrt{m}} \sum_{r \in S_i} a_r \sigma\left(\mathbf{w}_{k,r}^\top \mathbf{x}_i\right); \quad I_{i,k}^\perp = \frac{1}{\sqrt{m}} \sum_{r \in S_i^\perp} a_r \sigma\left(\mathbf{w}_{k,r}^\top \mathbf{x}_i\right)
$$

Then we have that $f\left(\boldsymbol{\theta}_k, \mathbf{x}_i\right) = I_{i,k} + I_{i,k}^\perp$. Therefore

$$
f\left(\boldsymbol{\theta}_{k+1}, \mathbf{x}_i\right) - f\left(\boldsymbol{\theta}_k, \mathbf{x}_i\right) = \left(I_{i,k+1} - I_{i,k}\right) + \left(I_{i,k+1}^\perp - I_{i,k}^\perp\right)
$$

By the 1-Lipschitzness of ReLU, we have that

$$
\begin{aligned}
\left| I_{i,k+1}^{\perp} - I_{i,k}^{\perp} \right| &= \left| \frac{1}{\sqrt{m}} \sum_{r \in S_i^{\perp}} a_r \left( \sigma \left( \mathbf{w}_{k+1,r}^{\top} \mathbf{x}_i \right) - \sigma \left( \mathbf{w}_{k,r}^{\top} \mathbf{x}_i \right) \right) \right| \\
&\leq \frac{1}{\sqrt{m}} \sum_{r \in S_i^{\perp}} \left| \sigma \left( \mathbf{w}_{k+1,r}^{\top} \mathbf{x}_i \right) - \sigma \left( \mathbf{w}_{k,r}^{\top} \mathbf{x}_i \right) \right| \\
&\leq \frac{1}{\sqrt{m}} \sum_{r \in S_i^{\perp}} \left| \left( \mathbf{w}_{k+1,r} - \mathbf{w}_{k,r} \right)^{\top} \mathbf{x}_i \right| \\
&\leq \frac{\eta}{\sqrt{m}} \sum_{r \in S_i^{\perp}} \left\| \nabla_{\mathbf{w}_r} \hat{\mathcal{L}} \left( \boldsymbol{\theta}_k, \boldsymbol{\xi}_k \right) \right\|_2 \\
&\leq \frac{\eta \sqrt{\gamma}}{\sqrt{m}} \left| S_i^{\perp} \right| \hat{\mathcal{L}} \left( \boldsymbol{\theta}_k, \boldsymbol{\xi}_k \right)^{\frac{1}{2}}
\end{aligned}
$$

Applying $\left| S_i^{\perp} \right| \leq \frac{4mR}{\tau}$ gives $\left| I_{i,k+1}^{\perp} - I_{i,k}^{\perp} \right| \leq \frac{4\eta R}{\tau} \sqrt{\gamma m} \hat{\mathcal{L}} \left( \boldsymbol{\theta}_k, \boldsymbol{\xi}_k \right)^{\frac{1}{2}}$. This gives that

$$
\begin{aligned}
\sum_{i=1}^{n} &\left( f \left( \boldsymbol{\theta}_{k+1}, \mathbf{x}_i \right) - f \left( \boldsymbol{\theta}_k, \mathbf{x}_i \right) \right) \left( f \left( \boldsymbol{\theta}_k, \mathbf{x}_i \right) - y_i \right) \\
&= \sum_{i=1}^{n} \left( I_{i,k+1} - I_{i,k} \right) \left( f \left( \boldsymbol{\theta}_k, \mathbf{x}_i \right) - y_i \right) + \sum_{i=1}^{n} \left( I_{i,k+1}^{\perp} - I_{i,k}^{\perp} \right) \left( f \left( \boldsymbol{\theta}_k, \mathbf{x}_i \right) - y_i \right) \\
&\leq \sum_{i=1}^{n} \left( I_{i,k+1} - I_{i,k} \right) \left( f \left( \boldsymbol{\theta}_k, \mathbf{x}_i \right) - y_i \right) + \left( \sum_{i=1}^{n} \left( I_{i,k+1}^{\perp} - I_{i,k}^{\perp} \right)^2 \right)^{\frac{1}{2}} \left( \sum_{i=1}^{n} \left( f \left( \boldsymbol{\theta}_k, \mathbf{x}_i \right) - y_i \right)^2 \right)^{\frac{1}{2}} \\
&\leq \sum_{i=1}^{n} \left( I_{i,k+1} - I_{i,k} \right) \left( f \left( \boldsymbol{\theta}_k, \mathbf{x}_i \right) - y_i \right) + \frac{4\eta R}{\tau} \sqrt{\gamma m n} \hat{\mathcal{L}} \left( \boldsymbol{\theta}_k, \boldsymbol{\xi}_k \right)^{\frac{1}{2}} \mathcal{L} \left( \boldsymbol{\theta}_k \right)^{\frac{1}{2}}
\end{aligned}
\tag{28}
$$

Plugging (26), (27), and (28) into (25) gives

$$
\begin{aligned}
\mathcal{L} \left( \boldsymbol{\theta}_{k+1} \right) &\leq \mathcal{L} \left( \boldsymbol{\theta}_k \right) + \eta^2 m n \gamma \hat{\mathcal{L}} \left( \boldsymbol{\theta}_k, \boldsymbol{\xi}_k \right) + \frac{4\eta R}{\tau} \sqrt{\gamma m n} \hat{\mathcal{L}} \left( \boldsymbol{\theta}_k, \boldsymbol{\xi}_k \right)^{\frac{1}{2}} \mathcal{L} \left( \boldsymbol{\theta}_k \right)^{\frac{1}{2}} \\
&\quad + \sum_{i=1}^{n} \left( I_{i,k+1} - I_{i,k} \right) \left( f \left( \boldsymbol{\theta}_k, \mathbf{x}_i \right) - y_i \right)
\end{aligned}
$$

Under Jensen's inequality, we have that $\mathbb{E}_{\boldsymbol{\xi}_k} \left[ \hat{\mathcal{L}} \left( \boldsymbol{\theta}_k, \boldsymbol{\xi}_k \right)^{\frac{1}{2}} \right] \leq \mathbb{E}_{\boldsymbol{\xi}_k} \left[ \hat{\mathcal{L}} \left( \boldsymbol{\theta}_k, \boldsymbol{\xi}_k \right) \right]^{\frac{1}{2}}$. Using the property that

$$
\mathbb{E}_{\boldsymbol{\xi}_k} \left[ \hat{\mathcal{L}} \left( \boldsymbol{\theta}_k, \boldsymbol{\xi}_k \right) \right] \leq 2 \mathcal{L} \left( \boldsymbol{\theta}_k \right) + \varepsilon_1
$$

from Assumption 5.1, we can also obtain that

$$
\mathbb{E}_{\boldsymbol{\xi}_k} \left[ \hat{\mathcal{L}} \left( \boldsymbol{\theta}_k, \boldsymbol{\xi}_k \right)^{\frac{1}{2}} \right] \leq \left( 2 \mathcal{L} \left( \boldsymbol{\theta}_k \right) + \varepsilon_1 \right)^{\frac{1}{2}}
$$

Therefore, taking the expectation of $\mathcal{L}(\boldsymbol{\theta}_{k+1})$ gives

$$
\begin{aligned}
\mathbb{E}_{\boldsymbol{\xi}_k}[\mathcal{L}(\boldsymbol{\theta}_{k+1})] &\leq \mathcal{L}(\boldsymbol{\theta}_k) + \eta^2 mn\gamma\left(2\mathcal{L}(\boldsymbol{\theta}_k) + \varepsilon\right) + \frac{4\eta R}{\tau}\sqrt{\gamma mn}\left(2\mathcal{L}(\boldsymbol{\theta}_k) + \varepsilon_1\right)^{\frac{1}{2}}\mathcal{L}(\boldsymbol{\theta}_k)^{\frac{1}{2}} \\
&\quad + \sum_{i=1}^{n}\mathbb{E}_{\boldsymbol{\xi}_k}\left[I_{i,k+1} - I_{i,k}\right]\left(f(\boldsymbol{\theta}_k, \mathbf{x}_i) - y_i\right) \\
&\leq \mathcal{L}(\boldsymbol{\theta}_k) + \eta^2 mn\gamma\left(2\mathcal{L}(\boldsymbol{\theta}_k) + \varepsilon_1\right) + \frac{10\eta R}{\tau}\sqrt{\gamma mn}\mathcal{L}(\boldsymbol{\theta}_k) + \frac{4\eta R}{\tau}\sqrt{\gamma mn}\cdot\varepsilon_1 \\
&\quad + \sum_{i=1}^{n}\mathbb{E}_{\boldsymbol{\xi}_k}\left[I_{i,k+1} - I_{i,k}\right]\left(f(\boldsymbol{\theta}_k, \mathbf{x}_i) - y_i\right) \\
&= \left(1 + 2\eta^2 mn\gamma + 10C\eta\lambda_0\sqrt{\frac{\gamma m}{n}}\right)\mathcal{L}(\boldsymbol{\theta}_k) + \left(\eta^2 mn\gamma + 4C\eta\lambda_0\sqrt{\frac{\gamma m}{n}}\right)\varepsilon_1 \\
&\quad + \sum_{i=1}^{n}\mathbb{E}_{\boldsymbol{\xi}_k}\left[I_{i,k+1} - I_{i,k}\right]\left(f(\boldsymbol{\theta}_k, \mathbf{x}_i) - y_i\right)
\end{aligned}
\tag{29}
$$

where in the last inequality we use the property that $\sqrt{a(a+b)} \leq \frac{5}{4}a + b$. Recall that $\mathbf{w}_{k+1,r}, \mathbf{w}_{k,r} \in \mathcal{B}(\mathbf{w}_{0,r}, R)$. Therefore, for $r \in S_i$, we must have that $\mathbb{I}\left\{\mathbf{w}_{k+1,r}^\top \mathbf{x}_i \geq 0\right\} = \mathbb{I}\left\{\mathbf{w}_{0,r}^\top \mathbf{x}_i \geq 0\right\} = \mathbb{I}\left\{\mathbf{w}_{k,r}^\top \mathbf{x}_i \geq 0\right\}$. Thus, we have

$$
\begin{aligned}
\mathbb{E}_{\boldsymbol{\xi}_k}\left[I_{i,k+1} - I_{i,k}\right] &= \frac{1}{\sqrt{m}}\sum_{r \in S_i} a_r\mathbb{E}_{\boldsymbol{\xi}_k}\left[\mathbf{w}_{k+1,r} - \mathbf{w}_{k,r}\right]^\top \mathbf{x}_i\mathbb{I}\left\{\mathbf{w}_{k,r}^\top \mathbf{x}_i \geq 0\right\} \\
&= -\frac{\eta}{\sqrt{m}}\sum_{r \in S_i} a_r\mathbb{E}_{\boldsymbol{\xi}_k}\left[\nabla_{\mathbf{w}_r}\hat{\mathcal{L}}(\boldsymbol{\theta}_k, \boldsymbol{\xi}_k)\right]^\top \mathbf{x}_i\mathbb{I}\left\{\mathbf{w}_{k,r}^\top \mathbf{x}_i \geq 0\right\}
\end{aligned}
\tag{30}
$$

Let $\mathbf{g}_{k,r} = \mathbb{E}_{\boldsymbol{\xi}_k}\left[\nabla_{\mathbf{w}_r}\hat{\mathcal{L}}(\boldsymbol{\theta}_k, \boldsymbol{\xi}_k)\right] - \nabla_{\mathbf{w}_r}\mathcal{L}(\boldsymbol{\theta}_k)$. Then by Assumption 5.1 we have that $\|\mathbf{g}_{k,r}\|_2 \leq \varepsilon_3\mathcal{L}(\boldsymbol{\theta})^{\frac{1}{2}} + \varepsilon_2$. Using $\mathbf{g}_{k,r}$, we can write (30) as

$$
\mathbb{E}_{\boldsymbol{\xi}_k}\left[I_{i,k+1} - I_{i,k}\right] = -\eta\sum_{r \in S_i}\frac{a_r}{\sqrt{m}}\nabla_{\mathbf{w}_r}\mathcal{L}(\boldsymbol{\theta}_k)^\top \mathbf{x}_i\mathbb{I}\left\{\mathbf{w}_{k,r}^\top \mathbf{x}_i \geq 0\right\} - \frac{\eta}{\sqrt{m}}\sum_{r \in S_i}a_r\mathbf{g}_{k,r}^\top \mathbf{x}_i\mathbb{I}\left\{\mathbf{w}_{k,r}^\top \mathbf{x}_i \geq 0\right\}
\tag{31}
$$

By definition, we have

$$
\nabla_{\mathbf{w}_r}\mathcal{L}(\boldsymbol{\theta}_k) = \frac{a_r}{\sqrt{m}}\sum_{j=1}^{n}\left(f(\boldsymbol{\theta}_k, \mathbf{x}_i) - y_i\right)\mathbf{x}_j\mathbb{I}\left\{\mathbf{w}_{k,r}^\top \mathbf{x}_j \geq 0\right\}
$$

Therefore, we have that

$$
\frac{a_r}{\sqrt{m}}\nabla_{\mathbf{w}_r}\mathcal{L}(\boldsymbol{\theta}_k)^\top \mathbf{x}_i\mathbb{I}\left\{\mathbf{w}_{k,r}^\top \mathbf{x}_i \geq 0\right\} = \frac{1}{m}\sum_{j=1}^{n}\left(f(\boldsymbol{\theta}_k, \mathbf{x}_i) - y_i\right)\mathbf{x}_i^\top \mathbf{x}_j\mathbb{I}\left\{\mathbf{w}_{k,r}^\top \mathbf{x}_i \geq 0\mathbf{w}_{k,r}^\top \mathbf{x}_j \geq 0\right\}
$$

Combining with (31), we have that

$$
\sum_{i=1}^{n} \mathbb{E}_{\boldsymbol{\xi}_k} \left[ I_{i,k+1} - I_{i,k} \right] \left( f\left(\boldsymbol{\theta}_k, \mathbf{x}_i\right) - y_i \right)
$$

$$
= -\frac{\eta}{m} \sum_{i,j=1}^{n} \sum_{r \in S_i} \left( f\left(\boldsymbol{\theta}_k, \mathbf{x}_i\right) - y_i \right) \left( f\left(\boldsymbol{\theta}_k, \mathbf{x}_j\right) - y_j \right) \mathbf{x}_i^\top \mathbf{x}_j \mathbb{I}\left\{ \mathbf{w}_{k,r}^\top \mathbf{x}_i \geq 0; \mathbf{w}_{k,r}^\top \mathbf{x}_j \geq 0 \right\}
$$

$$
- \frac{\eta}{\sqrt{m}} \sum_{i=1}^{n} \sum_{r \in S_i} \left( f\left(\boldsymbol{\theta}_k, \mathbf{x}_i\right) - y_i \right) a_r \mathbf{g}_{k,r}^\top \mathbf{x}_i \mathbb{I}\left\{ \mathbf{w}_{k,r}^\top \mathbf{x}_i \geq 0 \right\}
$$

$$
\leq -\eta \sum_{i,j=1}^{n} \left( f\left(\boldsymbol{\theta}_k, \mathbf{x}_i\right) - y_i \right) \underbrace{\left( \frac{\mathbf{x}_i^\top \mathbf{x}_j}{m} \sum_{r \in S_i} \mathbb{I}\left\{ \mathbf{w}_{k,r}^\top \mathbf{x}_i \geq 0; \mathbf{w}_{k,r}^\top \mathbf{x}_j \geq 0 \right\} \right)}_{\mathbf{H}_{k,ij}} \left( f\left(\boldsymbol{\theta}_k, \mathbf{x}_j\right) - y_j \right)
$$

$$
+ \frac{\eta}{\sqrt{m}} \sum_{i=1}^{m} \sum_{r=1}^{n} \left| f\left(\boldsymbol{\theta}_k, \mathbf{x}_i\right) - y_i \right| \left\| \mathbf{g}_{k,r} \right\|_2
$$

$$
\leq -\eta \lambda_{\min}\left(\mathbf{H}_k\right) \sum_{i=1}^{n} \left( f\left(\boldsymbol{\theta}_k, \mathbf{x}_i\right) - y_i \right)^2 + \eta \varepsilon_2 \sqrt{mn} \left( \sum_{r=1}^{n} \left( f\left(\boldsymbol{\theta}_k, \mathbf{x}_i\right) - y_i \right)^2 \right)^{\frac{1}{2}} + \eta \varepsilon_3 \sqrt{mn} \mathcal{L}\left(\boldsymbol{\theta}_k\right)
$$

$$
= - \left( 2\eta \lambda_{\min}\left(\mathbf{H}_k\right) + \eta \varepsilon_3 \sqrt{mn} \right) \mathcal{L}\left(\boldsymbol{\theta}_k\right) + 2\eta \varepsilon_2 \sqrt{mn} \mathcal{L}\left(\boldsymbol{\theta}_k\right)^{\frac{1}{2}}
$$

Using the property that $ab \leq \frac{a^2}{2} + \frac{b^2}{2}$, we have that for any $C' > 0$,

$$
2\eta \varepsilon_2 \sqrt{mn} \mathcal{L}\left(\boldsymbol{\theta}_k\right)^{\frac{1}{2}} \leq C' \eta \lambda_0 \sqrt{\frac{\gamma m}{n}} + \frac{\eta \varepsilon_2^2 n}{C' \lambda_0} \sqrt{\frac{mn}{\gamma}}
$$

Moreover, by Lemma C.1, we have that when $m = \Omega\left( \frac{n^2}{\lambda_0^2} \log \frac{n}{\delta} \right)$, with probability at least $1 - \delta - n^2 \exp -\frac{mR}{\tau}$, it holds that

$$
\left\| \mathbf{H}_k - \mathbf{H}^\infty \right\|_F \leq \frac{\lambda_0}{6} + \frac{1}{m} \left( \sum_{i,j=1}^{n} \left| S_i^\perp \right|^2 \right)^{\frac{1}{2}} + \frac{2nR}{\tau}
$$

Plugging in $\left| S_i^\perp \right| \leq \frac{4mR}{\tau}$ and $R \leq C_1 \cdot \frac{\tau \lambda_0}{n}$, we have that $\left\| \mathbf{H}_k - \mathbf{H}^\infty \right\|_F \leq \frac{\lambda_0}{2}$ for small enough $C_1$. Thus, we have that $\lambda_{\min}\left(\mathbf{H}_k\right) \geq \frac{\lambda_0}{2}$. Therefore, we have that

$$
\sum_{i=1}^{n} \mathbb{E}_{\boldsymbol{\xi}_k} \left[ I_{i,k+1} - I_{i,k} \right] \left( f\left(\boldsymbol{\theta}_k, \mathbf{x}_i\right) - y_i \right) \leq \left( C' \eta \lambda_0 \sqrt{\frac{\gamma m}{n}} + \eta \varepsilon_3 \sqrt{mn} - \eta \lambda_0 \right) \mathcal{L}\left(\boldsymbol{\theta}_k\right) + \frac{\eta \varepsilon_2^2 n}{C' \lambda_0} \sqrt{\frac{mn}{\gamma}}
$$

Plugging this back into (31) gives

$$
\mathbb{E}_{\boldsymbol{\xi}_k} \left[ \mathcal{L}\left(\boldsymbol{\theta}_{k+1}\right) \right] \leq \left( 1 + 2\eta^2 mn\gamma + 10C\eta\lambda_0 \sqrt{\frac{\gamma m}{n}} \right) \mathcal{L}\left(\boldsymbol{\theta}_k\right) + \left( \eta^2 mn\gamma + 4C\eta\lambda_0 \sqrt{\frac{\gamma m}{n}} \right) \varepsilon_1
$$

$$
+ \left( C' \eta \lambda_0 \sqrt{\frac{\gamma m}{n}} + \eta \varepsilon_3 \sqrt{mn} - \eta \lambda_0 \right) \mathcal{L}\left(\boldsymbol{\theta}_k\right) + \frac{\eta \varepsilon_2^2 n}{C' \lambda_0} \sqrt{\frac{mn}{\gamma}}
$$

$$
= \left( 1 - \eta \lambda_0 + \eta \varepsilon_3 \sqrt{mn} + 2\eta^2 mn\gamma + \left(10C + C'\right) \eta\lambda_0 \sqrt{\frac{\gamma m}{n}} \right) \mathcal{L}\left(\boldsymbol{\theta}_k\right)
$$

$$
+ \left( \eta^2 mn\gamma + 4C\eta\lambda_0 \sqrt{\frac{\gamma m}{n}} \right) \varepsilon_1 + \frac{\eta \varepsilon_2^2 n}{C' \lambda_0} \sqrt{\frac{mn}{\gamma}}
$$

Apply $\varepsilon_3 \leq C_\varepsilon \cdot \frac{\lambda_0}{\sqrt{mn}}, \gamma = C_1 \cdot \frac{n}{m}$ and $\eta = C_2 \cdot \frac{\lambda_0}{n^2}$ gives

$$\mathbb{E}_{\boldsymbol{\xi}_k}\left[\mathcal{L}\left(\boldsymbol{\theta}_{k+1}\right)\right] \leq \left(1 - \left(1 - 2C_1 C_2 - \left(10C + C'\right)\sqrt{C_1}\right)\eta\lambda_0\right)\mathcal{L}\left(\boldsymbol{\theta}_k\right)$$
$$+ \eta\lambda_0\left(\frac{mn\varepsilon_2^2}{C'\sqrt{C_1}\lambda_0^2} + \left(C_1 C_2 + 4C\sqrt{C_1}\right)\varepsilon_1\right)$$

Choosing a small enough $C_1, C_2, C, C'$ gives

$$\mathbb{E}_{\boldsymbol{\xi}_k}\left[\mathcal{L}\left(\boldsymbol{\theta}_{k+1}\right)\right] \leq \left(1 - \frac{\eta\lambda_0}{2}\right)\mathcal{L}\left(\boldsymbol{\theta}_k\right) + \frac{1}{2}\hat{C}\eta\lambda_0\left(\frac{mn}{\lambda_0^2}\cdot\varepsilon_2^2 + \varepsilon_1\right) \tag{32}$$

for a large enough $\hat{C}$. Thus, unrolling the iterations gives

$$\mathbb{E}_{\boldsymbol{\xi}_0,\ldots,\boldsymbol{\xi}_{k-1}}\left[\mathcal{L}\left(\mathbf{W}_k\right)\right] \leq \left(1 - \frac{\eta\lambda_0}{2}\right)^k\mathcal{L}\left(\mathbf{W}_0\right) + \hat{C}\left(\frac{mn}{\lambda_0^2}\cdot\varepsilon_2^2 + \varepsilon_1\right) \tag{33}$$

for all $k < K'$. Next, we shall lower bound $K'$. For all $k \leq K'$, we have that

$$\left\|\mathbf{w}_{k,r} - \mathbf{w}_{0,r}\right\|_2 \leq \sum_{t=0}^{k-1}\left\|\mathbf{w}_{t+1,r} - \mathbf{w}_{t,r}\right\|_2 = \eta\sum_{t=0}^{k-1}\left\|\nabla_{\mathbf{w}_r}\hat{\mathcal{L}}\left(\mathbf{W}_t, \boldsymbol{\xi}_t\right)\right\|_2 \leq \eta\sqrt{\gamma}\sum_{t=0}^{k-1}\hat{\mathcal{L}}\left(\mathbf{W}_t, \boldsymbol{\xi}_t\right)^{\frac{1}{2}}$$

By (33), we have

$$\mathbb{E}_{\boldsymbol{\xi}_0,\ldots,\boldsymbol{\xi}_{t-1}}\left[\hat{\mathcal{L}}\left(\mathbf{W}_t, \boldsymbol{\xi}_t\right)^{\frac{1}{2}}\right] \leq \left(\mathcal{L}\left(\mathbf{W}_t\right) + \varepsilon_1\right)^{\frac{1}{2}}$$
$$\leq \left(2\left(1 - \frac{\eta\lambda_0}{2}\right)^t\mathcal{L}\left(\mathbf{W}_0\right) + \left(\hat{C} + 1\right)\left(\frac{mn}{\lambda_0^2}\cdot\varepsilon_2^2 + \varepsilon_1\right)\right)^{\frac{1}{2}}$$
$$\leq 2\left(1 - \frac{\eta\lambda_0}{4}\right)^t\mathcal{L}\left(\mathbf{W}_0\right)^{\frac{1}{2}} + \sqrt{\hat{C}+1}\left(\frac{\varepsilon_2}{\lambda_0}\sqrt{mn} + \sqrt{\varepsilon_1}\right)$$

Therefore, we have

$$\mathbb{E}_{\boldsymbol{\xi}_0,\ldots,\boldsymbol{\xi}_{k-1}}\left[\left\|\mathbf{w}_{k,r} - \mathbf{w}_{0,r}\right\|_2\right] \leq \eta\sqrt{\gamma}\sum_{t=0}^{k-1}\mathbb{E}_{\boldsymbol{\xi}_0,\ldots,\boldsymbol{\xi}_{t-1}}\left[\hat{\mathcal{L}}\left(\mathbf{W}_t, \boldsymbol{\xi}_t\right)^{\frac{1}{2}}\right]$$
$$\leq 2\eta\sqrt{\gamma}\mathcal{L}\left(\mathbf{W}_0\right)^{\frac{1}{2}}\sum_{t=0}^{\infty}\left(1 - \frac{\eta\lambda_0}{4}\right)^t + k\sqrt{\left(\hat{C}+1\right)\gamma}\left(\frac{\varepsilon_2}{\lambda_0}\sqrt{mn} + \sqrt{\varepsilon_1}\right)$$
$$= \frac{8\sqrt{\gamma}}{\lambda_0}\mathcal{L}\left(\mathbf{W}_0\right)^{\frac{1}{2}} + k\sqrt{\left(\hat{C}+1\right)\gamma}\left(\frac{\varepsilon_2}{\lambda_0}\sqrt{mn} + \sqrt{\varepsilon_1}\right)$$

By Lemma 26 in , we have that $\mathbb{E}_{\mathbf{W}_0, \mathbf{a}}\left[\mathcal{L}\left(\mathbf{W}_0\right)^2\right] = O\left(n\right)$. $\gamma = C_1 \cdot \frac{n}{m}$, we have that

$$\mathbb{E}_{\mathbf{W}_0, \mathbf{a}, \boldsymbol{\xi}_0,\ldots,\boldsymbol{\xi}_{k-1}}\left[\left\|\mathbf{w}_{k,r} - \mathbf{w}_{0,r}\right\|_2\right] \leq O\left(\frac{n}{\lambda_0}\sqrt{m}\right) + O\left(k\left(\frac{\varepsilon_2 n}{\lambda_0} + \sqrt{\frac{\varepsilon_1 n}{m}}\right)\right)$$

Thus, by Markov's inequality, we have that with probability at least $1 - \frac{\delta}{3K}$,

$$\left\|\mathbf{w}_{k,r} - \mathbf{w}_{0,r}\right\|_2 \leq \underbrace{O\left(\frac{nK}{\lambda_0\delta\sqrt{m}}\right)}_{\mathcal{T}_1} + \underbrace{O\left(\frac{K^2}{\delta}\left(\frac{\varepsilon_2 n}{\lambda_0} + \sqrt{\frac{\varepsilon_1 n}{m}}\right)\right)}_{\mathcal{T}_2}$$

Setting $m = \Omega\left(\frac{n^4}{\lambda_0^4\delta^2\tau^2}\right)$ guarantees that $T_1 \leq \frac{C_1}{2}\cdot\frac{\tau\lambda_0}{n} = \frac{R}{2}$ and set $\varepsilon_2 \leq O\left(\frac{\delta\lambda_0}{nK^2}\right), \varepsilon_1 \leq O\left(\frac{\delta m}{K^4 n}\right)$ gives that $T_2 \leq \frac{C_1}{2}\cdot\frac{\tau\lambda_0}{n} = \frac{R}{2}$. Combining the bound on $\mathcal{T}_1$ and $\mathcal{T}_2$ and taking a union bound gives that, with probability at least $1 - \frac{\delta}{3}$, it holds that

$$\left\|\mathbf{w}_{k,r} - \mathbf{w}_{0,r}\right\|_2 \leq R; \quad \forall k \in [K]$$

This shows that we must have $K' > K$, which completes the proof. $\qquad\square$

**Lemma C.1.** *Let $\mathbf{H}^\infty$ be defined in (11), and let $\mathbf{H}_k$ be defined as*

$$\mathbf{H}_{k,ij} = \frac{\mathbf{x}_i^\top \mathbf{x}_j}{m} \sum_{r \in S_i} \mathbb{I}\left\{\mathbf{w}_{k,r}^\top \mathbf{x}_i \geq 0; \mathbf{w}_{k,r}^\top \mathbf{x}_j \geq 0\right\}$$

*Fix any $R$. Assume that $\mathbf{w}_{k,r} \in \mathcal{B}\left(\mathbf{w}_{0,r}, R\right)$ for all $r \in [m]$. If $\mathbf{w}_{0,r} \sim \mathcal{N}\left(\mathbf{0}, \tau^2 \mathbf{I}\right)$, and $m = \Omega\left(\right)$, then we have that*

$$\|\mathbf{H}_k - \mathbf{H}^\infty\|_F \leq \frac{\lambda_0}{6} + \frac{1}{m^2} \sum_{i,j=1}^n \left|S_i^\perp\right|^2 + \frac{2nR}{\tau}$$

*Proof.* We define $\hat{\mathbf{H}}_k$ as follows

$$\hat{\mathbf{H}}_{k,ij} = \frac{\mathbf{x}_i^\top \mathbf{x}_j}{m} \sum_{r=1}^m \mathbb{I}\left\{\mathbf{w}_{k,r}^\top \mathbf{x}_i \geq 0; \mathbf{w}_{k,r}^\top \mathbf{x}_j \geq 0\right\}$$

Then we have that

$$\|\mathbf{H}_k - \mathbf{H}^\infty\|_F \leq \left\|\mathbf{H}_k - \hat{\mathbf{H}}_k\right\|_F + \left\|\hat{\mathbf{H}}_k - \hat{\mathbf{H}}_0\right\|_F + \left\|\hat{\mathbf{H}}_0 - \mathbf{H}^\infty\right\|_F$$

By Lemma 3.1 in Du et al. (2018), we have that with probability at least $1-\delta$, we have that $\left\|\hat{\mathbf{H}}_0 - \mathbf{H}^\infty\right\|_F \leq \frac{\lambda_0}{6}$ when $m = \Omega\left(\frac{n^2}{\lambda_0^2}\log\frac{n}{\delta}\right)$. By Lemma 3.2 in Song & Yang (2020), we have that with probability at least $1 - n^2 \exp{-\frac{mR}{\tau}}$, it holds that $\left\|\hat{\mathbf{H}}_k - \hat{\mathbf{H}}_0\right\| \leq \frac{2nR}{\tau}$. Lastly, for the first term, we have

$$\left\|\mathbf{H}_k - \hat{\mathbf{H}}_k\right\|_F^2 \leq \sum_{i,j=1}^n \left(\mathbf{H}_{k,ij} - \hat{\mathbf{H}}_{k,ij}\right)^2$$

$$= \frac{1}{m^2} \sum_{i,j=1}^n \left(\mathbf{x}_i^\top \mathbf{x}_j \sum_{r \in S_i^\perp} \mathbb{I}\left\{\mathbf{w}_{k,r}^\top \mathbf{x}_i \geq 0; \mathbf{w}_{k,r}^\top \mathbf{x}_j \geq 0\right\}\right)^2$$

$$\leq \frac{1}{m^2} \sum_{i,j=1}^n \left|S_i^\perp\right|^2$$

Combining the three bounds gives the desired result. $\square$

## C.2 Proof of Theorem 5.6

We view Gaussian input masking as a special case of the general stochastic training framework in Section 5. Recall that in that framework, the randomness at iteration $k$ is denoted by $\boldsymbol{\xi}_k$, and the update rule is

$$\mathbf{W}_{k+1} = \mathbf{W}_k - \eta \nabla_{\mathbf{W}} \hat{\mathcal{L}}(\mathbf{W}_k, \boldsymbol{\xi}_k). \tag{7}$$

In the Gaussian-masked setting we take

$$\boldsymbol{\xi}_k \equiv \mathbf{C}_k, \qquad \hat{\mathcal{L}}(\mathbf{W}, \boldsymbol{\xi}_k) \equiv \mathcal{L}_{\mathbf{C}_k}(\mathbf{W}),$$

where $\mathbf{C}_k$ is the multiplicative Gaussian mask at iteration $k$ and $\mathcal{L}_{\mathbf{C}_k}$ is the masked loss. Thus

$$\nabla_{\mathbf{w}_r} \hat{\mathcal{L}}(\mathbf{W}, \boldsymbol{\xi}_k) \equiv \nabla_{\mathbf{w}_r} \mathcal{L}_{\mathbf{C}_k}(\mathbf{W}),$$

and the update (7) coincides with the masked gradient descent rule (1). Therefore, to apply Theorem 5.2 to training with Gaussian input masks, it suffices to verify that Assumption 5.1 holds with suitable $\varepsilon_1, \varepsilon_2, \varepsilon_3, \gamma$, and that these parameters satisfy the smallness conditions of Theorem 5.2 under the constraints (14)–(15).

Throughout the proof we condition on the high-probability NTK event of Theorem 5.2, on which

- the minimum eigenvalue of the empirical NTK satisfies $\lambda_{\min}(\mathbf{H}_k) \geq \lambda_0/2$ for all $k \in [K]$,

- all first-layer weights remain in a ball of radius $R = C_1 \tau \lambda_0/n$ around their initialization, i.e., $\|\mathbf{w}_{k,r} - \mathbf{w}_{0,r}\|_2 \leq R$ for all $k \in [K], r \in [m]$,

- the data are bounded as in Assumption 3.1.

The probability of this event is at least $1 - 2\delta - n^2 \exp(-n^3/(\delta^2\tau^2\lambda_0^3))$, as in Theorem 5.2. All inequalities below hold on this event.

Comparing Corollary 5.3 with Assumption 5.1(8), we identify

$$\varepsilon_1(\mathbf{W}) = 2mn\kappa^2 R_{\mathbf{u}}^2 + mn\big(\kappa^2 R_{\mathbf{u}}^2 + \kappa R_{\mathbf{w}}\big)\phi_{\max}(\mathbf{W})^2 + mn\kappa^2\big(R_{\mathbf{u}}^2 + 1\big)\psi_{\max}(\mathbf{W})^2. \tag{34}$$

On the NTK event, the weights stay close to initialization, hence their norms are uniformly bounded; using Assumption 3.1 and the definition of $R_{\mathbf{w}}$ and $R_{\mathbf{u}}$, we obtain:

$$R_{\mathbf{w}}(\mathbf{W}_k) := \max_{r \in [m]} \|\mathbf{w}_{k,r}\|_2 \leq C_w \tau\sqrt{d} \tag{35}$$

$$R_{\mathbf{u}}(\mathbf{W}_k) := \max_{r \in [m], i \in [n]} \|\mathbf{w}_{k,r} \odot \mathbf{x}_i\|_2 \leq C_u \tau\sqrt{d}. \tag{36}$$

for constants $C_w, C_u > 0$.

**Lemma C.2** (Bound on $R_{\mathbf{w}}$). *On the NTK event of Theorem 5.2, there exists an absolute constant $C_w > 0$ such that for all iterations $k \leq K$:*

$$R_{\mathbf{w}}(\mathbf{W}_k) := \max_{r \in [m]} \|\mathbf{w}_{k,r}\|_2 \leq C_w \tau\sqrt{d}. \tag{37}$$

*Proof.* Recall that the first–layer weights are initialized as $\mathbf{w}_{0,r} \sim \mathcal{N}(\mathbf{0}, \tau^2\mathbf{I}_d)$ for $r = 1, \ldots, m$. During training, the NTK event of Theorem 5.2 ensures that each row stays in a small ball around its initialization:

$$\|\mathbf{w}_{k,r} - \mathbf{w}_{0,r}\|_2 \leq R, \qquad R := C_1\tau\frac{\lambda_0}{n}, \qquad \forall k \leq K, r \in [m]. \tag{38}$$

We define $\mathbf{z}_r := \frac{1}{\tau}\mathbf{w}_{0,r}$. Each coordinate satisfies $(z_r)_j \sim \mathcal{N}(0,1)$, making $\mathbf{z}_r$ a standard Gaussian vector $\mathcal{N}(\mathbf{0}, \mathbf{I}_d)$. Its squared norm follows a chi-square distribution: $\|\mathbf{z}_r\|_2^2 \sim \chi_d^2$. Using the Laurent–Massart concentration inequality, for $t = d$:

$$\Pr\big(\|\mathbf{z}_r\|_2^2 \geq 5d\big) \leq e^{-d}.$$

Thus, with high probability, $\|\mathbf{z}_r\|_2 \leq \sqrt{5d}$. Defining $C_0 = \sqrt{5}$, we obtain the initialization bound:

$$\|\mathbf{w}_{0,r}\|_2 = \tau\|\mathbf{z}_r\|_2 \leq C_0\tau\sqrt{d}. \tag{39}$$

By a union bound over $r \in [m]$, this holds for all rows with probability at least $1 - me^{-d}$. Combining the triangle inequality with (38) and (39), we find:

$$\|\mathbf{w}_{k,r}\|_2 \leq \|\mathbf{w}_{0,r}\|_2 + \|\mathbf{w}_{k,r} - \mathbf{w}_{0,r}\|_2$$

$$\leq C_0\tau\sqrt{d} + C_1\tau\frac{\lambda_0}{n}$$

$$= \tau\sqrt{d}\left(C_0 + \frac{C_1\lambda_0}{n\sqrt{d}}\right).$$

Since $\lambda_0/n$ is $O(1)$ and $d \geq 1$, we define the absolute constant $C_w := C_0 + \frac{C_1\lambda_0}{n\sqrt{d}}$. Taking the maximum over $r \in [m]$ yields:

$$R_w(\mathbf{W}_k) = \max_{r \in [m]} \|\mathbf{w}_{k,r}\|_2 \leq C_w\tau\sqrt{d}. \tag{40}$$

Intuitively, since the movement term $\frac{C_1\lambda_0}{n\sqrt{d}}$ vanishes as $d \to \infty$, the weights remain on the same scale as their initialization throughout training. $\qquad\square$

**Lemma C.3** (Bound on $R_u$). *Suppose Assumption 3.1 holds, such that the input data is bounded in $\ell_\infty$-norm by $B_x := \max_{i \in [n]} \|x_i\|_\infty$. On the NTK event of Theorem 5.2, there exists an absolute constant $C_u > 0$ such that, for all iterations $k \leq K$:*

$$R_u(W_k) := \max_{r \in [m], i \in [n]} \|w_{k,r} \odot x_i\|_2 \leq C_u \tau \sqrt{d}. \tag{41}$$

*Proof.* Consider any iteration $k \leq K$, neuron $r \in [m]$, and sample index $i \in [n]$. We analyze the squared $\ell_2$-norm of the Hadamard product by pulling out the maximum coordinate of the input vector:

$$\|w_{k,r} \odot x_i\|_2^2 = \sum_{j=1}^d w_{k,r,j}^2 x_{i,j}^2$$

$$\leq \left(\max_{1 \leq j \leq d} x_{i,j}^2\right) \sum_{j=1}^d w_{k,r,j}^2$$

$$= \|x_i\|_\infty^2 \|w_{k,r}\|_2^2.$$

Taking the square root of both sides, we obtain the inequality:

$$\|w_{k,r} \odot x_i\|_2 \leq \|x_i\|_\infty \|w_{k,r}\|_2 \leq B_x \|w_{k,r}\|_2.$$

Taking the maximum over all $r \in [m]$ and $i \in [n]$ yields:

$$R_u(W_k) \leq B_x R_w(W_k). \tag{42}$$

From the weight stability bound previously established (Proof of $R_w$), we know that on the NTK event, the weights are bounded by $R_w(W_k) \leq C_w \tau \sqrt{d}$, where $C_w$ is an absolute constant. Substituting this into (42):

$$R_u(W_k) \leq B_x(C_w \tau \sqrt{d}).$$

Defining the absolute constant $C_u := B_x C_w$ completes the proof. Note that since $B_x$ is a fixed property of the dataset and $C_w$ is independent of $k$, $C_u$ is a valid absolute constant for the problem. $\square$

We an iteration $k \in [K]$ and denote, for brevity,

$$R_{\mathbf{w}} := R_{\mathbf{w}}(\mathbf{W}_k), \quad R_{\mathbf{u}} := R_{\mathbf{u}}(\mathbf{W}_k), \quad \phi_k := \phi_{\max}(\mathbf{W}_k), \quad \psi_k := \psi_{\max}(\mathbf{W}_k).$$

Then (34) becomes

$$\varepsilon_1(\mathbf{W}_k) = 2mn\kappa^2 R_{\mathbf{u}}^2 + mn(\kappa^2 R_{\mathbf{u}}^2 + \kappa R_{\mathbf{w}})\phi_k^2 + mn\kappa^2(R_{\mathbf{u}}^2 + 1)\psi_k^2. \tag{43}$$

We now bound each of the three terms on the right-hand side using (35),(36).

*(i) First term.* Using $R_{\mathbf{u}}^2 \leq C_u^2 \tau^2 d$ from (36), we get

$$2mn\kappa^2 R_{\mathbf{u}}^2 \leq 2mn\kappa^2 \cdot C_u^2 \tau^2 d$$

$$= (2C_u^2)\kappa^2 \tau^2 mnd = O(\kappa^2 \tau^2 mnd). \tag{44}$$

*(ii) Middle term.*

$$mn(\kappa^2 R_{\mathbf{u}}^2 + \kappa R_{\mathbf{w}})\phi_k^2 = mn\kappa^2 R_{\mathbf{u}}^2 \phi_k^2 + mn\kappa R_{\mathbf{w}}\phi_k^2.$$

For the $\kappa^2 R_{\mathbf{u}}^2 \phi_k^2$ component, we use $R_{\mathbf{u}}^2 \leq C_u^2 \tau^2 d$ from (36):

$$mn\kappa^2 R_{\mathbf{u}}^2 \phi_k^2 \leq mn\kappa^2 (C_u^2 \tau^2 d)\phi_k^2$$

$$= C_u^2 \kappa^2 \tau^2 mnd \phi_k^2 = O(\kappa^2 \tau^2 mnd \phi_k^2). \tag{45}$$

For the $\kappa R_{\mathbf{w}}\phi_k^2$ component, we use $R_{\mathbf{w}} \le C_w \tau \sqrt{d}$ from (35):

$$mn\kappa R_{\mathbf{w}}\phi_k^2 \le mn\kappa(C_w\tau\sqrt{d})\phi_k^2$$
$$= C_w\kappa\tau mn\sqrt{d}\phi_k^2 = O\big(\kappa\tau mn\sqrt{d}\phi_k^2\big). \tag{46}$$

*(iii) Last term.* For the last term, using (36), it is true that $R_{\mathbf{u}}^2 + 1 \le C_u^2\tau^2 d + 1$, so there exists a constant $C_u' > 0$ such that $R_{\mathbf{u}}^2 + 1 \le C_u'\tau^2 d$ for big enough $d$.

Hence,

$$mn\kappa^2\big(R_{\mathbf{u}}^2 + 1\big)\psi_k^2 \le mn\kappa^2(C_u'\tau^2 d)\psi_k^2$$
$$= C_u'\kappa^2\tau^2 mnd\psi_k^2 = O\big(\kappa^2\tau^2 mnd\psi_k^2\big). \tag{47}$$

Combining (43) with (44), (45), (46), and (47), we obtain

$$\varepsilon_1(\mathbf{W}_k) \le O\big(\kappa^2\tau^2 mnd\big) + O\big(\kappa^2\tau^2 mnd\phi_k^2\big) + O\big(\kappa\tau mn\sqrt{d}\phi_k^2\big) + O\big(\kappa^2\tau^2 mnd\psi_k^2\big)$$
$$= O\big(\kappa^2\tau^2 mnd\big) + O\big(\kappa^2\tau^2 mnd(\phi_k^2 + \psi_k^2)\big) + O\big(\kappa\tau mn\sqrt{d}\phi_k^2\big). \tag{48}$$

We denote

$$\hat{\phi}_{\max} := \max_{k \in [K]} \phi_{\max}(\mathbf{W}_k), \qquad \hat{\psi}_{\max} := \max_{k \in [K]} \psi_{\max}(\mathbf{W}_k),$$

so that for each $k$,

$$\phi_k^2 \le \hat{\phi}_{\max}^2, \qquad \psi_k^2 \le \hat{\psi}_{\max}^2.$$

Substituting these into (48) yields

$$\varepsilon_1(\mathbf{W}_k) \le O\big(\kappa^2\tau^2 mnd\big) + O\big(\kappa^2\tau^2 mnd(\hat{\phi}_{\max}^2 + \hat{\psi}_{\max}^2)\big) + O\big(\kappa\tau mn\sqrt{d}\hat{\phi}_{\max}^2\big), \qquad \forall k \in [K]. \tag{49}$$

Taking the maximum over $k \in [K]$ does not change the right-hand side, so

$$\varepsilon_1 := \max_{k \in [K]} \varepsilon_1(\mathbf{W}_k) \le O\big(\kappa^2\tau^2 mnd\big) + O\big(\kappa^2\tau^2 mnd(\hat{\phi}_{\max}^2 + \hat{\psi}_{\max}^2)\big) + O\big(\kappa\tau mn\sqrt{d}\hat{\phi}_{\max}^2\big). \tag{50}$$

or equivalently:

$$\boxed{\varepsilon_1 \le O\big(\kappa^2\tau^2 mnd(\hat{\phi}_{\max}^2 + \hat{\psi}_{\max}^2 + 1)\big) + O\big(\kappa\tau mn\sqrt{d}\hat{\phi}_{\max}^2\big)} \tag{51}$$

Matching Corollary 5.4 with Assumption 5.1(9), we read off

$$\varepsilon_3(\mathbf{W}) = O\Big(\frac{\sigma_{\max}(\mathbf{X})\phi_{\max}(\mathbf{W})}{\sqrt{m}}\Big)$$

and

$$\varepsilon_2(\mathbf{W}) = O\big((n\kappa^2 B_{\mathbf{x}}^2 R_{\mathbf{w}} + n\kappa R_{\mathbf{u}}\sqrt{d})\phi_{\max}(\mathbf{W})\big)$$
$$+ O\big(n\kappa R_{\mathbf{u}}\psi_{\max}(\mathbf{W}) + \kappa^2\sqrt{m}B_{\mathbf{x}}^2 R_{\mathbf{w}}\big). \tag{52}$$

Using $\|\mathbf{x}_i\|_2 \le 1$ from Assumption 3.1, we have $B_{\mathbf{x}} \le 1$. On the NTK event, for all $k \in [K]$ we have the uniform bounds

$$R_{\mathbf{w}}(\mathbf{W}_k) \le C_w\tau\sqrt{d}, \qquad R_{\mathbf{u}}(\mathbf{W}_k) \le C_u\tau\sqrt{d},$$

for some absolute constants $C_w, C_u > 0$ (cf. the bounds proved earlier for $R_{\mathbf{w}}$ and $R_{\mathbf{u}}$). Substituting these into (52) yields:

$$\varepsilon_2(\mathbf{W}_k) \le O\Big(\big(n\kappa^2 C_w\tau\sqrt{d} + n\kappa C_u\tau\sqrt{d}\sqrt{d}\big)\phi_{\max}(\mathbf{W}_k)\Big)$$
$$+ O\Big(n\kappa C_u\tau\sqrt{d}\psi_{\max}(\mathbf{W}_k) + \kappa^2\sqrt{m}C_w\tau\sqrt{d}\Big)$$
$$= O\big(n\kappa^2\tau\sqrt{d}\phi_{\max}(\mathbf{W}_k)\big) + O\big(n\kappa\tau d\phi_{\max}(\mathbf{W}_k)\big)$$
$$+ O\big(n\kappa\tau\sqrt{d}\psi_{\max}(\mathbf{W}_k)\big) + O\big(\kappa^2\tau\sqrt{md}\big). \tag{53}$$

Since $d \geq 1$, we have $\sqrt{d} \leq d$, and hence each term containing $\sqrt{d}$ can be upper bounded by the corresponding expression with $d$ in place of $\sqrt{d}$. Therefore,

$$
\begin{aligned}
\varepsilon_2(\mathbf{W}_k) &\leq O\big(n\kappa^2\tau d\phi_{\max}(\mathbf{W}_k)\big) + O\big(n\kappa\tau d\phi_{\max}(\mathbf{W}_k)\big) \\
&\quad + O\big(n\kappa\tau d\psi_{\max}(\mathbf{W}_k)\big) + O\big(\kappa^2\tau\sqrt{m}d\big) \\
&\leq O\big(\kappa\tau nd\big(\phi_{\max}(\mathbf{W}_k) + \psi_{\max}(\mathbf{W}_k)\big)\big) \\
&\quad + O\big(\kappa^2\tau d\big(n\phi_{\max}(\mathbf{W}_k) + \sqrt{m}\big)\big).
\end{aligned}
\tag{54}
$$

To obtain a uniform bound over the whole training trajectory, define

$$
\hat{\phi}_{\max} := \max_{k \in [K]} \phi_{\max}(\mathbf{W}_k), \qquad \hat{\psi}_{\max} := \max_{k \in [K]} \psi_{\max}(\mathbf{W}_k).
$$

Taking the maximum over $k$ in (54) yields

$$
\varepsilon_2 := \max_{k \in [K]} \varepsilon_2(\mathbf{W}_k) \leq O\big(\kappa\tau nd(\hat{\phi}_{\max} + \hat{\psi}_{\max})\big) + O\big(\kappa^2\tau d\big(n\hat{\phi}_{\max} + \sqrt{m}\big)\big).
$$

We can simplify $\varepsilon_2$ further as:

$$
\begin{aligned}
O\big(\kappa\tau nd(\hat{\phi}_{\max} + \hat{\psi}_{\max})\big) + O\big(\kappa^2\tau d\big(n\hat{\phi}_{\max} + \sqrt{m}\big)\big) &= O\big(\kappa\tau nd(\hat{\phi}_{\max} + \hat{\psi}_{\max}) + \kappa^2\tau d\big(n\hat{\phi}_{\max} + \sqrt{m}\big)\big) \\
&= O\big(\kappa\tau nd(\hat{\phi}_{\max} + \hat{\psi}_{\max}) + \kappa^2\tau dn\hat{\phi}_{\max} + \kappa^2\tau d\sqrt{m}\big) \\
&\leq O\big(\kappa\tau nd(\hat{\phi}_{\max} + \hat{\psi}_{\max}) + \kappa\tau dn(\hat{\phi}_{\max} + \hat{\psi}_{\max}) + \kappa^2\tau d\sqrt{m}\big) \\
&= O\big(2\kappa\tau nd(\hat{\phi}_{\max} + \hat{\psi}_{\max}) + \kappa^2\tau d\sqrt{m}\big)
\end{aligned}
$$

Thus,

$$
\boxed{\varepsilon_2 \leq O\big(\kappa\tau nd(\hat{\phi}_{\max} + \hat{\psi}_{\max})\big) + O\big(\kappa^2\tau d\sqrt{m}\big)}
\tag{55}
$$

Theorem 5.2 requires $\varepsilon_2$ to satisfy, for some absolute constant $c_0 > 0$,

$$
\varepsilon_2 \leq c_0 \frac{\delta\lambda_0}{nK^2}
\tag{56}
$$

Combining (55) and (56), we get that the following inequality must hold:

$$
C_1\kappa\tau nd\big(\hat{\phi}_{\max} + \hat{\psi}_{\max}\big) + C_2\kappa^2\tau d\sqrt{m} \leq c_0 \frac{\delta\lambda_0}{nK^2}.
\tag{57}
$$

Let us denote

$$
a := C_1\kappa\tau nd\big(\hat{\phi}_{\max} + \hat{\psi}_{\max}\big), \qquad b := C_2\kappa^2\tau d\sqrt{m}, \qquad R := c_0 \frac{\delta\lambda_0}{nK^2}.
$$

Then (57) can be written as $a + b \leq R$.

A sufficient way to enforce this inequality is to require that each term $a$ and $b$ is at most $R/2$:

$$
a \leq \frac{R}{2}, \qquad b \leq \frac{R}{2}
\tag{58}
$$

Indeed, if (58) holds, then

$$
a + b \leq \frac{R}{2} + \frac{R}{2} = R,
$$

so (57) is automatically satisfied.

Imposing $a \leq R/2$ yields:

$$
C_1\kappa\tau nd\big(\hat{\phi}_{\max} + \hat{\psi}_{\max}\big) \leq \frac{c_0}{2}\frac{\delta\lambda_0}{nK^2}.
$$

Thus,

$$\kappa \;\leq\; \kappa_{\mathrm{lin}_2} := \frac{c_0}{2C_1} \frac{\delta\lambda_0}{\tau n^2 dK^2 (\hat{\phi}_{\max} + \hat{\psi}_{\max})}. \tag{59}$$

Similarly, imposing $b \leq R/2$ gives:

$$C_2 \kappa^2 \tau d\sqrt{m} \;\leq\; \frac{c_0}{2}\frac{\delta\lambda_0}{nK^2} \quad\Rightarrow$$

$$\kappa^2 \;\leq\; \frac{c_0}{2C_2}\frac{\delta\lambda_0}{\tau d\sqrt{m}nK^2}$$

and therefore

$$\kappa \;\leq\; \kappa_{\mathrm{quad}_2} := \sqrt{\frac{c_0}{2C_2}}\sqrt{\frac{\delta\lambda_0}{\tau d\sqrt{m}nK^2}}. \tag{60}$$

To ensure (56) holds, it is sufficient that (59) and (60) both hold. Equivalently,

$$\kappa \;\leq\; \min\{\kappa_{\mathrm{lin}_2}, \kappa_{\mathrm{quad}_2}\}.$$

In big-$O$ notation we may write this as

$$\kappa = O\!\left(\frac{\delta\lambda_0}{\tau n^2 dK^2(\hat{\phi}_{\max} + \hat{\psi}_{\max})}\right) \quad\text{and}\quad \kappa = O\!\left(\sqrt{\frac{\delta\lambda_0}{\tau d\sqrt{m}nK^2}}\right). \tag{61}$$

Similarly, for $\varepsilon_1$ the general stochastic convergence theorem requires that

$$\varepsilon_1 \;\leq\; c_1\frac{\delta m}{nK^4}, \tag{4}$$

for some absolute constant $c_1 > 0$.

Combining (51) and (4) we get that the following must hold:

$$O\!\left(\kappa^2\tau^2 mnd(\hat{\phi}_{\max}^2 + \hat{\psi}_{\max}^2 + 1)\right) + O\!\left(\kappa\tau mn\sqrt{d}\hat{\phi}_{\max}^2\right) \;\leq\; c_1\frac{\delta m}{nK^4} \quad\Rightarrow$$

$$C_a\kappa^2\tau^2 nd\!\left(\hat{\phi}_{\max}^2 + \hat{\psi}_{\max}^2 + 1\right) + C_b\kappa\tau n\sqrt{d}\hat{\phi}_{\max}^2 \;\leq\; c_1\frac{\delta}{nK^4}.$$

Similarly, we impose

$$C_a\kappa^2\tau^2 nd\!\left(1 + \hat{\phi}_{\max}^2 + \hat{\psi}_{\max}^2\right) \;\leq\; \frac{c_1}{2}\frac{\delta}{nK^4} \quad\Rightarrow$$

$$\kappa^2 \;\leq\; \frac{c_1}{2C_a}\frac{\delta}{\tau^2 n^2 dK^4\!\left(\hat{\phi}_{\max}^2 + \hat{\psi}_{\max}^2 + 1\right)} \quad\Rightarrow$$

$$\kappa \;\leq\; \kappa_{\mathrm{quad}_1} := \sqrt{\frac{c_1}{2C_a}}\cdot\frac{\sqrt{\delta}}{\tau n\sqrt{d}K^2\sqrt{\hat{\phi}_{\max}^2 + \hat{\psi}_{\max}^2 + 1}}. \tag{62}$$

and

$$C_b\kappa\tau n\sqrt{d}\hat{\phi}_{\max}^2 \;\leq\; \frac{c_1}{2}\frac{\delta}{nK^4} \quad\Rightarrow$$

$$\kappa \;\leq\; \kappa_{\mathrm{lin}_1} := \frac{c_1}{2C_b}\frac{\delta}{\tau n^2\sqrt{d}K^4\hat{\phi}_{\max}^2}. \tag{63}$$

Thus, the $\varepsilon_1$ requirement (4) is guaranteed whenever

$$\kappa \;\leq\; \min\!\left\{\kappa_{\mathrm{quad}_1}, \kappa_{\mathrm{lin}_1}\right\}.$$

Combining all constraints from $\varepsilon_1$ and $\varepsilon_2$, we see that a sufficient set of conditions is

$$\kappa \;\leq\; \min\{\kappa_{\mathrm{lin}_1}, \kappa_{\mathrm{quad}_1}, \kappa_{\mathrm{lin}_2}, \kappa_{\mathrm{quad}_2}\}.$$

Equivalently, in big-$O$ notation,

$$\kappa = O\left(\min\left\{\frac{\delta\lambda_0}{\tau n^2 dK^2(\hat{\phi}_{\max} + \hat{\psi}_{\max})}, \;\frac{\delta}{\tau n^2 \sqrt{d}K^4 \hat{\phi}_{\max}^2}, \;\sqrt{\frac{\delta\lambda_0}{\tau d\sqrt{m}nK^2}}, \;\sqrt{\frac{\delta}{\tau^2 n^2 dK^4(\hat{\phi}_{\max}^2 + \hat{\psi}_{\max}^2 + 1)}}\right\}\right). \tag{64}$$

Instead of carrying this minimum in the theorem statement, we define a slightly more restrictive but cleaner condition that implies all of the above bounds:

$$\kappa \;=\; O\left(\frac{\sqrt{\delta\lambda_0}}{\tau^2 K^2\big(m^{1/4}\sqrt{d} + nd\big)\big(\hat{\phi}_{\max} + \hat{\psi}_{\max}\big)}\right). \tag{65}$$

For $\varepsilon_3$, we combine (9) and (5.4) which yields that:

$$\varepsilon_3 = O\left(\frac{\sigma_{\max}(\mathbf{X})\phi_{\max}}{\sqrt{m}}\right)$$

We assume that

$$\sigma_{\max}(\mathbf{X})\hat{\phi}_{\max} \leq C\lambda_0/\sqrt{n}$$

for some constant $C > 0$ (assumption (15)), and therefore

$$\varepsilon_3 \leq O\left(\frac{\lambda_0}{\sqrt{mn}}\right), \tag{66}$$

which is exactly the form required in Theorem 5.2.

Theorem 5.2 includes the factor

$$O\left(\frac{mn}{\lambda_0^2}\cdot\varepsilon_2^2 + \varepsilon_1\right) \overset{(55),(50)}{=}$$

So we study the quantity $\frac{mn}{\lambda_0^2}\cdot\varepsilon_2^2$.

$$\begin{aligned}
\frac{mn}{\lambda_0^2}\cdot\varepsilon_2^2 &= \frac{mn}{\lambda_0^2}\cdot\left(\kappa^2\tau^2 n^2 d^2(\hat{\phi}_{\max} + \hat{\psi}_{\max})^2 + \kappa^4\tau^2 d^2 m\right)\\
&\leq \frac{2}{\lambda_0^2}\kappa^2\tau^2 n^3 d^2 m(\hat{\phi}_{\max}^2 + \hat{\psi}_{\max}^2) + \frac{1}{\lambda_0^2}\kappa^4\tau^2 d^2 m^2 n\\
&= O\left(\kappa\tau^2 mn^3 d^2\big(\hat{\phi}_{\max}^2 + \hat{\psi}_{\max}^2\big)\right) + O\left(\kappa^2\tau^2 m^2 nd^2\right)
\end{aligned} \tag{67}$$

because $(a+b)^2 \leq 2(a^2 + b^2)$ and $\kappa^4 \leq \kappa^2 \leq \kappa$ for $\kappa \leq 1$

Furthermore,

$$\begin{aligned}
\varepsilon_1 &= O\big(\kappa^2\tau^2 mnd\big) + O\big(\kappa^2\tau^2 mnd(\hat{\phi}_{\max}^2 + \hat{\psi}_{\max}^2)\big) + O\big(\kappa\tau mn\sqrt{d}\hat{\phi}_{\max}^2\big)\\
&\leq O\big(\kappa^2\tau^2 m^2 nd^2\big) + O\big(\kappa\tau^2 mn^3 d^2(\hat{\phi}_{\max}^2 + \hat{\psi}_{\max}^2)\big) + O\big(\kappa\tau mn\sqrt{d}\hat{\phi}_{\max}^2\big)
\end{aligned} \tag{68}$$

Therefore,

$$O\left(\frac{mn}{\lambda_0^2}\cdot\varepsilon_2^2 + \varepsilon_1\right) \overset{(68),(67)}{=} O\big(\kappa^2\tau^2 m^2 nd^2\big) + O\big(\kappa\tau^2 mn^3 d^2(\hat{\phi}_{\max}^2 + \hat{\psi}_{\max}^2)\big) + O\big(\kappa\tau mn\sqrt{d}\hat{\phi}_{\max}^2\big)$$

Thus, the expected loss is bounded by:

$$\mathbb{E}[\mathcal{L}(\mathbf{W}_K)] \leq \left(1 - \frac{\eta\lambda_0}{2}\right)^K \mathcal{L}(\mathbf{W}_0) + O\big(\kappa^2\tau^2 m^2 nd^2\big) + O\big(\kappa\tau^2 mn^3 d^2(\hat{\phi}_{\max}^2 + \hat{\psi}_{\max}^2)\big) + O\big(\kappa\tau mn\sqrt{d}\hat{\phi}_{\max}^2\big)$$

### C.2.1 Proof of Corollary 5.3

*Proof.* By Theorem 4.2, we have that

$$
|\mathbb{E}_{\mathbf{C}}\left[\mathcal{L}_{\mathbf{C}}\left(\boldsymbol{\theta}\right)\right] - \mathcal{L}\left(\boldsymbol{\theta}\right)| \leq \underbrace{\left| \frac{1}{2} \sum_{i=1}^{n} \left( \left( \hat{f}\left(\boldsymbol{\theta}, \mathbf{x}_i\right) - y_i \right)^2 - \left( f\left(\mathbf{W}, \mathbf{x}_i\right) - y_i \right)^2 \right) \right|}_{\mathcal{T}_1}
$$
$$
+ \underbrace{\frac{\kappa^2}{2m} \sum_{i=1}^{n} \left\| \sum_{r=1}^{m} a_r \left(\mathbf{w}_r \odot \mathbf{x}_i\right) \boldsymbol{\Phi}_1 \left( \frac{\mathbf{w}_r^\top \mathbf{x}_i}{\kappa \left\| \mathbf{w}_r \odot \mathbf{x}_i \right\|_2} \right) \right\|_2^2}_{\mathcal{T}_2}
$$
$$
+ mn \left( \kappa^2 R_{\mathbf{u}}^2 \psi_{\max}^2 + \left( \kappa^2 R_{\mathbf{u}}^2 + \kappa R_{\mathbf{w}} \right) \phi_{\max}^2 \right)
$$

Now, we can bound $\mathcal{T}_1$ and $\mathcal{T}_2$ separately. For $\mathcal{T}_1$, we have

$$
\left| \frac{1}{2} \sum_{i=1}^{n} \left( \left( \hat{f}\left(\boldsymbol{\theta}, \mathbf{x}_i\right) - y_i \right)^2 - \left( f\left(\mathbf{W}, \mathbf{x}_i\right) - y_i \right)^2 \right) \right|
$$
$$
\leq \frac{1}{2} \sum_{i=1}^{n} \left( \left( \hat{f}\left(\boldsymbol{\theta}, \mathbf{x}_i\right) - f\left(\mathbf{W}, \mathbf{x}_i\right) \right)^2 + 2 \left| \left( \hat{f}\left(\boldsymbol{\theta}, \mathbf{x}_i\right) - f\left(\mathbf{W}, \mathbf{x}_i\right) \right) \left( \hat{f}\left(\boldsymbol{\theta}, \mathbf{x}_i\right) - f\left(\mathbf{W}, \mathbf{x}_i\right) \right) \right| \right)
$$
$$
\leq \sum_{i=1}^{n} \left( \hat{f}\left(\boldsymbol{\theta}, \mathbf{x}_i\right) - f\left(\mathbf{W}, \mathbf{x}_i\right) \right)^2 + \mathcal{L}\left(\boldsymbol{\theta}\right)
$$

Here, we can bound $\hat{f}\left(\mathbf{W}, \mathbf{x}_i\right) - f\left(\mathbf{W}, \mathbf{x}_i\right)$ as

$$
\left| \hat{f}\left(\mathbf{W}, \mathbf{x}_i\right) - f\left(\mathbf{W}, \mathbf{x}_i\right) \right| \leq \frac{1}{\sqrt{m}} \sum_{r=1}^{m} \left| \hat{\sigma}\left(\mathbf{w}_r, \mathbf{x}_i\right) - \sigma\left(\mathbf{w}_r^\top \mathbf{x}_i\right) \right|
$$
$$
\leq \frac{1}{\sqrt{m}} \sum_{r=1}^{m} \left| \mathbf{w}_r^\top \mathbf{x}_i \right| \left( \mathbb{I}\left\{ \mathbf{w}_r^\top \mathbf{x}_i \geq 0 \right\} - \boldsymbol{\Phi}_1 \left( \frac{\mathbf{w}_r^\top \mathbf{x}_i}{\kappa \left\| \mathbf{w}_r \odot \mathbf{x}_i \right\|_2} \right) \right)
$$
$$
\leq \frac{1}{\sqrt{m}} \sum_{r=1}^{m} \left| \mathbf{w}_r^\top \mathbf{x}_i \right| \exp\left( - \frac{\left( \mathbf{w}_r^\top \mathbf{x}_i \right)^2}{2\kappa^2 \left\| \mathbf{w}_r \odot \mathbf{x}_i \right\|_2} \right)
$$
$$
\leq \frac{1}{\sqrt{m}} \sum_{r=1}^{m} \psi \left( \frac{\mathbf{w}_r^\top \mathbf{x}_i}{2\kappa \left\| \mathbf{w}_r^\top \mathbf{x}_i \right\|_2} \right)
$$
$$
\leq \sqrt{m} \psi_{\max}
$$

where in the third inequality we used Lemma D.4. Therefore, $\mathcal{T}_1$ can be bounded as

$$
\mathcal{T}_1 \leq \mathcal{L}\left(\boldsymbol{\theta}\right) + mn\psi_{\max}
$$

For $\mathcal{T}_2$, we can bound it as

$$
\mathcal{T}_2 \leq 2\kappa^2 \sum_{i=1}^{n} \sum_{r=1}^{m} \left\| \mathbf{w}_r \odot \mathbf{x}_i \right\| \leq 2\kappa^2 mn R_{\mathbf{u}}^2
$$

Plugging in $\mathcal{T}_1$ and $\mathcal{T}_2$ gives the desired result.  $\square$

### C.2.2  Proof of Corollary 5.4

*Proof.* By Theorem 4.9, we have that

$$
\begin{aligned}
\|\mathbb{E}_{\mathbf{C}}\left[\nabla_{\mathbf{w}_r}\mathcal{L}\left(\boldsymbol{\theta}\right)\right] - \nabla_{\mathbf{w}_r}\mathcal{L}\left(\boldsymbol{\theta}\right)\|_2 &= \left\| \mathbf{g}_r + \frac{3\kappa^2}{\sqrt{m}}\sum_{r'=1}^{m} a_{r'}\mathrm{Diag}\left(\mathbf{x}_i\right)^2 \mathbf{w}_{r'}\mathbb{I}\left\{\mathbf{w}_r^\top \mathbf{x}_i \geq 0; \mathbf{w}_{r'}^\top\mathbf{x}_i \geq 0\right\}\right\|_2 \\
&\leq \|\mathbf{g}_r\|_2 + \frac{3\kappa^2}{\sqrt{m}}\sum_{r'=1}^{m}\left\| a_{r'}\mathrm{Diag}\left(\mathbf{x}_i\right)^2 \mathbf{w}_{r'}\mathbb{I}\left\{\mathbf{w}_r^\top \mathbf{x}_i \geq 0; \mathbf{w}_{r'}^\top\mathbf{x}_i \geq 0\right\}\right\|_2 \\
&\leq \|\mathbf{g}_r\|_2 + \frac{3\kappa^2}{\sqrt{m}}\sum_{r'=1}^{m}\|\mathbf{x}_i\|_\infty^2 \|\mathbf{w}_r\|_2 \\
&\leq \left(\frac{\sigma_{\max}(\mathbf{X})}{\sqrt{m}}\mathcal{L}\left(\boldsymbol{\theta}\right)^{\frac{1}{2}} + 6n\kappa^2 B_{\mathbf{x}}^2 R_{\mathbf{w}} + 5n\kappa R_{\mathbf{u}}\sqrt{d}\right)\phi_{\max} \\
&\qquad + 6n\kappa R_{\mathbf{u}}\psi_{\max} + 3\kappa^2\sqrt{m}B_{\mathbf{x}}^2 R_{\mathbf{w}}
\end{aligned}
$$

Theorem 4.9 states that the norm $\|\mathbf{g}_r\|_2$ satisfies

$$
\|\mathbf{g}_r\|_2 \leq \left(6n\kappa^2 B_{\mathbf{x}}^2 R_{\mathbf{w}} + 5n\kappa R_{\mathbf{u}}\sqrt{d}\right)\phi_{\max} + \frac{\sigma_{\max}(\mathbf{X})}{\sqrt{m}}\phi_{\max}\mathcal{L}\left(\boldsymbol{\theta}\right)^{\frac{1}{2}} + 6n\kappa R_{\mathbf{u}}\psi_{\max}
$$

Moreover, by Assumption 3.1, we have that $\|\mathbf{x}\|_\infty \leq B_{\mathbf{x}}$. Therefore, we obtain that

$$
\begin{aligned}
\|\mathbb{E}_{\mathbf{C}}\left[\nabla_{\mathbf{w}_r}\mathcal{L}\left(\boldsymbol{\theta}\right)\right] - \nabla_{\mathbf{w}_r}\mathcal{L}\left(\boldsymbol{\theta}\right)\|_2 &\leq \left(\frac{\sigma_{\max}(\mathbf{X})}{\sqrt{m}}\mathcal{L}\left(\boldsymbol{\theta}\right)^{\frac{1}{2}} + 6n\kappa^2 B_{\mathbf{x}}^2 R_{\mathbf{w}} + 5n\kappa R_{\mathbf{u}}\sqrt{d}\right)\phi_{\max} \\
&\qquad + 6n\kappa R_{\mathbf{u}}\psi_{\max} + 3\kappa^2\sqrt{m}B_{\mathbf{x}}^2 R_{\mathbf{w}}
\end{aligned}
$$

$\square$

### C.2.3  Proof of Lemma D.21

*Proof.* By the form of $\nabla_{\mathbf{w}_r}\mathcal{L}_{\mathbf{C}}\left(\mathbf{W}\right)$ in (2), we have:

$$
\begin{aligned}
\|\nabla_{\mathbf{w}_r}\mathcal{L}_{\mathbf{C}}\left(\mathbf{W}\right)\|_2 &= \left\| \frac{a_r}{\sqrt{m}}\sum_{i=1}^{n}\left(f\left(\mathbf{W}, \mathbf{x}_i \odot \mathbf{c}_i\right) - y_i\right)\left(\mathbf{x}_i \odot \mathbf{c}_i\right)\underbrace{\mathbb{I}\left\{\mathbf{w}_r^\top\left(\mathbf{x}_i \odot \mathbf{c}_i\right) \geq 0\right\}}_{\leq 1}\right\|_2 \\
&\leq \frac{|a_r|}{\sqrt{m}}\sum_{i=1}^{n}\left|f\left(\mathbf{W}, \mathbf{x}_i \odot \mathbf{c}_i\right) - y_i\right| \cdot \|\mathbf{x}_i \odot \mathbf{c}_i\|_2 \\
&\leq \frac{\sqrt{n}}{\sqrt{m}}\left(\sum_{i=1}^{n}\left(f\left(\mathbf{W}, \mathbf{x}_i \odot \mathbf{c}_i\right) - y_i\right)^2\right)^{1/2} \cdot \|\mathbf{c}_i\|_\infty\|\mathbf{x}_i\|_2 \text{ using D.20} \\
&\leq C\frac{\sqrt{n}}{\sqrt{m}}\left(2\mathcal{L}_{\mathbf{C}}\left(\mathbf{W}\right)\right)^{1/2} \text{ using } \|\mathbf{c}_i\|_\infty \leq C \text{ and } \|\mathbf{x}_i\|_2 \leq 1 \\
&\leq C\sqrt{2}\frac{\sqrt{n}}{\sqrt{m}}\mathcal{L}_{\mathbf{C}}\left(\mathbf{W}\right)^{1/2}
\end{aligned}
$$

where, in the first inequality, we use the fact that the indicator function is upper-bounded by 1 and in the second inequality, we use the fact that $a_r = \pm 1$.

and so,

$$
\|\nabla_{\mathbf{w}_r}\mathcal{L}_{\mathbf{C}}\left(\mathbf{W}\right)\|_2^2 \leq 2C^2 \cdot \frac{n}{m}\mathcal{L}_{\mathbf{C}}\left(\mathbf{W}\right)
$$

$\square$

# D Auxiliary Results

## D.1 Gaussian Random Variables

### D.1.1 Conditional Expectation and Covariance

**Lemma D.1.** *Let $\mathbf{c} \sim \mathcal{N}\left(\boldsymbol{\mu}, \kappa^2 \mathbf{I}\right)$, and let $z = \mathbf{c}^\top \mathbf{u}, z' = \mathbf{c}^\top \mathbf{v}$. Then we have that*

$$\mathbb{E}_{\mathbf{c}}\left[\mathbf{c} \mid z\right] = \boldsymbol{\mu} + \frac{\mathbf{u}}{\|\mathbf{u}\|_2^2}\left(z - \boldsymbol{\mu}^\top \mathbf{u}\right); \quad \mathbb{E}_{\mathbf{c}}\left[\mathbf{c} \mid z, z'\right] = \boldsymbol{\mu} + \mathbf{s}_1\left(z - \boldsymbol{\mu}^\top \mathbf{u}\right) + \mathbf{s}_2\left(z' - \boldsymbol{\mu}^\top \mathbf{v}\right)$$

*where the vectos $\mathbf{s}_1, \mathbf{s}_2$ are defined as*

$$\mathbf{s}_1 = \frac{\|\mathbf{v}\|_2^2 \mathbf{u} - \mathbf{u}^\top \mathbf{v} \cdot \mathbf{v}}{\|\mathbf{u}\|_2^2 \|\mathbf{v}\|_2^2 - (\mathbf{u}^\top \mathbf{v})^2}; \quad \mathbf{s}_2 = \frac{\|\mathbf{u}\|_2^2 \mathbf{v} - \mathbf{u}^\top \mathbf{v} \cdot \mathbf{u}}{\|\mathbf{u}\|_2^2 \|\mathbf{v}\|_2^2 - (\mathbf{u}^\top \mathbf{v})^2}$$

*Proof.* By the formula of conditional expectation, we have

$$\mathbb{E}_{\mathbf{c}}\left[\mathbf{c} \mid z\right] = \mathbb{E}[\mathbf{c}] + \mathrm{Cov}\left(\mathbf{c}, z\right) \mathrm{Var}\left(z\right)^{-1}\left(z - \mathbb{E}[z]\right)$$

By the definition of $\mathbf{c}$, we have $\mathbb{E}[\mathbf{c}] = \boldsymbol{\mu}$. Moreover, since $z = \mathbf{c}^\top \mathbf{u}$, by Lemma D.13, we have $\mathbb{E}[z] = \boldsymbol{\mu}^\top \mathbf{u}$ and $\mathrm{Var}\left(z\right) = \kappa^2 \|\mathbf{u}\|_2^2$. Therefore, the covariance between $\mathbf{c}$ and $z$ is given by

$$\mathrm{Cov}\left(\mathbf{c}, z\right) = \mathbb{E}[(\mathbf{c} - \boldsymbol{\mu})\left(z - \langle \boldsymbol{\mu}, \mathbf{v}\rangle\right)] = \mathbb{E}\left[z\mathbf{c}\right] - \boldsymbol{\mu}\langle \boldsymbol{\mu}, \mathbf{v}\rangle$$

where

$$\mathbb{E}[z\mathbf{c}]_j = \sum_{j'=1}^{d} \mathbb{E}[c_j c_{j'} u_{j'}] = \sum_{j'=1}^{d}\left(\mu_j \mu_{j'} + \mathbb{I}\{j = j'\}\kappa^2\right)u_{j'} = \mu_j \boldsymbol{\mu}^\top \mathbf{u} + \kappa^2 u_j$$

This gives

$$\mathbb{E}[z\mathbf{c}] = \boldsymbol{\mu}^\top \mathbf{u} \cdot \boldsymbol{\mu} + \kappa^2 \mathbf{u}$$

Therefore

$$\mathbb{E}_{\mathbf{c}}\left[\mathbf{c} \mid z\right] = \boldsymbol{\mu} + \frac{\mathbf{u}}{\|\mathbf{u}\|_2^2}\left(z - \boldsymbol{\mu}^\top \mathbf{u}\right)$$

Similarly, since $z' = \mathbf{c}^\top \mathbf{v}$. Then

$$\mathbb{E}_{\mathbf{c}}\left[\mathbf{c} \mid z, z'\right] = \mathbb{E}[\mathbf{c}] + \mathrm{Cov}\left(\mathbf{c}, [z, z']\right)\mathrm{Cov}\left(z, z'\right)^{-1}\left([z, z']^\top - \mathbb{E}[[z, z']]^\top\right)$$

where

$$\mathrm{Cov}\left(\mathbf{c}, [z, z']\right) = \kappa^2 \begin{bmatrix} \mathbf{u} \\ \mathbf{v} \end{bmatrix}; \quad \mathrm{Cov}\left(z, z'\right) = \kappa^2 \begin{bmatrix} \|\mathbf{u}\|_2^2 & \mathbf{u}^\top \mathbf{v} \\ \mathbf{u}^\top \mathbf{v} & \|\mathbf{v}\|_2^2 \end{bmatrix}$$

This gives

$$
\begin{aligned}
\mathbb{E}_{\mathbf{c}}\left[\mathbf{c} \mid z, z'\right] &= \boldsymbol{\mu} + \begin{bmatrix} \mathbf{u} \\ \mathbf{v} \end{bmatrix}\begin{bmatrix} \|\mathbf{u}\|_2^2 & \mathbf{u}^\top \mathbf{v} \\ \mathbf{u}^\top \mathbf{v} & \|\mathbf{v}\|_2^2 \end{bmatrix}^{-1}\begin{bmatrix} z - \boldsymbol{\mu}^\top \mathbf{u} \\ z' - \boldsymbol{\mu}^\top \mathbf{v} \end{bmatrix} \\
&= \boldsymbol{\mu} + \frac{1}{\|\mathbf{u}\|_2^2 \|\mathbf{v}\|_2^2 - (\mathbf{u}^\top \mathbf{v})^2}\begin{bmatrix} \mathbf{u} \\ \mathbf{v} \end{bmatrix}\begin{bmatrix} \|\mathbf{v}\|_2^2 & -\mathbf{u}^\top \mathbf{v} \\ -\mathbf{u}^\top \mathbf{v} & \|\mathbf{u}\|_2^2 \end{bmatrix}^{-1}\begin{bmatrix} z - \boldsymbol{\mu}^\top \mathbf{u} \\ z' - \boldsymbol{\mu}^\top \mathbf{v} \end{bmatrix} \\
&= \boldsymbol{\mu} + \frac{\left(\|\mathbf{v}\|_2^2 \mathbf{u} - \mathbf{u}^\top \mathbf{v} \cdot \mathbf{v}\right)\left(z - \boldsymbol{\mu}^\top \mathbf{u}\right) + \left(\|\mathbf{u}\|_2^2 \mathbf{v} - \mathbf{u}^\top \mathbf{v} \cdot \mathbf{u}\right)\left(z' - \boldsymbol{\mu}^\top \mathbf{v}\right)}{\|\mathbf{u}\|_2^2 \|\mathbf{v}\|_2^2 - (\mathbf{u}^\top \mathbf{v})^2} \\
&= \boldsymbol{\mu} + \mathbf{s}_1\left(z - \boldsymbol{\mu}^\top \mathbf{u}\right) + \mathbf{s}_2\left(z' - \boldsymbol{\mu}^\top \mathbf{v}\right)
\end{aligned}
$$

where the vectos $\mathbf{s}_1, \mathbf{s}_2$ are defined as

$$\mathbf{s}_1 = \frac{\|\mathbf{v}\|_2^2 \mathbf{u} - \mathbf{u}^\top \mathbf{v} \cdot \mathbf{v}}{\|\mathbf{u}\|_2^2 \|\mathbf{v}\|_2^2 - (\mathbf{u}^\top \mathbf{v})^2}; \quad \mathbf{s}_2 = \frac{\|\mathbf{u}\|_2^2 \mathbf{v} - \mathbf{u}^\top \mathbf{v} \cdot \mathbf{u}}{\|\mathbf{u}\|_2^2 \|\mathbf{v}\|_2^2 - (\mathbf{u}^\top \mathbf{v})^2}$$

$\square$

**Lemma D.2.** *Let* $\mathbf{c} \sim \mathcal{N}\left(\boldsymbol{\mu}, \kappa^2 \mathbf{I}\right)$, *and let* $z = \mathbf{c}^\top \mathbf{u}, z' = \mathbf{c}^\top \mathbf{v}$. *Then we have*

$$Cov\left(\mathbf{c} \mid z, z'\right) = \kappa^2 \mathbf{I} - \frac{\kappa^2 \left(\mathbf{u}\mathbf{v}^\top - \mathbf{v}\mathbf{u}^\top\right)^2}{\|\mathbf{u}\|_2^2 \|\mathbf{v}\|_2^2 - \left(\mathbf{u}^\top \mathbf{v}\right)^2}$$

*Proof.* The conditional covariance of Gaussian random variables is given by

$$\text{Cov}\left(\mathbf{c} \mid z, z'\right) = \text{Cov}\left(\mathbf{c}\right) - \text{Cov}\left(\mathbf{c}, [z, z']\right) \text{Cov}\left(z, z'\right)^{-1} \text{Cov}\left(\mathbf{c}, [z, z']\right)^\top$$

Recall that in the previos lemma we have computed

$$\text{Cov}\left(\mathbf{c}, [z, z']\right) = \kappa^2 \begin{bmatrix} \mathbf{u} \\ \mathbf{v} \end{bmatrix}; \quad \text{Cov}\left(z, z'\right) = \kappa^2 \begin{bmatrix} \|\mathbf{u}\|_2^2 & \mathbf{u}^\top \mathbf{v} \\ \mathbf{u}^\top \mathbf{v} & \|\mathbf{v}\|_2^2 \end{bmatrix}$$

Thus, we have

$$\begin{aligned}
\text{Cov}\left(\mathbf{c} \mid z, z'\right) &= \kappa^2 \mathbf{I} - \kappa^2 \begin{bmatrix} \mathbf{u} \\ \mathbf{v} \end{bmatrix} \begin{bmatrix} \|\mathbf{u}\|_2^2 & \mathbf{u}^\top \mathbf{v} \\ \mathbf{u}^\top \mathbf{v} & \|\mathbf{v}\|_2^2 \end{bmatrix}^{-1} \begin{bmatrix} \mathbf{u} & \mathbf{v} \end{bmatrix} \\
&= \kappa^2 \mathbf{I} - \frac{\kappa^2}{\|\mathbf{u}\|_2^2 \|\mathbf{v}\|_2^2 - \left(\mathbf{u}^\top \mathbf{v}\right)^2} \begin{bmatrix} \mathbf{u} \\ \mathbf{v} \end{bmatrix} \begin{bmatrix} \|\mathbf{v}\|_2^2 & -\mathbf{u}^\top \mathbf{v} \\ -\mathbf{u}^\top \mathbf{v} & \|\mathbf{u}\|_2^2 \end{bmatrix} \begin{bmatrix} \mathbf{u} & \mathbf{v} \end{bmatrix} \\
&= \kappa^2 \mathbf{I} - \frac{\kappa^2 \left(\mathbf{u}\mathbf{v}^\top - \mathbf{v}\mathbf{u}^\top\right)^2}{\|\mathbf{u}\|_2^2 \|\mathbf{v}\|_2^2 - \left(\mathbf{u}^\top \mathbf{v}\right)^2}
\end{aligned}$$

$\square$

### D.1.2  Approximation of CDF

In approximation of the Gaussian CDF, we use the following property

**Lemma D.3 (?).** *Let* $x \geq 0$ *be given, then the following inequality holds*

$$\frac{1}{2}\left(1 - \exp\left(-\frac{x^2}{2}\right)\right)^{\frac{1}{2}} \leq \frac{1}{\sqrt{2\pi}} \int_0^x \exp\left(-\frac{t^2}{2}\right) dt \leq \frac{1}{2}\left(1 - \exp\left(-\frac{2x^2}{\pi}\right)\right)^{\frac{1}{2}}$$

To start, we analyze the CDF of a single variable Gaussian random variable

**Lemma D.4.** *Let* $\boldsymbol{\Phi}_1$ *be the CDF of a standard Gaussian random variable. Then we have*

$$\left|\boldsymbol{\Phi}_1\left(\alpha\right) - \mathbb{I}\left\{\alpha \geq 0\right\}\right| \leq \exp\left(-\frac{\alpha^2}{2}\right)$$

*Proof.* Let $\nu(x) = \frac{1}{\sqrt{2\pi}} \int_0^x \exp\left(-\frac{t^2}{2}\right) dt$ for $x \geq 0$. We study the case of $\alpha \geq 0$ and $\alpha < 0$ separately. For $\alpha \geq 0$, we have

$$\begin{aligned}
\boldsymbol{\Phi}_1\left(\alpha\right) &= \frac{1}{\sqrt{2\pi}} \int_{-\infty}^\alpha \exp\left(-\frac{t^2}{2}\right) dt \\
&= \frac{1}{2} + \nu(\alpha) \\
&\geq \frac{1}{2} + \frac{1}{2}\left(1 - \exp\left(-\frac{\alpha^2}{2}\right)\right)^{\frac{1}{2}} \\
&\geq 1 - \exp\left(-\frac{\alpha^2}{2}\right) \\
&= \mathbb{I}\left\{\alpha \geq 0\right\} - \exp\left(-\frac{\alpha^2}{2}\right)
\end{aligned}$$

Since $\boldsymbol{\Phi}_1 \leq 1 = \mathbb{I}\{\alpha \geq 0\}$ when $\alpha \geq 0$, we must have that, for $\alpha \geq 0$, $|\boldsymbol{\Phi}_1(\alpha) - \mathbb{I}\{\alpha \geq 0\}| \leq \exp\left(-\frac{\alpha^2}{2}\right)$. Similarly, for $\alpha < 0$, we have

$$
\begin{aligned}
\boldsymbol{\Phi}_1(\alpha) &= \frac{1}{\sqrt{2\pi}} \int_{-\infty}^{\alpha} \exp\left(-\frac{t^2}{2}\right) dt \\
&= \frac{1}{2} - \nu(\alpha) \\
&\leq \frac{1}{2} - \frac{1}{2}\left(1 - \exp\left(-\frac{\alpha^2}{2}\right)\right)^{\frac{1}{2}} \\
&\leq \exp\left(-\frac{\alpha^2}{2}\right) \\
&= \mathbb{I}\{\alpha \geq 0\} + \exp\left(-\frac{\alpha^2}{2}\right)
\end{aligned}
$$

Since $\boldsymbol{\Phi}_1 \geq 0 = \mathbb{I}\{\alpha \geq 0\}$ when $\alpha < 0$, we must have that, for $\alpha < 0$, $|\boldsymbol{\Phi}_1(\alpha) - \mathbb{I}\{\alpha \geq 0\}| \leq \exp\left(-\frac{\alpha^2}{2}\right)$. $\square$

**Lemma D.5.** *Let $\boldsymbol{\Phi}_2(\alpha_1, \alpha_2, \rho)$ be the joint CDF of two standard Gaussian random variables with covariance $\rho$ at $\alpha_1, \alpha_2$. Then we have*

$$
|\boldsymbol{\Phi}_2(\alpha_1, \alpha_2, \rho) - \mathbb{I}\{\alpha_1 \geq 0; \alpha_2 \geq 0\}| \leq 2\exp\left(-\frac{\min\{\alpha_1, \alpha_2\}^2}{2}\right)
$$

*Proof.* Since $z_1, z_2$ are standard Gaussian random variables with covariance $\rho$, we have $z_1 \mid z_2 = \zeta \sim \mathcal{N}(\rho\zeta, 1 - \rho^2)$. According to the definition of CDF,

$$
\begin{aligned}
\boldsymbol{\Phi}_2(\alpha_1, \alpha_2, \rho) &= \int_{-\infty}^{\alpha_2} \int_{-\infty}^{\alpha_1} f_{z_1, z_2}(\zeta_1, \zeta_2, \rho) \, d\zeta_1 d\zeta_2 \\
&= \int_{-\infty}^{\alpha_2} \int_{-\infty}^{\alpha_1} f_{z_1 \mid z_2 = \zeta_2}(\zeta_1) \, d\zeta_1 f_{z_2}(\zeta_2) \, d\zeta_2
\end{aligned}
$$

Focusing on the inner integral, we substitute $\zeta' = \frac{\zeta_1 - \rho\zeta_2}{\sqrt{1 - \rho^2}}$. Then we have $d\zeta_1 = \sqrt{1 - \rho^2} d\zeta'$. Thus

$$
\begin{aligned}
\int_{-\infty}^{\alpha_1} f_{z_1 \mid z_2 = \zeta_2}(\zeta_1) \, d\zeta_1 &= \frac{1}{\sqrt{2\pi(1 - \rho^2)}} \int_{-\infty}^{\alpha_1} \exp\left(-\frac{(\zeta_1 - \rho\zeta_2)^2}{2(1 - \rho^2)}\right) d\zeta_1 \\
&= \frac{1}{\sqrt{2\pi}} \int_{-\infty}^{\frac{\alpha_1 - \rho\zeta_2}{\sqrt{1 - \rho^2}}} \exp\left(-\frac{\zeta'^2}{2}\right) d\zeta' \\
&= \Phi_1\left(\frac{\alpha_1 - \rho\zeta_2}{\sqrt{1 - \rho^2}}\right) \\
&= \mathbb{I}\left\{\frac{\alpha_1 - \rho\zeta_2}{\sqrt{1 - \rho^2}} \geq 0\right\} + \varepsilon\left(\frac{\alpha_1 - \rho\zeta_2}{\sqrt{1 - \rho^2}}\right) \\
&= \mathbb{I}\{\alpha_1 \geq \rho\zeta_2\} + \varepsilon\left(\frac{\alpha_1 - \rho\zeta_2}{\sqrt{1 - \rho^2}}\right)
\end{aligned}
$$

Where $\left| \varepsilon \left( \frac{\alpha_1 - \rho \zeta_2}{\sqrt{1-\rho^2}} \right) \right| \leq \exp \left( -\frac{(\alpha_1 - \rho \zeta_2)^2}{2(1-\rho^2)} \right)$. Therefore

$$\mathbf{\Phi}_2 (\alpha_1, \alpha_2, \rho) = \int_{-\infty}^{\alpha_2} \left( \mathbb{I} \{ \alpha_1 \geq \rho \zeta_2 \} + \varepsilon \left( \frac{\alpha_1 - \rho \zeta_2}{\sqrt{1-\rho^2}} \right) \right) f_{z_2} (\zeta_2) \, d\zeta_2$$

$$= \underbrace{\int_{-\infty}^{\alpha_2} \mathbb{I} \{ \alpha_1 \geq \rho \zeta_2 \} f_{z_2} (\zeta_2) \, d\zeta_2}_{\mathcal{I}_1} + \underbrace{\int_{-\infty}^{\alpha_2} \varepsilon \left( \frac{\alpha_1 - \rho \zeta_2}{\sqrt{1-\rho^2}} \right) f_{z_2} (\zeta_2) \, d\zeta_2}_{\mathcal{I}_2}$$

For $\mathcal{I}_1$, we have

$$\mathcal{I}_1 = \int_{-\infty}^{\alpha_2} \mathbb{I} \{ \alpha_1 \geq \rho \zeta_2 \} f_{z_2} (\zeta_2) \, d\zeta_2$$

$$= \int_{-\infty}^{\min \left\{ \frac{\alpha_1}{\rho}, \alpha_2 \right\}} f_{z_2} (\zeta_2) \, d\zeta_2$$

$$= \Phi_1 \left( \min \left\{ \frac{\alpha_1}{\rho}, \alpha_2 \right\} \right)$$

$$= \mathbb{I} \left\{ \min \left\{ \frac{\alpha_1}{\rho}, \alpha_2 \right\} \geq 0 \right\} + \varepsilon \left( \min \left\{ \frac{\alpha_1}{\rho}, \alpha_2 \right\} \right)$$

Notice that, when $\alpha_1 \geq 0$ and $\alpha_2 \geq 0$, we must have $\min \left\{ \frac{\alpha_1}{\rho}, \alpha_2 \right\} \geq 0$. Conversely, when $\min \left\{ \frac{\alpha_1}{\rho}, \alpha_2 \right\} \geq 0$, we must have that $\alpha_1 \geq 0$ and $\alpha_2 \geq 0$. Therefore $\mathbb{I} \left\{ \min \left\{ \frac{\alpha_1}{\rho}, \alpha_2 \right\} \geq 0 \right\} = \mathbb{I} \{ \alpha_1 \geq 0; \alpha_2 \geq 0 \}$. Thus, we have

$$\mathbf{\Phi}_2 (\alpha_1, \alpha_2, \rho) = \mathbb{I} \{ \alpha_1 \geq 0; \alpha_2 \geq 0 \} + \varepsilon \left( \min \left\{ \frac{\alpha_1}{\rho}, \alpha_2 \right\} \right) + \mathcal{I}_2$$

For $\mathcal{I}_2$, we have

$$|\mathcal{I}_2| \leq \int_{-\infty}^{\alpha_2} \left| \varepsilon \left( \frac{\alpha_1 - \rho \zeta_2}{\sqrt{1-\rho^2}} \right) \right| f_{z_2} (\zeta_2) \, d\zeta_2$$

$$\leq \int_{-\infty}^{\alpha_2} \exp \left( -\frac{(\alpha_1 - \rho \zeta_2)^2}{2(1-\rho^2)} \right) f_{z_2} (\zeta_2) \, d\zeta_2$$

$$= \frac{1}{\sqrt{2\pi}} \int_{-\infty}^{\alpha_2} \exp \left( -\frac{(\alpha_1 - \rho \zeta_2)^2}{2(1-\rho^2)} - \frac{\zeta_2^2}{2} \right) d\zeta_2$$

$$= \frac{1}{\sqrt{2\pi}} \exp \left( -\frac{\alpha_1^2}{2} \right) \int_{-\infty}^{\alpha_2} \exp \left( -\frac{(\zeta_2 - \rho \alpha_1)^2}{2 (1-\rho^2)} \right) d\zeta_2$$

$$\leq \exp \left( -\frac{\alpha_1^2}{2} \right)$$

Moreover, since $\left| \varepsilon \left( \min \left\{ \frac{\alpha_1}{\rho}, \alpha_2 \right\} \right) \right| \leq \exp \left( -\frac{1}{2} \min \left\{ \frac{\alpha_1}{\rho}, \alpha_2 \right\}^2 \right)$, we have that

$$|\mathbf{\Phi}_2 (\alpha_1, \alpha_2, \rho) - \mathbb{I} \{ \alpha_1 \geq 0; \alpha_2 \geq 0 \}| \leq \exp \left( -\frac{1}{2} \min \left\{ \frac{\alpha_1}{\rho}, \alpha_2 \right\}^2 \right) + \exp \left( -\frac{\alpha_1^2}{2} \right)$$

$$\leq \exp \left( -\frac{\min \{ \alpha_1, \alpha_2 \}^2}{2} \right) + \exp \left( -\frac{\alpha_1^2}{2} \right)$$

Exchanging $\alpha_1$ and $\alpha_2$, we have

$$|\mathbf{\Phi}_2\left(\alpha_1, \alpha_2, \rho\right) - \mathbb{I}\left\{\alpha_1 \geq 0; \alpha_2 \geq 0\right\}| \leq \exp\left(-\frac{1}{2}\min\left\{\frac{\alpha_1}{\rho}, \alpha_2\right\}^2\right) + \exp\left(-\frac{\alpha_1^2}{2}\right)$$

$$\leq \exp\left(-\frac{\min\left\{\alpha_1, \alpha_2\right\}^2}{2}\right) + \exp\left(-\frac{\alpha_2^2}{2}\right)$$

Thus, we have

$$|\mathbf{\Phi}_2\left(\alpha_1, \alpha_2, \rho\right) - \mathbb{I}\left\{\alpha_1 \geq 0; \alpha_2 \geq 0\right\}| \leq 2\exp\left(-\frac{\min\left\{\alpha_1, \alpha_2\right\}^2}{2}\right)$$

$\square$

### D.1.3 Uni-variate Coupled Expectation

**Lemma D.6.** *let* $z_1, z_2 \sim \mathcal{N}\left(0, 1\right)$ *with* $Cov\left(z_1, z_2\right) = \rho$. *Let* $a, b \in \mathbb{R}$. *Define*

$$T_1 = \exp\left(-\frac{a^2}{2}\right)\mathbf{\Phi}_1\left(\frac{\rho a - b}{\sqrt{1 - \rho^2}}\right); \quad T_2 = \exp\left(-\frac{b^2}{2}\right)\mathbf{\Phi}_1\left(\frac{\rho b - a}{\sqrt{1 - \rho^2}}\right)$$

*Then we have that*

$$\mathbb{E}\left[z_1\mathbb{I}\left\{z_1 \geq a; z_2 \geq b\right\}\right] = \frac{1}{\sqrt{2\pi}}\left(T_1 + \rho T_2\right); \quad \mathbb{E}\left[z_2\mathbb{I}\left\{z_1 \geq a; z_2 \geq b\right\}\right] = \frac{1}{\sqrt{2\pi}}\left(T_2 + \rho T_1\right)$$

*Proof.* Writing the expectation in integral form, we have that

$$\mathbb{E}\left[z_1\mathbb{I}\left\{z_1 \geq a; z_2 \geq b\right\}\right] = \int_b^\infty \int_a^\infty z_1 f\left(z_1, z_2\right) dz_1 dz_2 = \int_b^\infty \left(\int_a^\infty z_1 f\left(z_1 \mid z_2\right) dz_1\right) f\left(z_2\right) dz_2$$

Since $z_1, z_2 \sim \mathcal{N}\left(0, 1\right)$ with covariance $\rho$, we have that $z_1 \mid z_2 \sim \mathcal{N}\left(\rho z_2, 1 - \rho^2\right)$. Therefore, let $\rho' = 1 - \rho^2$, we have

$$f\left(z_1 \mid z_2\right) = \frac{1}{\sqrt{2\pi\rho'}}\exp\left(-\frac{\left(z_1 - \rho z_2\right)^2}{2\rho'}\right)$$

Thus, by Lemma D.14, we have that

$$\int_a^\infty z_1 f\left(z_1 \mid z_2\right) dz_1 = \frac{1}{\sqrt{2\pi\rho'}}\int_a^\infty z_1 \exp\left(-\frac{\left(z_1 - \rho z_2\right)^2}{2\rho'}\right) dz_1$$

$$= \frac{1}{\sqrt{2\pi\rho'}}\left(\rho'\exp\left(-\frac{\left(a - \rho z_2\right)^2}{2\rho'}\right) + \rho z_2\sqrt{2\pi\rho'}\mathbf{\Phi}_1\left(\frac{\rho z_2 - a}{\sqrt{\rho'}}\right)\right)$$

$$= \sqrt{\frac{\rho'}{2\pi}}\exp\left(-\frac{\left(a - \rho z_2\right)^2}{2\rho'}\right) + \rho z_2\mathbf{\Phi}_1\left(\frac{\rho z_2 - a}{\sqrt{\rho'}}\right)$$

where we set $\kappa = \sqrt{\rho'}$ and $\mu = \rho z_2$ in Lemma D.14. Plugging into the original integral gives

$$\mathbb{E}\left[z_1\mathbb{I}\left\{z_1 \geq a; z_2 \geq b\right\}\right] = \int_b^\infty \left(\sqrt{\frac{\rho'}{2\pi}}\exp\left(-\frac{\left(a - \rho z_2\right)^2}{2\rho'}\right) + \rho z_2\mathbf{\Phi}_1\left(\frac{\rho z_2 - a}{\sqrt{\rho'}}\right)\right) f\left(z_2\right) dz_2$$

$$= \frac{\sqrt{\rho'}}{2\pi}\int_b^\infty \exp\left(-\frac{\left(a - \rho z_2\right)^2}{2\rho'} - \frac{z_2^2}{2}\right) dz_2 + \rho\int_b^\infty z_2\mathbf{\Phi}_1\left(\frac{\rho z_2 - a}{\sqrt{\rho'}}\right) f\left(z_2\right) dz_2$$

Notice that

$$\exp\left(-\frac{(a-\rho z_2)^2}{2\rho'}-\frac{z_2^2}{2}\right)=\exp\left(-\frac{a^2-2\rho a z_2+z_2^2}{2\rho'}\right)=\exp\left(-\frac{(z_2-\rho a)^2}{2\rho'}\right)\cdot\exp\left(-\frac{a^2}{2}\right)$$

From a previous result, we have an identity for the conditional probability, which expresses the CDF term as an integral:

$$\mathbf{\Phi}_1\left(\frac{\rho z_2-a}{\sqrt{1-\rho^2}}\right)=\int_a^\infty f(z_1\mid z_2)dz_1 \tag{69}$$

and so:

$$\rho\int_b^\infty z_2\mathbf{\Phi}_1\left(\frac{\rho z_2-a}{\sqrt{1-\rho^2}}\right)f(z_2)dz_2=\rho\int_b^\infty z_2\int_a^\infty f(z_1\mid z_2)dz_1 f(z_2)dz_2$$

Moreover, by applying Lemma D.16, we have

$$\mathbb{E}\left[z_1\mathbb{I}\left\{z_1\geq a;z_2\geq b\right\}\right]=\frac{\sqrt{\rho'}}{2\pi}\exp\left(-\frac{a^2}{2}\right)\int_b^\infty\exp\left(-\frac{(z_2-\rho a)^2}{2\rho'}\right)dz_2+\rho\int_b^\infty z_2\int_a^\infty f(z_1\mid z_2)dz_1 f(z_2)dz_2$$

$$=\frac{\rho'}{\sqrt{2\pi}}\exp\left(-\frac{a^2}{2}\right)\int_b^\infty f\left(z_2\mid z_1=a\right)dz_2+\rho\int_b^\infty\int_a^\infty z_2\underbrace{f(z_1\mid z_2)f(z_2)}_{f(z_1,z_2)}dz_1 dz_2$$

$$=\frac{\rho'}{\sqrt{2\pi}}\exp\left(-\frac{a^2}{2}\right)\mathbf{\Phi}_1\left(\frac{\rho a-b}{\sqrt{1-\rho^2}}\right)+\rho\mathbb{E}\left[z_2\mathbb{I}\left\{z_1\geq a;z_2\geq b\right\}\right]$$

Therefore, we can conclude that

$$\mathbb{E}\left[z_1\mathbb{I}\left\{z_1\geq a;z_2\geq b\right\}\right]-\rho\mathbb{E}\left[z_2\mathbb{I}\left\{z_1\geq a;z_2\geq b\right\}\right]=\frac{\rho'}{\sqrt{2\pi}}\exp\left(-\frac{a^2}{2}\right)\mathbf{\Phi}_1\left(\frac{\rho a-b}{\sqrt{1-\rho^2}}\right)=\frac{\rho'}{\sqrt{2\pi}}T_1 \tag{70}$$

Switching $z_1,z_2$ and $a,b$ gives

$$\mathbb{E}\left[z_2\mathbb{I}\left\{z_1\geq a;z_2\geq b\right\}\right]-\rho\mathbb{E}\left[z_1\mathbb{I}\left\{z_1\geq a;z_2\geq b\right\}\right]=\frac{\rho'}{\sqrt{2\pi}}\exp\left(-\frac{b^2}{2}\right)\mathbf{\Phi}_1\left(\frac{\rho b-a}{\sqrt{1-\rho^2}}\right)=\frac{\rho'}{\sqrt{2\pi}}T_2 \tag{71}$$

Solving for $\mathbb{E}\left[z_1\mathbb{I}\left\{z_1\geq a;z_2\geq b\right\}\right]$ and $\mathbb{E}\left[z_2\mathbb{I}\left\{z_1\geq a;z_2\geq b\right\}\right]$ from (70) and (71) gives

$$\mathbb{E}\left[z_1\mathbb{I}\left\{z_1\geq a;z_2\geq b\right\}\right]=\frac{1}{\sqrt{2\pi}}\left(T_1+\rho T_2\right);\quad\mathbb{E}\left[z_2\mathbb{I}\left\{z_1\geq a;z_2\geq b\right\}\right]=\frac{1}{\sqrt{2\pi}}\left(T_2+\rho T_1\right)$$

$\square$

**Lemma D.7.** *Let* $z_1,z_2\sim\mathcal{N}\left(0,1\right)$ *with* $Cov\left(z_1,z_2\right)=\rho$. *Let* $a,b\in\mathbb{R}$. *Define*

$$T_1=\exp\left(-\frac{a^2}{2}\right)\mathbf{\Phi}_1\left(\frac{\rho a-b}{\sqrt{1-\rho^2}}\right);\quad T_2=\exp\left(-\frac{b^2}{2}\right)\mathbf{\Phi}_1\left(\frac{\rho b-a}{\sqrt{1-\rho^2}}\right)$$

*Then we have that*

$$\mathbb{E}\left[z_1^2\mathbb{I}\left\{z_1\geq a;z_2\geq b\right\}\right]=\frac{\rho\sqrt{1-\rho^2}}{2\pi}\exp\left(-\frac{a^2-2\rho ab+b^2}{2\left(1-\rho^2\right)}\right)+\mathbf{\Phi}_2\left(-a,-b,\rho\right)+\frac{1}{\sqrt{2\pi}}\left(aT_1+\rho^2 bT_2\right)$$

$$\mathbb{E}\left[z_2^2\mathbb{I}\left\{z_1\geq a;z_2\geq b\right\}\right]=\frac{\rho\sqrt{1-\rho^2}}{2\pi}\exp\left(-\frac{a^2-2\rho ab+b^2}{2\left(1-\rho^2\right)}\right)+\mathbf{\Phi}_2\left(-a,-b,\rho\right)+\frac{1}{\sqrt{2\pi}}\left(bT_2+\rho^2 aT_1\right)$$

*Proof.* Again, we write the expectation in the integral form to get that

$$\mathbb{E}\left[z_1^2 \mathbb{I}\{z_1 \geq a; z_2 \geq b\}\right] = \int_b^\infty \int_a^\infty z_1^2 f(z_1, z_2) \, dz_1 dz_2 = \int_b^\infty \left(\int_a^\infty z_1^2 f(z_1 \mid z_2) \, dz_1\right) f(z_2) \, dz_2$$

Since $z_1, z_2 \sim \mathcal{N}(0,1)$ with $\mathrm{Cov}(z_1, z_2) = \rho$, we have that $z_1 \mid z_2 \sim \mathcal{N}(\rho z_2, 1 - \rho^2)$. Define $\rho' = 1 - \rho^2$. By Lemma D.15, we have that

$$\int_a^\infty z_1^2 f(z_1 \mid z_2) \, dz_1 = \frac{1}{\sqrt{2\pi\rho'}} \int_a^\infty z_1^2 \exp\left(-\frac{(z_1 - \rho z_2)^2}{2\rho'}\right) dz_1$$

$$= \frac{1}{\sqrt{2\pi\rho'}} \left(\rho'(a + \rho z_2) \exp\left(-\frac{(a - \rho z_2)^2}{2\rho'}\right) + \sqrt{2\pi\rho'}\left(\rho' + \rho^2 z_2^2\right) \mathbf{\Phi}_1\left(\frac{\rho z_2 - a}{\sqrt{\rho'}}\right)\right)$$

$$= \rho^2 z_2^2 \mathbf{\Phi}_1\left(\frac{\rho z_2 - a}{\sqrt{\rho'}}\right) + \rho z_2 \sqrt{\frac{\rho'}{2\pi}} \exp\left(-\frac{(a - \rho z_2)^2}{2\rho'}\right)$$

$$+ a\sqrt{\frac{\rho'}{2\pi}} \exp\left(-\frac{(a - \rho z_2)^2}{2\rho'}\right) + \rho' \mathbf{\Phi}_1\left(\frac{\rho z_2 - a}{\sqrt{\rho'}}\right)$$

where we set $\kappa = \sqrt{\rho'}$ and $\mu = \rho z_2$ in Lemma D.15. Therefore, we have

$$\mathbb{E}\left[z_1^2 \mathbb{I}\{z_1 \geq a; z_2 \geq b\}\right] = \rho^2 \int_b^\infty z_2^2 \mathbf{\Phi}_1\left(\frac{\rho z_2 - a}{\sqrt{\rho'}}\right) f(z_2) \, dz_2$$

$$+ \rho\sqrt{\frac{\rho'}{2\pi}} \int_b^\infty z_2 \exp\left(-\frac{(a - \rho z_2)^2}{2\rho'}\right) f(z_2) \, dz_2$$

$$+ a\sqrt{\frac{\rho'}{2\pi}} \int_b^\infty \exp\left(-\frac{(a - \rho z_2)^2}{2\rho'}\right) f(z_2) \, dz_2$$

$$+ \rho' \int_b^\infty \mathbf{\Phi}_1\left(\frac{\rho z_2 - a}{\sqrt{\rho'}}\right) f(z_2) \, dz_2$$

To start, by Lemma D.16, we have

$$\mathbf{\Phi}_1\left(\frac{\rho z_2 - a}{\sqrt{\rho'}}\right) f(z_2) = \int_a^\infty f(z_1 \mid z_2) f(z_2) \, dz_1 = \int_a^\infty f(z_1, z_2) \, dz_1$$

Therefore, for the first term, we have

$$\int_b^\infty z_2^2 \mathbf{\Phi}_1\left(\frac{\rho z_2 - a}{\sqrt{\rho'}}\right) f(z_2) \, dz_2 = \int_b^\infty \int_a^\infty z_2^2 f(z_1, z_2) \, dz_2 = \mathbb{E}\left[z_2^2 \mathbb{I}\{z_1 \geq a; z_2 \geq b\}\right]$$

For the last term, we have

$$\int_b^\infty \mathbf{\Phi}_1\left(\frac{\rho z_2 - a}{\sqrt{\rho'}}\right) f(z_2) \, dz_2 = \int_b^\infty \int_a^\infty f(z_1, z_2) \, dz_1 dz_2 = \mathbf{\Phi}_2(-a, -b, \rho)$$

Next, we notice that

$$\exp\left(-\frac{(a - \rho z_2)^2}{2\rho'}\right) f(z_2) = \frac{1}{\sqrt{2\pi}} \exp\left(-\frac{a^2 - 2\rho a z_2 + z_2^2}{2\rho'}\right) = \frac{1}{\sqrt{2\pi}} \exp\left(-\frac{a^2}{2}\right) \exp\left(-\frac{(z_2 - \rho a)^2}{2\rho'}\right)$$

Therefore, for the second term, we apply Lemma D.14 to get that

$$\int_b^\infty z_2 \exp\left(-\frac{(a - \rho z_2)^2}{2\rho'}\right) f(z_2) \, dz_2 = \frac{1}{\sqrt{2\pi}} \exp\left(-\frac{a^2}{2}\right) \int_b^\infty z_2 \exp\left(-\frac{(z_2 - \rho a)^2}{2\rho'}\right) dz_2$$

$$= \frac{1}{\sqrt{2\pi}} \exp\left(-\frac{a^2}{2}\right) \left(\rho' \exp\left(-\frac{(\rho a - b)^2}{2\rho'}\right) + \sqrt{2\pi\rho'}\rho a \mathbf{\Phi}_1\left(\frac{\rho a - b}{\sqrt{\rho'}}\right)\right)$$

$$= \frac{\rho'}{\sqrt{2\pi}} \exp\left(-\frac{a^2 - 2\rho ab + b^2}{2\rho'}\right) + \rho a\sqrt{\rho'} \exp\left(-\frac{a^2}{2}\right) \mathbf{\Phi}_1\left(\frac{\rho a - b}{\sqrt{\rho'}}\right)$$

where we set $\kappa = \sqrt{\rho'}$ and $\mu = \rho a$ in Lemma D.14. For the third term, we have

$$\int_b^\infty \exp\left(-\frac{(a - \rho z_2)^2}{2\rho'}\right) f(z_2)\, dz_2 = \frac{1}{\sqrt{2\pi}} \exp\left(-\frac{a^2}{2}\right) \int_b^\infty \exp\left(-\frac{(z_2 - \rho a)^2}{2\rho'}\right) dz_2$$

$$= \sqrt{\rho'} \exp\left(-\frac{a^2}{2}\right) \int_b^\infty f(z_2 \mid z_1 = a)\, dz_2$$

$$= \sqrt{\rho'} \exp\left(-\frac{a^2}{2}\right) \mathbf{\Phi}_1\left(\frac{\rho a - b}{\sqrt{\rho'}}\right)$$

Combining all four terms gives

$$\mathbb{E}\left[z_1^2 \mathbb{I}\{z_1 \geq a; z_2 \geq b\}\right] = \rho\sqrt{\frac{\rho'}{2\pi}}\left(\frac{\rho'}{\sqrt{2\pi}} \exp\left(-\frac{a^2 - 2\rho ab + b^2}{2\rho'}\right) + \rho a \sqrt{\rho'} \exp\left(-\frac{a^2}{2}\right) \mathbf{\Phi}_1\left(\frac{\rho a - b}{\sqrt{\rho'}}\right)\right)$$

$$+ a\sqrt{\frac{\rho'}{2\pi}} \cdot \sqrt{\rho'} \exp\left(-\frac{a^2}{2}\right) \mathbf{\Phi}_1\left(\frac{\rho a - b}{\sqrt{\rho'}}\right)$$

$$+ \rho^2 \mathbb{E}\left[z_2^2 \mathbb{I}\{z_1 \geq a; z_2 \geq b\}\right] + \rho' \mathbf{\Phi}_2(-a, -b, \rho)$$

$$= \frac{\rho'^{\frac{3}{2}} \rho}{2\pi} \exp\left(-\frac{a^2 - 2\rho ab + b^2}{2\rho'}\right) + \frac{\rho' a}{\sqrt{2\pi}} (\rho^2 + 1) \exp\left(-\frac{a^2}{2}\right) \mathbf{\Phi}_1\left(\frac{\rho a - b}{\sqrt{\rho'}}\right)$$

$$+ \rho^2 \mathbb{E}\left[z_2^2 \mathbb{I}\{z_1 \geq a; z_2 \geq b\}\right] + \rho' \mathbf{\Phi}_2(-a, -b, \rho)$$

Therefore, we can conclude that

$$\mathbb{E}\left[z_1^2 \mathbb{I}\{z_1 \geq a; z_2 \geq b\}\right] - \rho^2 \mathbb{E}\left[z_2^2 \mathbb{I}\{z_1 \geq a; z_2 \geq b\}\right]$$

$$= \frac{\rho'^{\frac{3}{2}} \rho}{2\pi} \exp\left(-\frac{a^2 - 2\rho ab + b^2}{2\rho'}\right) + \frac{\rho' a}{\sqrt{2\pi}} (\rho^2 + 1) T_1 + \rho' \mathbf{\Phi}_2(-a, -b, \rho)$$

$$\mathbb{E}\left[z_2^2 \mathbb{I}\{z_1 \geq a; z_2 \geq b\}\right] - \rho^2 \mathbb{E}\left[z_1^2 \mathbb{I}\{z_1 \geq a; z_2 \geq b\}\right]$$

$$= \frac{\rho'^{\frac{3}{2}} \rho}{2\pi} \exp\left(-\frac{a^2 - 2\rho ab + b^2}{2\rho'}\right) + \frac{\rho' b}{\sqrt{2\pi}} (\rho^2 + 1) T_2 + \rho' \mathbf{\Phi}_2(-a, -b, \rho)$$

Solving for $\mathbb{E}\left[z_1^2 \mathbb{I}\{z_1 \geq a; z_2 \geq b\}\right]$ and $\mathbb{E}\left[z_2^2 \mathbb{I}\{z_1 \geq a; z_2 \geq b\}\right]$ gives

$$\mathbb{E}\left[z_1^2 \mathbb{I}\{z_1 \geq a; z_2 \geq b\}\right] = \frac{\rho\sqrt{\rho'}}{2\pi} \exp\left(-\frac{a^2 - 2\rho ab + b^2}{2\rho'}\right) + \mathbf{\Phi}_2(-a, -b, \rho) + \frac{1}{\sqrt{2\pi}}\left(aT_1 + \rho^2 bT_2\right)$$

$$\mathbb{E}\left[z_2^2 \mathbb{I}\{z_1 \geq a; z_2 \geq b\}\right] = \frac{\rho\sqrt{\rho'}}{2\pi} \exp\left(-\frac{a^2 - 2\rho ab + b^2}{2\rho'}\right) + \mathbf{\Phi}_2(-a, -b, \rho) + \frac{1}{\sqrt{2\pi}}\left(bT_2 + \rho^2 aT_1\right)$$

Write $\rho' = 1 - \rho^2$ gives the desired result. $\qquad\square$

**Lemma D.8.** *Let $z_1, z_2 \sim \mathcal{N}(0, 1)$ with $Cov(z_1, z_2) = \rho$. Let $a, b \in \mathbb{R}$. Define*

$$T_1 = \exp\left(-\frac{a^2}{2}\right) \mathbf{\Phi}_1\left(\frac{\rho a - b}{\sqrt{1 - \rho^2}}\right); \quad T_2 = \exp\left(-\frac{b^2}{2}\right) \mathbf{\Phi}_1\left(\frac{\rho b - a}{\sqrt{1 - \rho^2}}\right)$$

*Then we have that*

$$\mathbb{E}\left[z_1 z_2 \mathbb{I}\{z_1 \geq a; z_2 \geq b\}\right] = \frac{\sqrt{1 - \rho^2}}{2\pi} \exp\left(-\frac{a^2 - 2\rho ab + b^2}{2(1 - \rho^2)}\right) + \rho \mathbf{\Phi}_2(-a, -b, \rho) + \frac{\rho}{\sqrt{2\pi}}\left(aT_1 + bT_2\right)$$

*Proof.* To start, we write out the integral form of the expectation as

$$\mathbb{E}\left[z_1 z_2 \mathbb{I}\{z_1 \geq a; z_2 \geq b\}\right] = \int_b^\infty \int_a^\infty z_1 z_2 f(z_1, z_2)\, dz_1 dz_2 = \int_b^\infty \left(\int_a^\infty z_1 f(z_1 \mid z_2)\, dz_1\right) z_2 f(z_2)\, dz_2$$

Since $z_1, z_2 \sim \mathcal{N}(0,1)$ with covariance $\rho$, we have that $z_1 \mid z_2 \sim \mathcal{N}(\rho z_2, 1 - \rho^2)$. Therefore, let $\rho' = 1 - \rho^2$, we have

$$f(z_1 \mid z_2) = \frac{1}{\sqrt{2\pi\rho'}} \exp\left(-\frac{(z_1 - \rho z_2)^2}{2\rho'}\right)$$

Thus, by Lemma D.14, we have that

$$
\int_a^\infty z_1 f(z_1 \mid z_2) \, dz_1 = \frac{1}{\sqrt{2\pi\rho'}} \int_a^\infty z_1 \exp\left(-\frac{(z_1 - \rho z_2)^2}{2\rho'}\right) dz_1
$$

$$
= \frac{1}{\sqrt{2\pi\rho'}} \left(\rho' \exp\left(-\frac{(a - \rho z_2)^2}{2\rho'}\right) + \rho z_2 \sqrt{2\pi\rho'} \, \Phi_1\left(\frac{\rho z_2 - a}{\sqrt{\rho'}}\right)\right)
$$

$$
= \sqrt{\frac{\rho'}{2\pi}} \exp\left(-\frac{(a - \rho z_2)^2}{2\rho'}\right) + \rho z_2 \Phi_1\left(\frac{\rho z_2 - a}{\sqrt{\rho'}}\right)
$$

where we set $\kappa = \sqrt{\rho'}$ and $\mu = \rho z_2$. Therefore

$$
\mathbb{E}\left[z_1 z_2 \mathbb{I}\{z_1 \geq a; z_2 \geq b\}\right] = \int_b^\infty \left(\sqrt{\frac{\rho'}{2\pi}} \exp\left(-\frac{(a - \rho z_2)^2}{2\rho'}\right) + \rho z_2 \Phi_1\left(\frac{\rho z_2 - a}{\sqrt{\rho'}}\right)\right) z_2 f(z_2) \, dz_2
$$

$$
= \sqrt{\frac{\rho'}{2\pi}} \int_b^\infty z_2 \exp\left(-\frac{(a - \rho z_2)^2}{2\rho'}\right) f(z_2) \, dz_2 + \rho \int_b^\infty z_2^2 \Phi_1\left(\frac{\rho z_2 - a}{\sqrt{\rho'}}\right) f(z_2) \, dz_2
$$

To start, by Lemma D.16, we have

$$
\Phi_1\left(\frac{\rho z_2 - a}{\sqrt{\rho'}}\right) f(z_2) = \int_a^\infty f(z_1 \mid z_2) \, dz_1 f(z_2) = \int_a^\infty f(z_1, z_2) \, dz_1
$$

Therefore, for the second term, we have

$$
\int_b^\infty z_2^2 \Phi_1\left(\frac{\rho z_2 - a}{\sqrt{\rho'}}\right) f(z_2) \, dz_2 = \int_b^\infty \int_a^\infty z_2^2 f(z_1, z_2) \, dz_1 dz_2 = \mathbb{E}\left[z_2^2 \mathbb{I}\{z_1 \geq a; z_2 \geq b\}\right]
$$

For the first term, we have

$$
\exp\left(-\frac{(a - \rho z_2)^2}{2\rho'}\right) f(z_2) = \frac{1}{\sqrt{2\pi}} \exp\left(-\frac{z_2^2 - 2\rho a z_2 + a^2}{2\rho'}\right) = \frac{1}{\sqrt{2\pi}} \exp\left(-\frac{a^2}{2}\right) \exp\left(-\frac{(z_2 - \rho a)^2}{2\rho'}\right)
$$

Therefore, by Lemma D.14, the first term can be written as

$$
\int_b^\infty z_2 \exp\left(-\frac{(a - \rho z_2)^2}{2\rho'}\right) f(z_2) \, dz_2 = \frac{1}{\sqrt{2\pi}} \exp\left(-\frac{a^2}{2}\right) \int_b^\infty z_2 \exp\left(-\frac{(z_2 - \rho a)^2}{2\rho'}\right) dz_2
$$

$$
= \frac{1}{\sqrt{2\pi}} \exp\left(-\frac{a^2}{2}\right) \left(\rho' \exp\left(-\frac{(\rho a - b)^2}{2\rho'}\right) + \sqrt{2\pi\rho'} \rho a \Phi_1\left(\frac{\rho a - b}{\sqrt{\rho'}}\right)\right)
$$

$$
= \frac{\rho'}{\sqrt{2\pi}} \exp\left(-\frac{a^2 - 2\rho ab + b^2}{2\rho'}\right) + \rho a \sqrt{\rho'} \exp\left(-\frac{a^2}{2}\right) \Phi_1\left(\frac{\rho a - b}{\sqrt{\rho'}}\right)
$$

where we set $\kappa = \sqrt{\rho'}$ and $\mu = \rho a$ in Lemma D.14. Combining the two terms, we have

$$
\mathbb{E}\left[z_1 z_2 \mathbb{I}\{z_1 \geq a; z_2 \geq b\}\right] = \sqrt{\frac{\rho'}{2\pi}} \left(\frac{\rho'}{\sqrt{2\pi}} \exp\left(-\frac{a^2 - 2\rho ab + b^2}{2\rho'}\right) + \rho a \sqrt{\rho'} \exp\left(-\frac{a^2}{2}\right) \Phi_1\left(\frac{\rho a - b}{\sqrt{\rho'}}\right)\right)
$$

$$
+ \rho \mathbb{E}\left[z_2^2 \mathbb{I}\{z_1 \geq a; z_2 \geq b\}\right]
$$

$$
= \frac{\rho'^{\frac{3}{2}}}{2\pi} \exp\left(-\frac{a^2 - 2\rho ab + b^2}{2\rho'}\right) + \frac{a\rho\rho'}{\sqrt{2\pi}} T_1 + \rho \mathbb{E}\left[z_2^2 \mathbb{I}\{z_1 \geq a; z_2 \geq b\}\right]
$$

From Lemma D.7, we have that

$$\mathbb{E}\left[z_2^2 \mathbb{I}\left\{z_1 \geq a; z_2 \geq b\right\}\right] = \frac{\rho\sqrt{\rho'}}{2\pi}\exp\left(-\frac{a^2 - 2\rho ab + b^2}{2\rho'}\right) + \mathbf{\Phi}_2\left(-a, -b, \rho\right) + \frac{1}{\sqrt{2\pi}}\left(bT_2 + \rho^2 aT_1\right)$$

Therefore

$$\mathbb{E}\left[z_1 z_2 \mathbb{I}\left\{z_1 \geq a; z_2 \geq b\right\}\right] = \frac{\sqrt{\rho'}}{2\pi}\exp\left(-\frac{a^2 - 2\rho ab + b^2}{2\rho'}\right) + \rho\mathbf{\Phi}_2\left(-a, -b, \rho\right) + \frac{\rho}{\sqrt{2\pi}}\left(aT_1 + bT_2\right)$$

Write $\rho' = 1 - \rho^2$ gives the desired result. $\qquad\square$

**Lemma D.9.** Let $z_1 \sim \mathcal{N}\left(\mu_1, \kappa_1^2\right)$ and $z_2 \sim \mathcal{N}\left(\mu_2, \kappa_2^2\right)$, with $Cov(z_1, z_2) = \kappa_1\kappa_2\rho$. Let $a, b \in \mathbb{R}$. Then we have

$$\mathbb{E}\left[z_1 z_2 \mathbb{I}\left\{z_1 \geq a; z_2 \geq b\right\}\right] = \left(\mu_1\mu_2 + \kappa_1\kappa_2\rho\right)\mathbf{\Phi}\left(\frac{\mu_1 - a}{\kappa_1}, \frac{\mu_2 - b}{\kappa_2}, \rho\right)$$

$$+ \frac{\kappa_1\kappa_2}{2\pi}\exp\left(-\frac{1}{2\left(1 - \rho^2\right)}\left(\frac{\left(\mu_1 - a\right)^2}{\kappa_1^2} - \frac{2\rho}{\kappa_1\kappa_2}\left(\mu_1 - a\right)\left(\mu_2 - b\right) + \frac{\left(\mu_2 - b\right)^2}{\kappa_2^2}\right)\right)$$

$$+ \frac{1}{\sqrt{2\pi}}\left(\left(\kappa_2\rho a + \kappa_1\mu_2\right)T_1 + \left(\kappa_1\rho b + \kappa_2\mu_1\right)T_2\right)$$

*Here $T_1, T_2$ are defined as*

$$T_1 = \exp\left(-\frac{\left(a - \mu_1\right)^2}{2\kappa_1^2}\right)\mathbf{\Phi}_1\left(\frac{1}{\sqrt{1 - \rho^2}}\left(\frac{\rho\left(a - \mu_1\right)}{\kappa_1} - \frac{b - \mu_2}{\kappa_2}\right)\right)$$

$$T_2 = \exp\left(-\frac{\left(b - \mu_2\right)^2}{2\kappa_2^2}\right)\mathbf{\Phi}_1\left(\frac{1}{\sqrt{1 - \rho^2}}\left(\frac{\rho\left(b - \mu_2\right)}{\kappa_2} - \frac{a - \mu_1}{\kappa_1}\right)\right)$$

*Proof.* Let $\hat{z}_1 = \frac{z_1 - \mu_1}{\kappa_1}$ and $\hat{z}_1 = \frac{z_2 - \mu_2}{\kappa_2}$. Then we have $\hat{z}_1, \hat{z}_2 \sim \mathcal{N}(0, 1)$. Moreover,

$$\mathrm{Cov}\left(\hat{z}_1, \hat{z}_2\right) = \mathbb{E}\left[\hat{z}_1\hat{z}_2\right] = \frac{1}{\kappa_1\kappa_2}\mathbb{E}\left[\left(z_1 - \mu_1\right)\left(z_2 - \mu_2\right)\right] = \rho$$

Since $z_1 = \kappa_1\hat{z}_1 + \mu_1$ and $z_2 = \kappa_2\hat{z}_2 + \mu_2$, we have

$$\mathbb{E}\left[z_1 z_2 \mathbb{I}\left\{z_1 \geq a; z_2 \geq b\right\}\right] = \mathbb{E}\left[\left(\kappa_1\hat{z}_1 + \mu_1\right)\left(\kappa_2\hat{z}_2 + \mu_2\right)\mathbb{I}\left\{\hat{z}_1 \geq \frac{a - \mu_1}{\kappa_1}; \hat{z}_2 \geq \frac{b - \mu_2}{\kappa_2}\right\}\right]$$

$$= \kappa_1\kappa_2\mathbb{E}\left[\hat{z}_1\hat{z}_2\mathbb{I}\left\{\hat{z}_1 \geq \hat{a}; \hat{z}_2 \geq \hat{b}\right\}\right] + \mu_1\mu_2\mathbb{E}\left[\mathbb{I}\left\{\hat{z}_1 \geq \hat{a}; \hat{z}_2 \geq \hat{b}\right\}\right]$$

$$+ \kappa_1\mu_2\mathbb{E}\left[\hat{z}_1\mathbb{I}\left\{\hat{z}_1 \geq \hat{a}; \hat{z}_2 \geq \hat{b}\right\}\right] + \kappa_2\mu_1\mathbb{E}\left[\hat{z}_2\mathbb{I}\left\{\hat{z}_1 \geq \hat{a}; \hat{z}_2 \geq \hat{b}\right\}\right]$$

where we re-defined $\hat{a} = \frac{a - \mu_1}{\kappa_1}$ and $\hat{b} = \frac{b - \mu_2}{\kappa_2}$. By Lemma D.16, Lemma D.6, and Lemma D.8, we have

$$\mathbb{E}\left[\hat{z}_1\mathbb{I}\left\{\hat{z}_1 \geq \hat{a}; \hat{z}_2 \geq \hat{b}\right\}\right] = \frac{1}{\sqrt{2\pi}}\left(T_1 + \rho T_2\right); \quad \mathbb{E}\left[\hat{z}_2\mathbb{I}\left\{\hat{z}_1 \geq \hat{a}; \hat{z}_2 \geq \hat{b}\right\}\right] = \frac{1}{\sqrt{2\pi}}\left(T_2 + \rho T_1\right)$$

$$\mathbb{E}\left[\hat{z}_1\hat{z}_2\mathbb{I}\left\{\hat{z}_1 \geq \hat{a}; \hat{z}_2 \geq \hat{b}\right\}\right] = \frac{\sqrt{1 - \rho^2}}{2\pi}\exp\left(-\frac{\hat{a}^2 - 2\rho\hat{a}\hat{b} + \hat{b}^2}{2\left(1 - \rho^2\right)}\right) + \rho\mathbf{\Phi}_2\left(-\hat{a}, -\hat{b}, \rho\right) + \frac{\rho}{\sqrt{2\pi}}\left(\hat{a}T_1 + \hat{b}T_2\right)$$

and $\mathbb{E}\left[\mathbb{I}\left\{\hat{z}_1 \geq \hat{a}; \hat{z}_2 \geq \hat{b}\right\}\right] = \mathbf{\Phi}_2\left(-\hat{a}, -\hat{b}, \rho\right)$. Here, $T_1, T_2$ are defined as

$$T_1 = \exp\left(-\frac{\hat{a}^2}{2}\right)\mathbf{\Phi}\left(\frac{\rho\hat{a} - \hat{b}}{\sqrt{1 - \rho^2}}\right); \quad T_2 = \exp\left(-\frac{\hat{b}^2}{2}\right)\mathbf{\Phi}\left(\frac{\rho\hat{b} - \hat{a}}{\sqrt{1 - \rho^2}}\right)$$

Plugging in the value of $\hat{a}$ and $\hat{b}$ gives the desired result. $\qquad\square$

### D.1.4 Multi-variate Coupled Expectation

**Lemma D.10.** *Let* $\mathbf{c} \sim \mathcal{N}\left(\boldsymbol{\mu}, \kappa^2 \mathbf{I}\right)$, *and let* $\mathbf{u} \in \mathbb{R}^d$. *Define* $z = \mathbf{c}^\top \mathbf{u}$. *Then we have*

$$\mathbb{E}\left[\mathbf{c}\mathbb{I}\left\{z \geq 0\right\}\right] = \boldsymbol{\mu}\Phi_1\left(-\frac{\boldsymbol{\mu}^\top \mathbf{u}}{\kappa\left\|\mathbf{u}\right\|_2}\right) + \frac{\kappa}{\sqrt{2\pi}}\exp\left(-\frac{\left(\boldsymbol{\mu}^\top \mathbf{u}\right)^2}{2\kappa^2\left\|\mathbf{u}\right\|_2^2}\right) \cdot \frac{\mathbf{u}}{\left\|\mathbf{u}\right\|_2}$$

*Proof.* According to the law of total expectation,

$$\mathbb{E}\left[\mathbf{c}\mathbb{I}\left\{\mathbf{c}^\top \mathbf{u} \geq 0\right\}\right] = \mathbb{E}_z\left[\mathbb{E}_{\mathbf{c}}\left[\mathbf{c}\mathbb{I}\left\{z \geq 0\right\} \mid z\right]\right] = \mathbb{E}_z\left[\mathbb{E}_{\mathbf{c}}\left[\mathbf{c} \mid z\right]\mathbb{I}\left\{z \geq 0\right\}\right]$$

By Lemma D.1, we have that $\mathbb{E}_{\mathbf{c}}\left[\mathbf{c} \mid z\right] = \boldsymbol{\mu} + \frac{\mathbf{u}}{\|\mathbf{u}\|_2^2}\left(z - \boldsymbol{\mu}^\top \mathbf{u}\right)$. Therefore

$$\mathbb{E}\left[\mathbf{c}\mathbb{I}\left\{\mathbf{c}^\top \mathbf{u} \geq 0\right\}\right] = \frac{\mathbf{u}}{\left\|\mathbf{u}\right\|_2^2}\mathbb{E}_z\left[z\mathbb{I}\left\{z \geq 0\right\}\right] + \left(\boldsymbol{\mu} - \frac{\boldsymbol{\mu}^\top \mathbf{u} \cdot \mathbf{u}}{\left\|\mathbf{u}\right\|_2^2}\right)\mathbb{E}_z\left[\mathbb{I}\left\{z \geq 0\right\}\right]$$

By definition, $\mathbb{E}_z\left[\mathbb{I}\left\{z \geq 0\right\}\right] = \Pr\left(z \geq 0\right) = 1 - \Pr\left(z \leq 0\right)$. Since, by Lemma D.13, $z \sim \mathcal{N}\left(\boldsymbol{\mu}^\top \mathbf{u}, \kappa^2\left\|\mathbf{u}\right\|_2^2\right)$, we have that

$$\Pr\left(z \leq 0\right) = \Pr\left(\frac{z - \boldsymbol{\mu}^\top \mathbf{u}}{\kappa\left\|\mathbf{u}\right\|_2} \leq -\frac{\boldsymbol{\mu}^\top \mathbf{u}}{\left\|\mathbf{u}\right\|_2}\right) = \Phi_1\left(-\frac{\boldsymbol{\mu}^\top \mathbf{u}}{\kappa\left\|\mathbf{u}\right\|_2}\right)$$

Moreover, let $\hat{z} = \frac{z - \boldsymbol{\mu}^\top \mathbf{u}}{\kappa\|\mathbf{u}\|_2}$, then we have

$$\mathbb{E}_z\left[z\mathbb{I}\left\{z \geq 0\right\}\right] = \mathbb{E}_{\hat{z}}\left[\left(\kappa\left\|\mathbf{u}\right\|_2\hat{z} + \boldsymbol{\mu}^\top \mathbf{u}\right)\mathbb{I}\left\{\hat{z} \geq -\frac{\boldsymbol{\mu}^\top \mathbf{u}}{\kappa\left\|\mathbf{u}\right\|_2}\right\}\right]$$

$$= \kappa\left\|\mathbf{u}\right\|_2\mathbb{E}_{\hat{z}}\left[\hat{z}\mathbb{I}\left\{\hat{z} \geq -\frac{\boldsymbol{\mu}^\top \mathbf{u}}{\kappa\left\|\mathbf{u}\right\|_2}\right\}\right] + \boldsymbol{\mu}^\top \mathbf{u}\mathbb{E}_z\left[z \geq 0\right]$$

By the PDF of $\hat{z}$, we have

$$\mathbb{E}_{\hat{z}}\left[\hat{z}\mathbb{I}\left\{z \geq 0\right\}\right] = \frac{1}{\sqrt{2\pi}}\int_a^\infty z\exp\left(-\frac{z^2}{2}\right)dz = -\frac{1}{\sqrt{2\pi}}\exp\left(-\frac{z^2}{2}\right)\Big|_a^\infty = \frac{1}{\sqrt{2\pi}}\exp\left(-\frac{a^2}{2}\right)$$

Therefore

$$\mathbb{E}_z\left[z\mathbb{I}\left\{z \geq 0\right\}\right] = \frac{\kappa\left\|\mathbf{u}\right\|_2}{\sqrt{2\pi}}\exp\left(-\frac{\left(\boldsymbol{\mu}^\top \mathbf{u}\right)^2}{2\kappa^2\left\|\mathbf{u}\right\|_2^2}\right) + \boldsymbol{\mu}^\top \mathbf{u}\mathbb{E}_z\left[z \geq 0\right]$$

Plugging in gives

$$\mathbb{E}\left[\mathbf{c}\mathbb{I}\left\{\mathbf{c}^\top \mathbf{u} \geq 0\right\}\right] = \boldsymbol{\mu}\Phi_1\left(-\frac{\boldsymbol{\mu}^\top \mathbf{u}}{\kappa\left\|\mathbf{u}\right\|_2}\right) + \frac{\kappa}{\sqrt{2\pi}}\exp\left(-\frac{\left(\boldsymbol{\mu}^\top \mathbf{u}\right)^2}{2\kappa^2\left\|\mathbf{u}\right\|_2^2}\right) \cdot \frac{\mathbf{u}}{\left\|\mathbf{u}\right\|_2}$$

$\square$

**Lemma D.11.** *Let* $\mathbf{c} \sim \mathcal{N}\left(\boldsymbol{\mu}, \kappa^2 \mathbf{I}\right)$ *with* $\kappa \leq 1$. *Let* $\mathbf{u}, \mathbf{v}$ *be given, and let* $z_1 = \mathbf{c}^\top \mathbf{u}, z_2 = \mathbf{c}^\top \mathbf{v}$. *Then we have that*

$$\left\|\mathbb{E}_{\mathbf{c}}\left[\mathbf{c}\mathbb{I}\left\{z_1 \geq 0\right\}\right] - \boldsymbol{\mu}\Phi_1\left(\frac{\boldsymbol{\mu}^\top \mathbf{u}}{\kappa\left\|\mathbf{u}\right\|_2}\right)\right\|_\infty \leq \kappa\left\|\mathbf{u}\right\|_2\exp\left(-\frac{\left(\boldsymbol{\mu}^\top \mathbf{u}\right)^2}{2\kappa^2\left\|\mathbf{u}\right\|_2}\right)$$

*Proof.* Given the form of the conditional expectation, we have

$$\mathbb{E}_{\mathbf{c}}\left[\mathbf{c}\mathbb{I}\left\{z_1 \geq 0\right\}\right] = \int_0^\infty \mathbb{E}_{\mathbf{c}}\left[\mathbf{c} \mid z_1\right]f_1(z_1)dz_1 = \int_0^\infty \left(\boldsymbol{\mu} + \frac{\mathbf{u}}{\left\|\mathbf{u}\right\|_2^2}\left(z_1 - \boldsymbol{\mu}^\top \mathbf{u}\right)\right)f_1(z_1)dz_1$$

Since $z_1 = \mathbf{c}^\top \mathbf{u}$, we must have that $z_1 \sim \mathcal{N}\left(\boldsymbol{\mu}^\top \mathbf{u}, \kappa^2 \|\mathbf{u}\|_2^2\right)$. Define $z' = \frac{z_1 - \boldsymbol{\mu}^\top \mathbf{u}}{\kappa \|\mathbf{u}\|_2}$, then we have that $z_1 = \kappa \|\mathbf{u}\|_2 z' + \boldsymbol{\mu}^\top \mathbf{u}$

$$
\mathbb{E}_{\mathbf{c}}\left[\mathbf{c}\mathbb{I}\{z_1 \geq 0\}\right] = \int_{\frac{-\boldsymbol{\mu}^\top \mathbf{u}}{\kappa \|\mathbf{u}\|_2}}^{\infty} (\mu + \kappa \mathbf{u} \cdot z') f(z') \, dz'
$$

$$
= \mu \cdot \int_{-\infty}^{\frac{\boldsymbol{\mu}^\top \mathbf{u}}{\kappa \|\mathbf{u}\|_2}} f(z') \, dz' + \kappa \mathbf{u} \cdot \int_{-\infty}^{\frac{\boldsymbol{\mu}^\top \mathbf{u}}{\kappa \|\mathbf{u}\|_2}} z' f(z') \, dz'
$$

$$
= \boldsymbol{\mu}\boldsymbol{\Phi}_1\left(\frac{\boldsymbol{\mu}^\top \mathbf{u}}{\kappa \|\mathbf{u}\|_2}\right) + \kappa \mathbf{u} \exp\left(-\frac{\left(\boldsymbol{\mu}^\top \mathbf{u}\right)^2}{2\kappa^2 \|\mathbf{u}\|_2}\right)
$$

where in the last equality we applied Lemma D.14 with $a = 0$ and $\kappa = 1$ in the lemma. Therefore, we have that

$$
\left\| \mathbb{E}_{\mathbf{c}}\left[\mathbf{c}\mathbb{I}\{z_1 \geq 0\}\right] - \boldsymbol{\mu}\boldsymbol{\Phi}_1\left(\frac{\boldsymbol{\mu}^\top \mathbf{u}}{\kappa \|\mathbf{u}\|_2}\right) \right\|_\infty \leq \kappa \|\mathbf{u}\|_\infty \exp\left(-\frac{\left(\boldsymbol{\mu}^\top \mathbf{u}\right)^2}{2\kappa^2 \|\mathbf{u}\|_2}\right)
$$

$\square$

Now, we shall dive into $\mathbb{E}_{\mathbf{c}}\left[\mathbf{c}\mathbf{c}^\top \mathbb{I}\{z_1 \geq 0; z_2 \geq 0\}\right]$.

**Lemma D.12.** *Let $\mathbf{c} \sim \mathcal{N}\left(\boldsymbol{\mu}, \kappa^2 \mathbf{I}\right)$ with $\|\boldsymbol{\mu}\|_2 \geq 2$ and $\kappa \leq 1$. Let $\mathbf{u}, \mathbf{v}$ be given, and let $z_1 = \boldsymbol{\mu}^\top \mathbf{u}, z_2 = \boldsymbol{\mu}^\top \mathbf{v}$. Then we have that*

$$
\left\| \mathbb{E}_{\mathbf{c}}\left[\mathbf{c}\mathbf{c}^\top \mathbb{I}\{z_1 \geq 0; z_2 \geq 0\}\right] \mathbf{v} - \left(\boldsymbol{\mu}\boldsymbol{\mu}^\top \mathbf{v} + 3\kappa^2 \mathbf{v}\right) \boldsymbol{\Phi}_1\left(\frac{\boldsymbol{\mu}^\top \mathbf{u}}{2\kappa \|\mathbf{v}\|_2}\right) \boldsymbol{\Phi}_1\left(\frac{\boldsymbol{\mu}^\top \mathbf{v}}{2\kappa \|\mathbf{v}\|_2}\right) \right\| \leq \Delta
$$

*where $\Delta$ is given by*

$$
\Delta = 2\kappa \|\mathbf{v}\|_2 \left( \|\boldsymbol{\mu}\|_2 \left( \phi\left(\frac{\boldsymbol{\mu}^\top \mathbf{u}}{2\kappa \|\mathbf{u}\|_2}\right) + \phi\left(\frac{\boldsymbol{\mu}^\top \mathbf{u}}{2\kappa \|\mathbf{v}\|_2}\right) \right) + \|\boldsymbol{\mu}\|_\infty \left( \psi\left(\frac{\boldsymbol{\mu}^\top \mathbf{u}}{2\kappa \|\mathbf{u}\|_2}\right) + \psi\left(\frac{\boldsymbol{\mu}^\top \mathbf{u}}{2\kappa \|\mathbf{u}\|_2}\right) \right) \right)
$$

*Proof.* We have

$$
\mathbb{E}_{\mathbf{c}}\left[\mathbf{c}\mathbf{c}^\top \mathbb{I}\{z_1 \geq 0; z_2 \geq 0\}\right] = \int_{z_1, z_2 \geq 0} \mathbb{E}_{\mathbf{c}}\left[\mathbf{c}\mathbf{c}^\top \mid z_1, z_2\right] f(z_1, z_2) \, dz_1 dz_2
$$

Notice that

$$
\mathbb{E}_{\mathbf{c}}\left[\mathbf{c}\mathbf{c}^\top \mid z_1, z_2\right] = \mathrm{Cov}\left(\mathbf{c} \mid z_1, z_2\right) + \mathbb{E}[\mathbf{c} \mid z_1, z_2]\mathbb{E}[\mathbf{c} \mid z_1, z_2]^\top
$$

By the form of $\mathrm{Cov}\left(\mathbf{c} \mid z_1, z_2\right)$, we have

$$
\mathrm{Cov}\left(\mathbf{c} \mid z_1, z_2\right) = \kappa^2 \mathbf{I} - \frac{\kappa^2 \left(\mathbf{u}\mathbf{v}^\top - \mathbf{v}\mathbf{u}^\top\right)^2}{\|\mathbf{v}\|_2^2 \|\mathbf{u}\|_2^2 - \langle \mathbf{v}, \mathbf{u}\rangle^2} := \mathbf{M}
$$

By the form of $\mathbb{E}[\mathbf{c} \mid z_1, z_2]$ we have

$$
\mathbb{E}[\mathbf{c} \mid z_1, z_2] = z_1 \cdot \mathbf{s}_1 + z_2 \cdot \mathbf{s}_2 + \mathbf{s}_3
$$

where

$$
\mathbf{s}_1 = \frac{\|\mathbf{v}\|_2^2 \mathbf{u} - \langle \mathbf{u}, \mathbf{v}\rangle \mathbf{v}}{\|\mathbf{v}\|_2^2 \|\mathbf{u}\|_2^2 - \langle \mathbf{v}, \mathbf{u}\rangle^2}; \mathbf{s}_2 = \frac{\|\mathbf{u}\|_2^2 \mathbf{v} - \langle \mathbf{u}, \mathbf{v}\rangle \mathbf{u}}{\|\mathbf{v}\|_2^2 \|\mathbf{u}\|_2^2 - \langle \mathbf{v}, \mathbf{u}\rangle^2}
$$

$$
\mathbf{s}_3 = \boldsymbol{\mu} - \frac{\left(\|\mathbf{v}\|_2^2 \mathbf{u} - \langle \mathbf{u}, \mathbf{v}\rangle \mathbf{v}\right) \langle \boldsymbol{\mu}, \mathbf{u}\rangle + \left(\|\mathbf{u}\|_2^2 \mathbf{v} - \langle \mathbf{u}, \mathbf{v}\rangle \mathbf{u}\right) \langle \boldsymbol{\mu}, \mathbf{v}\rangle}{\|\mathbf{v}\|_2^2 \|\mathbf{u}\|_2^2 - \langle \mathbf{v}, \mathbf{u}\rangle^2}
$$

Therefore

$$\mathbb{E}\left[\mathbf{c} \mid z_1, z_2\right] \mathbb{E}\left[\mathbf{c} \mid z_1, z_2\right]^\top = \left(z_1 \cdot \mathbf{s}_1 + z_2 \cdot \mathbf{s}_2 + \mathbf{s}_3\right)\left(z_1 \cdot \mathbf{s}_1 + z_2 \cdot \mathbf{s}_2 + \mathbf{s}_3\right)^\top$$

Recall that we are interested in

$$\mathbb{E}_{\mathbf{c}}\left[\mathbf{cc}^\top \mathbb{I}\{z_1 \geq 0; z_2 \geq 0\}\right] = \int_{z_1, z_2 \geq 0} \left(\mathrm{Cov}\left(\mathbf{c} \mid z_1, z_2\right) + \mathbb{E}[\mathbf{c} \mid z_1, z_2]\mathbb{E}[\mathbf{c} \mid z_1, z_2]^\top\right) f(z_1, z_2) dz_1 dz_2$$

Let

$$\hat{z}_1 = \frac{z_1 - \boldsymbol{\mu}^\top \mathbf{u}}{\kappa \left\|\mathbf{u}\right\|_2}; \quad \hat{z}_2 = \frac{z_1 - \boldsymbol{\mu}^\top \mathbf{v}}{\kappa \left\|\mathbf{v}\right\|_2}$$

Then we have

$$\hat{z}_1, \hat{z}_2 \sim \mathcal{N}(0, 1); \quad \mathrm{Cov}\left(\hat{z}_1, \hat{z}_2\right) = \frac{\mathbf{u}^\top \mathbf{v}}{\left\|\mathbf{u}\right\|_2 \left\|\mathbf{v}\right\|_2} := \rho$$

Thus,

$$\hat{z}_1 \mid \hat{z}_2 = \gamma_2 \sim \mathcal{N}\left(\rho\gamma_2, 1 - \rho^2\right); \quad \hat{z}_2 \mid \hat{z}_1 = \gamma_1 \sim \mathcal{N}\left(\rho\gamma_1, 1 - \rho^2\right)$$

Moreover,

$$\begin{aligned}\mathbb{E}[\mathbf{c} \mid z_1, z_2] &= z_1 \cdot \mathbf{s}_1 + z_2 \cdot \mathbf{s}_2 + \mathbf{s}_3 \\ &= \left(\kappa \left\|\mathbf{u}\right\|_2 \hat{z}_1 + \boldsymbol{\mu}^\top \mathbf{u}\right) \mathbf{s}_1 + \left(\kappa \left\|\mathbf{v}\right\|_2 \hat{z}_2 + \boldsymbol{\mu}^\top \mathbf{v}\right) \mathbf{s}_2 + \mathbf{s}_3\end{aligned}$$

Redefine

$$\hat{\mathbf{s}}_1 = \kappa \left\|\mathbf{u}\right\|_2 \mathbf{s}_1; \quad \hat{\mathbf{s}}_2 = \kappa \left\|\mathbf{v}\right\|_2 \mathbf{s}_2; \quad \hat{\mathbf{s}}_3 = \boldsymbol{\mu}^\top \mathbf{u} \cdot \mathbf{s}_1 + \boldsymbol{\mu}^\top \mathbf{v} \cdot \mathbf{s}_2 + \mathbf{s}_3 = \boldsymbol{\mu}$$

Then we have

$$\mathbb{E}[\mathbf{c} \mid z_1, z_2] = \hat{\mathbf{s}}_1 \hat{z}_1 + \hat{\mathbf{s}}_2 \hat{z}_2 + \hat{\mathbf{s}}_3$$

In this case, let $\hat{f}$ be the joint PDF of $\hat{z}_1$ and $\hat{z}_2$, then we have

$$\begin{aligned}\hat{f}\left(\hat{z}_1, \hat{z}_2\right) &= \frac{1}{2\pi\sqrt{1-\rho^2}} \exp\left(-\frac{1}{2(1-\rho^2)}\left(z_1^2 - 2\rho z_1 z_2 + z_2^2\right)\right) \\ &= \frac{\kappa^2 \left\|\mathbf{u}\right\|_2 \left\|\mathbf{v}\right\|_2}{2\pi\kappa^2 \left\|\mathbf{u}\right\|_2 \left\|\mathbf{v}\right\|_2 \sqrt{1-\rho^2}} \exp\left(-\frac{1}{2(1-\rho^2)}\left(z_1^2 - 2\rho z_1 z_2 + z_2^2\right)\right) \\ &= \kappa^2 \left\|\mathbf{u}\right\|_2 \left\|\mathbf{v}\right\|_2 f\left(\hat{z}_1, \hat{z}_2\right)\end{aligned}$$

Therefore, since $dz_1 = \kappa \left\| \mathbf{u} \right\|_2 d\hat{z}_1$, and $dz_2 = \kappa \left\| \mathbf{v} \right\|_2 d\hat{z}_2$, we have

$$
\begin{aligned}
\mathbb{E}_{\mathbf{c}} \left[ \mathbf{c}\mathbf{c}^\top \mathbb{I} \{ z_1 \geq 0; z_2 \geq 0 \} \right] &= \int_{z_1, z_2 \geq 0} \left( \mathrm{Cov}\left( \mathbf{c} \mid z_1, z_2 \right) + \mathbb{E}[\mathbf{c} \mid z_1, z_2] \mathbb{E}[\mathbf{c} \mid z_1, z_2]^\top \right) f(z_1, z_2) dz_1 dz_2 \\
&= \int_{-\frac{\boldsymbol{\mu}^\top \mathbf{v}}{\kappa \|\mathbf{v}\|_2}}^{\infty} \int_{-\frac{\boldsymbol{\mu}^\top \mathbf{u}}{\kappa \|\mathbf{u}\|_2}}^{\infty} \left( \mathrm{Cov}\left( \mathbf{c} \mid z_1, z_2 \right) + \mathbb{E}[\mathbf{c} \mid z_1, z_2] \mathbb{E}[\mathbf{c} \mid z_1, z_2]^\top \right) \hat{f}(\hat{z}_1, \hat{z}_2) d\hat{z}_1 d\hat{z}_2 \\
&= \hat{\mathbf{s}}_1 \hat{\mathbf{s}}_1^\top \underbrace{\int_{-\frac{\boldsymbol{\mu}^\top \mathbf{v}}{\kappa \|\mathbf{v}\|_2}}^{\infty} \int_{-\frac{\boldsymbol{\mu}^\top \mathbf{u}}{\kappa \|\mathbf{u}\|_2}}^{\infty} \hat{z}_1^2 \hat{f}(\hat{z}_1, \hat{z}_2) d\hat{z}_1 d\hat{z}_2}_{\mathcal{I}_1} \\
&\quad + \hat{\mathbf{s}}_2 \hat{\mathbf{s}}_2^\top \underbrace{\int_{-\frac{\boldsymbol{\mu}^\top \mathbf{v}}{\kappa \|\mathbf{v}\|_2}}^{\infty} \int_{-\frac{\boldsymbol{\mu}^\top \mathbf{u}}{\kappa \|\mathbf{u}\|_2}}^{\infty} \hat{z}_2^2 \hat{f}(\hat{z}_1, \hat{z}_2) d\hat{z}_1 d\hat{z}_2}_{\mathcal{I}_2} \\
&\quad + \left( \hat{\mathbf{s}}_1 \hat{\mathbf{s}}_2^\top + \hat{\mathbf{s}}_2 \hat{\mathbf{s}}_1^\top \right) \underbrace{\int_{-\frac{\boldsymbol{\mu}^\top \mathbf{v}}{\kappa \|\mathbf{v}\|_2}}^{\infty} \int_{-\frac{\boldsymbol{\mu}^\top \mathbf{u}}{\kappa \|\mathbf{u}\|_2}}^{\infty} \hat{z}_1 \hat{z}_2 \hat{f}(\hat{z}_1, \hat{z}_2) d\hat{z}_1 d\hat{z}_2}_{\mathcal{I}_3} \\
&\quad + \left( \hat{\mathbf{s}}_1 \hat{\mathbf{s}}_3 + \hat{\mathbf{s}}_3 \hat{\mathbf{s}}_1^\top \right) \underbrace{\int_{-\frac{\boldsymbol{\mu}^\top \mathbf{v}}{\kappa \|\mathbf{v}\|_2}}^{\infty} \int_{-\frac{\boldsymbol{\mu}^\top \mathbf{u}}{\kappa \|\mathbf{u}\|_2}}^{\infty} \hat{z}_1 \hat{f}(\hat{z}_1, \hat{z}_2) d\hat{z}_1 d\hat{z}_2}_{\mathcal{I}_4} \\
&\quad + \left( \hat{\mathbf{s}}_2 \hat{\mathbf{s}}_3^\top + \hat{\mathbf{s}}_3 \hat{\mathbf{s}}_2^\top \right) \underbrace{\int_{-\frac{\boldsymbol{\mu}^\top \mathbf{v}}{\kappa \|\mathbf{v}\|_2}}^{\infty} \int_{-\frac{\boldsymbol{\mu}^\top \mathbf{u}}{\kappa \|\mathbf{u}\|_2}}^{\infty} \hat{z}_2 \hat{f}(\hat{z}_1, \hat{z}_2) d\hat{z}_1 d\hat{z}_2}_{\mathcal{I}_5} \\
&\quad + \left( \hat{\mathbf{s}}_3 \hat{\mathbf{s}}_3^\top + \mathbf{M} \right) \underbrace{\int_{-\frac{\boldsymbol{\mu}^\top \mathbf{v}}{\kappa \|\mathbf{v}\|_2}}^{\infty} \int_{-\frac{\boldsymbol{\mu}^\top \mathbf{u}}{\kappa \|\mathbf{u}\|_2}}^{\infty} \hat{f}(\hat{z}_1, \hat{z}_2) d\hat{z}_1 d\hat{z}_2}_{\mathcal{I}_6}
\end{aligned}
$$

Since our goal is to study the term $\mathbb{E}_{\mathbf{c}} \left[ \mathbf{c}\mathbf{c}^\top \mathbb{I} \{ z_1 \geq 0; z_2 \geq 0 \} \right] \mathbf{v}$, we need to understand the terms $\mathcal{I}_1$ to $\mathcal{I}_6$, as well as understanding the matrix-vector product in front of these terms. To start, under some standard computation, we have

$$
\begin{aligned}
\hat{\mathbf{s}}_1^\top \mathbf{u} &= \kappa \left\| \mathbf{u} \right\|_2 \cdot \frac{\|\mathbf{v}\|_2^2 \mathbf{u}^\top \mathbf{u} - \mathbf{v}^\top \mathbf{u} \cdot \mathbf{v}^\top \mathbf{u}}{\|\mathbf{v}\|_2^2 \|\mathbf{u}\|_2^2 - (\mathbf{v}^\top \mathbf{u})^2} = \kappa \left\| \mathbf{u} \right\|_2 \\
\hat{\mathbf{s}}_1^\top \mathbf{v} &= \kappa \left\| \mathbf{u} \right\|_2 \cdot \frac{\|\mathbf{v}\|_2^2 \mathbf{u}^\top \mathbf{v} - \mathbf{v}^\top \mathbf{u} \cdot \mathbf{v}^\top \mathbf{v}}{\|\mathbf{v}\|_2^2 \|\mathbf{u}\|_2^2 - (\mathbf{v}^\top \mathbf{u})^2} = 0 \\
\hat{\mathbf{s}}_2^\top \mathbf{u} &= \kappa \left\| \mathbf{v} \right\|_2 \cdot \frac{\|\mathbf{u}\|_2^2 \mathbf{v}^\top \mathbf{u} - \mathbf{v}^\top \mathbf{u} \cdot \mathbf{u}^\top \mathbf{u}}{\|\mathbf{v}\|_2^2 \|\mathbf{u}\|_2^2 - (\mathbf{v}^\top \mathbf{u})^2} = 0 \\
\hat{\mathbf{s}}_2^\top \mathbf{v} &= \kappa \left\| \mathbf{v} \right\|_2 \cdot \frac{\|\mathbf{u}\|_2^2 \mathbf{v}^\top \mathbf{v} - \mathbf{v}^\top \mathbf{u} \cdot \mathbf{u}^\top \mathbf{v}}{\|\mathbf{v}\|_2^2 \|\mathbf{u}\|_2^2 - (\mathbf{v}^\top \mathbf{u})^2} = \kappa \left\| \mathbf{v} \right\|_2
\end{aligned}
$$

Therefore, the following must holds

$$
\begin{aligned}
\hat{\mathbf{s}}_1 \hat{\mathbf{s}}_1^\top \mathbf{v} &= \mathbf{0}; \quad \hat{\mathbf{s}}_2 \hat{\mathbf{s}}_2^\top \mathbf{v} = \kappa \left\| \mathbf{v} \right\|_2 \hat{\mathbf{s}}_2; \quad \left( \hat{\mathbf{s}}_1 \hat{\mathbf{s}}_2^\top + \hat{\mathbf{s}}_2 \hat{\mathbf{s}}_1^\top \right) \mathbf{v} = \kappa \left\| \mathbf{v} \right\|_2 \hat{\mathbf{s}}_1; \\
\left( \hat{\mathbf{s}}_1 \hat{\mathbf{s}}_3^\top + \hat{\mathbf{s}}_3 \hat{\mathbf{s}}_1^\top \right) \mathbf{v} &= \boldsymbol{\mu}^\top \mathbf{v} \cdot \hat{\mathbf{s}}_1; \quad \left( \hat{\mathbf{s}}_2 \hat{\mathbf{s}}_3^\top + \hat{\mathbf{s}}_3 \hat{\mathbf{s}}_2^\top \right) \mathbf{v} = \boldsymbol{\mu}^\top \mathbf{v} \cdot \hat{\mathbf{s}}_2 + \kappa \left\| \mathbf{v} \right\|_2 \boldsymbol{\mu}
\end{aligned}
$$

Lastly, we have

$$
\begin{aligned}
\left(\hat{\mathbf{s}}_3\hat{\mathbf{s}}_3^\top + \mathbf{M}\right)\mathbf{v} &= \boldsymbol{\mu}^\top\mathbf{v}\cdot\boldsymbol{\mu} + \kappa^2\mathbf{v} - \frac{\kappa^2}{\|\mathbf{v}\|_2^2\|\mathbf{u}\|_2^2 - \left(\mathbf{v}^\top\mathbf{u}\right)^2}\left(\mathbf{u}\mathbf{v}^\top - \mathbf{v}\mathbf{u}^\top\right)^2\mathbf{v} \\
&= \boldsymbol{\mu}^\top\mathbf{v}\cdot\boldsymbol{\mu} + \kappa^2\mathbf{v} - \frac{\kappa^2}{\|\mathbf{v}\|_2^2\|\mathbf{u}\|_2^2 - \left(\mathbf{v}^\top\mathbf{u}\right)^2}\left(\mathbf{u}\mathbf{v}^\top - \mathbf{v}\mathbf{u}^\top\right)\left(\mathbf{u}^\top\mathbf{v}\cdot\mathbf{v} - \|\mathbf{v}\|_2^2\mathbf{u}\right) \\
&= \boldsymbol{\mu}^\top\mathbf{v}\cdot\boldsymbol{\mu} + \kappa^2\mathbf{v} \\
&\quad - \frac{\kappa^2}{\|\mathbf{v}\|_2^2\|\mathbf{u}\|_2^2 - \left(\mathbf{v}^\top\mathbf{u}\right)^2}\left(\mathbf{u}^\top\mathbf{v}\cdot\|\mathbf{v}\|_2^2\mathbf{u} - \left(\mathbf{u}^\top\mathbf{v}\right)^2\mathbf{v} - \mathbf{u}^\top\mathbf{v}\cdot\|\mathbf{v}\|_2^2\mathbf{u} + \|\mathbf{v}\|_2^2\|\mathbf{u}\|_2^2\mathbf{v}\right) \\
&= \boldsymbol{\mu}^\top\mathbf{v}\cdot\boldsymbol{\mu} + \kappa^2\mathbf{v} + \kappa^2\mathbf{v} \\
&= \boldsymbol{\mu}^\top\mathbf{v}\cdot\boldsymbol{\mu} + 2\kappa^2\mathbf{v}
\end{aligned}
$$

Therefore, we can write $\mathbb{E}_\mathbf{c}\left[\mathbf{c}\mathbf{c}^\top\mathbb{I}\left\{z_1\geq 0; z_2\geq 0\right\}\right]\mathbf{v}$ as

$$
\begin{aligned}
\mathbb{E}_\mathbf{c}\left[\mathbf{c}\mathbf{c}^\top\mathbb{I}\left\{z_1\geq 0; z_2\geq 0\right\}\right]\mathbf{v} &= \kappa\|\mathbf{v}\|_2\,\hat{\mathbf{s}}_2\cdot\mathcal{I}_2 + \kappa\|\mathbf{v}\|_2\,\hat{\mathbf{s}}_1\cdot\mathcal{I}_3 + \boldsymbol{\mu}^\top\mathbf{v}\cdot\hat{\mathbf{s}}_1\cdot\mathcal{I}_4 \\
&\quad + \left(\boldsymbol{\mu}^\top\mathbf{v}\cdot\hat{\mathbf{s}}_2 + \kappa\|\mathbf{v}\|_2\,\boldsymbol{\mu}\right)\mathcal{I}_5 + \left(\boldsymbol{\mu}^\top\mathbf{v}\cdot\boldsymbol{\mu} + 2\kappa^2\mathbf{v}\right)\mathcal{I}_6 \\
&= \kappa\|\mathbf{v}\|_2\left(\mathcal{I}_2\cdot\hat{\mathbf{s}}_2 + \mathcal{I}_3\cdot\hat{\mathbf{s}}_1\right) + \boldsymbol{\mu}^\top\mathbf{v}\left(\mathcal{I}_4\cdot\hat{\mathbf{s}}_1 + \mathcal{I}_5\cdot\hat{\mathbf{s}}_2\right) \\
&\quad + \left(\kappa\|\mathbf{v}\|_2\cdot\mathcal{I}_5 + \boldsymbol{\mu}^\top\mathbf{v}\cdot\mathcal{I}_6\right)\boldsymbol{\mu} + 2\kappa^2\mathcal{I}_6\mathbf{v}
\end{aligned}
\tag{72}
$$

By the definition of $\mathcal{I}_2$ to $\mathcal{I}_6$, we first notice that

$$
\begin{aligned}
\mathcal{I}_6 &= \int_{-\frac{\boldsymbol{\mu}^\top\mathbf{u}}{\kappa\|\mathbf{u}\|_2}}^\infty\int_{-\frac{\boldsymbol{\mu}^\top\mathbf{u}}{\kappa\|\mathbf{u}\|_2}}^\infty f(\hat{z}_1,\hat{z}_2)d\hat{z}_1 d\hat{z}_2 \\
&= \mathbb{P}\left(\hat{z}_1\geq -\frac{\boldsymbol{\mu}^\top\mathbf{u}}{\kappa\|\mathbf{u}\|_2}; \hat{z}_2\geq -\frac{\boldsymbol{\mu}^\top\mathbf{v}}{\kappa\|\mathbf{v}\|_2}\right) \\
&= \mathbb{P}\left(\hat{z}_1\leq \frac{\boldsymbol{\mu}^\top\mathbf{u}}{\kappa\|\mathbf{u}\|_2}; \hat{z}_2\leq \frac{\boldsymbol{\mu}^\top\mathbf{v}}{\kappa\|\mathbf{v}\|_2}\right) \\
&= \Phi_2\left(\frac{\boldsymbol{\mu}^\top\mathbf{u}}{\kappa\|\mathbf{u}\|_2}, \frac{\boldsymbol{\mu}^\top\mathbf{v}}{\kappa\|\mathbf{v}\|_2}, \rho\right)
\end{aligned}
$$

Moreover, we can invoke Lemma D.6, Lemma D.7, and Lemma D.8 to get that

$$
\mathcal{I}_4 = \frac{1}{\sqrt{2\pi}}\left(T_1 + \rho T_2\right); \quad \mathcal{I}_5 = \frac{1}{\sqrt{2\pi}}\left(T_2 + \rho T_1\right)
$$

$$
\mathcal{I}_2 = \frac{\rho\sqrt{1-\rho^2}}{2\pi}\exp\left(-\frac{a^2 - 2\rho ab + b^2}{2\left(1-\rho^2\right)}\right) + \Phi_2\left(-a, -b, \rho\right) + \frac{1}{\sqrt{2\pi}}\left(bT_2 + \rho^2 aT_1\right)
$$

$$
\mathcal{I}_3 = \frac{\sqrt{1-\rho^2}}{2\pi}\exp\left(-\frac{a^2 - 2\rho ab + b^2}{2\left(1-\rho^2\right)}\right) + \rho\Phi_2\left(-a, -b, \rho\right) + \frac{\rho}{\sqrt{2\pi}}\left(aT_1 + bT_2\right)
$$

where $T_1, T_2$ and $a, b$ are defined as

$$
T_1 = \exp\left(-\frac{a^2}{2}\right)\Phi_1\left(\frac{\rho a - b}{\sqrt{1-\rho^2}}\right); \quad T_2 = \exp\left(-\frac{b^2}{2}\right)\Phi_1\left(\frac{\rho b - a}{\sqrt{1-\rho^2}}\right)
$$

$$
a = -\frac{\boldsymbol{\mu}^\top\mathbf{u}}{\kappa\|\mathbf{u}\|_2}; \quad b = -\frac{\boldsymbol{\mu}^\top\mathbf{v}}{\kappa\|\mathbf{v}\|_2}; \quad \rho = \frac{\mathbf{u}^\top\mathbf{v}}{\|\mathbf{u}\|_2\|\mathbf{v}\|_2}
$$

To ease our computation, we define

$$
E = \frac{1-\rho^2}{2\pi}\exp\left(-\frac{a^2 - 2\rho ab + b^2}{2\left(1-\rho^2\right)}\right); F = \Phi_2\left(-a, -b, \rho\right)
$$

Then the terms $\mathcal{I}_2$ to $\mathcal{I}_6$ can be written as

$$
\begin{aligned}
\mathcal{I}_2 = \rho E + F + \frac{1}{\sqrt{2\pi}}\left(bT_2 + \rho^2 aT_1\right); \quad \mathcal{I}_3 = E + \rho F + \frac{1}{\sqrt{2\pi}}\left(bT_2 + aT_1\right) \\
\mathcal{I}_4 = \frac{1}{\sqrt{2\pi}}\left(T_1 + \rho T_2\right); \quad \mathcal{I}_5 = \frac{1}{\sqrt{2\pi}}\left(T_2 + \rho T_1\right); \quad \mathcal{I}_6 = F
\end{aligned}
\tag{73}
$$

Now, the trick of evaluating (72) is to re-write $\hat{\mathbf{s}}_1$ and $\hat{\mathbf{s}}_2$ as below

$$
\begin{aligned}
\hat{\mathbf{s}}_1 &= \frac{\kappa \|\mathbf{u}\|_2}{\|\mathbf{u}\|_2^2 \|\mathbf{v}\|_2^2 - (\mathbf{u}^\top \mathbf{v})^2} \cdot \left(\|\mathbf{v}\|_2^2 \mathbf{u} - \mathbf{u}^\top \mathbf{v} \cdot \mathbf{v}\right) \\
&= \frac{\kappa \left(\|\mathbf{v}\|_2^2 \mathbf{u} - \mathbf{u}^\top \mathbf{v} \cdot \mathbf{v}\right)}{\|\mathbf{u}\|_2 \|\mathbf{v}\|_2^2 (1 - \rho^2)} \\
&= \frac{\kappa}{1 - \rho^2} \cdot \frac{\mathbf{u}}{\|\mathbf{u}\|_2} - \frac{\kappa \rho}{1 - \rho^2} \cdot \frac{\mathbf{v}}{\|\mathbf{v}\|_2} \\
&= \frac{\kappa}{1 - \rho^2} \left(\frac{\mathbf{u}}{\|\mathbf{u}\|_2} - \rho \cdot \frac{\mathbf{v}}{\|\mathbf{v}\|_2}\right) \\
\hat{\mathbf{s}}_2 &= \frac{\kappa \|\mathbf{v}\|_2}{\|\mathbf{u}\|_2^2 \|\mathbf{v}\|_2^2 - (\mathbf{u}^\top \mathbf{v})^2} \cdot \left(\|\mathbf{u}\|_2^2 \mathbf{v} - \mathbf{u}^\top \mathbf{v} \cdot \mathbf{u}\right) \\
&= \frac{\kappa \left(\|\mathbf{u}\|_2^2 \mathbf{v} - \mathbf{u}^\top \mathbf{v} \cdot \mathbf{u}\right)}{\|\mathbf{u}\|_2 \|\mathbf{v}\|_2^2 (1 - \rho^2)} \\
&= \frac{\kappa}{1 - \rho^2} \cdot \frac{\mathbf{v}}{\|\mathbf{v}\|_2} - \frac{\kappa \rho}{1 - \rho^2} \cdot \frac{\mathbf{u}}{\|\mathbf{u}\|_2} \\
&= \frac{\kappa}{1 - \rho^2} \left(\frac{\mathbf{v}}{\|\mathbf{v}\|_2} - \rho \cdot \frac{\mathbf{u}}{\|\mathbf{u}\|_2}\right)
\end{aligned}
\tag{74}
$$

Now, we can simplify (72) with (73) and (74). To start, for the terms $\mathcal{I}_2 \cdot \hat{\mathbf{s}}_2 + \mathcal{I}_3 \cdot \hat{\mathbf{s}}_1$ we have

$$
\begin{aligned}
\mathcal{I}_2 \cdot \hat{\mathbf{s}}_2 + \mathcal{I}_3 \cdot \hat{\mathbf{s}}_1 &= \frac{\kappa}{1 - \rho^2}\left(\rho E + F + \frac{1}{\sqrt{2\pi}}\left(bT_2 + \rho^2 aT_1\right)\right)\left(\frac{\mathbf{v}}{\|\mathbf{v}\|_2} - \rho \cdot \frac{\mathbf{u}}{\|\mathbf{u}\|_2}\right) \\
&\quad + \frac{\kappa}{1 - \rho^2}\left(E + \rho F + \frac{\rho}{\sqrt{2\pi}}\left(bT_2 + aT_1\right)\right)\left(\frac{\mathbf{u}}{\|\mathbf{u}\|_2} - \rho \cdot \frac{\mathbf{v}}{\|\mathbf{v}\|_2}\right) \\
&= \frac{\kappa E}{1 - \rho^2}\left(\rho\left(\frac{\mathbf{v}}{\|\mathbf{v}\|_2} - \rho \cdot \frac{\mathbf{u}}{\|\mathbf{u}\|_2}\right) + \left(\frac{\mathbf{u}}{\|\mathbf{u}\|_2} - \rho \cdot \frac{\mathbf{v}}{\|\mathbf{v}\|_2}\right)\right) \\
&\quad + \frac{\kappa F}{1 - \rho^2}\left(\left(\frac{\mathbf{v}}{\|\mathbf{v}\|_2} - \rho \cdot \frac{\mathbf{u}}{\|\mathbf{u}\|_2}\right) + \rho\left(\frac{\mathbf{u}}{\|\mathbf{u}\|_2} - \rho \cdot \frac{\mathbf{v}}{\|\mathbf{v}\|_2}\right)\right) \\
&\quad + \frac{\kappa \rho aT_1}{\sqrt{2\pi}(1 - \rho^2)}\left(\rho\left(\frac{\mathbf{v}}{\|\mathbf{v}\|_2} - \rho \cdot \frac{\mathbf{u}}{\|\mathbf{u}\|_2}\right) + \left(\frac{\mathbf{u}}{\|\mathbf{u}\|_2} - \rho \cdot \frac{\mathbf{v}}{\|\mathbf{v}\|_2}\right)\right) \\
&\quad + \frac{\kappa bT_2}{\sqrt{2\pi}(1 - \rho^2)}\left(\left(\frac{\mathbf{v}}{\|\mathbf{v}\|_2} - \rho \cdot \frac{\mathbf{u}}{\|\mathbf{u}\|_2}\right) + \rho\left(\frac{\mathbf{u}}{\|\mathbf{u}\|_2} - \rho \cdot \frac{\mathbf{v}}{\|\mathbf{v}\|_2}\right)\right) \\
&= \kappa E \cdot \frac{\mathbf{u}}{\|\mathbf{u}\|_2} + \kappa F \cdot \frac{\mathbf{v}}{\|\mathbf{v}\|_2} + \frac{\kappa \rho aT_1}{\sqrt{2\pi}} \cdot \frac{\mathbf{u}}{\|\mathbf{u}\|_2} + \frac{\kappa bT_2}{\sqrt{2\pi}} \cdot \frac{\mathbf{v}}{\|\mathbf{v}\|_2} \\
&= \kappa \left(\left(E + \frac{\rho aT_1}{\sqrt{2\pi}}\right)\frac{\mathbf{u}}{\|\mathbf{u}\|_2} + \left(F + \frac{bT_2}{\sqrt{2\pi}}\right)\frac{\mathbf{v}}{\|\mathbf{v}\|_2}\right)
\end{aligned}
$$

Similarly, for the term $\mathcal{I}_4 \hat{\mathbf{s}}_1 + \mathcal{I}_4 \hat{\mathbf{s}}_2$, we have

$$
\begin{aligned}
\mathcal{I}_4 \hat{\mathbf{s}}_1 + \mathcal{I}_4 \hat{\mathbf{s}}_2 &= \frac{\kappa}{\sqrt{2\pi}\,(1-\rho^2)}\left((T_1 + \rho T_2)\left(\frac{\mathbf{v}}{\|\mathbf{v}\|_2} - \rho \cdot \frac{\mathbf{u}}{\|\mathbf{u}\|_2}\right) + (T_2 + \rho T_1)\left(\frac{\mathbf{u}}{\|\mathbf{u}\|_2} - \rho \cdot \frac{\mathbf{v}}{\|\mathbf{v}\|_2}\right)\right) \\
&= \frac{\kappa T_1}{\sqrt{2\pi}\,(1-\rho^2)}\left(\left(\frac{\mathbf{v}}{\|\mathbf{v}\|_2} - \rho \cdot \frac{\mathbf{u}}{\|\mathbf{u}\|_2}\right) + \rho\left(\frac{\mathbf{u}}{\|\mathbf{u}\|_2} - \rho \cdot \frac{\mathbf{v}}{\|\mathbf{v}\|_2}\right)\right) \\
&\quad + \frac{\kappa T_2}{\sqrt{2\pi}\,(1-\rho^2)}\left(\rho\left(\frac{\mathbf{v}}{\|\mathbf{v}\|_2} - \rho \cdot \frac{\mathbf{u}}{\|\mathbf{u}\|_2}\right) + \left(\frac{\mathbf{u}}{\|\mathbf{u}\|_2} - \rho \cdot \frac{\mathbf{v}}{\|\mathbf{v}\|_2}\right)\right) \\
&= \frac{\kappa T_1}{\sqrt{2\pi}} \cdot \frac{\mathbf{v}}{\|\mathbf{v}\|_2} + \frac{\kappa T_2}{\sqrt{2\pi}} \cdot \frac{\mathbf{u}}{\|\mathbf{u}\|_2} \\
&= \frac{\kappa}{\sqrt{2\pi}}\left(T_1 \cdot \frac{\mathbf{v}}{\|\mathbf{v}\|_2} + T_2 \cdot \frac{\mathbf{u}}{\|\mathbf{u}\|_2}\right)
\end{aligned}
$$

Applying these evaluations, (72) becomes

$$
\begin{aligned}
\mathbb{E}_{\mathbf{c}}\left[\mathbf{c}\mathbf{c}^\top \mathbb{I}\{z_1 \geq 0; z_2 \geq 0\}\right]\mathbf{v} &= \kappa^2 \|\mathbf{v}\|_2 \left(\left(E + \frac{\rho a T_1}{\sqrt{2\pi}}\right)\frac{\mathbf{u}}{\|\mathbf{u}\|_2} + \left(F + \frac{b T_2}{\sqrt{2\pi}}\right)\frac{\mathbf{v}}{\|\mathbf{v}\|_2}\right) \\
&\quad + \frac{\kappa \boldsymbol{\mu}^\top \mathbf{v}}{\sqrt{2\pi}}\left(T_1 \cdot \frac{\mathbf{v}}{\|\mathbf{v}\|_2} + T_2 \cdot \frac{\mathbf{u}}{\|\mathbf{u}\|_2}\right) \\
&\quad + \frac{\kappa \|\mathbf{v}\|_2}{\sqrt{2\pi}}(T_2 + \rho T_1)\boldsymbol{\mu} + \boldsymbol{\mu}^\top \mathbf{v} \cdot F \cdot \boldsymbol{\mu} + 2\kappa^2 F \mathbf{v} \\
&= \underbrace{\kappa^2 \|\mathbf{v}\|_2 \left(E \cdot \frac{\mathbf{u}}{\|\mathbf{u}\|_2} + \frac{\rho a T_1}{\sqrt{2\pi}} \cdot \frac{\mathbf{u}}{\|\mathbf{u}\|_2} + \frac{b T_2}{\sqrt{2\pi}} \cdot \frac{\mathbf{v}}{\|\mathbf{v}\|_2}\right)}_{\mathbf{g}_1} \\
&\quad + \underbrace{\frac{\kappa}{\sqrt{2\pi}}\left(\boldsymbol{\mu}^\top \mathbf{v}\left(T_1 \cdot \frac{\mathbf{v}}{\|\mathbf{v}\|_2} + T_2 \cdot \frac{\mathbf{u}}{\|\mathbf{u}\|_2}\right) + \|\mathbf{v}\|_2 (T_2 + \rho T_1)\boldsymbol{\mu}\right)}_{\mathbf{g}_2} \\
&\quad + F\left(\boldsymbol{\mu}\boldsymbol{\mu}^\top \mathbf{v} + 3\kappa^2 \mathbf{v}\right)
\end{aligned}
\tag{75}
$$

Then we have that

$$
\begin{aligned}
&\mathbb{E}_{\mathbf{c}}\left[\mathbf{c}\mathbf{c}^\top \mathbb{I}\{z_1 \geq 0; z_2 \geq 0\}\right]\mathbf{v} - \left(\boldsymbol{\mu}\boldsymbol{\mu}^\top \mathbf{v} + 3\kappa^2 \mathbf{v}\right)\boldsymbol{\Phi}_1\left(\frac{\boldsymbol{\mu}^\top \mathbf{u}}{\kappa\|\mathbf{u}\|_2}\right)\boldsymbol{\Phi}_1\left(\frac{\boldsymbol{\mu}^\top \mathbf{v}}{\kappa\|\mathbf{v}\|_2}\right) \\
&= \mathbf{g}_1 + \mathbf{g}_2 + \mathcal{C}\left(\frac{\boldsymbol{\mu}^\top \mathbf{u}}{\kappa\|\mathbf{u}\|_2}, \frac{\boldsymbol{\mu}^\top \mathbf{v}}{\kappa\|\mathbf{v}\|_2}, \frac{\mathbf{u}^\top \mathbf{v}}{\|\mathbf{u}\|_2\|\mathbf{v}\|_2}\right)
\end{aligned}
\tag{76}
$$

The proof then proceed by estimating the magnitude of the three terms. To start, we need to bound $T_1$ and $T_2$. In particular, since $\boldsymbol{\Phi}_1$ is the CDF, its magnitude must be bounded by 1. Therefore

$$
0 \leq T_1 \leq \exp\left(-\frac{a^2}{2}\right); \quad 0 \leq T_2 \leq \exp\left(-\frac{b^2}{2}\right)
$$

Therefore, the $\ell_\infty$ norm of $\mathbf{g}_2$ is bounded by

$$
\begin{aligned}
\|\mathbf{g}_2\|_\infty &\leq \frac{\kappa}{\sqrt{2\pi}}\left(\boldsymbol{\mu}^\top \mathbf{v}\left(\exp\left(-\frac{a^2}{2}\right) + \exp\left(-\frac{b^2}{2}\right)\right)\right) + \|\mathbf{v}\|_2 \|\boldsymbol{\mu}\|_2\left(\exp\left(-\frac{a^2}{2}\right) + \rho\exp\left(-\frac{b^2}{2}\right)\right) \\
&\leq \frac{2\kappa}{\sqrt{2\pi}}\|\mathbf{v}\|_2 \|\boldsymbol{\mu}\|_2\left(\exp\left(-\frac{a^2}{2}\right) + \exp\left(-\frac{b^2}{2}\right)\right) \\
&\leq \kappa\|\mathbf{v}\|_2 \|\boldsymbol{\mu}\|_2\left(\phi\left(\frac{a}{2}\right) + \phi\left(\frac{b}{2}\right)\right)
\end{aligned}
\tag{77}
$$

Next, for $E$, we have

$$
\begin{aligned}
E &= \frac{1-\rho^2}{2\pi} \exp\left(-\frac{a^2 - 2\rho ab + b^2}{2\left(1-\rho^2\right)}\right) \\
&= \frac{1-\rho^2}{4\pi} \left(\exp\left(-\frac{a^2 - 2\rho ab + \rho^2 b^2}{2\left(1-\rho^2\right)} - \frac{b^2}{2}\right) + \exp\left(-\frac{\rho^2 a^2 - 2\rho ab + b^2}{2\left(1-\rho^2\right)} - \frac{a^2}{2}\right)\right) \\
&\leq \frac{1}{4\pi} \left(\exp\left(-\frac{a^2}{2}\right) + \exp\left(-\frac{b^2}{2}\right)\right) \\
&\leq \frac{1}{4\pi} \left(\phi\left(\frac{a}{2}\right) + \phi\left(\frac{b}{2}\right)\right)
\end{aligned}
$$

Therefore, the magnitude of $\mathbf{g}_1$ can be bounded by

$$
\begin{aligned}
\|\mathbf{g}_1\|_2 &\leq \kappa^2 \|\mathbf{v}\|_2 \left(|E| + \frac{|a|\,T_1}{\sqrt{2\pi}} + \frac{|b|\,T_2}{\sqrt{2\pi}}\right) \\
&\leq \kappa^2 \|\mathbf{v}\|_2 \left(\frac{1}{4\pi}\left(\phi\left(\frac{a}{2}\right) + \phi\left(\frac{b}{2}\right)\right) + \frac{|a|}{\sqrt{2\pi}}\exp\left(-\frac{a^2}{2}\right) + \frac{|b|}{\sqrt{2\pi}}\exp\left(-\frac{b^2}{2}\right)\right) \qquad (78) \\
&\leq \kappa^2 \|\mathbf{v}\|_2 \left(\frac{1}{4\pi}\left(\phi\left(\frac{a}{2}\right) + \phi\left(\frac{b}{2}\right)\right) + \psi\left(\frac{a}{2}\right) + \psi\left(\frac{b}{2}\right)\right)
\end{aligned}
$$

Moreover, by the bound of the Gaussian Copula function, we have that

$$
|\mathcal{C}\left(a, b, \rho\right)| \leq \frac{1}{4}\exp\left(-\frac{a^2 + b^2}{4}\right)
$$

Therefore, we have that

$$
\mathcal{C}\left(\frac{\boldsymbol{\mu}^\top \mathbf{u}}{\kappa\|\mathbf{u}\|_2}, \frac{\boldsymbol{\mu}^\top \mathbf{v}}{\kappa\|\mathbf{v}\|_2}, \frac{\mathbf{u}^\top \mathbf{v}}{\|\mathbf{u}\|_2\|\mathbf{v}\|_2}\right) \leq \frac{1}{4}\exp\left(-\frac{a^2}{4}\right)\exp\left(-\frac{b^2}{4}\right) = \frac{1}{4}\phi\left(\frac{a}{2}\right)\phi\left(\frac{b}{2}\right)
$$

Combining the results gives

$$
\begin{aligned}
&\left\|\mathbb{E}_{\mathbf{c}}\left[\mathbf{c}\mathbf{c}^\top \mathbb{I}\{z_1 \geq 0; z_2 \geq 0\}\right]\mathbf{v} - \left(\boldsymbol{\mu}\boldsymbol{\mu}^\top \mathbf{v} + 3\kappa^2 \mathbf{v}\right)\boldsymbol{\Phi}_1\left(\frac{\boldsymbol{\mu}^\top \mathbf{u}}{\kappa\|\mathbf{u}\|_2}\right)\boldsymbol{\Phi}_1\left(\frac{\boldsymbol{\mu}^\top \mathbf{v}}{\kappa\|\mathbf{v}\|_2},\right)\right\|_2 \\
&\leq \kappa^2 \|\mathbf{v}\|_2 \left(\frac{1}{4\pi}\left(\phi\left(\frac{a}{2}\right) + \phi\left(\frac{b}{2}\right)\right) + \psi\left(\frac{a}{2}\right) + \psi\left(\frac{b}{2}\right)\right) + \kappa\|\mathbf{v}\|_2\|\boldsymbol{\mu}\|_2\left(\phi\left(\frac{a}{2}\right) + \phi\left(\frac{b}{2}\right)\right) \\
&\quad + \frac{1}{4}\left(|\boldsymbol{\mu}^\top \mathbf{v}|\,\|\boldsymbol{\mu}\|_\infty + 3\kappa^2\|\mathbf{v}\|_2\right)\phi\left(\frac{a}{2}\right)\phi\left(\frac{b}{2}\right) \\
&= \|\mathbf{v}\|_2 \left(\left(\kappa^2 + \kappa\|\boldsymbol{\mu}\|_2\right)\left(\phi\left(\frac{a}{2}\right) + \phi\left(\frac{b}{2}\right)\right) + \left(\kappa^2 + \kappa\|\boldsymbol{\mu}\|_\infty\right)\left(\psi\left(\frac{a}{2}\right) + \psi\left(\frac{b}{2}\right)\right)\right) \\
&\leq 2\kappa\|\mathbf{v}\|_2 \left(\|\boldsymbol{\mu}\|_2\left(\phi\left(\frac{a}{2}\right) + \phi\left(\frac{b}{2}\right)\right) + \|\boldsymbol{\mu}\|_\infty\left(\psi\left(\frac{a}{2}\right) + \psi\left(\frac{b}{2}\right)\right)\right)
\end{aligned}
$$

$\square$

### D.1.5  Other Results

**Lemma D.13.** *Let* $\mathbf{c} \sim \mathcal{N}\left(\boldsymbol{\mu}, \kappa^2\mathbf{I}\right)$*, and let* $\mathbf{u} \in \mathbb{R}^d$ *be a vector. Define* $z = \mathbf{c}^\top \mathbf{u}$*. Then we have that* $z \sim \mathcal{N}\left(\boldsymbol{\mu}^\top \mathbf{u}, \kappa^2\|\mathbf{u}\|_2^2\right)$*.*

*Proof.* Since $z = \mathbf{c}^\top \mathbf{u}$ where $\mathbf{c} \sim \mathcal{N}\left(\boldsymbol{\mu}, \kappa^2 \mathbf{I}\right)$. Then the moment generating function of $z$ is given by

$$M_z(t) = \mathbb{E}\left[\exp\left(zt\right)\right]$$

$$= \mathbb{E}\left[\prod_{j=1}^{d} \exp\left(c_j u_j t\right)\right]$$

$$= \prod_{j=1}^{d} \mathbb{E}\left[\exp\left(c_j u_j t\right)\right]$$

$$= \prod_{j=1}^{d} \exp\left(u_j \mu_j t + \frac{1}{2} u_j^2 \kappa^2 t^2\right)$$

$$= \exp\left(\left(\sum_{j=1}^{d} u_j \mu_j\right) t + \frac{1}{2}\left(\sum_{j=1}^{d} u_j^2\right)\kappa^2 t^2\right)$$

$$= \exp\left(\boldsymbol{\mu}^\top \mathbf{u} \cdot t + \frac{1}{2}\|\mathbf{u}\|_2^2 \kappa^2 t^2\right)$$

Therefore, $z \sim \mathcal{N}\left(\boldsymbol{\mu}^\top \mathbf{u}, \kappa^2 \|\mathbf{u}\|_2^2\right)$. $\qquad\square$

**Lemma D.14.** *Let $\kappa, \mu, a \in \mathbb{R}$ be given such that $\kappa > 0$. Then we have that*

$$\int_a^\infty z \exp\left(-\frac{(z-\mu)^2}{2\kappa^2}\right) dz = \kappa^2 \exp\left(-\frac{(\mu-a)^2}{2\kappa^2}\right) + \kappa\mu\sqrt{2\pi}\,\boldsymbol{\Phi}_1\left(\frac{\mu-a}{\kappa}\right)$$

*Proof.* We use a change of variable $z' = \frac{z-\mu}{\kappa}$. Then we have that $z = \kappa z' + \mu$, and $dz = \kappa dz'$. Therefore

$$\int_a^\infty z \exp\left(-\frac{(z-\mu)^2}{2\kappa^2}\right) dz = \int_{\frac{a-\mu}{\kappa}}^\infty \left(\kappa z' + \mu\right)\exp\left(-\frac{z'^2}{2}\right)\kappa dz'$$

$$= \kappa^2 \int_{\frac{a-\mu}{\kappa}}^\infty z' \exp\left(-\frac{z'^2}{2}\right) dz' + \kappa\mu \int_{\frac{a-\mu}{\kappa}}^\infty \exp\left(-\frac{z'^2}{2}\right) dz'$$

$$= \kappa^2 \exp\left(-\frac{z'^2}{2}\right)\Big|_\infty^{\frac{a-\mu}{\kappa}} + \kappa\mu\sqrt{2\pi}\left(1 - \boldsymbol{\Phi}_1\left(\frac{a-\mu}{\kappa}\right)\right)$$

$$= \kappa^2 \exp\left(-\frac{(\mu-a)^2}{2\kappa^2}\right) + \kappa\mu\sqrt{2\pi}\,\boldsymbol{\Phi}_1\left(\frac{\mu-a}{\kappa}\right)$$

$\qquad\square$

**Lemma D.15.** *Let $\kappa, \mu, a \in \mathbb{R}$ be given such that $\kappa > 0$. Then we have that*

$$\int_a^\infty z^2 \exp\left(-\frac{(z-\mu)^2}{2\kappa^2}\right) dz = \kappa^2 (a+\mu)\exp\left(-\frac{(\mu-a)^2}{2\kappa^2}\right) + \sqrt{2\pi}\kappa\left(\kappa^2 + \mu^2\right)\boldsymbol{\Phi}_1\left(\frac{\mu-a}{\kappa}\right)$$

*Proof.* To start, let $z' = \frac{z-\mu}{\kappa}$. Then we have that $z = \kappa z' + \mu$, and $dz = \kappa dz'$. Therefore

$$\int_a^\infty z^2 \exp\left(-\frac{(z-\mu)^2}{2\kappa^2}\right) dz = \kappa \int_{\frac{a-\mu}{\kappa}}^\infty \left(\kappa z' + \mu\right)^2 \exp\left(-\frac{z'^2}{2}\right) dz'$$

$$= \kappa^3 \int_{\frac{a-\mu}{\kappa}}^\infty z'^2 \exp\left(-\frac{z'^2}{2}\right) dz' + 2\kappa^2 \mu \int_{\frac{a-\mu}{\kappa}}^\infty z' \exp\left(-\frac{z'^2}{2}\right) dz'$$

$$+ \kappa\mu^2 \int_{\frac{a-\mu}{\kappa}}^\infty \exp\left(-\frac{z'^2}{2}\right) dz'$$

Notice that for the third term, we have that

$$\int_{\frac{a-\mu}{\kappa}}^{\infty} \exp\left(-\frac{z'^2}{2}\right) dz' = \sqrt{2\pi}\left(1 - \mathbf{\Phi}_1\left(\frac{a-\mu}{\kappa}\right)\right) = \sqrt{2\pi}\mathbf{\Phi}_1\left(\frac{\mu-a}{\kappa}\right)$$

For the second term, we can directly apply Lemma D.14 with $\kappa = 1, \mu = 0$ to get that

$$\int_{\frac{a-\mu}{\kappa}}^{\infty} z' \exp\left(-\frac{z'^2}{2}\right) dz' = \exp\left(-\frac{(a-\mu)^2}{2\kappa^2}\right)$$

For the first term, we apply integration by parts with $u(z') = -z'$ and $v(z') = \exp\left(-\frac{z'^2}{2}\right)$. In particular, notice that $v'(z') = -z'\exp\left(-\frac{z'^2}{2}\right)$ and $u'(z') = -1$. Therefore

$$
\begin{aligned}
\int_{\frac{a-\mu}{\kappa}}^{\infty} z'^2 \exp\left(-\frac{z'^2}{2}\right) dz' &= \int_{\frac{a-\mu}{\kappa}}^{\infty} u(z') dv(z') \\
&= u(z')v(z')\,|_{\frac{a-\mu}{\kappa}}^{\infty} - \int_{\frac{a-\mu}{\kappa}}^{\infty} v(z') du(z') \\
&= -z'\exp\left(-\frac{z'^2}{2}\right)\,|_{\frac{a-\mu}{\kappa}}^{\infty} + \int_{\frac{a-\mu}{\kappa}}^{\infty} \exp\left(-\frac{z'^2}{2}\right) dz' \\
&= \frac{a-\mu}{\kappa}\exp\left(-\frac{(\mu-a)^2}{2\kappa^2}\right) + \sqrt{2\pi}\left(1 - \mathbf{\Phi}_1\left(\frac{a-\mu}{\kappa}\right)\right) \\
&= \frac{a-\mu}{\kappa}\exp\left(-\frac{(\mu-a)^2}{2\kappa^2}\right) + \sqrt{2\pi}\mathbf{\Phi}_1\left(\frac{\mu-a}{\kappa}\right)
\end{aligned}
$$

Putting things together, we have that

$$
\begin{aligned}
\int_{a}^{\infty} z^2 \exp\left(-\frac{(z-\mu)^2}{2\kappa^2}\right) dz &= \kappa^3\left(\frac{a-\mu}{\kappa}\exp\left(-\frac{(\mu-a)^2}{2\kappa^2}\right) + \sqrt{2\pi}\mathbf{\Phi}_1\left(\frac{\mu-a}{\kappa}\right)\right) \\
&\quad + 2\kappa^2\mu\exp\left(-\frac{(a-\mu)^2}{2\kappa^2}\right) + \kappa\mu^2\sqrt{2\pi}\mathbf{\Phi}_1\left(\frac{\mu-a}{\kappa}\right) \\
&= \left(\kappa^2(a-\mu) + 2\kappa^2\mu\right)\exp\left(-\frac{(\mu-a)^2}{2\kappa^2}\right) + \sqrt{2\pi}\left(\kappa^3 + \kappa\mu^2\right)\mathbf{\Phi}_1\left(\frac{\mu-a}{\kappa}\right) \\
&= \kappa^2(a+\mu)\exp\left(-\frac{(\mu-a)^2}{2\kappa^2}\right) + \sqrt{2\pi}\kappa\left(\kappa^2 + \mu^2\right)\mathbf{\Phi}_1\left(\frac{\mu-a}{\kappa}\right)
\end{aligned}
$$

$\square$

**Lemma D.16.** *Let $z_1, z_2 \sim \mathcal{N}(0,1)$ with $\text{Cov}(z_1, z_2) = \rho$. Then we have that*

$$\int_{a}^{\infty} f(z_1 \mid z_2) dz_1 = \mathbf{\Phi}_1\left(\frac{\rho z_2 - a}{\sqrt{1-\rho^2}}\right)$$

*Proof.* Since $z_1, z_2 \sim \mathcal{N}(0, 1)$ with $\mathrm{Cov}(z_1, z_2) = \rho$, we have that $z_1 \mid z_2 \sim \mathcal{N}(\rho z_2, 1 - \rho^2)$. Therefore, using a change of variable $z' = \frac{z_1 - \rho z_2}{\sqrt{1 - \rho^2}}$, we have

$$\int_a^\infty f(z_1 \mid z_2) \, dz_1 = \frac{1}{\sqrt{2\pi(1-\rho^2)}} \int_a^\infty \exp\left(-\frac{(z_1 - \rho z_2)^2}{2(1-\rho^2)}\right) dz_1$$

$$= \frac{1}{\sqrt{2\pi}} \int_{\frac{a - \rho z_2}{\sqrt{1-\rho^2}}}^\infty \exp\left(-\frac{z'^2}{2}\right) dz'$$

$$= 1 - \mathbf{\Phi}_1\left(\frac{a - \rho z_2}{\sqrt{1 - \rho^2}}\right)$$

$$= \mathbf{\Phi}_1\left(\frac{\rho z_2 - a}{\sqrt{1 - \rho^2}}\right)$$

□

**Lemma D.17.** *Let $z_1, z_2 \sim \mathcal{N}(0, 1)$ with $\mathrm{Cov}(z_1, z_2) = \rho$. Then we have that*

$$\mathbf{\Phi}_2(-a, -b, \rho) = \int_a^\infty \int_b^\infty f(z_1, z_2) \, dz_2 dz_1 = \int_b^\infty \mathbf{\Phi}_1\left(\frac{\rho z_2 - a}{\sqrt{1 - \rho^2}}\right) f(z_2) \, dz_2$$

*Proof.* Let $z_1', z_2' \sim \mathcal{N}(0, 1)$ with $\mathrm{Cov}(z_1, z_2) = \rho$, and define $z_1 = -z_1', z_2 = -z_2'$. Then we have that $z_1, z_2 \sim \mathcal{N}(0, 1)$ with $\mathrm{Cov}(z_1, z_2) = \rho$. By symmetry, we have $f(z_1, z_2) = f(-z_1, -z_2) = f(z_1', z_2')$. Moreover, $dz_2' dz_1' = (-dz_2)(-dz_1) = dz_2 dz_1$. Thus

$$\mathbf{\Phi}_2(-a, -b, \rho) = \int_{-\infty}^{-a} \int_{-\infty}^{-b} f(z_1', z_2') \, dz_2' dz_1' = \int_a^\infty \int_b^\infty f(z_1, z_2) \, dz_2 dz_1$$

Recall that $f(z_1, z_2) = f(z_1 \mid z_2) f(z_2)$. Then we can apply Lemma D.16 to get that

$$\int_a^\infty \int_b^\infty f(z_1, z_2) \, dz_2 dz_1 = \int_b^\infty \left(\int_a^\infty f(z_1 \mid z_2) \, dz_1\right) f(z_2) \, dz_2 = \int_b^\infty \mathbf{\Phi}_1\left(\frac{\rho z_2 - a}{\sqrt{1 - \rho^2}}\right) f(z_2) \, dz_2$$

□

**Lemma D.18.** *Let $z \sim \mathcal{N}(\mu, \kappa^2)$, and let $a \in \mathbb{R}$. Then we have*

$$\mathbb{E}[z\mathbb{I}\{z \geq a\}] = \frac{\kappa}{\sqrt{2\pi}} \exp\left(-\frac{(\mu - a)^2}{2\kappa^2}\right) + \mu\mathbf{\Phi}_1\left(\frac{\mu - a}{\kappa}\right)$$

*Proof.* Define $\hat{z} = \frac{z - \mu}{\kappa}$. Then we have that $\hat{z} \sim \mathcal{N}(0, 1)$. Since $z = \kappa\hat{z} + \mu$, we have

$$\mathbb{E}[z\mathbb{I}\{z \geq 0\}] = \mathbb{E}\left[(\kappa\hat{z} + \mu)\mathbb{I}\left\{\hat{z} \geq \frac{a - \mu}{\kappa}\right\}\right]$$

$$= \kappa\mathbb{E}\left[\hat{z}\mathbb{I}\left\{\hat{z} \geq \frac{a - \mu}{\kappa}\right\}\right] + \mu\mathbb{E}\left[\mathbb{I}\left\{\hat{z} \geq \frac{a - \mu}{\kappa}\right\}\right]$$

Notice that $\mathbb{E}\left[\mathbb{I}\left\{\hat{z} \geq \frac{a - \mu}{\kappa}\right\}\right] = \mathrm{Pr}\left(\hat{z} \geq \frac{a - \mu}{\kappa}\right) = \mathbf{\Phi}_1\left(\frac{\mu - a}{\kappa}\right)$. Moreover

$$\mathbb{E}\left[\hat{z}\mathbb{I}\left\{\hat{z} \geq -\frac{\mu}{\kappa}\right\}\right] = \frac{1}{\sqrt{2\pi}} \int_{\frac{a - \mu}{\kappa}}^\infty \hat{z} \exp\left(-\frac{z^2}{2}\right) dz = \frac{1}{\sqrt{2\pi}} \exp\left(-\frac{(a - \mu)^2}{2\kappa^2}\right)$$

Therefore

$$\mathbb{E}[z\mathbb{I}\{z \geq 0\}] = \frac{\kappa}{\sqrt{2\pi}} \exp\left(-\frac{(a - \mu)^2}{2\kappa^2}\right) + \mu\mathbf{\Phi}_1\left(\frac{\mu - a}{\kappa}\right)$$

□

**Lemma D.19.** . *Let $x \in [-1, 1]$. Then we have that $|\arcsin x| \leq \frac{\pi}{2} \cdot |x|$.*

*Proof.* To start, consider the case $x > 0$. Define $f(x) = \frac{\arcsin x}{x}$. Then we have that

$$f'(x) = x^{-2}\left(\frac{x}{\sqrt{1-x^2}} - \arcsin x\right); \qquad f''(x) = x^{-3}\left(\frac{3x^3 - 2x}{(1-x^2)^{\frac{3}{2}}} + 2\arcsin x\right)$$

For all $x \in (0, 1]$, we have that $1 - x^2 \geq 0$. Notice that by the Taylor expansion of $\arcsin x$, we have

$$\arcsin x \leq x + \frac{x^3}{6}$$

when $x \in (0, 1]$. Therefore

$$\frac{3x^3 - 2x}{(1-x^2)^{\frac{3}{2}}} + 2\arcsin x \geq \frac{3x^3 - 2x + 2x\left(1-x^2\right)^{\frac{3}{2}} + \frac{x^3}{3}\left(1-x^2\right)^{\frac{3}{2}}}{(1-x^2)^{\frac{3}{2}}} \geq \frac{3x^3 - 2x\left(1-\left(1-x^2\right)^{\frac{3}{2}}\right)}{(1-x^2)^{\frac{3}{2}}}$$

Since $\left(1-x^2\right)^{\frac{3}{2}} \geq \left(1-x^2\right)^3 \geq 1 - 3x^4 + 2x^6$, we must have that

$$3x^3 - 2x\left(1 - \left(1-x^2\right)^{\frac{3}{2}}\right) \leq 3x^3 - 6x^5 + 4x^7 = 3x^3\left(1 - 2x^2 + x^4\right) + x^7 = 3x^3\left(1-x^2\right)^2 + x^7 \geq 0$$

Thus, we must have that $f''(x) \geq 0$. Therefore, for any $\epsilon \in (0, 0.1]$, we have that for $x \in [\epsilon, 1]$

$$\begin{aligned}
f(x) &\leq f\left(((1+\epsilon)x - \epsilon) \cdot 1 + (1-x) \cdot \epsilon\right) \\
&\leq (1-x) \cdot f(\epsilon) + ((1+\epsilon)x - \epsilon) \cdot f(1) \\
&= (1-x) \cdot f(\epsilon) + \frac{\pi}{2} \cdot x - \frac{\pi}{2} \cdot \epsilon(1-x) \\
&\leq \frac{\pi}{2} \cdot x + 1.002(1-x) \\
&\leq \frac{\pi}{2}
\end{aligned}$$

This gives that $f(x) \leq \frac{\pi}{2}$ for all $x \in (0, 1]$. Since $f(x)$ is an even function, we have $f(x) \leq \frac{\pi}{2}$ for all $x \in [-1, 0)$. Therefore, $|\arcsin x| \leq \frac{\pi}{2}|x|$ when $x \in [-1, 1] \setminus \{0\}$. When $x = 0$, we have $\arcsin x = 0$. This completes the proof. $\square$

**Lemma D.20.** *Let $\mathbf{u} := (|e_1|, ..., |e_n|) \in \mathbb{R}^n$ and $\mathbf{1} := (1, ..., 1) \in \mathbb{R}^n$. Then*

$$\sum_{i=1}^{n} |e_i| = u^\top \mathbf{1}.$$

*By the Cauchy–Schwarz inequality,*

$$u^\top \mathbf{1} \leq \|u\|_2 \|\mathbf{1}\|_2.$$

*Moreover,*

$$\|u\|_2 = \left(\sum_{i=1}^{n} u_i^2\right)^{1/2} = \left(\sum_{i=1}^{n} |e_i|^2\right)^{1/2} = \left(\sum_{i=1}^{n} e_i^2\right)^{1/2}, \qquad \|\mathbf{1}\|_2 = \left(\sum_{i=1}^{n} 1^2\right)^{1/2} = \sqrt{n}.$$

*Combining the above gives*

$$\sum_{i=1}^{n} |e_i| \leq \sqrt{n}\left(\sum_{i=1}^{n} e_i^2\right)^{1/2}.$$

**Lemma D.21.** *Assume that Assumption 3.1 holds. Then, we have:*

$$\|\nabla_{\mathbf{w}_r}\mathcal{L}_\mathbf{C}(\mathbf{W})\|_2^2 \leq \frac{\sqrt{n}}{\sqrt{m}}\mathcal{L}_\mathbf{C}(\mathbf{W})^{\frac{1}{2}}.$$

*Proof.* By the form of $\nabla_{\mathbf{w}_r} \mathcal{L}_\mathbf{C}(\mathbf{W})$ in (2), we have:

$$\|\nabla_{\mathbf{w}_r} \mathcal{L}_\mathbf{C}(\mathbf{W})\|_2 = \left\| \frac{a_r}{\sqrt{m}} \sum_{i=1}^{n} (f(\mathbf{W}, \mathbf{x}_i \odot \mathbf{c}_i) - y_i)(\mathbf{x}_i \odot \mathbf{c}_i) \underbrace{\mathbb{I}\left\{\mathbf{w}_r^\top (\mathbf{x}_i \odot \mathbf{c}_i) \geq 0\right\}}_{\leq 1} \right\|_2$$

$$\leq \frac{|a_r|}{\sqrt{m}} \sum_{i=1}^{n} |f(\mathbf{W}, \mathbf{x}_i \odot \mathbf{c}_i) - y_i| \cdot \|\mathbf{x}_i \odot \mathbf{c}_i\|_2$$

$$\leq \frac{\sqrt{n}}{\sqrt{m}} \left(\sum_{i=1}^{n} (f(\mathbf{W}, \mathbf{x}_i \odot \mathbf{c}_i) - y_i)^2\right)^{1/2} \cdot \|\mathbf{c}_i\|_\infty \|\mathbf{x}_i\|_2 \text{ using D.20}$$

$$\leq C\frac{\sqrt{n}}{\sqrt{m}} (2\mathcal{L}_\mathbf{C}(\mathbf{W}))^{1/2} \text{ using D.22 with } \|\mathbf{c}_i\|_\infty \leq 1 + \kappa\sqrt{2\log\left(\frac{2d}{\delta}\right)} = C \text{ and } \|\mathbf{x}_i\|_2 \leq 1$$

$$\leq C\sqrt{2}\frac{\sqrt{n}}{\sqrt{m}} \mathcal{L}_\mathbf{C}(\mathbf{W})^{1/2}$$

where, in the first inequality, we use the fact that the indicator function is upper-bounded by 1 and in the second inequality, we use the fact that $a_r = \pm 1$.

and so,

$$\|\nabla_{\mathbf{w}_r} \mathcal{L}_\mathbf{C}(\mathbf{W})\|_2^2 \leq 2C^2 \cdot \frac{n}{m} \mathcal{L}_\mathbf{C}(\mathbf{W})$$

$\square$

**Lemma D.22** (High-probability $\ell_\infty$ bound for Gaussian masks). *Fix $\delta \in (0, 1)$. Let $\mathbf{c}_i \in \mathbb{R}^d$ be a Gaussian mask with independent coordinates*

$$\mathbf{c}_i \sim \mathcal{N}(\mathbf{1}, \kappa^2 \mathbf{I}_d), \qquad i.e., \qquad c_{i,j} = 1 + \kappa g_{i,j}, \;\; g_{i,j} \overset{i.i.d.}{\sim} \mathcal{N}(0, 1).$$

*Then, with probability at least $1 - \delta$, we have*

$$\|\mathbf{c}_i\|_\infty \leq 1 + \kappa\sqrt{2\log\left(\frac{2d}{\delta}\right)}.$$

*Proof.* Since $\mathbf{c}_i \sim \mathcal{N}(\mathbf{1}, \kappa^2 \mathbf{I}_d)$ with independent coordinates, each coordinate can be written as

$$c_{i,j} = 1 + \kappa g_{i,j}, \qquad \text{where } g_{i,j} \sim \mathcal{N}(0, 1) \;\; i.i.d.$$

Hence

$$\|\mathbf{c}_i\|_\infty = \max_{j \in [d]} |c_{i,j}| = \max_{j \in [d]} |1 + \kappa g_{i,j}|.$$

We observe that

$$|c_{i,j}| = |1 + (c_{i,j} - 1)|$$
$$\leq |1| + |c_{i,j} - 1| \text{ triangle inequality}$$
$$= 1 + |\kappa g_{i,j}|$$

We will first bound $\max_j |c_{i,j} - 1| = \max_j |\kappa g_{i,j}|$, and then convert this into a bound on $\|\mathbf{c}_i\|_\infty$.

For brevity, we denote $g_{i,j}$ as $g \sim \mathcal{N}(0, 1)$ and we are going to show that

$$\boxed{\Pr(|g| \geq t) \leq 2e^{-t^2/2}} \tag{79}$$

By symmetry of the standard normal distribution,

$$\Pr(|g| \geq t) = \Pr(g \geq t) + \Pr(g \leq -t) = 2\Pr(g \geq t).$$

So it suffices to upper bound $\Pr(g \geq t)$.

For any $\lambda > 0$, since the exponential is monotone increasing we have:

$$g \geq t \Rightarrow \lambda g \geq \lambda t \Rightarrow e^{\lambda g} \geq e^{\lambda t}$$

and so

$$\Pr(g \geq t) = \Pr(e^{\lambda g} \geq e^{\lambda t})$$

By Markov's inequality, for any nonnegative random variable $X$ and any $a > 0$,

$$\Pr(X \geq a) \leq \frac{\mathbb{E}[X]}{a}$$

Applying this with $X = e^{\lambda g}$ and $a = e^{\lambda t}$ gives

$$\Pr(g \geq t) = \Pr\left(e^{\lambda g} \geq e^{\lambda t}\right) \leq \frac{\mathbb{E}[e^{\lambda g}]}{e^{\lambda t}} = \mathbb{E}[e^{\lambda g}]e^{-\lambda t} \tag{80}$$

Computation of the moment generating function $\mathbb{E}[e^{\lambda g}]$:

The standard normal density is

$$\varphi(x) = \frac{1}{\sqrt{2\pi}} e^{-x^2/2}, \qquad x \in \mathbb{R}.$$

Therefore,

$$\mathbb{E}[e^{\lambda g}] = \int_{-\infty}^{\infty} e^{\lambda x} \varphi(x) dx = \frac{1}{\sqrt{2\pi}} \int_{-\infty}^{\infty} \exp\left(\lambda x - \frac{x^2}{2}\right) dx. \tag{81}$$

We now study the exponent:

$$\lambda x - \frac{x^2}{2} = -\frac{1}{2}\left(x^2 - 2\lambda x\right) = -\frac{1}{2}\left((x - \lambda)^2 - \lambda^2\right) = \frac{\lambda^2}{2} - \frac{(x - \lambda)^2}{2}.$$

Plugging this into (81) yields

$$\mathbb{E}[e^{\lambda g}] = \frac{1}{\sqrt{2\pi}} \int_{-\infty}^{\infty} \exp\left(\frac{\lambda^2}{2} - \frac{(x - \lambda)^2}{2}\right) dx$$

$$= e^{\lambda^2/2} \cdot \frac{1}{\sqrt{2\pi}} \int_{-\infty}^{\infty} \exp\left(-\frac{(x - \lambda)^2}{2}\right) dx.$$

We set $u = x - \lambda$ (change of variables with $dx = du$)

$$\int_{-\infty}^{\infty} \exp\left(-\frac{(x - \lambda)^2}{2}\right) dx = \int_{-\infty}^{\infty} \exp\left(-\frac{u^2}{2}\right) du = \sqrt{2\pi}.$$

Hence

$$\mathbb{E}[e^{\lambda g}] = e^{\lambda^2/2}. \tag{82}$$

Substituting (82) into (80) gives

$$\Pr(g \geq t) \leq \exp\left(\frac{\lambda^2}{2} - \lambda t\right), \qquad \forall \lambda > 0.$$

The right-hand side is a valid bound for *every* $\lambda > 0$, so we choose $\lambda$ to make it as small as possible. Define

$$f(\lambda) := \frac{\lambda^2}{2} - \lambda t.$$

Then $f'(\lambda) = \lambda - t$, so the unique minimizer is $\lambda = t$ (and $f''(\lambda) = 1 > 0$ confirms it is a minimum). Plugging $\lambda = t$ gives

$$\Pr(g \geq t) \leq \exp\left(\frac{t^2}{2} - t^2\right) = e^{-t^2/2}.$$

Using $\Pr(|g| \geq t) = 2 \Pr(g \geq t)$ proves (79).

Now, we focus on one of the mask coordinates by fixing a coordinate $j \in [d]$. Since $c_{i,j} - 1 = \kappa g_{i,j}$ with $g_{i,j} \sim \mathcal{N}(0, 1)$, for any $u \geq 0$ we have

$$\Pr\left(|c_{i,j} - 1| \geq u\right) = \Pr\left(|g_{i,j}| \geq u/\kappa\right) \leq 2 \exp\left(-\frac{u^2}{2\kappa^2}\right),$$

where we applied (79) with $t = u/\kappa$.

Let's define the event

$$\mathcal{E}_i(u) := \left\{ \max_{j \in [d]} |c_{i,j} - 1| \leq u \right\}.$$

Its complement is the event that at least one coordinate deviates by more than $u$:

$$\mathcal{E}_i(u)^c = \left\{ \exists j \in [d] \text{ s.t. } |c_{i,j} - 1| > u \right\}.$$

By the union bound,

$$\Pr\left(\mathcal{E}_i(u)^c\right) = \Pr\left(\bigcup_{j=1}^{d} \{|c_{i,j} - 1| > u\}\right) \leq \sum_{j=1}^{d} \Pr\left(|c_{i,j} - 1| > u\right)$$

$$\leq \sum_{j=1}^{d} 2 \exp\left(-\frac{u^2}{2\kappa^2}\right) = 2d \exp\left(-\frac{u^2}{2\kappa^2}\right).$$

Choose $u$ so that the right-hand side is at most $\delta$:

$$2d \exp\left(-\frac{u^2}{2\kappa^2}\right) \leq \delta \iff -\frac{u^2}{2\kappa^2} \leq \log\left(\frac{\delta}{2d}\right) \iff u \geq \kappa \sqrt{2 \log\left(\frac{2d}{\delta}\right)}.$$

Set

$$u := \kappa \sqrt{2 \log\left(\frac{2d}{\delta}\right)}.$$

Then $\Pr(\mathcal{E}_i(u)^c) \leq \delta$, i.e. $\Pr(\mathcal{E}_i(u)) \geq 1 - \delta$, and on $\mathcal{E}_i(u)$ we have

$$|c_{i,j} - 1| \leq u \quad \forall j \in [d].$$

On $\mathcal{E}_i(u)$, for each coordinate $j$,

$$|c_{i,j}| = |1 + (c_{i,j} - 1)| \leq |1| + |c_{i,j} - 1| \leq 1 + u.$$

Taking the maximum over $j$ yields, with probability at least $1 - \delta$,

$$\|\mathbf{c}_i\|_\infty = \max_{j \in [d]} |c_{i,j}| \leq 1 + u = 1 + \kappa \sqrt{2 \log\left(\frac{2d}{\delta}\right)}.$$

$\square$

