# OpenReview forum: "Convergence Analysis of Two-Layer Neural Networks under Gaussian Input Masking"
_TMLR — Under review for TMLR_

### Review · Reviewer_fD53 · 2026-04-25

**Summary Of Contributions:**

The paper provides a convergence analysis for neural network training under Gaussian multiplicative input masking.  The paper provides a theoretical analysis of input masking, general stochastic training framework, convergence guarantees, and empirical validation through simulations.

The main weaknesses of the proposal are the assumptions of overparameterization needed for the NTK regime, the limitation of two-layer MLPs which further limits the generalization of the theoretical guarantees, the privacy accounting is also limited as noted by the authors.

**Audience:**

Yes

**Audience Explanation:**

While limited in the setup and having strong assumptions, the proposal sheds some light on convergence two layer networks.

**Broader Impact Concerns:**

No concerns

**Claims And Evidence:**

Yes

**Claims Explanation:**

Despite the paper limitations, the main theoretical claims are supported by the descriptions and proofs in the paper.  Moreover the experimental results in the corroborate them as well.

**Requested Changes:**

Sections 4 and 5 would benefit from having descriptions about the intuition of the method and the proofs, and how they can be used as well.

While the paper is focused on the derivations, it will help to have more analysis and discussion of the relevance of the convergence guarantees and how they can be translated to practical scenarios.

Overall the derivations are sound, but I have some minor questions about the paper:
- Is the subscript of the $\tau_1$ term supposed to be there?  It seems to be a typo.
- In remark 4.5, shouldn't the gradient w.r.t. $w_r$ remove the summation since the gradient w.r.t. any other $w_s$ for $s \neq r$ should vanish?

---

> ### Author Response · Authors · 2026-06-14
> **Response to Reviewer fD53**
>
> We thank the reviewer for the careful reading and constructive suggestions. We are grateful for the acknowledgment of the paper's main contribution and for the clear identification of areas where the exposition can be improved. We address each of the requested changes below:
> - The main weaknesses...as noted by the authors
>
> We agree these are genuine limitations, and we state them explicitly in Section 7 (Conclusion). The NTK overparameterization and the two-layer restriction are not specific to our work: they are inherited from the line of convergence analyses we build on  [Du et al, 2018, Liao and Kyrillidis, 2022, Mianjy and Arora, 2020], which operate in the same regime. Our contribution is to handle, within that established setting, the additional difficulty of randomness applied before the nonlinearity, so that the mask is carried through the ReLU, which to our knowledge had not previously been analyzed. Relaxing these assumptions toward deeper networks and beyond the NTK regime is an open direction we flag as future work. Regarding privacy, the membership-inference experiments are an empirical illustration of one setting where the analyzed noise is deliberately applied, and we make no formal privacy claim.
> - Sections 4 and 5 would benefit...can be used as well
>
> For Theorem 4.2 and Theorem 4.9, the main technique we use is an explicit integration over the Gaussian distribution of the mask by carefully applying integration by parts. Then, we separate the dominating term that is determined by the Gaussian CDF and an exponentially decaying error term. Lastly, we bound the error term to get the desired result. We have added this discussion in updated version of the paper. For more details of the proof, we kindly refer the reviewer to the sketch of he proofs provided after the statement of the theorems. Regarding Theorem 5.2, the proof structure follows a standard NTK type argument as in [Du et al.,2018, Song and Yang, 2020], where the minimum eigenvalue of the finite-width NTK is lower-bounded to give explicit convergence rate, and the neurons are separated into "changing" and "unchanging" subsets to account for the nonlinearity. The expectation and gradient bias are injected at proper steps to handle the stochasticity introduced by the masks. Theorem 5.6 is obtained by instantiating Theorem 5.2 with the following collection of results: Corollary 5.3 supplies $\varepsilon_1$, Corollary 5.4 supplies $\varepsilon_2,\varepsilon_3$ (both from the Section 4  analysis), and Lemma 5.5 supplies the relaxed-smoothness constant γ. The κ-dependence of the final error floor in (16) is then inherited directly from the κ-dependence of these three error terms. We have updated the paper with the discussion.
> - While the paper..practical scenarios.
>
> We thank the reviewer for the advice to discuss the practical significance. Though our work is primarily theoretical, we have added three practitioner takeaways to the Conclusion (Section 7, marked in purple): (1) Our paper derives the expected masked activation (up to a small correction term), which is a closed-form smoothed ReLU whose smoothness is controlled by κ, giving an intuitive link between Gaussian dropout and GELU; (2) Multiplicative masking induces a data-dependent regularizer, the $\mathcal{T}_2$ term in Theorem 4.2, shaped by the data and current weights rather than the isotropic penalty of weight decay; and (3) Convergence reaches a noise-dependent floor: a geometric decrease followed by a κ-controlled plateau, whose qualitative shape is observable beyond the NTK regime (Figure 3(a)). We refer the reviewer to Section 7 for the full discussion.
>
> - Is the subscript of the  $\tau_1$ term supposed to be there? Seems to be a typo.
>
> We would like to politely ask which $\tau_1$ the reviewer is referring to, as we did not find such a symbol in our paper after some attempt.
>
> - In remark 4.5, shouldn't the gradient...
>
> We thank the reviewer for pointing out the typo. The gradient in Remark 4.5 was indeed incorrect: the summation should be removed, because the gradient with respect to $\mathbf{w}_r$ of any term involving $\mathbf{w}_s$ for $s \neq r$ vanishes. The correct expression is
> $\nabla\_{\mathbf{w}\_r}\hat{f}(\mathbf{W},\mathbf{x}\_i) \approx \frac{1}{\sqrt{m}} a\_r \mathbf{x}\_i \Phi\_1\left(\frac{\mathbf{w}\_r^\top\mathbf{x}\_i}{\kappa\||\mathbf{u}\_{i,r}\||\_2}\right).$
>
> We thank the reviewer again for the careful reading and the constructive suggestions.
>
> References
>
> Du, S. S., Zhai, X., Poczos, B., and Singh, A. Gradient Descent Provably Optimizes Over-Parameterized Neural Networks. arXiv:1810.02054, 2018.
>
> Liao, F. and Kyrillidis, A. On the Convergence of Shallow Neural Network Training with Randomly Masked Neurons. TMLR, 2022.
>
> Mianjy, P. and Arora, R. On Convergence and Generalization of Dropout Training. NeurIPS, 2020.
>
> Song, Z. and Yang, X. Quadratic Suffices for Over-Parametrization via Matrix Chernoff Bound. arXiv:1906.03593, 2020.

---

### Review · Reviewer_JoR6 · 2026-04-26

**Summary Of Contributions:**

The authors provide a convergence analysis for Gaussian input masking, which consists of the following:
- analysis of training dynamics for two-layer ReLU networks with Gaussian input masking, with the noise injection preceding the non-linear activation
- convergence theorem, which establishes generic linear convergence, i.e. without assumption of the Gaussian masking, to an error ball parameterised with the noise
- applications to privacy: the authors demonstrate how this can be applied to the setting of defence against membership inference attacks (MIA)

**Additional Comments:**

No additional comments

**Audience:**

Yes

**Audience Explanation:**

I put 'no' for now, because I believe this question requires additional clarification.

(1) I think this work, through literature review, could be put in a wider context of works. The current literature review tells about well-known papers, such as Dropout ((Srivastava et al., 2014)) and  Neural Tangent Kernel (Jacot et al. (2018)). There is, however, as authors mention, rich literature on the analysis of training dynamics and convergence of neural networks. There is a need, therefore, to contrast this work against the existing convergence analyses.

(2) In addition to showing the exact place of this work in the literature (what is the closest theoretical result that the community has achieved?), I think it would be important to show why this particular setting could inform practical models. That would mean: how do  the insights of this work generalise to realistic, larger-scale neural networks, and if it does not generalise, how does it inform the users?

(3) The MIA discussion needs to be joined with the rest of this work. This might be linked to the overall premise by saying that the proposed analysis is developed for the purpose of MIA defence, or alternatively, present other use cases where this setting is useful.

(4) Overall, I see the work has three strands: (1) analysis of the  two-layer ReLU networks with Gaussian input masking (2) generic linear convergence theorem (3) MIA defence. The authors should clearly outline the motivation of this work so that one could see it as a single contribution.

**Claims And Evidence:**

Yes

**Claims Explanation:**

The proofs are, to my knowledge, sound and correct.

I am putting no due to the following reason:
- "Furthermore, we demonstrate the practical utility of this training regime as a defense against Membership Inference Attacks (MIA), highlighting a favorable trade-off between privacy and utility." It would be important to see whether it is possible to see how it generalises to any datasets beyond toy examples, and what would be the trade-off in that case. Otherwise, the claim of favourable trade-off does not sound correct: we may know that the trade-off is favourable for the toy examples, but do we know for the larger datasets (e.g., ImageNet) for when it actually might be practically relevant?

**Requested Changes:**

It is crucial to address the questions above.

---

> ### Author Response · Authors · 2026-06-14
> **Response to Reviewer JoR6**
>
> We thank the reviewer for the careful reading and for pushing us to clarify the paper's positioning and scope. We address each concern below, and the revised manuscript has incorporated the corresponding changes.
>
> - It would be important to see...for the larger datasets (e.g., ImageNet) for when it actually might be practically relevant?
>
> We agree with the reviewer that the current evidence does not support the asserted favorable trade-off. We have removed relevant claims about the ``favorable trade-off'' from both the introduction and the discussion about the MIA results, and state explicitly that we do \emph{not} interpret this as evidence that multiplicative Gaussian noise is competitive with formal privacy mechanisms (e.g.\ DP-SGD).
> We also provided additional experimental results on SVHN and Fashion-MNIST in the paper to better support the quantitative behavior predicted by the analysis. We respectfully note that our contribution is primarily theoretical, and we use the experiments only as a validation for the theoretical results.
>
> - I think this work, through literature review...contrast this work against the existing convergence analyses.
>
> Thank you for your advice. We have re-written our related works section to also include more recent works on the implicit bias of neural network training and feature learning (marked in purple).
>
> - In addition to showing the exact place of this work in the literature...how does it inform the users?
>
> We thank the reviewer for the advice to discuss the practical significance.Though our work is theoretical by its nature, the results could inform practitioners in the following ways:
> 1. A closed-form for how multiplicative input noise smooths ReLU. Our paper derives expected masked activation (up to a small correction term), which can be seen as a smoothed ReLU or a more general form of the popular GELU activation, whose smoothness is controlled by the noise scale κ. This result offers a way to conceptualize what input-level multiplicative noise does to a ReLU unit, and provides an intuitive connection between Gaussian drop-out and GELU activation.
> 2. Multiplicative masking induces a data-dependent regularizer. The $\mathcal{T}_2$ term in Theorem 4.2 has the form of a regularization that depends on the tangent-feature outer-product matrix (as discussed in Remark 4.5). Qualitatively, this is a penalty shaped by the data and the current weights, unlike weight decay's isotropic Frobenius norm penalty.
> 3. Convergence reaches a noise-dependent floor. Theorem 5.6 gives a linear convergence up to an error term where the error is a sum of κ-dependent terms. The exact constants are tied to the NTK regime, but the qualitative shape, a geometric decrease followed by a plateau whose height increases with κ, is observable beyond that regime. Figure 3(a) shows it on the synthetic regression task the theory describes.
> We have added these three takeaways in our conclusion.
>
> - The MIA discussion...other use cases where this setting is useful.
>
> While the central contribution of our paper is a theoretical convergence analysis for NNs trained under multiplicative Gaussian input masking, the MIA section aims to provide a use case that embeds the scheme of Gaussian input masking into practical scenarios and study the behavior that falls both within and beyond the scope of our theoretical study. We do not claim formal privacy or systems contributions; the experiments are intended to corroborate concrete settings where the analysis applies.
>
> - Overall, I see the work has three strands...could see it as a single contribution.
>
> Strand (1) analyzes the expected masked objective and the expected gradient, which deviates from traditional SGD analysis that assumes an unbiased gradient. Strand (2) resolves this difficulty by considering a convergence framework that incorporates gradient bias. Combined, Strand (1) and (2) give the convergence guarantee of training under Gaussian mask as in Theorem 5.6. Strand (3) is one instantiation of the analyzed noise model, and is included to illustrate the empirical behavior of the noise model.
> We hope that this instantiation gives a more complete picture of the Gaussian mask scheme.
>
> References
>
> Jacot, A., Gabriel, F., and Hongler, C. Neural Tangent Kernel: Convergence and Generalization in Neural Networks. NeurIPS, 2018.
>
> Du, S. S., Zhai, X., Poczos, B., and Singh, A. Gradient Descent Provably Optimizes Over-Parameterized Neural Networks. arXiv:1810.02054, 2018.
>
> Oymak, S. and Soltanolkotabi, M. Towards Moderate Overparameterization: Global Convergence Guarantees for Training Shallow Neural Networks. arXiv:1902.04674, 2019.
>
> Song, Z. and Yang, X. Quadratic Suffices for Over-Parametrization via Matrix Chernoff Bound. arXiv:1906.03593, 2020.
>
> Liao, F. and Kyrillidis, A. On the Convergence of Shallow Neural Network Training with Randomly Masked Neurons. TMLR, 2022.
>
> Mianjy, P. and Arora, R. On Convergence and Generalization of Dropout Training. NeurIPS, 2020.

---

> > ### Comment · Reviewer_JoR6 · 2026-06-24
> >
> > Dear authors,
> >
> > Many thanks for your response, and I acknowledge the substantial improvements in the paper that address my comments.
> >
> > " We also provided additional experimental results on SVHN and Fashion-MNIST in the paper to better support the quantitative behavior predicted by the analysis. We respectfully note that our contribution is primarily theoretical, and we use the experiments only as a validation for the theoretical results."
> >
> > Thank you, amendments of the claims and the experiments do improve the standing of the paper. Just to note that the intention was not to make the theoretical paper into an empirical one, but to support the theoretical claims, including showing how it would translate into the real world scenario. This is because in this case there is a need to show that these theoretical claims do represent the actual processes.

---

### Review · Reviewer_qSyb · 2026-05-31

**Summary Of Contributions:**

The paper investigates the convergence guarantee of two-layer ReLU neural network training with Gaussian randomly masked inputs. Using the NTK framework, the paper proves linear convergence despite input corruption, with a residual error depending on the masking variance. The main technical novelty lies in handling randomness inside the nonlinear activation function during the convergence analysis.

Strengths
1. The paper resolves the challenging problem of handling randomness inside non-linear ReLU activations and shows that the expected loss can be decomposed into a smoothed objective and an adaptive regularizer.
2. The paper provides a convergence guarantee for two-layer ReLU networks trained with Gaussian input masking, showing that linear convergence can still be achieved despite the injected noise.
3. The experiments are conducted on both synthetic and real datasets. The experimental results validate the theoretical guarantees by demonstrating linear convergence and a final error radius that grows with the masking variance.

Weaknesses
1. The experiments are conducted with small-noise constraint and limited synthetic sample (n = 5000), and only on real dataset is used (CIFAR-10 dataset).

**Audience:**

Yes

**Audience Explanation:**

The findings are likely to interest researchers in machine learning theory, neural network optimization.

**Claims And Evidence:**

Yes

**Claims Explanation:**

Experiments on both synthetic and real datasets support the theoretical claims and confirm the predicted convergence behavior.

**Requested Changes:**

Additional experiments on more datasets and under broader noise settings would further strengthen the empirical validation.

---

> ### Author Response · Authors · 2026-06-14
> **Response to Reviewer qSyb**
>
> We thank the reviewer for the careful reading and the positive assessment of the paper's contributions. We are glad the technical novelty of handling randomness inside the nonlinear activation and the convergence guarantee came through clearly. We address the requested change below:
>
> We agree with the reviewer and have added experiments on three further datasets in Appendix A.3 (Additional Datasets and Broader Noise Range): Fashion-MNIST, SVHN, and CIFAR-100. For Fashion-MNIST and SVHN we report final test accuracy versus κ for both a CNN and an MLP for a bigger range of κs (up to κ=3.0). We also added a similar CIFAR-100 experiment for a deeper CNN. We additionally include clean training-loss trajectories for a CNN and an MLP on Fashion-MNIST, reproducing Figure 3(a) using a real dataset.
>
> We thank the reviewer again.